

# LegacyVegetation 1.0: Global reconstruction of vegetation composition and forest cover from pollen archives of the last 50 ka

Laura Schild[1,2], Peter Ewald[1,2], Chenzhi Li[1,2], Raphaël Hébert[1], Thomas Laepple[1,3], and Ulrike Herzschuh[1,2,4]

[1]Helmholtz Centre for Polar and Marine Research, Research Unit Potsdam, Alfred Wegener Institute (AWI), Germany
[2]Institute of Environmental Sciences and Geography, University of Potsdam, Karl-Liebknecht-Straße 24-25, Potsdam, Germany
[3]MARUM-Center for Marine Environmental Sciences and Faculty of Geosciences, University of Bremen, Germany
[4]Institute of Biochemistry and Biology, University of Potsdam, Karl-Liebknecht-Straße 24-25, Potsdam, Germany

**Correspondence:** Ulrike Herzschuh (ulrike.herzschuh@awi.de)

**Abstract.** With rapid anthropogenic climate change future vegetation trajectories are uncertain. Climate-vegetation models can be useful for predictions but need extensive data on past vegetation for validation and improving systemic understanding. Even though pollen data provide a great source of this information, the data is compositionally biased due to differences in taxon-specific relative pollen productivity (RPP) and dispersal.

Here we present a reconstruction of quantitative regional vegetation cover from a global sedimentary pollen data set for the last 50 ka using the REVEALS model to correct for taxon- and basin-specific biases. In a first reconstruction, we used previously published, continental RPP values. For a second reconstruction, we statistically optimized RPP values for common taxa with the goal of improving the fit of reconstructed forest cover from modern pollen samples with remote sensing forest cover.

The data sets include taxonomic compositions as well as reconstructed forest cover for each original pollen sample. Relative
pollen sources areas were also calculated and are included in the data set of the original REVEALS run. Additional metadata includes modeled ages, age model sources, basin locations, types and sizes.

The improvements in forest cover reconstructions with the REVEALS reconstruction using original/optimized parameters range from 1/0% (Australia and Oceania/Australia and Oceania) to 58/65% (Europe/North America) relative to the mean absolute error (MAE) in the pollen-based reconstruction. Optimizations were considerably more successful in reducing MAE
when more records and RPP estimates were available. The optimizations were purely statistical and only partly ecologically informed and should, therefore, be used with caution depending on the study matter.

This improved quantitative reconstruction of vegetation cover is invaluable for the investigation of past vegetation dynamics and modern model validation. By collecting more RPP estimates for taxa in the Southern Hemisphere and adding more records to existing pollen data syntheses, reconstructions may be improved even further. Both reconstructions are freely available on
PANGAEA (see Data availability section).



# 1 Introduction

Anthropogenic climate change is driving vegetation shifts that could lead to disruptions in ecosystem functions and services, and even trigger feedback effects with other earth system elements (IPCC, 2023; Armstrong McKay et al., 2022). Predicting these changes through modeling is challenging. A sufficient mechanistic understanding of vegetation dynamics and interac-
tions with climate is needed, which requires validation and testing of model data with extensive vegetation data across climatic transitions akin to those anticipated in the future (Dearing et al., 2012). Given the relatively brief duration of available instrumental climate and vegetation data, there is a clear need for long-term environmental records derived from paleoecological archives that cover broader climatic gradients than modern datasets (Dearing et al., 2010; Dallmeyer et al., 2023).

Pollen data as a direct proxy for paleo-vegetation is especially useful for comparisons with modeled data as it can be used to reconstruct land-use (Fyfe et al., 2015; Davis et al., 2015), biomes (Woodbridge et al., 2014; Prentice et al., 1996), and climate (Herzschuh et al., 2023a, b; Bartlein et al., 2011; Viau et al., 2012). The compilation of pollen data syntheses is essential to aid this purpose (Anderson et al., 2006; Gaillard et al., 2010; Strandberg et al., 2014). Several subcontinental and continental collections of pollen data already exist, spanning regions such as Europe, North America, Africa, Siberia, and China (Fyfe
et al., 2009; Whitmore et al., 2005; Vincens et al., 2007; Cao et al., 2014, 2020) and have been integrated into the global database Neotoma (Williams et al., 2018). To allow for a broader application of pollen data, LegacyPollen 2.0 (Li et al., 2024b) offers a global, harmonized pollen dataset that underwent taxonomic standardization, metadata verification and consistent age modeling (Li et al., 2022a, 2021; Herzschuh et al., 2022). Despite advances in harmonization, the use of pollen data remains limited due to the fact that pollen compositions do not accurately reflect vegetation (Davis, 1963; Prentice, 1985; Prentice and Webb III, 1986). This limitation arises from variations in taxon-specific parameters like relative pollen productivity (RPP) and pollen dispersal characteristics, leading to discrepancies between the pollen record and real past vegetation. This hinders quantitative vegetation assessment as taxa with high pollen productivity and efficient pollen dispersal tend to be overrepresented in the pollen record, while those with low pollen productivity and less effective dispersal are underrepresented. These factors, together with the compositional nature of pollen data, result in a non-linear relationship between pollen and vegetation (Prentice and Webb III, 1986). Approaches such as the R-value model (Davis, 1963; Webb et al., 1981) and the extended R-value model (Parsons and Prentice, 1981) were created to address this issue and were refined with Sugita's (2007) model for "Regional Estimates of Vegetation Abundance from Large Sites" (REVEALS) . By accounting for taxon-specific RPP and fall speed values, as well as basin-specific parameters such as basin size and type, REVEALS models quantitative vegetation cover in relevant pollen source areas from pollen compositions. The model has been applied in several regional-scale studies (Nielsen et al., 2012; Mazier et al., 2015; Hellman et al., 2008; Nielsen and Odgaard, 2010) and multiple validations have demonstrated its accuracy in approximating actual vegetation (Sugita et al., 2010; Hellman et al., 2008; Soepboer et al., 2010; Mazier et al., 2012), even though the model's performance heavily relies on accurate taxon-specific parameters. While Wieczorek and Herzschuh (2020) provide a comprehensive compilation of RPP and fall speed values for taxa of the Northern Hemisphere, the overall availability of RPP studies is still limited and regional variations in RPP values exist (Harris et al.,

2020; Broström et al., 2008; Li et al., 2017; Mazier et al., 2012). This makes the application of REVEALS on larger scales particularly challenging. Only some (sub-) continental REVEALS reconstructions are available for Europe (Trondman et al., 2015; Roberts et al., 2018; Githumbi et al., 2021; Serge et al., 2023), Asia (Cao et al., 2019; Li et al., 2022b, 2023, 2024a), and North America (Dawson et al., 2018). Currently, no global quantitative vegetation cover reconstructions using REVEALS exist.

With its importance for the assessment of biome stability, carbon storage, climatic feedbacks, and land-use-change, forest cover is an often reconstructed variable (e.g. Fyfe et al., 2015; Githumbi et al., 2021; Serge et al., 2023). Due to the global availability of remote sensing data on contemporary forest cover, it also offers good opportunities for the validation of reconstructions (Hjelle et al., 2015; Roberts et al., 2018). Yet, only Serge et al. (2023) use this opportunity for extensive validation and even improvement of reconstructions from European pollen records. No site-wise validations or attempts at improvements 65 of forest cover reconstructions by adjusting RPP values exist for other regions or on global scales.

Here we present global reconstructed quantitative vegetation cover from the LegacyPollen2.0 dataset - an updated global taxonomically and temporally standardized fossil pollen dataset of 3728 palynological records - using REVEALS spanning primarily the last 50k years, with some records reaching back even further. The data sets were created using existing estimates 70 of taxon-specific parameters and also applied an optimization approach to improve parameters. Using remote sensing forest cover we adjust RPP values for the ten most common taxa on each continent for better agreement of reconstructed with remote sensing forest cover. The REVEALS reconstructions with original and optimized parameters include corrected vegetation compositions as well as reconstructed forest cover.

## 2 Methods

### 2.1 Pollen Data Set

The pollen data synthesis LegacyPollen2.0 (Li et al., 2024b) includes 3728 temporally resolved records (time-series) distributed globally. Sediment and peat cores used for the creation of pollen data are of lacustrine, peat and marine origin. Analogous to the preceding LegacyPollen 1.0 dataset (Herzschuh et al., 2022), the data synthesis involved revising age modeling and taxonomic harmonization for consistency of records. Spatial data coverage of records in the reconstruction is densest in North America 80 (1132 records) and Europe (1451), sparser in Asia (706) and very scattered in South America (191), Africa (164) and Australia and Oceania (84, see Fig. 1). The records primarily span the last 50 ka with temporal coverage being a lot sparser before 20 ka BP (see Fig. 2).

### 2.2 Implementing REVEALS

The REVEALS model ("Regional Estimates of Vegetation Abundance from Large Sites") estimates quantitative vegetation 85 coverage from pollen assemblages using site and taxon-specific parameters (Sugita, 2007). Based on wind speed and taxon-

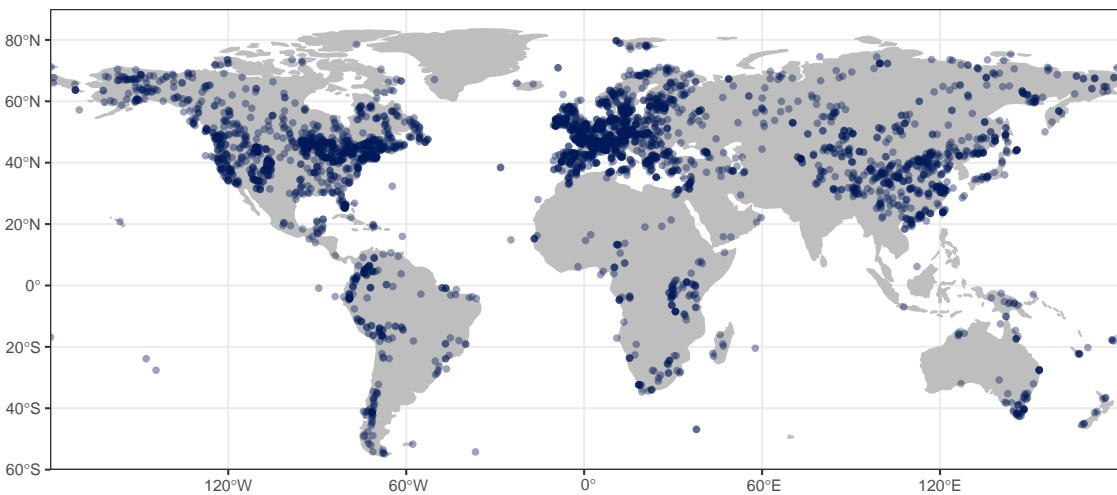

**Figure 1.** Pollen record locations in the LegacyVegetation dataset. Record density is highes in Europe and North America, and lowest in Africa and Australia and Oceania.

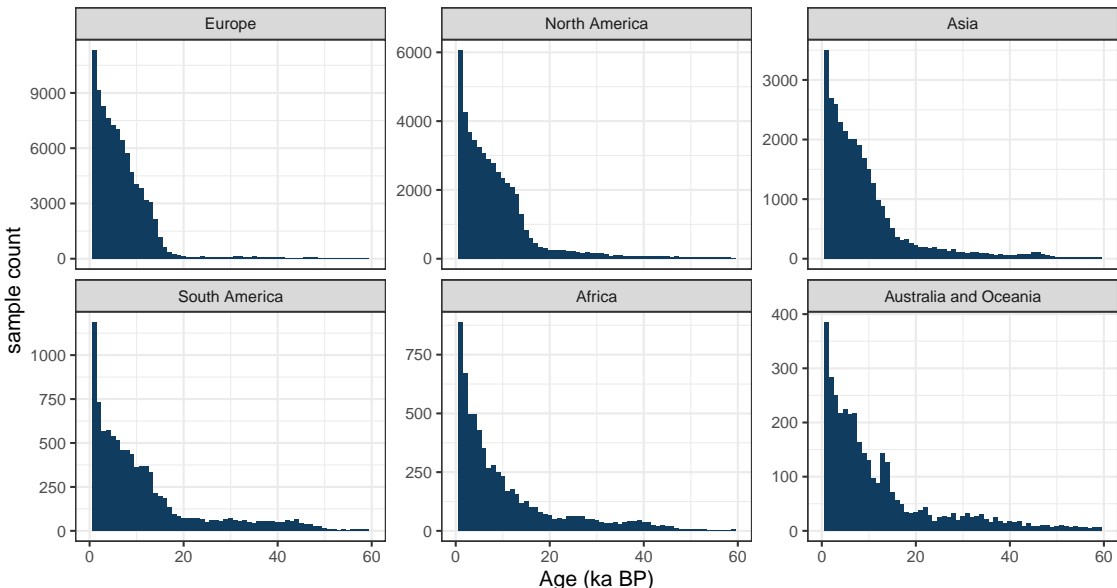

**Figure 2.** Temporal coverage of records in the LegacyVegetation dataset per continent. Bins are 1000 years wide. Sample count decreases with age with a noticeable drop in samples at 20 ka BP

specific fall speed, pollen dispersal is modeled in ring sources around the basin and deposition over the basin is integrated to give pollen influx. Together with RPP this dispersal factor is used to correct original pollen counts to better represent real



vegetation (see Equation 1 and Table 1). By running the model with variations of relative pollen productivity (RPP) values, a statistical distribution of results is calculated.

$$90 \quad \hat{V}_i = \frac{n_{i,k}/\hat{\alpha}_i \int_R^{Z_{max}} g_i(z)dz}{\sum_{j=1}^{m}(n_{j,k}/\hat{\alpha}_j \int_R^{Z_{max}} g_i(z)dz)} \tag{1}$$

The REVEALS model follows a set of assumptions. Firstly, neither directionality nor pollen transport through agents other than

**Table 1.** Algebraic terms in the REVEALS equation (see Equation 1)

| Function term | explanation |
| --- | --- |
| $\hat{V}_i$ | vegetation estimate of taxon i |
| $n_{i,k}$ | pollen counts of taxon i at site k |
| $\alpha_i$ | relative pollen productivity of taxon i |
| $R$ | basin radius |
| $Z_{max}$ | maximum extent of regional vegetation |
| $z$ | distance from a point in the center of a basin |
| $g_i$ | dispersal and deposition function for taxon i |

wind are considered in the model. Additionally, it is assumed that the basin is circular with no source of pollen within the basin radius. The peatland and bog sites used in our reconstructions inherently violate this assumption. Nevertheless, the quantitative reconstruction of vegetation cover from peatland cores is possible by using Prentice's deposition model (Prentice, 1985, 1988)
instead of Sugita's deposition model (Sugita, 1993) in the dispersal and deposition function (see Eq. 1; Sugita, 2007). Previous studies show that results from small bogs are still reliable, while results from large bogs tend to deviate from those of large lakes (Trondman et al., 2015; Mazier et al., 2012). Using peatland records for reconstructions is, therefore, appropriate. All sites that were not classified as lakes were run with peatland settings. We use the implementation of REVEALS from the R package REVEALSinR (Theuerkauf et al., 2016).

### 2.2.1 Parameters

For each site, the REVEALS model also requires information on basin type, basin size and original pollen counts, all of which were collected in the LegacyPollen 2.0 dataset (Li et al., 2024b). For each taxon, values for RPP (with uncertainties provided as standard deviation) and fall speeds are used. When available, we use continent-specific values in our reconstruction following the synthesis of Northern Hemisphere RPP and fall speed values by Wieczorek and Herzschuh (2020). For taxa with no
continental values present, we use northern hemispheric values. If no values exist for a taxon, RPP is set to a constant (RPP = 1, $\sigma$=0.25) and fall speeds are filled with mean continental fall speeds (see Appendix A: Original RPP and fall speed values per continent). The fraction of pollen counts for which RPP estimates are available are much higher in the Northern Hemisphere





than in the Southern Hemipshere (see Fig. 3). Apart from taxon- and basin-specific parameters the REVEALS model requires several constant parameters to be set, which can be found in Table 2.

**Table 2.** Static model parameters for REVEALS runs using REVEALSinR (Theuerkauf et al., 2016).

| Parameter | Values and settings used in model run |
|---|---|
| atmospheric model | unstable atmosphere |
| dispersal model | gaussian plume |
| wind speed | $3m \times s^{-1}$ |
| maximum extent of regional vegetation (region cutoff) | 1000 km |
| number of RPP variations | 2000 |
| peatland basin radius | 100 m |
| function to randomize pollen counts | rmultinom_reveals |

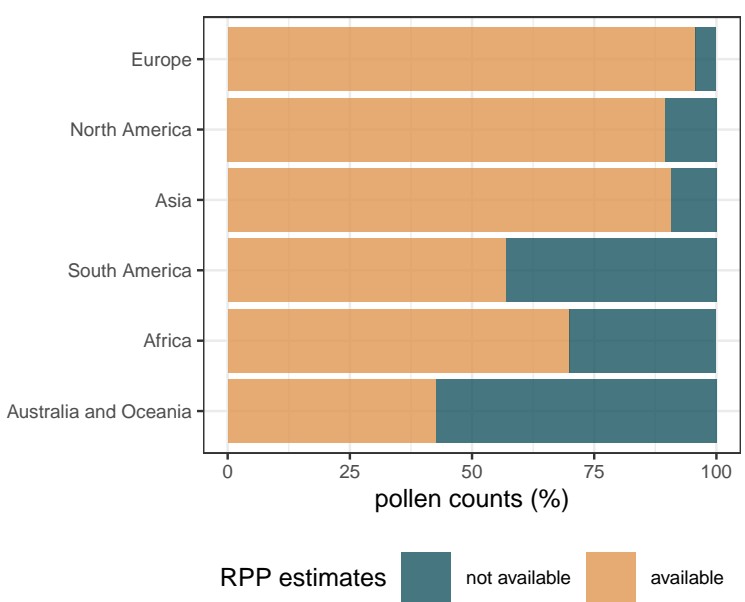

**Figure 3.** Percentage of pollen counts per continent for which RPP estimates are available. A higher percentage of pollen counts has RPP information in the Northern Hemisphere compared to the continents of the Southern Hemisphere.



### 2.2.2 Modifications in REVEALSinR

We calculate the radius of relevant pollen source area by finding the radius in which the median influx of all taxa is 80% of the total influx (as defined by the total influx in the maximum extent of regional vegetation chosen). We also reduced computational effort by implementing a maximum number of steps in the lake model used to model mixing in the basin. The number of steps was set to 500 unless $n$ falls below that maximum value for $n = basin\,radius/10$ for basins with a radius of at least 1000 m and $n = basin\,radius/2$ for basins with a radius smaller than 1000 m.

### 2.3 Reconstruction of forest cover and validation

Forest cover was reconstructed by summing up percentages of arboreal taxa (see S1: List of arboreal taxa) with Betulaceae being classified as arboreal at sites below 70° N. The mean reconstructed compositional coverages from the REVEALS results were used for the forest cover reconstructions. For validation, the reconstructed forest cover of the past 500 years was compared to modern remote sensing forest cover. Average tree canopy cover within pollen source areas of all sites was extracted from the Landsat Global Forest Cover Change (GFCC) data set from the temporal average of the years 2000, 2005, 2010 and 2015 (Sexton et al., 2013; Townshend, 2016). An openness correction was applied to sites containing urban areas and paved surfaces within the pollen source areas (PSA) to correct for areas without any pollen sources and thus improve comparability to modern remote sensing forest cover (see Equations 2-4). For this, the percentage of unvegetated land cover classes for the year 2015 in the ESA CCI land cover data set was used (ESA, 2017, see Table 3). Areas covered by water or ice are already considered as missing values in the remote sensing forest cover data set and do not need to be corrected for. Forest cover was validated site-wise and mean absolute error (MAE) calculated for each continent.

**Table 3.** Unvegetated land cover classes in ESA CCI LC chosen for the openness correction.

| Name | Code |
|------|------|
| Urban areas | 190 |
| Bare areas | 200 |
| Consolidated bare areas | 201 |
| Unconsolidated bare areas | 202 |

$$unvegetated\ classes = \{190, 200, 201, 202\} \tag{2}$$

$$unvegetated\ (\%) = \frac{\sum cells\ in\ PSA \in open\ classes}{\sum cells\ in\ PSA} \tag{3}$$



$$corrected\ tree\ cover = reconstructed\ tree\ cover \times (1 - unvegetated) \tag{4}$$

### 2.4 Optimization

In addition to the REVEALS approach, which is motivated by a biophysical model but also based on a large number of model choices and parameters, we also apply a statistical approach. Here, RPP values for common taxa are estimated by minimizing the misfit of reconstructed and remote sensing forest cover. For the optimization we rely on the "L-BFGS-B" method (Byrd et al., 1995), which allows for box constraints, and minimize the residual sum of squares (RSS) of reconstructed forest cover with remote sensing forest cover. RPP values were bound by upper and lower limits based on original RPP values (see Equation

5). Fall speeds and standard deviations of RPP were kept constant to the REVEALS approach.

$$original\ RPP \times 0.25 < new\ RPP < original\ RPP \times 4 \tag{5}$$

The RPP values were optimized for the ten most common taxa in the REVEALS reconstruction for all sites on a continent, forest cover reconstructed, and the residual sum of squares (RSS) with remote sensing forest cover calculated. The results

were validated using a spatial leave-one-out (SLOO) cross-validation. In this cross-validation one site and all sites within a predefined radius (exclusion buffer) were excluded from the optimization to account for spatial autocorrelation. The optimized RPP values were then applied to the forest cover reconstruction of the site left-out and the absolute error with remote sensing forest cover recorded. This was repeated with 20 sites to estimate the spread of MAE. The exclusion buffer around the validation site was set to 200 km. Due to computational limitations (roughly 3 hours for one continental SLOO fold using 20 threads with

1.2 GHz CPU each), the number of sites used per continental optimization during the cross-validation was limited to 100, leading to a rather conservative estimate of the true error.

## 3 Data summary

### 3.1 Pollen Source Areas

Using REVEALS and original RPP values, radii of relevant pollen source areas were calculated for all sites (see Fig. 4). The

155 relevant pollen source areas indicate in which area 80% of the deposited pollen originated from (see Section 2.2.2) and yield an understanding of which area the pollen record is representative of. The pollen source areas are roughly a function of basin size (see Fig. 5) and range between 68 km and 729 km. The median pollen source radius is 86 km including all basins and 138 km including only lakes.

### 3.2 Comparison of original and optimized RPP values

The calculated pollen source areas (see section 3.1) were used to extract modern remote sensing forest cover per site. Within the optimization, RPP values were adjusted for the ten most common taxa per continent to improve the fit between reconstructed

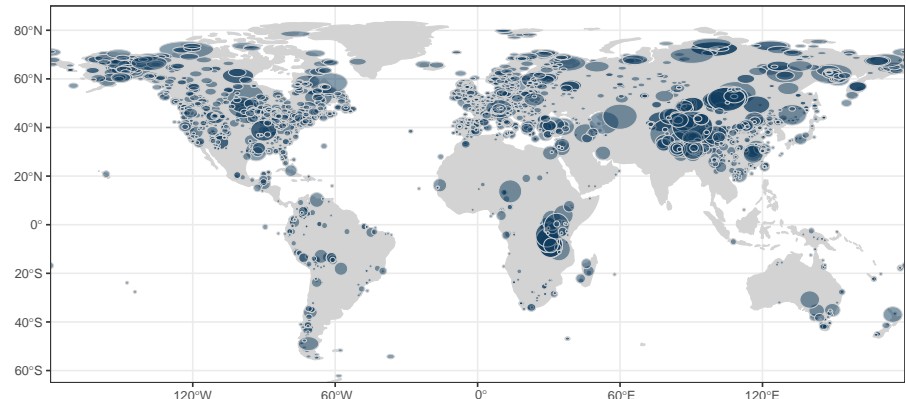

**Figure 4.** Map indicating the size of relevant pollen source areas for all records. Many small basins in Europe lead to smaller pollen source areas. Several large basins and correspondingly large pollen source areas exist in Asia.

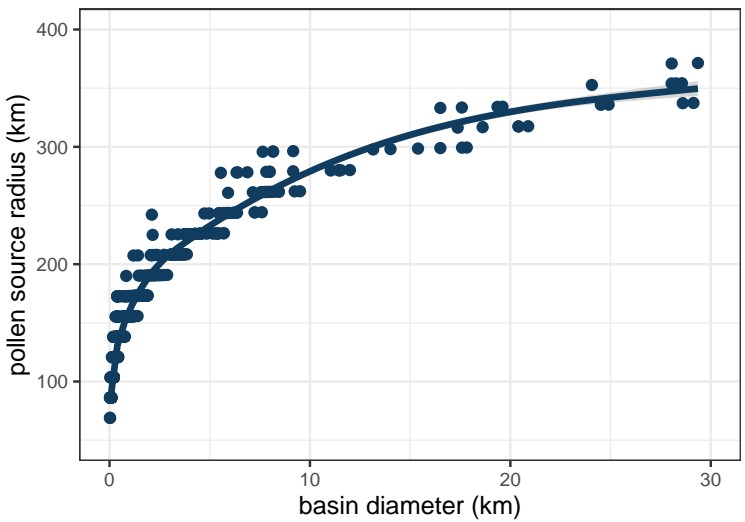

**Figure 5.** Scatter plot of basin diameter and pollen source radius of a subset of REVEALS records with original RPP values. Larger basins have larger pollen source areas with the relationship between basin diameter and pollen source radius being roughly logarithmic.

and remotely sensed modern forest cover. The RPP values are one of the main correction factors applied in REVEALS. Here we compare original and optimized RPP values for the relevant continental taxa.

The magnitude of adjustment from original to optimized RPP values differs between continents (see Fig. 6). The highest and lowest absolute change respectively occurred for *Quercus* (4.08) and Fabaceae (0.09) in Africa, for *Picea* (87.81) and *Ephedra* (0.43) in Asia, for *Pinus* (32.58) and Asteraceae (0.16) in Europe, for *Alnus* (1.79) and Amaranthaceae (in which we





included Chenopodiaceae, 0.02) in Australia and Oceania, for Amaranthaceae (63.81) and *Tsuga* (0.43) in North America, and
for Amaranthaceae (15.91) and Melastomataceae (0.74) in South America (see Appendix B). Relative change of RPP values
is mostly positive with many taxa reaching an increase of three times the original RPP value. This is the maximum RPP value
that can be reached, as the upper constraint for RPP optimization was set as 4 times the original RPP value (see Section 2.4).
In most cases RPP values for arboreal taxa are increased. This increase represents reconstructed forest cover being regulated
down as can be seen in the validations (see Fig. 10).

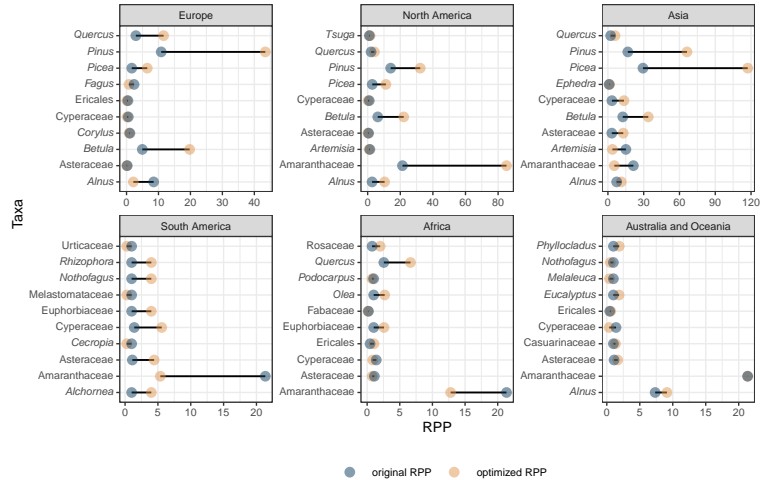

**Figure 6.** Dumbbell graph illustrating original and optimized RPP values per continent and taxon. Arboreal taxa such as Pinus, Picea,
Quercus have increases that are especially large.

### 3.3   Reconstructed compositions

Both the original and optimized RPP values were used to run REVEALS and reconstruct quantitative vegetation cover. Due to
the differences in RPP values the reconstructed compositions differ between both REVEALS runs. Here we compared these
reconstructed compositions among each other and with the original pollen composition.

Differences in composition are especially apparent for continents of the Northern Hemisphere. For example, compared to the
original pollen composition REVEALS runs with the original and the optimized RPP values both increase *Larix* cover in Asia,
Ericales cover in Europe, and decrease *Picea* cover in North America, although the version with optimized RPP values does
so more strongly (see Fig. 7). The original and the optimized version also diverge in the adjustment of some taxa. *Artemisia*
cover in Asia is reduced by the original version and increased by the optimized one. *Picea* cover stays roughly the same with
original RPP values in North America and decreases with optimized ones and while Asteraceae cover in Europe  is increased
in the REVEALS version with original RPP values, it is considerably higher in the optimized one.



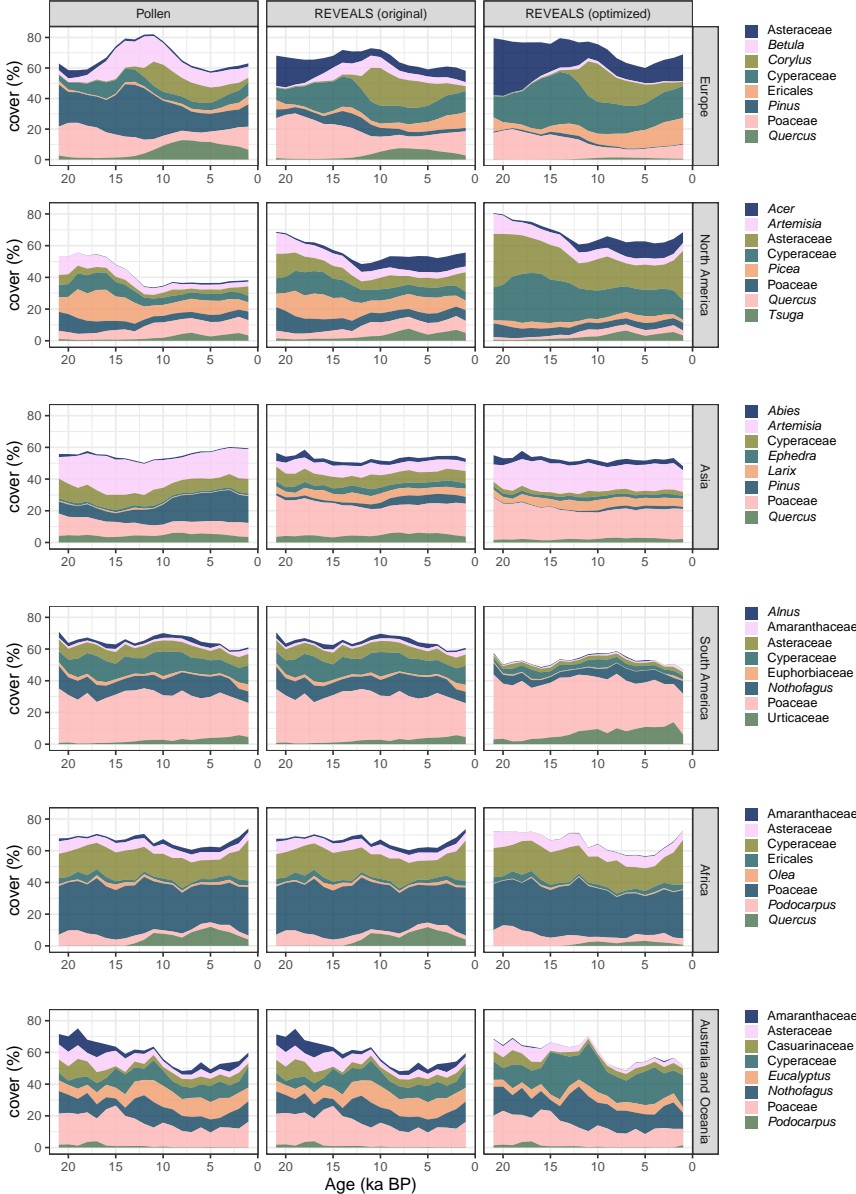

**Figure 7.** Average continental taxonomic coverages per reconstruction for the 8 most common taxa per continent. Compositional differences are more pronounced in the Northern Hemisphere due to the availability of more RPP values.

In the Southern Hemisphere the differences between reconstructions are much less pronounced (see Fig. 7). The REVEALS reconstruction with original RPP values is almost indistinguishable from the original pollen spectra and adjustments in the optimized version are also much smaller than in the Northern Hemisphere. An increase in Cyperaceae cover in Australia and
190 Oceania, decreases of Asteraceae and Cyperaceae in South America, and decreases of *Quercus* in Africa are evident in the



REVEALS run with optimized RPP values.

The difference in reconstructions between the hemispheres is most likely due to the availability of regional RPP and fall speed values. For South American taxa many RPP values are unknown and for remaining taxa average values of Northern

Hemispheric studies were used (see Fig. 3 and Appendix A). These are often close to 1 and, therefore, do not change the original compositions drastically. Improving reconstructions without more available RPP estimates for Southern Hemispheric taxa is unrealistic.

### 3.4   Reconstructed forest cover

Using the compositional data available from the original pollen data, the REVEALS run with original RPP values, and the

REVEALS run with optimized RPP values (see section 3.3), we reconstructed forest cover for all sites and samples. The temporal trend in forest cover is the same for all three reconstructions. Forest cover increases from 20 ka BP until roughly 6 ka BP and decreases again towards the present (see Fig. 8). REVEALS reconstructed forest cover is generally lower than forest cover from original pollen compositions. On average forest cover values from the REVEALS run with original/optimized RPP values are roughly 11/19% lower than values from original pollen compositions.

Forest cover is higher in the Northern Hemisphere in all time slices and reconstructions with the exception of the Eurasian

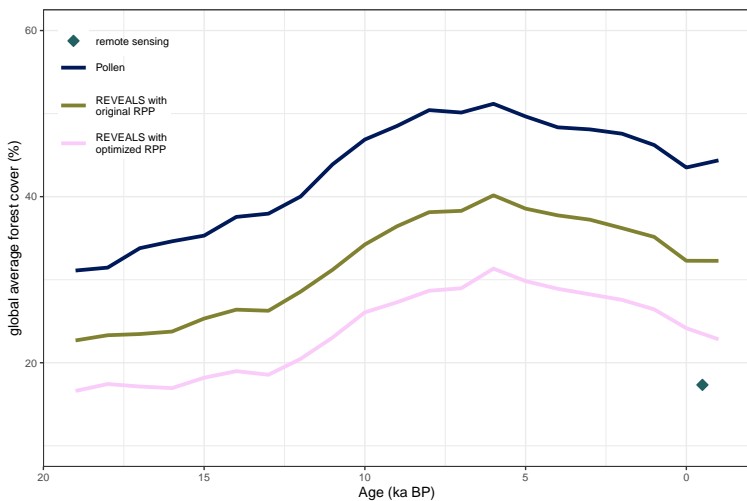

**Figure 8.** Global average forest cover from 10x10° grid cell means for raw pollen data, the REVEALS reconstruction with original RPP values, and the REVEALS reconstruction with optimized RPP values. Remotely sensed global average forest cover for the pollen record locations is indicated with the diamond. Temporal trends are the same, but absolute forest cover reduced in REVEALS reconstructions compared to the original pollen data. Forest cover from REVEALS reconstructions with optimized RPP is lowest.

Steppe, which is always characterized by a low reconstructed forest cover (see Fig. 9). Within REVEALS reconstructions, forest cover is reduced more in the Northern Hemisphere than in the Southern Hemisphere. A continuous band of highly forested





boreal forest is visible in the REVEALS reconstructions using original RPP values. The intensity of this band is reduced in the REVEALS reconstruction using optimized RPP values. However, areas in northeastern Siberia, China, and eastern North America remain strongly forested.

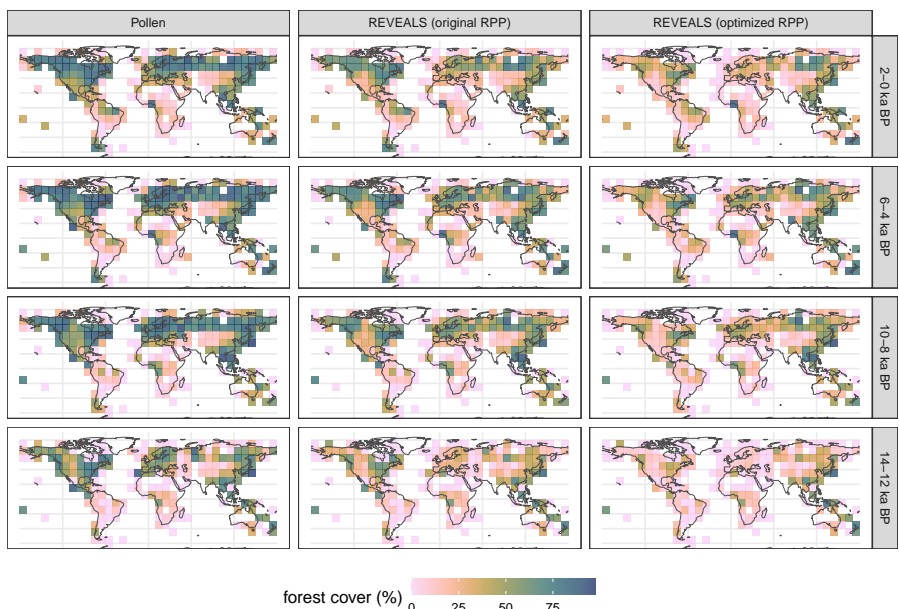

**Figure 9.** Reconstructed forest cover in 10x10° grid cells from raw pollen data, the REVEALS reconstruction with original RPP values, and the REVEALS reconstruction with optimized RPP values. Forest cover is generally higher in the Northern Hemisphere. Reductions of forest cover with the REVEALS reconstructions are higher in the Northern Hemisphere.

## 3.5 Validation

### 3.5.1 Validation with complete data sets

Remote sensing forest cover within relevant pollen source areas was used to validate the modern, reconstructed forest cover from the original pollen data and both REVEALS runs for each site. As the true error for the optimization results will be underestimated here, we also present results from the SLOO validation is Section 3.5.2. Forest cover reconstructed from original pollen data is predominantly higher than remote sensing forest cover with a global mean absolute error (MAE) of 34.39% (see Fig. 10a). As reconstructed forest cover is much lower for both REVEALS runs (see Fig.8), MAE values are reduced for both REVEALS reconstructions. Using the original RPP values yields an MAE of 20.35% of reconstructed to


remotely sensed forest cover. This is further reduced to 14.36% using the optimized RPP values (see Fig. 10a).

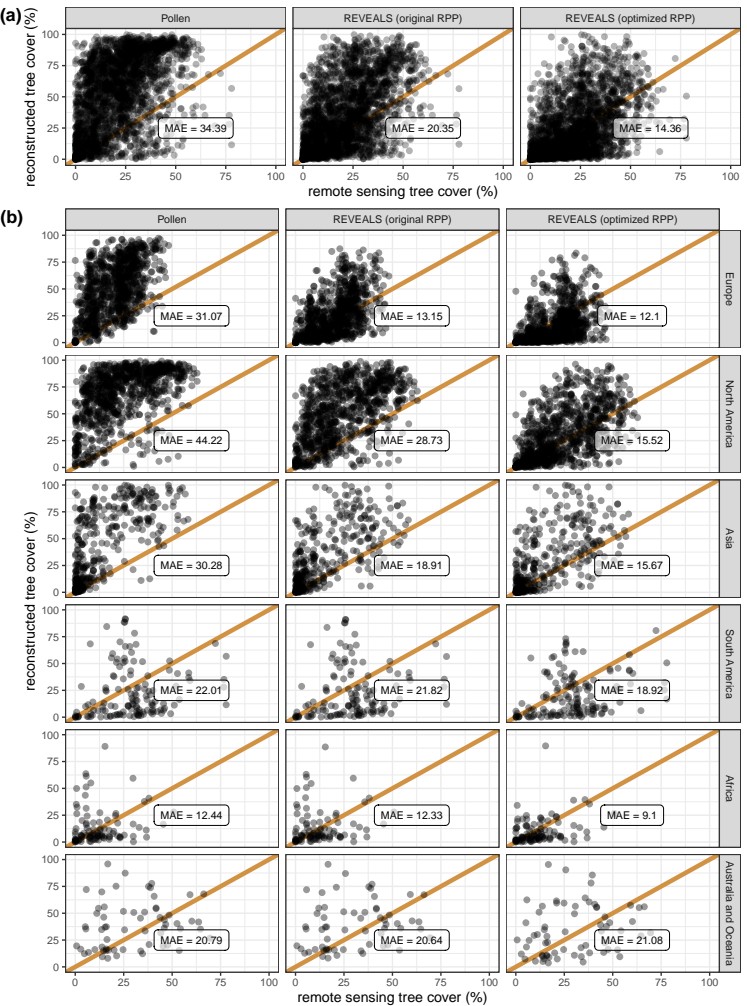

**Figure 10.** Remote sensing forest cover (LANDSAT) and reconstructed forest cover from Pollen, REVEALS with original RPP values, and REVEALS with optimized RPP values globally (a) and for all continents (b). Reconstructed forest cover from the original pollen data tends to overestimate observed (remote sensing) forest cover. This is improved with the REVEALS run using original RPP values and even more so with the REVEALS run using optimized RPP values.

Continental mean absolute errors (MAE) in forest cover from original pollen data range from 12.44% (Africa) to 44.22% forest cover (North America, see Fig. 10b). All continental MAE values are lower for the REVEALS reconstruction with original RPP values and range from 12.33% (Africa) to 28.73% (North America). The improvement is largest in Europe (58% relative to the initial MAE of the pollen-based reconstruction, see Fig. 11) and smallest in Africa (1%). Forest cover

from the REVEALS reconstruction with optimized RPP values reduces continental MAE values even further with values





ranging between 9.1% (Africa) and 21.08% forest cover (South America). MAE are generally improved more with optimized RPP values with the exception of records in Australia and Oceania. The largest improvement (relative to the pollen-based forest cover MAE) was achieved in North America (65%) but reconstructions in Europe (61%) and Asia (48%) also reduced the original MAE by more than or roughly half. The REVEALS run with optimized RPP values, therefore, produced the

230 reconstructed forest cover that corresponds best with remote sensing forest cover, with the exception of records from Australia and Oceania. Additionally, the reduction of forest cover MAE, and therefore the reconstruction improvement, was much larger in the continents of the Northern Hemisphere for both REVEALS runs.

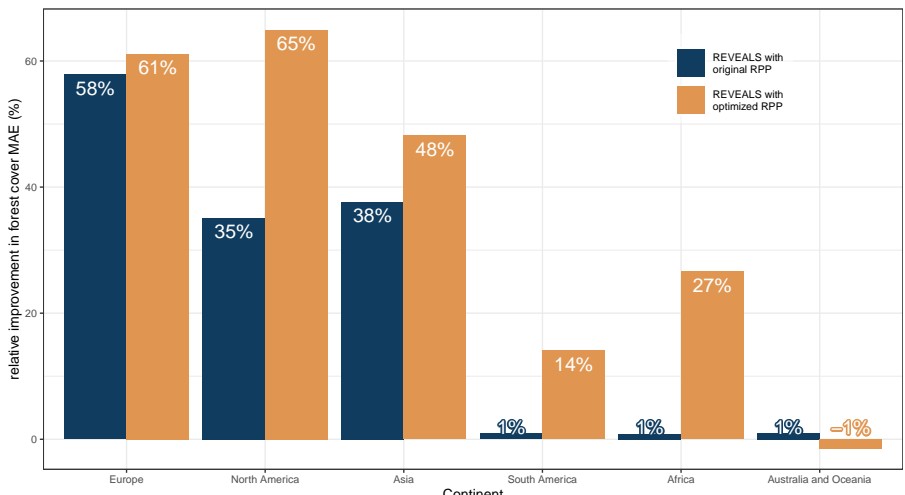

**Figure 11.** Bar graph of MAE improvement relative to the MAE of the pollen-based reconstruction per continent and REVEALS reconstruction. The absolute MAE reduction is shown in the text labels. Except for Australia and Oceania, the REVEALS reconstruction with optimized RPP values achieves higher improvements. Improvements are generally higher in the Northern Hemisphere.

Spatial patterns are present for the errors of all three forest cover reconstructions (see Fig. 12). In the Southern Hemisphere,

especially western South America, forest cover is predominantly underestimated by the reconstructions. The highest errors in reconstructed forest cover occur in continents of the Northern Hemisphere where forest cover is predominantly overestimated by the pollen-based reconstruction. In Europe the REVEALS reconstructions manage to reduce errors extensively. In eastern North America some records still tend to overestimate forest cover, even with the application of REVEALS and after optimizing RPP values. The same is the case for several records in eastern Asia.

The large difference between forest cover reconstructed from original pollen compositions and remote sensing forest cover could be due to the difference in the signal that is recorded. Remote sensing forest cover records the canopy, whereas pollen data also records the vegetation present below the tallest canopy. Several layers of trees could, therefore, increase the percentage of arboreal taxa recorded. Even though this comparison between these data sources may not be straightforward, it is still





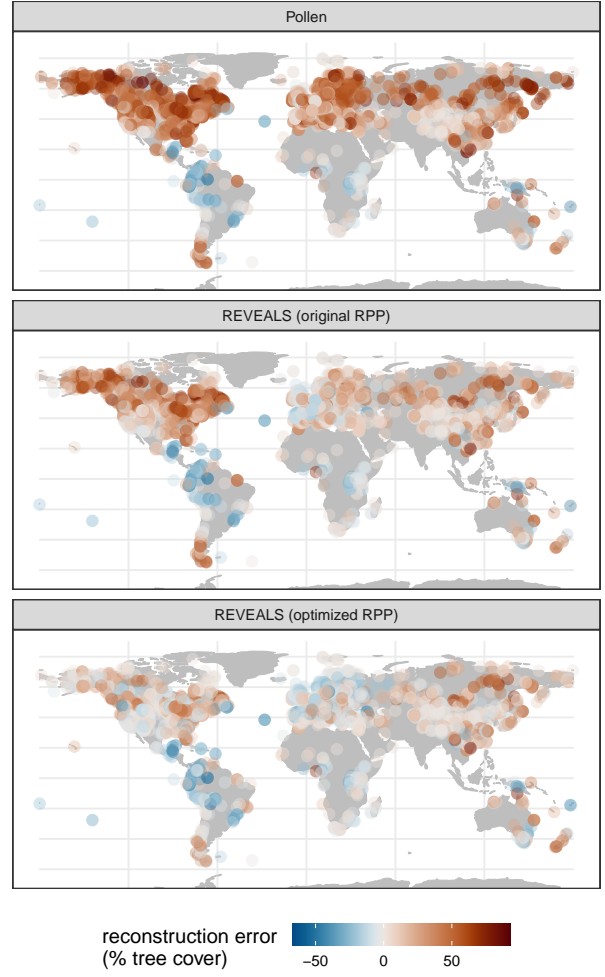

**Figure 12.** Map reconstruction error (in % forest cover) for forest cover reconstructed from Pollen, REVEALS with original RPP values and REVEALS with optimized RPP values.

necessary for this large-scale validation of reconstruction as few other vegetation data is available globally. Additionally it is more likely that the overestimation of forest cover in the initial pollen data is due to the higher production of pollen by trees than by non-arboreal taxa. This leads to an overrepresentation of arboreal taxa in the pollen record. By using REVEALS, the pollen productivity of taxa is taken into account and corrected for. The proportion of arboreal taxa is therefore strongly reduced in the vegetation compositions reconstructed using REVEALS.

     The reasons for the difference in reconstruction improvements between the hemispheres could lie both in the smaller number of records available and the lack of regional RPP estimates for continents of the Southern Hemisphere. The latter play an important role as the optimization is based on the original RPP estimates and can only determine better values if these are in





the range of the original RPP values described in Equation 5 (see Sect. 2.6). An effective optimization of RPP values may,
therefore, rely on some existing continental RPP estimates that can be refined with the optimization approach.

Optimizing more RPP could also solve the lack of regional improvements in eastern North America. This area is, amongst
others, dominated by *Acer* which is not one of the ten most common taxa in the RPP optimization in North America. Optionally,
this could also be solved by optimizing on subcontinental scales, though this requires a sufficient amount of regional records.

### 3.5.2 SLOO Validation of Optimization

A spatial leave-one-out validation was conducted by excluding a subset of available records in the optimization (see Sect.
2.4). By separating testing and training sites, the true spread of forest cover error from the optimization of RPP values can be
evaluated. This also indicates the potential error if the optimized parameters were to be applied to new records. The distribution
of absolute error from the SLOO validation is comparable to that of the reconstruction utilizing the complete optimization for
Africa, Asia, Europe and South America (see Fig. 13). In North America, the absolute error spread and media are larger in the
SLOO validation than in both REVEALS reconstructions. As errors in North America were comparably large to begin with
(see Fig. 10 and 12), this could be due to the small number of folds conducted in the SLOO validation (n = 20) as well as the
small number of records used (n = 100). The same could be the case for Australia and Oceania. Additionally, the spatial buffer
in the SLOO validation leads to even fewer records being available for optimization. This could further decrease improvements
in Australia and Oceania optimization. Overall the SLOO validation results indicate that the optimization success is relatively
stable in Africa, Asia, Europe and South America. In North America, the spatial variability leads to higher uncertainty and in
Australia and Oceania the optimization is not able to decrease absolute errors considerably.

## 4 Dataset applications and limitations

Our reconstructed quantitative vegetation cover datasets using REVEALS provide global coverage of taxonomic compositions
as well as forest cover and extend to 50 ka BP and beyond. The reconstructions made use of taxon-specific parameters and
were, thus, able to correct some of the compositional biases present in pollen compositions. Notably, the error in modern recon-
structed forest cover was reduced compared to pollen-based reconstructions on all continents which shows that improvements
in forest cover reconstructions from both REVEALS applications are considerable.

Reconstruction results are also similar to available large-scale pollen-based vegetation reconstructions. Increases in forest
cover in northern and eastern Asia up until the Holocene thermal maximum as seen in our results are consistent with recon-
structions by Cao et al. (2019) and Tian et al. (2016). The reconstructed spatial patterns of forest cover in China with low forest
cover in the North China plain and the Tibetan Plateau and a higher forest cover along the east coast and the south agree with
previous reconstructions as well (Li et al., 2023, 2022b, 2024a). Results for European forest cover also roughly correspond
with previous REVEALS applications and show an increase of forest cover after the last glacial maximum until roughly 4 ka



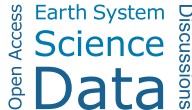

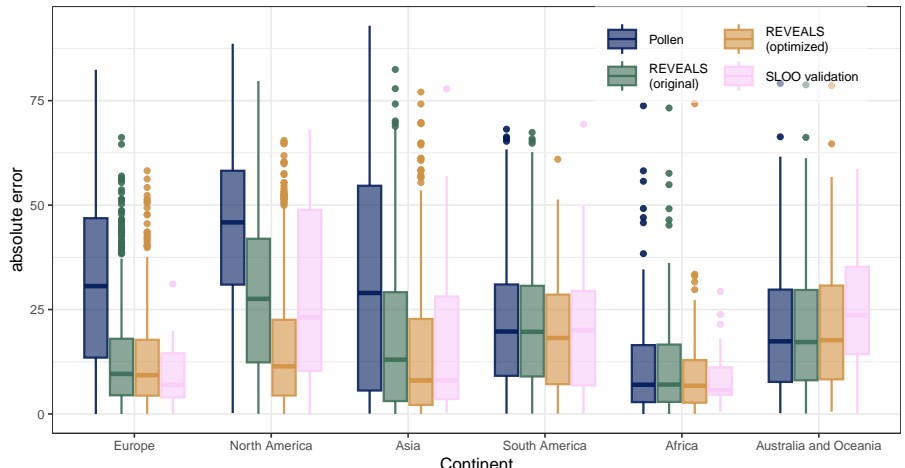

**Figure 13.** Boxplot of absolute errors from continental SLOO validations (20 folds) and from validations with complete Pollen, REVEALS (original RPP) and REVEALS (optimized RPP) data sets. The SLOO validation shows how reliable the optimized parameters are when testing sites were not included in the optimization. Variance and averages of absolute errors are comparable to the entire optimization dataset for Africa, Europe, Asia and South America. Errors are larger in Australia and Oceania and North America.

BP (Githumbi et al., 2021; Fyfe et al., 2015; Serge et al., 2023). The gridded reconstruction by Serge et al. (2023) was even validated with modern remote sensing forest cover and showed a good fit.

The REVEALS forest cover reconstructions presented here offer valuable insight into past vegetation changes. The global
dataset provides an opportunity to explore past vegetation dynamics, gaining a deeper understanding of responses, trajectories, and potential feedback mechanisms. Given the increasing discussions surrounding the possibility of tipping events in vegetation cover (Armstrong McKay et al., 2022; Lenton and Williams, 2013), this could be of considerable use. Additionally, this dataset can address unanswered questions about Holocene vegetation dynamics, including the deglacial forest conundrum (Dallmeyer et al., 2022). It also serves as a valuable tool for validating models with coupled climate and vegetation, relying
on extensive time series and vegetation data for accurate predictions (Dallmeyer et al., 2023). New insights gained from these applications could enhance our ability to predict future changes.

However, the reconstructions are associated with some of the limitations of sedimentary pollen data. This includes age uncertainty, temporal mixing, and irregular spatial and temporal resolution of records. Age uncertainty is already treated as best
as possible through consistent age modeling of the pollen dataset (Li et al., 2022a, 2021). Nevertheless, in general, replicating sediment and peat cores could provide more accurate estimates.





Moreover, there is uncertainty surrounding the success of the compositional reconstructions. As global compositional vegetation data is not readily available, using remote sensing forest cover poses as the best option for validation. Even with an accurate forest cover reconstruction, uncertainties persist regarding the abundance of individual taxa due to the aggregated nature of the forest cover measure. To address this, global syntheses of forest and other plant inventories or compositional remote sensing products could offer better validation. The optimized RPP set can produce very unrealistic compositions, for example regarding Asteraceae in Europe. The optimization was conducted purely statistically and limited ecological information was provided as input. The use of original RPP values, originating from physical studies, is, therefore, the more conservative approach for compositional reconstructions and the optimized data set should be used with caution for compositional applications. Although, many missing RPP and fall speed values, especially for taxa in the Southern Hemisphere, result in uncertainties in the original REVEALS reconstruction as well. A higher number of RPP estimates could help increase not only the confidence in compositional reconstructions, but also the optimization success in continents of the Southern Hemisphere, where the small amount of information led to lower improvements in forest cover reconstruction.

Another challenge lies in validating the results with past vegetation data. It is uncertain whether RPP values have remained stable over time, and historical compositional data are not only scarce but likely too recent to test this assumption (Baker et al., 2016). Vegetational compositions from sedimentary ancient DNA could provide a solution. Local aDNA vegetation signals could be averaged across multiple records within a pollen source area to generate a comparable reconstructed vegetation composition using a different proxy and to compare to pollen-based results (Niemeyer et al., 2017).

## 5 Conclusions

We present data sets of reconstructed compositional vegetation and forest cover from a globally distributed sedimentary pollen data set using the REVEALS model. We used published (original), continental RPP values for one reconstruction, while in a second reconstruction, we optimized continental RPP values for common taxa by incorporating remote sensing forest cover data. This approach allowed us to address some of the inherent biases in pollen compositions and suggests a method for enhancing taxon-specific RPP estimates. Considerable improvement in the reconstruction of forest cover is especially achieved in the continents of the Northern Hemisphere. Even though improvements of reconstructions in the Southern Hemisphere were largely possible as well, the collection of more regional RPP values is indispensable for better reconstructions.

Accurate data on past vegetation is invaluable for the validation of coupled climate-vegetation models and the testing of hypotheses on feedback effects and vegetation dynamics. This knowledge is essential for modeling and predicting vegetation trajectories under anthropogenic climate change.



## 6 Code and data availability

The produced datasets are freely available from PANGAEA (https://doi.pangaea.de/10.1594/PANGAEA.961699, https://doi.pangaea.de/10.1594/PANGAEA.961588, Herzschuh et al. 2023c; Schild et al. 2023).

Input data from LegacyPollen 2.0 is available on PANGAEA as well (https://doi.pangaea.de/10.1594/PANGAEA.965907, Li et al. 2024b).

The code used to produce the datasets is freely available from Zenodo (https://doi.org/10.5281/zenodo.10191859, Schild and Ewald 2023).



## Appendix A: Original RPP and fall speed values per continent

| Taxon | Continent | RPP | RPP SD | Fallspeed |
|---|---|---|---|---|
| Acer | Asia | 0.23 | 0.04255715 | 0.056 |
| Acardiaceae | Asia | 0.45 | 0.07 | 0.027 |
| Salix | Asia | 0.5366667 | 0.02995367 | 0.0218125 |
| Rosaceae | Asia | 0.53 | 0.04924429 | 0.0165 |
| Tilia | Asia | 0.4 | 0.1 | 0.02966667 |
| Moraceaea | Asia | 1.1 | 0.55 | 0.016 |
| Cupressaceae | Asia | 1.11 | 0.09 | 0.01 |
| Larix | Asia | 1.6033333 | 0.20374276 | 0.1194 |
| Rubiaceae | Asia | 1.23 | 0.36 | 0.019 |
| Corylus | Asia | 3.17 | 0.2 | 0.012 |
| Populus | Asia | 1.5866667 | 0.5363353 | 0.02566667 |
| Ulmus | Asia | 2.24 | 0.46179 | 0.02433333 |
| Fagus | Asia | 2.35 | 0.10692677 | 0.056 |
| Fraxinus | Asia | 1.05 | 0.17755281 | 0.0195 |
| Quercus | Asia | 2.284 | 0.07116179 | 0.02125 |
| Juglans | Asia | 2.8033333 | 0.11259564 | 0.0315 |
| Carpinus | Asia | 3.0933333 | 0.28446949 | 0.0415 |
| Castanea | Asia | 5.87 | 0.24505102 | 0.014 |
| Picea | Asia | 29.4 | 0.87 | 0.0819 |
| Abies | Asia | 6.875 | 1.44191713 | 0.12 |
| Betula | Asia | 12.45 | 0.1459452 | 0.0164 |
| Alnus | Asia | 7.334 | 0.17397803 | 0.021 |
| Pinus | Asia | 16.684 | 0.50916009 | 0.032425 |
| Juniperus | Asia | 14.305 | 1.00124922 | 0.016 |
| Thymelaceae | Asia | 33.05 | 3.78 | 0.009 |
| wild.herbs | Asia | 0.07 | 0.07 | 0.03425 |
| Equisetum | Asia | 0.09 | 0.02 | 0.021 |
| Convolvulaceae | Asia | 0.18 | 0.03 | 0.043 |
| Fabaceae | Asia | 0.2033333 | 0.05259911 | 0.0195 |
| Orobanchaceae | Asia | 0.33 | 0.04 | 0.038 |
| Ericales | Asia | 0.4475 | 0.01328768 | 0.03165 |



| Taxon | Continent | RPP | RPP SD | Fallspeed |
|---|---|---|---|---|
| Brassicaceae | Asia | 0.89 | 0.18 | 0.02 |
| Poaceae | Asia | 1 | 0.03166667 | 0.0211625 |
| Lamiaceae | Asia | 1.235 | 0.18668155 | 0.015 |
| Asteraceae | Asia | 3.2725 | 0.18848077 | 0.02911667 |
| Sambucus nigra-type | Asia | 1.3 | 0.12 | 0.013 |
| Cyperaceae | Asia | 3.3666667 | 0.12712243 | 0.02853333 |
| Rumex | Asia | 1.462 | 0.07139076 | 0.0148 |
| Liliaceae | Asia | 1.49 | 0.11 | 0.0135 |
| Amaryllidaceae | Asia | 1.64 | 0.09 | 0.0125 |
| Corceae | Asia | 1.72 | 0.14 | 0.044 |
| Apiaceae | Asia | 2.1266667 | 0.41013548 | 0.042 |
| Campanulaceae | Asia | 2.29 | 0.14 | 0.022 |
| Cerealia | Asia | 2.3625 | 0.42228545 | 0.069 |
| Ranunculaceae | Asia | 7.86 | 2.65 | 0.007 |
| Platagiceae | Asia | 2.8722222 | 0.10746231 | 0.0255 |
| Caryophyllaceae | Asia | 4.075 | 0.09899495 | 0.02573333 |
| Thalictrum | Asia | 4.65 | 0.3 | 0.013 |
| Chenopodiaceae | Asia | 5.5566667 | 0.6647413 | 0.01418333 |
| Urtica | Asia | 10.52 | 0.31 | 0.007 |
| Artemisia | Asia | 15.065 | 0.38084336 | 0.01016667 |
| Elaeagnaceae | Asia | 13.64 | 0.68622154 | 0.0124 |
| Humulus | Asia | 16.43 | 1 | 0.01 |
| Amaranthaceae | Asia | 21.35 | 2.34 | 0.0104 |
| Sanguisorba | Asia | 24.07 | 3.5 | 0.012 |
| Acer | Europe | 0.23 | 0.04255715 | 0.056 |
| Acardiaceae | Europe | 0.45 | 0.07 | 0.027 |
| Salix | Europe | 0.39 | 0.05840472 | 0.028125 |
| Rosaceae | Europe | 0.9725 | 0.10908712 | 0.012 |
| Tilia | Europe | 0.93 | 0.08736367 | 0.032 |
| Moraceaea | Europe | 1.1 | 0.55 | 0.016 |
| Cupressaceae | Europe | 1.11 | 0.09 | 0.01 |
| Larix | Europe | 0.16 | 0.05 | 0.126 |
| Rubiaceae | Europe | 1.56 | 0.11789826 | 0.019 |



| Taxon | Continent | RPP | RPP SD | Fallspeed |
|---|---|---|---|---|
| Corylus | Europe | 1.0533333 | 0.02947964 | 0.025 |
| Populus | Europe | 3.42 | 1.6 | 0.025 |
| Ulmus | Europe | 2.24 | 0.46179 | 0.032 |
| Fagus | Europe | 2.35 | 0.10692677 | 0.056 |
| Fraxinus | Europe | 2.972 | 0.25196031 | 0.022 |
| Quercus | Europe | 2.924 | 0.09826495 | 0.035 |
| Juglans | Europe | 2.8033333 | 0.11259564 | 0.0315 |
| Carpinus | Europe | 3.0933333 | 0.28446949 | 0.0415 |
| Castanea | Europe | 5.87 | 0.24505102 | 0.014 |
| Picea | Europe | 1.645 | 0.15323593 | 0.056 |
| Abies | Europe | 6.875 | 1.44191713 | 0.12 |
| Betula | Europe | 4.94 | 0.44296664 | 0.024 |
| Alnus | Europe | 8.4925 | 0.21539337 | 0.021 |
| Pinus | Europe | 10.86 | 0.79845945 | 0.036 |
| Juniperus | Europe | 7.94 | 1.28 | 0.016 |
| Thymelaceae | Europe | 33.05 | 3.78 | 0.009 |
| wild.herbs | Europe | 0.07 | 0.07 | 0.03425 |
| Equisetum | Europe | 0.09 | 0.02 | 0.021 |
| Convolvulaceae | Europe | 0.18 | 0.03 | 0.043 |
| Fabaceae | Europe | 0.4 | 0.07 | 0.021 |
| Orobanchaceae | Europe | 0.33 | 0.04 | 0.038 |
| Ericales | Europe | 0.4357143 | 0.01518592 | 0.0300625 |
| Brassicaceae | Europe | 0.07 | 0.04 | 0.022 |
| Poaceae | Europe | 1 | 0.01231474 | 0.035 |
| Lamiaceae | Europe | 1.0633333 | 0.12727922 | 0.019 |
| Asteraceae | Europe | 0.21875 | 0.01777287 | 0.032 |
| Sambucus nigra-type | Europe | 1.3 | 0.12 | 0.013 |
| Cyperaceae | Europe | 0.555 | 0.01892969 | 0.035 |
| Rumex | Europe | 0.5766667 | 0.03076073 | 0.018 |
| Liliaceae | Europe | 1.49 | 0.11 | 0.0135 |
| Amaryllidaceae | Europe | 1.64 | 0.09 | 0.0125 |
| Corceae | Europe | 1.72 | 0.14 | 0.044 |
| Apiaceae | Europe | 2.1266667 | 0.41013548 | 0.042 |

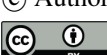



| Taxon | Continent | RPP | RPP SD | Fallspeed |
|---|---|---|---|---|
| Campanulaceae | Europe | 2.29 | 0.14 | 0.022 |
| Cerealia | Europe | 2.3625 | 0.42228545 | 0.069 |
| Ranunculaceae | Europe | 0.9933333 | 0.12064641 | 0.014 |
| Platagiceae | Europe | 2.48625 | 0.11451665 | 0.02766667 |
| Caryophyllaceae | Europe | 2.9166667 | 0.06806859 | 0.03164 |
| Thalictrum | Europe | 4.65 | 0.3 | 0.0125 |
| Chenopodiaceae | Europe | 4.28 | 0.27 | 0.019 |
| Urtica | Europe | 10.52 | 0.31 | 0.007 |
| Artemisia | Europe | 4.33 | 1.59198775 | 0.014 |
| Elaeagnaceae | Europe | 13.64 | 0.68622154 | 0.0124 |
| Humulus | Europe | 16.43 | 1 | 0.01 |
| Amaranthaceae | Europe | 21.35 | 2.34 | 0.0104 |
| Sanguisorba | Europe | 24.07 | 3.5 | 0.012 |
| Acer | North America | 0.23 | 0.04255715 | 0.056 |
| Acardiaceae | North America | 0.45 | 0.07 | 0.027 |
| Salix | North America | 0.6833333 | 0.01333333 | 0.0155 |
| Rosaceae | North America | 0.35 | 0.03 | 0.0145 |
| Tilia | North America | 0.7975 | 0.0701301 | 0.03025 |
| Moraceaea | North America | 1.1 | 0.55 | 0.016 |
| Cupressaceae | North America | 1.11 | 0.09 | 0.01 |
| Larix | North America | 1.2425 | 0.15331748 | 0.126 |
| Rubiaceae | North America | 1.4775 | 0.12616953 | 0.019 |
| Corylus | North America | 1.5825 | 0.05467028 | 0.0185 |
| Populus | North America | 0.67 | 0.085 | 0.026 |
| Ulmus | North America | 2.24 | 0.46179 | 0.02625 |
| Fagus | North America | 2.35 | 0.10692677 | 0.056 |
| Fraxinus | North America | 2.4228571 | 0.18698467 | 0.02033333 |
| Quercus | North America | 2.08 | 0.43 | 0.035 |
| Juglans | North America | 2.8033333 | 0.11259564 | 0.0315 |
| Carpinus | North America | 3.0933333 | 0.28446949 | 0.0415 |
| Castanea | North America | 5.87 | 0.24505102 | 0.014 |
| Picea | North America | 2.8 | 0.1773728 | 0.056 |
| Abies | North America | 6.875 | 1.44191713 | 0.12 |



| Taxon | Continent | RPP | RPP SD | Fallspeed |
|---|---|---|---|---|
| Betula | North America | 6.1875 | 0.14926905 | 0.05066667 |
| Alnus | North America | 2.7 | 0.12 | 0.021 |
| Pinus | North America | 14.0955556 | 0.45381374 | 0.03314 |
| Juniperus | North America | 20.67 | 1.54 | 0.016 |
| Thymelaceae | North America | 33.05 | 3.78 | 0.009 |
| wild.herbs | North America | 0.07 | 0.07 | 0.03425 |
| Equisetum | North America | 0.09 | 0.02 | 0.021 |
| Convolvulaceae | North America | 0.18 | 0.03 | 0.043 |
| Fabaceae | North America | 0.02 | 0.02 | 0.021 |
| Orobanchaceae | North America | 0.33 | 0.04 | 0.038 |
| Ericales | North America | 0.53 | 0.01328768 | 0.038 |
| Brassicaceae | North America | 0.48 | 0.09219544 | 0.021 |
| Poaceae | North America | 1 | 0.04828302 | 0.026 |
| Lamiaceae | North America | 0.72 | 0.08 | 0.031 |
| Asteraceae | North America | 0.5866667 | 0.13148722 | 0.02525 |
| Sambucus nigra-type | North America | 1.3 | 0.12 | 0.013 |
| Cyperaceae | North America | 0.975 | 0.025 | 0.0305 |
| Rumex | North America | 2.79 | 0.1724094 | 0.014 |
| Liliaceae | North America | 1.49 | 0.11 | 0.0135 |
| Amaryllidaceae | North America | 1.64 | 0.09 | 0.0125 |
| Corceae | North America | 1.72 | 0.14 | 0.044 |
| Apiaceae | North America | 2.1266667 | 0.41013548 | 0.042 |
| Campanulaceae | North America | 2.29 | 0.14 | 0.022 |
| Cerealia | North America | 2.3625 | 0.42228545 | 0.069 |
| Ranunculaceae | North America | 1.95 | 0.1 | 0.0145 |
| Platagiceae | North America | 5.96 | 0.31 | 0.019 |
| Caryophyllaceae | North America | 0.6 | 0.05 | 0.0405 |
| Thalictrum | North America | 4.65 | 0.3 | 0.012 |
| Chenopodiaceae | North America | 5.2375 | 0.50310467 | 0.011 |
| Urtica | North America | 10.52 | 0.31 | 0.007 |
| Artemisia | North America | 1.35 | 0.24 | 0.016 |
| Elaeagnaceae | North America | 13.64 | 0.68622154 | 0.0124 |
| Humulus | North America | 16.43 | 1 | 0.01 |



| Taxon | Continent | RPP | RPP SD | Fallspeed |
|---|---|---|---|---|
| Amaranthaceae | North America | 21.35 | 2.34 | 0.0104 |
| Sanguisorba | North America | 24.07 | 3.5 | 0.012 |
| Acer | Southern Hemisphere | 0.23 | 0.04255715 | 0.056 |
| Acardiaceae | Southern Hemisphere | 0.45 | 0.07 | 0.027 |
| Salix | Southern Hemisphere | 0.5366667 | 0.02995367 | 0.0218125 |
| Rosaceae | Southern Hemisphere | 0.7571429 | 0.06404718 | 0.01433333 |
| Tilia | Southern Hemisphere | 0.7975 | 0.0701301 | 0.03025 |
| Moraceaea | Southern Hemisphere | 1.1 | 0.55 | 0.016 |
| Cupressaceae | Southern Hemisphere | 1.11 | 0.09 | 0.01 |
| Larix | Southern Hemisphere | 1.2425 | 0.15331748 | 0.1216 |
| Rubiaceae | Southern Hemisphere | 1.4775 | 0.12616953 | 0.019 |
| Corylus | Southern Hemisphere | 1.5825 | 0.05467028 | 0.0185 |
| Populus | Southern Hemisphere | 1.5866667 | 0.5363353 | 0.02566667 |
| Ulmus | Southern Hemisphere | 2.24 | 0.46179 | 0.02625 |
| Fagus | Southern Hemisphere | 2.35 | 0.10692677 | 0.056 |
| Fraxinus | Southern Hemisphere | 2.4228571 | 0.18698467 | 0.02033333 |
| Quercus | Southern Hemisphere | 2.5563636 | 0.0675975 | 0.024 |
| Juglans | Southern Hemisphere | 2.8033333 | 0.11259564 | 0.0315 |
| Carpinus | Southern Hemisphere | 3.0933333 | 0.28446949 | 0.0415 |
| Castanea | Southern Hemisphere | 5.87 | 0.24505102 | 0.014 |
| Picea | Southern Hemisphere | 6.4633333 | 0.1773728 | 0.06463333 |
| Abies | Southern Hemisphere | 6.875 | 1.44191713 | 0.12 |
| Betula | Southern Hemisphere | 7.0569231 | 0.21223103 | 0.02781818 |
| Alnus | Southern Hemisphere | 7.334 | 0.17397803 | 0.021 |
| Pinus | Southern Hemisphere | 14.0955556 | 0.45381374 | 0.03314 |
| Juniperus | Southern Hemisphere | 14.305 | 1.00124922 | 0.016 |
| Thymelaceae | Southern Hemisphere | 33.05 | 3.78 | 0.009 |
| wild.herbs | Southern Hemisphere | 0.07 | 0.07 | 0.03425 |
| Equisetum | Southern Hemisphere | 0.09 | 0.02 | 0.021 |
| Convolvulaceae | Southern Hemisphere | 0.18 | 0.03 | 0.043 |
| Fabaceae | Southern Hemisphere | 0.206 | 0.03475629 | 0.01992857 |
| Orobanchaceae | Southern Hemisphere | 0.33 | 0.04 | 0.038 |
| Ericales | Southern Hemisphere | 0.4475 | 0.01328768 | 0.03165 |





| Taxon | Continent | RPP | RPP SD | Fallspeed |
|---|---|---|---|---|
| Brassicaceae | Southern Hemisphere | 0.48 | 0.09219544 | 0.021 |
| Poaceae | Southern Hemisphere | 1 | 0.01231474 | 0.0233 |
| Lamiaceae | Southern Hemisphere | 1.0633333 | 0.12727922 | 0.019 |
| Asteraceae | Southern Hemisphere | 1.1066667 | 0.05751197 | 0.02883571 |
| Sambucus nigra-type | Southern Hemisphere | 1.3 | 0.12 | 0.013 |
| Cyperaceae | Southern Hemisphere | 1.3981818 | 0.03645908 | 0.02968889 |
| Rumex | Southern Hemisphere | 1.462 | 0.07139076 | 0.0148 |
| Liliaceae | Southern Hemisphere | 1.49 | 0.11 | 0.0135 |
| Amaryllidaceae | Southern Hemisphere | 1.64 | 0.09 | 0.0125 |
| Corceae | Southern Hemisphere | 1.72 | 0.14 | 0.044 |
| Apiaceae | Southern Hemisphere | 2.1266667 | 0.41013548 | 0.042 |
| Campanulaceae | Southern Hemisphere | 2.29 | 0.14 | 0.022 |
| Cerealia | Southern Hemisphere | 2.3625 | 0.42228545 | 0.069 |
| Ranunculaceae | Southern Hemisphere | 2.558 | 0.53529431 | 0.0125 |
| Platagiceae | Southern Hemisphere | 2.8722222 | 0.10746231 | 0.0255 |
| Caryophyllaceae | Southern Hemisphere | 2.9166667 | 0.06806859 | 0.03164 |
| Thalictrum | Southern Hemisphere | 4.65 | 0.3 | 0.0125 |
| Chenopodiaceae | Southern Hemisphere | 5.2375 | 0.50310467 | 0.0143875 |
| Urtica | Southern Hemisphere | 10.52 | 0.31 | 0.007 |
| Artemisia | Southern Hemisphere | 11.1555556 | 0.43626926 | 0.01188889 |
| Elaeagnaceae | Southern Hemisphere | 13.64 | 0.68622154 | 0.0124 |
| Humulus | Southern Hemisphere | 16.43 | 1 | 0.01 |
| Amaranthaceae | Southern Hemisphere | 21.35 | 2.34 | 0.0104 |
| Sanguisorba | Southern Hemisphere | 24.07 | 3.5 | 0.012 |





**Appendix B: Optimized RPP values per continent**

| Taxa | optimized RPP value | original RPP value | Continent |
|---|---|---|---|
| Cyperaceae | 0.84654833 | 1.3981818 | Africa |
| Asteraceae | 0.76957547 | 1.1066667 | Africa |
| Quercus | 6.63958404 | 2.5563636 | Africa |
| Ericales | 1.04432639 | 0.4475 | Africa |
| Podocarpus | 0.75657208 | 1 | Africa |
| Amaranthaceae | 12.7898744 | 21.35 | Africa |
| Euphorbiaceae | 2.58335787 | 1 | Africa |
| Olea | 2.68441315 | 1 | Africa |
| Rosaceae | 1.99969879 | 0.7571429 | Africa |
| Fabaceae | 0.11735178 | 0.206 | Africa |
| Artemisia | 3.76625 | 15.065 | Asia |
| Pinus | 66.2779324 | 16.684 | Asia |
| Amaranthaceae | 5.34429663 | 21.35 | Asia |
| Cyperaceae | 13.4666668 | 3.3666667 | Asia |
| Betula | 33.8326975 | 12.45 | Asia |
| Quercus | 6.00064546 | 2.284 | Asia |
| Alnus | 11.1999651 | 7.334 | Asia |
| Asteraceae | 12.8740069 | 3.2725 | Asia |
| Picea | 117.210682 | 29.4 | Asia |
| Ephedra | 1.42698032 | 1 | Asia |
| Pinus | 43.44 | 10.86 | Europe |
| Cyperaceae | 0.18727252 | 0.555 | Europe |
| Betula | 19.7593317 | 4.94 | Europe |
| Quercus | 11.6005902 | 2.924 | Europe |
| Alnus | 2.12408706 | 8.4925 | Europe |
| Ericales | 0.10892858 | 0.4357143 | Europe |
| Picea | 6.48965812 | 1.645 | Europe |
| Fagus | 0.75915903 | 2.35 | Europe |
| Corylus | 0.83090779 | 1.0533333 | Europe |
| Asteraceae | 0.0546875 | 0.21875 | Europe |
| Cyperaceae | 0.34954545 | 1.3981818 | Indopacific |



| Taxa | optimized RPP value | original RPP value | Continent |
|---|---|---|---|
| Nothofagus | 0.53271905 | 1 | Indopacific |
| Eucalyptus | 1.86489233 | 1 | Indopacific |
| Asteraceae | 1.65106629 | 1.1066667 | Indopacific |
| Alnus | 9.12264565 | 7.334 | Indopacific |
| Amaranthaceae | 21.3676454 | 21.35 | Indopacific |
| Melaleuca | 0.39986185 | 1 | Indopacific |
| Casuarinaceae | 1.32091314 | 1 | Indopacific |
| Ericales | 0.59118499 | 0.4475 | Indopacific |
| Phyllocladus | 1.88815046 | 1 | Indopacific |
| Pinus | 32.245235 | 14.0955556 | North America |
| Betula | 22.1069251 | 6.1875 | North America |
| Quercus | 4.14832091 | 2.08 | North America |
| Asteraceae | 0.14668529 | 0.5866667 | North America |
| Picea | 11.1892262 | 2.8 | North America |
| Alnus | 10.3752134 | 2.7 | North America |
| Cyperaceae | 0.24375 | 0.975 | North America |
| Tsuga | 1.43191981 | 1 | North America |
| Artemisia | 0.85660575 | 1.35 | North America |
| Amaranthaceae | 85.1564704 | 21.35 | North America |
| Cyperaceae | 5.58206159 | 1.3981818 | South America |
| Nothofagus | 3.99593442 | 1 | South America |
| Asteraceae | 4.4266668 | 1.1066667 | South America |
| Urticaceae | 0.25 | 1 | South America |
| Euphorbiaceae | 3.99999539 | 1 | South America |
| Amaranthaceae | 5.36450324 | 21.35 | South America |
| Rhizophora | 3.99998911 | 1 | South America |
| Melastomataceae | 0.25682559 | 1 | South America |
| Alchornea | 4 | 1 | South America |
| Cecropia | 0.25293954 | 1 | South America |

*Author contributions.* UH conceptualized the data set production. CL curated the pollen dataset supervised by UH. CL revised age models supervised by UH. CL, PE and LS collected metadata for pollen records supervised by. PE set up, improved and tested code to run the RE-



VEALS model and run the initial Reveals reconstructions supervised by UH. LS, TL, RH, and UH developed the optimization methodology. LS wrote optimization code, curated remote sensing data and executed optimization, final reconstructions and validations. TL, RH and UH provided supervision for LS. LS prepared the original draft supervised by UH. All authors reviewed and edited the manuscript.

*Competing interests.* The authors declare that they have no conflict of interest.

*Acknowledgements.* We thank Thomas Böhmer for support with dataset curation and harmonization. The project was supported by the Bundesministerium für Bildung, Wissenschaft, Forschung und Technologie through the German Climate Modeling Initiative PALMOD (grant no. 01LP1510C to UH), the European Union (ERC, GlacialLegacy grant no. 772852 to UH), and the China Scholarship Council (grant no. 201908130165 to CL).





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
