# Peer review of "LegacyVegetation 1.0: Global reconstruction of vegetation composition and forest cover from pollen archives of the last 50 ka"

_Earth System Science Data, 2023_

## Community Comment (CC1)

[supplement omitted: unrelated document]

---

## Author Comment (AC1)

**Reply to Gaillard**

Laura Schild & Ulrike Herzschuh

**General reply**

Dear Marie-José Gaillard,

We thank you for your extensive review of our manuscript and the constructive comments. While you welcome our work towards global pollen-based vegetation reconstructions and do not criticize the general method you stated helpful comments regarding some concerns. We think that all of them are easily remediable and changes are feasible or have already been implemented by us .

Large basins provide great value to large-scale reconstructions and site-wise reconstructions will only be kept if the original basin is of sufficient size (>= 50ha). We do agree with and recognize the limited suitability of small basins for site-wise reconstructions. Our previous intention was to create a dataset that could be gridded at different resolutions desired by the user, which we did not emphasize enough in our manuscript. We believe this flexibility would improve the usefulness of the reconstructions. To still acknowledge and highlight potential downfalls here, it is pertinent to both flag unreliable, small basins in the data set and provide a detailed description of potential uses and an adjustable script for rasterization. We will implement this in our revisions. Additionally, any non-lake or peat basins will be removed from the data set as they are unfitting for reconstruction by REVEALS.
The naming of our calculated source area is indeed unfortunate and will be changed. We will also expand on its calculation and usability in the manuscript text. Additionally, new RPP studies will be added to our synthesis and the descriptor "original values" changed to "synthesis values".

We are confident in the feasibility of these adjustments, as we have already been able to implement the majority of them, and believe it will improve the clarity of the manuscript greatly. Below we respond in detail to each major issue raised and the detailed comments.

Best regards
Laura Schild and Ulrike Herzschuh

**Specific replies**

**Major Issues**

**Pollen records appropriate for the application of the REVEALS model to reconstruct REGIONAL plant cover**

Original comment

1. The REVEALS model was developed to reconstruct REGIONAL plant cover using pollen records from LARGE LAKES, alternatively multiple SMALL LAKES (Sugita 2007a, REVEALS model). Trondman et al. (2016) (VHA) tested the REVEALS model using MULTIPLE SMALL SITES (lakes and bogs) and concluded that pollen records from MULTIPLE SMALL BOGS could be used, ideally in mixture with pollen records from LARGE and/or SMALL lakes. Thus:
2. The REVEALS model is NOT appropriate to reconstruct regional plant cover using pollen records from SINGLE SMALL sites (lakes or bogs) and from LARGE BOGS (single or multiple). See Sugita (2007a, REVEALS model) for the definition of large lake, and Trondman et al. (2015) for the choice of 50 ha as a "practical" delimitation between small (< 50 ha) and large ( >50 ha) sites.
3. The REVEALS model IS appropriate to reconstruct plant cover using pollen records from SINGLE LARGE LAKES (however always better with records from SEVERAL LARGE LAKES in the same vegetation region); and it is also appropriate using pollen records from MULTIPLE SMALL LAKES (Sugita 2007a REVEALS model) and from a mixture of SMALL SITES (bogs and lakes) (Trondman et al., 2016).
4. In the LandCover6k protocol, LARGE BOGS are used, but the reconstructions are considered as not or less reliable (information provided in the publications) if they include: (1) only one large bog record, (2) several large bog records and no lake record or too few lake records relative to the number of large bog records.
5. The REVEALS model is NOT appropriate using pollen records from marine sediments or other types of sites receiving large amounts of pollen from rivers or surface run-off. The LandCover6k reconstructions have excluded marine and large deltas pollen records. Pollen records from lagunes that are sufficiently sheltered from the sea can be used.
6. **All the points made above, and the first point made below, imply that** (1) the dataset of single site REVEALS estimates of plant cover CANNOT BE USED AS SUCH as each REVEALS reconstruction from a single small site (bog or lake) and a single large bog is incorrect; (2) Only REVEALS estimates using pollen records from single LARGE LAKES or REVEALS **MEAN** ESTIMATES based on the REVEALS estimates from **MULTIPLE SITES** are correct and can therefore be used. This also implies that IF THIS DATASET IS MADE OPEN ACCESS FOR USE, **it MUST BE CLARIFIED for the user what can be done AND NOT DONE with these single sites REVEALS estimates**, i.e. (1) one CANNOT use the original single site REVEALS plant cover if the pollen record is from a SMALL SITE (lake or bog) or a LARGE BOG. (2) one **CAN** calculate **MEAN REVEALS estimates** within regions from ca. 50 km x 50 km (se Hellman et al., 2008b in VHA and Trondman et al., 2015 in GCB) up to whole regions or continents (the latter continental scale is provided in Schmid et al.'s dataset, but nothing else).

Reply

Our site-wise reconstructions using large lakes are still valid and their information can be used in gridded versions of this data set. We agree and recognize that reconstructions from small lakes and peatland sites should not be used alone as site-wise reconstructions. Our validation

will be adjusted to also use these sites aggregated in grid cells and not individually. Figures 1-3 below show different validations, some of which were , one with all records used as a site-wise reconstruction (incorrect validation, Fig 1, newly created), one where only large lakes are used for a site-wise validation (Fig 2, newly created) and one where all data is aggregated in 5x5° grid cells, which are used for validation (Fig 3, newly created). With the exception of Asia, aggregation in grid cells lowers the calculated mean absolute error even further and **highlights the reliability and validity of these aggregated reconstructions**.

Our aim is for the data set to be used flexibly, meaning that users can set their own temporal and spatial resolution for rasterization.This is why we did not prepare a set rasterization. To highlight this use case we will **provide a script to rasterize the dataset dynamically and classify grid cell reliability by record availability**. Additionally, we will expand on this in our data usability section and clarify how we intend the data set to be used reliably. Small sites and peatland will also receive an additional flag in the data set.

[Figure]

Fig 1: *(newly created)* Site-wise validation of reconstructed forest cover at all lake and peat records in Asia, Europe, and North America. Two age intervals were used for the calculation of a "modern" site average forest cover. The label shows the mean absolute error for each subset. This is the validation akin to the one used in our manuscript (with an exclusion of non-peat and non-lake sites). **The smaller modern interval changes error values only marginally. These values can be compared to the following two validations, which follow better practice, to see how the calculated MAE was potentially impacted.**

[Figure]

Fig 2: *(newly created)* Site-wise validation of reconstructed forest cover at all large lake sites (>= 50ha) in Asia, Europe, and North America. Two age intervals were used for the calculation of a "modern" site average forest cover. The label shows the mean absolute error for each subset. **This site-wise validation even shows slightly smaller errors for European sites. MAE values in Asia and North America are slightly larger. The now smaller sample size could also impact this. Error values are slightly larger in the shorter modern interval with the exception of North America. Removing small sites from the site-wise validation does not impact it significantly.**

[Figure]

**Validation of REVEALS reconstruction**
**gridded data (5°x5°)**

[Figure]

Fig 3: *(newly created)* Grid-cell wise validation of reconstructed forest cover in Asia, Europe, and North America. Two age intervals were used for the calculation of a "modern" cell average forest cover. The label shows the mean absolute error for each subset. **This gridded validation shows smaller errors compared to the original site-wise validation (Fig 1) and the corrected site-wise validation (Fig 2). Only MAE values in Asia are slightly larger. Spatial averages of our data set, therefore, do not impact the quality.**

[Figure]

**Modern forest cover and cell reliability**
**gridded REVEALS data set (5x5,°)**

[Figure]

Fig 4: Reconstructed forest cover and reliability of grid cell results for two modern intervals. The cell fill indicates the forest cover as reconstructed with REVEALS results without the openness correction applied for validation. Grid cells outlined in red include no large lake and less than two small basins (small lakes or peatlands) and are therefore considered to be less reliable. This example rasterization has a resolution of 5x5°. **A gridded version of our data set still achieves good spatial coverage with only few cells having a lower reliability. A figure showing the reliability through time could also be added to this.**

**Implementation of the REVEALS model and pollen source areas**

**Original comment**

7. This first point is also relevant for the issue discussed above: The authors  (Schild et al.) use Theuerkauf et al. (2016) "REVEALSinR" to implement the  REVEALS model as an alternative to Sugita's REVEALS programme (last  revised in 2022) or the R REVEALS program by Petr Kunes. The use of  "REVEALSinR" implies that the REVEALS model assumptions (Sugita 2007a; further discussed and explained in e.g., Githumbi et al. (2022), Li et  al. (2020; 2023) must be considered while implementing the model. For  example, the selection of appropriate sites and the number of pollen records  used for the reconstruction are essential. If using pollen records from SMALL  sites, the larger the number of sites/pollen records the better. The use of a  single small site or a single LARGE BOG will provide biased reconstructions that won't be useful for the analysis of past plant cover, neither at the regional  scale nor at the local scale. For instance, Theuerkauf et al. 2016 write: "*Like  the original REVEALS programme, 'REVEALSinR' includes a function to  address deposition in lakes (for details see ESM). Both the original REVEALS  programme and 'REVEALSinR' only consider atmospheric pollen deposition  (and lake mixing); neither model is applicable to sites that receive significant  amounts of pollen from rivers, streams or surface run-off*". Theuerkauf et al.  (2016) do not say explicitly that REVEALS can be used with pollen records  from SINGLE small sites. But, very unfortunately, they confuse the reader by introducing severe misunderstandings in their description of the REVEALS  model and its application (selection of pollen records). For instance (under  "Principles of 'REVEALSinR'": "*The REVEALS model (Sugita 2007a) is  based on the assumption that pollen deposition of a plant taxon in a large  lake or **peatland** is equal to the mean abundance of that taxon in the region, multiplied by its pollen productivity and its 'pollen dispersal-deposition  coefficient' K….etc*". Sugita 2007a calls his model REVEALS, i.e. Regional  Estimates of Vegetation Abundance from Large **Sites" BUT** it is developed  for pollen deposited in **LAKES** and tested theoretically with simulated pollen  records from **LAKES**. Further, one of the assumptions of the REVEALS  model is that **the deposition basin is NOT COVERED BY VEGETATION**.  It follows, therefore, that REVEALS is not appropriate for pollen records  from large bogs. Another unfortunate issue in Theuerkauf et al. (2016) is the  use of small sites in one of the tests of the effect of different pollen dispersal  models on the REVEALS reconstruction, **although the second experiment  uses a pollen record from a LARGE LAKE, which is correct!**, i.e., (under  Materials and Methods, in relation to the first experiment): "*We associate the  record with lakes and peatlands of different size (100–10,000 m in diameter),  using different cut-off distances for the tail of the GPM (50 km to infinity).  This cut-off sets an arbitrary limit to the maximum distance pollen may travel  (the region considered as pollen source area). The cut-off for the LSM is set  to 100 km, which is the calculated average distance at which 95 % of the  pollen has settled (cf. Fig. 1)*." The latter implies that the authors use  REVEALS for single sites from 1 to 100 ha. Moreover, they use the fact that  the "Radius of the 80 % source area of pollen" for sites of 1 ha or 100 ha are  not significantly different to argue that what makes the largest difference  between sites of different size is the pollen dispersal model. This is true for  the "pollen source area" defined as the characteristic radius for 80% or 90%  etc…. of the pollen reaching the site, but it is NOT true for the size of the area  a quantitative pollen reconstruction of plant cover represents when pollen records are from **SMALL SITES. See the LOVE model (Sugita 2007b) and  definition of RSAP (Sugita, 1994). This is a typical example of what a  published paper having got weak**

**reviews may lead to in studies by scientists that do not go back to the sources, in this case the description   of the REVEALS model by its author!..... A good scientist should know  that all what's published is not necessarily correct, especially today as  the review system is close to collapse due to a too large article production  in comparison to the number of reviewers that have the appropriate   expertise to evaluate a new study. GO BACK TO THE ORIGINAL  SOURCES!**

**Reply**

As outlined above, we agree that peatland sites as well as small lakes are unsuitable for a site-wise reconstruction, which is why we will **aggregate them in grid cells and highlight in the manuscript that this is the intended use case**. This will be supplemented by a script to create flexibel rasterized data sets.
Additionally, the 80% pollen source area is indeed less informative if calculated from small sites. We will **remove calculated values from the small lake and peatland sites** in our data set.
Of course we do not intend to recreate the RSAP as defined by Sugita in our data set creation, but the area where 80% of pollen originates from. The overlap in naming is indeed unfortunate, which is why we see no problem **changing the name of our calculated parameter** to "80% pollen source area".

**Original Comment**

8. The authors of the discussed paper (Schmid et al.) claim that they are  calculating the "RELEVANT SOURCE AREA" of each site, small and large (although it says RELATIVE pollen source area in the abstract". It is unclear  how this is calculated. Under 2.2.2 it says, *"We calculate the radius of relevant pollen source area by FINDING THE RADIUS IN WHICH THE  MEDIAN INFLUX OF ALL TAXA IS 80% OF THE TOTAL INFLUX (as  defined by the total influx in the MAXIMUM extent of REGIONAL  VEGETATION CHOSEN)"*. This seems to be the source area of pollen as defined by Theuerkauf et al (2016). **This in any case NOT the RELEVANT  SOURCE AREA OF POLLEN (RSAP)**. RSAP was defined originally by  Sugita (1994, Ecology) and can only be estimated for SMALL SITES using  the LOVE model (backwards modelling approach; Sugita 2007b The LOVE  model) or the ERV model and a forward modelling approach (Hellman et al.,  2009, R. Pal. Pal). **The RSAP is the minimum size of the area for which  the LOVE estimates of plant cover using pollen records from SMALL  SITES is valid.** The maximum size of the area cannot be calculated. The  definition of the pollen source area by Schmid et al. mentioned above seems  to correspond to the "characteristic radius" approach first described by  Prentice (1988). This method is generally used to estimate the parameter  $Zmax$ needed to apply the REVEALS model (see examples in Hellman et al.  2008b in VHA; and in Gaillard et al. 2022, see Figure below). Zmax is  defined as the maximum extent of the regional vegetation and is not estimated  in the REVEALS programme by Sugita. $Zmax$ is not the same as RSAP and  it is not either necessarily the size of the area for which a REVEALS plant  cover reconstruction (using appropriate pollen records!) is valid (or most  valid). See point 9 below.

From Supplementary Material for Gaillard, M.-J. Githumbi, E., Achoundong, G.,  Lézine, A.-M., Hély, C., Lebamba, J., Marquer, L., Mazier, F., Li, F., and Sugita,  S. (2021). "The challenge of pollen-based quantitative reconstruction of  Holocene plant cover in tropical regions: A pilot study in Cameroon." In: Runge,   J., Gosling, W., Lézine, A.-M., and Scott, L. (eds) Quaternary Vegetation  Dynamics. The African Pollen Database, pp. 183- 1518 205. CRC Press. eBook  ISBN9781003162766, Taylor and Francis Group. https://doi.org/10.1201/9781003162766-12

*"$Z_{max}$ (distance within which most pollen comes from) is a parameter needed to apply  the ERV model (see equation above). A way to estimate this distance is to calculate  the "characteristic radius" (CR) sensu Prentice (1988) for each taxon involved in the  ERV analysis and for the "basin size" (or radius) of*

*the sample site (0.5 m for soil samples, lake size for sediment samples) using the taxa FSP (e.g. Hellman et al., 2008b). We calculated CR using Prentice's bog model (GPM) and the Sutton's parameters $c_z$ (vertical diffusion coefficient, 0.12); $c_y$ (horizontal diffusion coefficient, 0.21), n (empirical coefficient, 0.25), and u (wind speed, 3 m/s). The CR of the 12 taxa used in this study (Table 2, above) for a basin size of 0.5 m (soil sample) (Figure 1) implies that 90% of three pollen taxa are coming from > 200 km (e.g. Moraceae, ca. 250 km) and 90% of nine pollen taxa are coming from < 200 km (e.g. Syzygium, ca. 290 km (max CR); Macaranga, ca. 150 km; Podocarpus, ca. 100 km; Poaceae, ca. 20 km (min CR)). ≤ 85% of all 26 taxa used in the first ERV model run come from ≤ 200 km (all results not shown here). Therefore, $Z_{max}$ was set to 200 km."*

9. Theuerkauf et al. (2016) also discuss the size of the area represented by REVEALS estimated plant cover (under Discussion): "*REVEALS output is commonly interpreted as representing the regional vegetation composition— but how large is this region? Or, where does the pollen come from? There is no simple answer because pollen arrives from nearby as well as far away, with nearby sources contributing (much) more (Janssen 1966). Prentice and Webb (1986) suggested approximating the source area as the area outside the basin from which e.g. 80 % of total pollen deposition arrives. For large lakes and peatlands with 1,000 m diameter (**MJG: e.g. large sites**!), the LSM predicts that the size of the 80 % source area is \*55 km for all taxa, whether with high or low fall speed. In contrast, the conventional GPM for neutral conditions predicts a large difference in the 80 % source area of taxa with low (\*120 km) and taxa with high fall speed (12 km; Table 1). Whereas the unrealistic GPM defies definition of a distinct source area, the realistic LSM offers a clear delineation." (……).* The latter result is perfectly logical but does not mean that it is not possible to define a pollen source area with the definition "*the area outside the basin from which e.g. 80 % of total pollen deposition arrives".* In the results presented by Theuerkauf et al. (2016) **the pollen source area is ca 55 km in diameter with the LSM and for the GPM it is maximum 120 km (i.e. distance for the low fall speed pollen, the distance for the high fall speed pollen being smaller, but for ALL the pollen types together, the max distance becomes 120 km.**

10. On the subject of "the size of the area represented by REVEALS estimated plant cover", Shinya Sugita writes in Li et al. (2022; pages 4-5): "*When REVEALS is applied using pollen records from multiple sites, one of the important assumptions is that there is no spatial gradients in vegetation composition within the multiple sites region (Sugita, 2007a). In addition, it is assumed (and computer simulations support it) that, **when the basin size is >100 ha, the site-to-site variation of pollen assemblages becomes negligible even if the spatial structure of vegetation is highly patchy (Sugita, 2007a).** Accordingly, the averaged values of the REVEALS estimates using pollen records from multiple large sites (**MJG: and multiple small sites, see Hellman et al., 2016**) **approximate the species composition of the regional vegetation reasonably well** as simulations and empirical studies have demonstrated (e.g. Hellman et al., 2008a, b). In theory and practice, however, the strict definition of the pollen source area is difficult for REVEALS application. **Sugita (2007a) defined it as the area within which most of the pollen comes from (Zmax).** Simulations and previous empirical studies (e.g. Sugita, 2007a, b; Hellman et al., 2008b; Sugita et al., 2010; Mazier et al., 2012) **have indicated that, when the radius of the source area defined varies from 50 km to 400 km, the REVEALS results of regional vegetation reconstruction do not change significantly.** The basin size is potentially important for REVEALS-based estimate of regional vegetation because differences in basin size among sites can lead to a significant site-to-site variation in the pollen assemblages. **However, as long as the multiple study sites are located within a region that satisfies the first assumption as described above (no gradients in the overall vegetation composition), the averaged REVEALS estimates effectively represent the regional vegetation composition as demonstrated in Hellman et al., 2008a.** The accuracy of the reconstructed vegetation against the observed vegetation composition was assessed for areas of 50 km × 50 km and 100 km × 100 km around each site in two regions of southern Sweden. The pollen records used are from 5 large lakes in each region, thus 10 lakes in total, that vary in size between 76 ha and 1965 ha. The results support the main conclusions and implications for the REVEALS application based on the theory and the simulations described in Sugita (2007a). Such evaluation is an essential step for credible application of the REVEALS model. Unfortunately, no other evaluation studies*

*following the strategy of Hellman et al., 2008a have been published  so far for other regions of the world.*"

11. Theuerkauf et al. also write: "*Therefore, in situations where **regional  vegetation is expected to be patchy**, approaches that do not rely on  **homogeneity** are preferable to REVEALS. For **a single site**, multiple  scenario approaches allow the detection of vegetation mosaics (Fyfe 2006; Bunting et al. 2008).*" . "Patchy" is not the same as "non homogenous" (see  e.g., Hellman et al., 2009a in Rev. Pal. Pal.) and above. The regional
vegetation can be patchy for a REVEALS application as long as the  patchiness is homogenous, see also point 10 above.

**12. All the points above imply that the authors of the discussed paper  (Schmid et al.) MUST clarify how their calculate the "pollen source area" of each site (lake or bog, large or small), what the definition of that source  area is, and what it can be used for when it is calculated  for a small site,  given that it is not the same as the RSAP and does not necessarily define  the size of the area for which the REVEALS reconstruction of plant cover is valid.**

**Reply**

As outlined above, we will **adapt our terminology** and call the parameter calculated by us "80% pollen source area". We define it as the area from which the median relative influx of all taxa is 80%. This is calculated by employing the lake deposition model in Theuerkauf et al.'s REVEALSinR. Starting from zmax the deposited pollen is calculated per taxon. This is assumed to be the total pollen each taxon deposits. This is of course not the reality as pollen can originate from even much further and fluvial inputs into lakes are inevitable as well. But this is an assumption that REVEALS makes as well. In a step-wise process the radius around the basin is increased and the deposited pollen relative to the total influx at zmax calculated for each taxon. We define our 80% pollen source radius as the radius where the median of the relative influx of all taxa reaches 80%. The aim of this calculation is mainly to give an idea of the scale of source area to users not familiar with pollen data. It emphasizes the regional character of lacustrine pollen data and showcases how lake size influences this source area. **We will include this expanded explanation in the manuscript.**

**Selection of RPP dataset and RPP values**

**Original comment**

1. The RPP dataset used is the one published by Wieczorek and Herzschuh  (2020). The RPP used in Schmid et al (this paper) are mean RPPs based on 1  to n original RPP values, **they are NOT original values**.

2. The Wieczorek and Herzschuh (2020) (WH) dataset does not include original  RPP values from the southern hemisphere and doesn't use RPPs published   since 2020 in China, Europe, and subtropical/tropical regions as well as  Australia. As far as I can see the WH dataset uses only the original values in  Commerford et al. (2013) for N America, but not the values from Calcote and  others (?).

3. Both points MUST be clarified. As it stands now it looks like a) there are no  original RPP values from the southern hemisphere/sub-tropical and tropical  regions and b) northern hemisphere RPP values are used for the southern  hemisphere and when the SH taxa do not exist in NH the RPP is put to 1.  Moreover, Tables A1 and A2 may give the impression to the reader that all  these taxa have original RPP values in all continents. **It is VERY  CONFUSING! CLARIFY. Do not call the values in Table A1 "Original  RPP", these are means of original RPPs AS SELECTED AND  CALCULATED BY WH! And provide the Tables A1 and A2 in a  different format, with the list of taxa only ONCE in the first column and ascribe the following columns to the**

different continents/regions you have defined. Indicate for each continent whether the RPP value you use is a mean of original values (1-n) from a single continent or several (for instance with an asterisk for the mean value used and indicate with e.g. a cross the continents in which those values are used although there are no original values in those continents. Also indicate what RPP value used is based on a single original RPP value! It would also be useful if the taxa within a family for which a RPP value exists are named, for instance Thymelaceae (only one value and it is for *Stellara* (China), and Orobanchaceae, only one value for *Rhinanthus* type (Europe), etc. This will make the RPP dataset much more transparent, the reader will have the direct information of whether the RPP value is robust for the continent in question or not, without having to go back to WH (2020) and the ESM in there. You could also indicate for the taxa you have put RPP=1 in case there is an original value published since WH (2020) that can be used as an alternative to 1 or your "optimized values", for instance for Alchornea, Melastomataceae, Podocarpus etc (e.g., Gaillard et al., 2021). NOTE: correct the spelling errors of the plant taxa names!

Finally, the points mentioned on the first page of this comment related to other existing continental REVEALS reconstructions and their use could be included the introduction of the paper (or the Discussion). I.e. better describe the difference between this "Global" REVEALS reconstruction and the existing continental REVEALS reconstructions.

Reply

We thank you for making us aware of this confusion. We used "original" here to describe the synthesis values and distinguish them from values that we tried to optimize. **We will change the name for these values to "WH synthesis values".** The synthesis by Wieczorek and Herzschuh does indeed take the 1996 publication by Calcote into account.

Concerning syntheses from reconstructions following Wieczorek and Herzschuh (2020), the synthesis used in Githumbi et al. (2021) was finalized in 2019, before the publication of Wieczorek and Herzschuh, and the RPP studies used overlap with the ones used in Wieczorek and Herzschuh. A set of RPP values in southern France by Mazier was not used by WH, but is cited as "unpublished" in Githumbi et al. (2021) so we were unable to acquire these values or a description of the study. The preprint of Dawson et al.'s reconstruction unfortunately only became available after submission of this manuscript. The authors detail the use of the synthesis by Wieczorek and Herzschuh and the addition of RPP values for *Ambrosia* and *Tsuga* from a previous synthesis. **We will look into the original publications for these values and include them as long as they fill the requirements** for RPP studies as stated in Wieczorek and Herzschuh. We were also made aware of some new **RPP estimates for China and Siberia, which we will include in our synthesis**. The **formatting of our RPP table will also be adjusted** to improve its usability.

Additionally, we will extend our introduction and discussion to further include existing continental scale reconstructions and compare them to our product. As we perceive a high uncertainty for reconstructions in the southern hemisphere, due to a lack of RPP values, **we decide to exclude reconstructions from these records from the data set.**

**Detailed Comments**

**Abstract**

adjust after having considered all major comments above. Clarify how the REVEALS dataset of plant-cover reconstructions for single sites of any type and size should be used! Clarify the definition of your "relative pollen source area" that I would rather call "characteristic pollen source area" or simply "pollen source area". See above.

Lines Comment

**Introduction**

26 akin ??

- We will change this wording to "not unlike".

41 "real", perhaps "actual vegetation" is better4

- We will implement this.

47-49 "By accounting for ….. REVEALS models quantitative vegetation cover in relevant pollen source areas …." WRONG! Correct

63 "Yet, only Serge et al. …use the opportunity for extensive validation…" WRONG! See Pirzamanbein et al. (2014) for Europe: Pirzamanbein, B., Lindström, J., Poska, A., Sugita, S., Trondman, A. K., Fyfe, R., Mazier, F., Nielsen, A.B., Kaplan, J.O., Bjune, A.E., Birks, H.J.B., Giesecke, T., Kangur, M., Latałowa, M., Marquer, L., Smith, B., and Gaillard, M.J. (2014). "Creating spatially continuous maps of past land cover from point 1780 estimates: A new statistical approach applied to pollen data." Ecological Complexity 20: 127–141. https://doi.org/10.1016/j.ecocom.2014.09.005

- We will add this reference here as well.

64 "No site-wise validations …. exist…." What do you mean? What about the the validations by Hellman et al., 2008a and b, and Sugita et al., 2010, and others?

- Here we refer to site-wise in the sense of a large-scale reconstructions that still aims to validate at a site-level as opposed to a gridded level, which is what we were aiming to do here for reconstructed forest cover from large basins.

**Mention somewhere in the Introduction the available syntheses of RPP without forgetting the latest RPP synthesis for Europe published in Githumbi et al. (2022a) and the new REVEALS reconstruction for northern America: Dawson, A., Williams, J. W., Gaillard, M.-J., Goring, S. J., Pirzamanbein, B., Lindstrom, J., Anderson, R. S., Brunelle, A., Foster, D., Gajewski, K., Gavin, D. G., Lacourse, T., Minckley, T. A., Oswald, W., Shuman, B., and Whitlock, C. (2024). "Holocene land cover change in North America: continental 1410 trends, regional drivers, and implications for vegetation-atmosphere feedbacks." Climate of the Past 1411 Discussion [preprint]. https://doi.org/10.5194/cp-2024-6 , in review, 2024.**

**Reply**

Even though the reconstruction by Githumbi et al. was published in 2022, the RPP synthesis was completed in 2018 prior to Wieczorek and Herzschuh (2020). The same studies were considered in Githumbi et al. and Wieczorek and Herzschuh with the exception of one unpublished study concerning RPP values in southern France by Mazier. We were not able to acquire the values from this study. Githumbi et al. provide more RPP values because a combination of taxa was not done as in Wieczorek and Herzschuh.

The preprint by Dawson et al. (2024) was unfortunately only available after our submission. We did find that the authors used RPP values as synthesized by Wieczorek and Herzschuh for Northern America. RPP values for two additional taxa were added and as stated above we will look at the original publications for these values to see if they fulfill the prerequisites to be added to the synthesis previously prepared by Wieczorek and Herzschuh.

**Methods**

Figure 1 Explain in the Figure caption what the different colours of dots mean,  I guess lakes versus bogs
- All of the points are slightly transparent to see where many records are leading to a higher density of records. We will add an explanation to the figure caption.

86-87 page 4 last lines until ca line 95 page 5
Explain the REVEALS model better and correctly OR simply  refer to Sugita (2007a).
- We feel that a concise description of the model is adequate and cite Sugita in line 85. We will add an additional reference to Githumbi et al. 2022 for an additional description of the REVEALS model.

97 "using peatland records for reconstructions is, therefore, appropriate." NOT CORRECT. You MUST clarify that only pollen records from  small bogs can be used if the mean REVEALS estimates from SEVERAL single small bogs is used, and even better, if the mean  REVEALS estimates are from a mix of SEVERAL small bogs and  small LAKES. Etc…. see major issues above
- "using peatland records for reconstructions is, therefore, appropriate when spatially averaged…"

8 (12)
Table 2 Several of these are not parameters but models, methods or function …..
- We will change the table title to "model settings".

Figure 3 Specify that the "available" RPP values are not necessarily original  values obtained in these continents, but can be values obtained in other  continents, right?
- We will clarify this in the figure caption and add additional information in the figure to highlight the fraction of continental values.

110 Modifications in REVEALSinR:
a. What are these modifications? Are they modifications compared to  REVEALSinR published in Theuerkauf et al (2016) or modifications  compared to the REVEALS program by Sugita? Or anything else?
- These are modifications compared to Theuerkauf et al.'s REVEALSinR. We will clarify this in the text.

b. You did not calculate the "relevant source area of pollen" (RSAP)  but something else that you define as "the radius in which etc…..total  nflux (…..)." RSAP is something else, see my major comments  above. Moreover the definition you provide is badly written, use  instead the wording for the definition given in Theuerkauf et al. (2016)
- We will adjust the naming as described above and edit the description of the definition to contain more detail.

117-127 This validation is problematic as it uses the REVEALS estimates for  individual sites which implies that the reconstructions using pollen  records from small sites (lakes or bogs" will be biased compared to the  REVEALS estimates obtained with pollen records from large sites. If  you keep this "validation", you MUST clarify that the REVEALS  estimates for the small sites can be strongly biased and therefore  the correlation with the modern vegetation might be less good than  if you would use the mean

REVEALS estimates from several sites within a given area size (e.g., grid cells of 1 degree, or vegetation regions, biomes, or continents).

- We will add an additional validation using spatially gridded forest cover to check for this potential bias and discuss it in the manuscript (also see Figures 1-3 above).

134 I do not understand this equation, it doesn't make sense to me. This perhaps because it is not clear what you mean with "reconstructed tree cover", and "corrected tree cover". It is clear what "unvegetated" cover is, and it is clear that you have to adjust the modern vegetation cover by using the sum of open vegetated cover as 100% or (1) (i.e. total open cover – unvegetated cover = 1). Is your "reconstructed cover" the total open cover including the unvegetated cover? The use of "reconstructed" here is confusing

- Reconstructed tree cover pertains to the tree cover reconstructed from a pollen product. In our manuscript this includes the original pollen counts, REVEALS reconstructions and optimized REVEALS reconstructions. It is created by summing the percentage of arboreal taxa. Because of the closed compositional nature of pollen records, there is no way to reconstruct unvegetated areas with pollen counts. The value we get can be defined as the percentage of forested area in the vegetated source area around the basin. The forest cover available in remote sensing products is differently defined as the percentage of forested area in the total source area around the basin, which includes unvegetated areas. In order to correct for this within the validation we make use of land cover maps and extract the percentage of unvegetated area in the source area around the basin (Equation 2). This way we are able to convert the reconstructed forest cover to a value which also represents the percentage of forested area in the total source area around the basin and enables a comparison between remote sensing forest cover and corrected, reconstructed forest cover.

General comment for Methods: I do not understand from this description what is the time resolution of the reconstructions, from the Figure 8 it looks to be 1000 years. In this case, this should also be mentioned as a difference in comparison with the continental PAGES LandCover6k that use 500 years resolution up to recent times, and 3 shorter time windows, 350, 250 and ca 100 years between 0.7 k BP up to "present" (see e.g. Trondman et al., 2015).

- The temporal resolution of the reconstruction is that of the original records. For the gridded data set users can choose their own temporal resolution. We will supplement a script to the manuscript that allows this and will expand on the use of the data set in the data usability section of the manuscript.

**Data Summary**

3.1 Pollen source area Adjust according to my major comments above

- We will change the name to "80% pollen source area" as described above.

3.2 Comparison of original and optimized values Do not use the term "original" for the RPP values you are using but rather "WH mean RPP values" or something similar. The values you are using are not "original values from specific studies unless there is only ONE value that you are using. See my major comment re Tables A1 and A2

- We will change the descriptor to "WH synthesis values".

Figure 4 Map indicating the size of relevant pollen source areas: CORRECT! It is not RSAP!

- We will change the name to "80% pollen source area" as described above.

165-169 "The highest and lowest absolute change respectively occurred for Quercus (4.08) and

Fabaceae (0.09) in Africa, etc….” What do you mean? Is it a +/- change or only + change, specify! I see that it is often a + change. I would write: "The highest respectively lowest absolute change (highest/lower) occurred for Quercus (+4.08)/Fabaceae (-0.09) in Africa. If this is not what you meant, CLARIFY!

- We will add signs to the change values to clarify the direction of change here.

175-197 The comparison presented in Figure 7 is fine as you have calculated average REVEALS-estimated cover for whole continents, which is OK even if you used pollen records from small sites. See my major comments above. My question is: did you calculate errors for the average REVEALS estimates using the errors produced by REVEALSinR from each individual pollen record?

- We did not calculate errors for this figure and only show mean values. Errors would have to be calculated using the delta method, which will be appended in our rasterization script.

199-209 Similarly, Figures 8 and 9 present average forest cover using REVEALS estimates from pollen records available within 10 degrees grid cells, which means that most grid cells are represented by REVEALS estimates using several pollen records. As these data are also made accessible, it would be useful for the user if you added a file that provides the identity code of the grid cells for which the "average" REVEALS estimate is based on the reconstruction from a single pollen record from one-several large bog(2), or 1-2 small bog(2), or 1-2 small lakes, or 1 small bog + 1 small lake. See example in Githumbi et al. (2022a).

- We will add this classification in this figure and include it in the script for rasterization to be used by the user.

3.5 validation It is not correct to validate the REVEALS model with modern vegetation data SITE BY SITE, given that a REVEALS reconstruction using a pollen record from ONE large bog or ONE small site (bog or lake) will in most cases be biased. **A proper revision of this paper should/MUST use the 10 degree grid cell reconstructions to validate these new REVEALS reconstructions (using WH RPP dataset or optimized RPP), and use the cover of modern vegetation within those same grid cells for comparison. Even the SLOO validation should be redone using 10 degree grid cells as the basis for the validation.**

- We will redo the validation using rasterized data as already prepared in figure 3 above.

251-258 The major difference between N hemispheric vegetation and sub tropical-tropical vegetation is that: in northern and temperate (mediterranean) regions a majority of the tree species are wind pollinated and produce large quantities of pollen per unit area, while pollen of herbaceous plants use to be insect pollinated or both wind and insect pollinated and produce less quantities of pollen per unit area, which implies that trees often are overrepresented by pollen compared with herbs; in (sub) tropical regions it is the inverse, many trees are insect pollinated and often produce small quantities of pollen which implies that herbs may be overrepresented by pollen compared to trees. The latter is well illustrated by Figure 10 pollen % versus remote sensed plant cover.

**In this section you MUST clarify that you have not used the RPP values that have been obtained from modern pollen-vegetation datasets in (sub) tropical regions and are available today in published articles (China, Africa, southern America) and provide**
**example references** (you do not need to do a literature search given that you do not use them). It's however important that you inform the reader that such values exist. For instance, in Gaillard et al. (2021) the obtained RPP in Cameroon for 13 taxa are compared with values obtained for these taxa in Africa and China, which already provide

a  significant number of existing values. Another useful paper is that by  Wan et al. (2020): Wan, Q., Zhang, Y., Huang, K., Sun, Q., Zhang, X.,  Gaillard, M.-J., Xu, Q., Li, F. and Zheng, Z., 2020, Evaluating  quantitative pollen representation of vegetation in the tropics:  A case  study on the Hainan Island, tropical China. Ecological Indicators, 114,  article: 106297, 10.1016/j.ecolind.2020.106297.

- We decided to exclude the southern hemisphere from this reconstruction as missing RPP and high fraction of insect pollination make our reconstruction too unreliable here.

**Dataset applications and limitations and Conclusions**

**Adjust these two sections according to the major comments explained in the first  part of this review in addressing all the issues implied by your  dataset, in particular the REVEALS estimates for single sites.**

285-286 "…with previous REVEALS applications and show an increase  ….until roughly 4 ka BP (references). **This is not correct, the  REVEALS reconstructions mentioned show an increase of forest  cover/respectively a decrease in openland cover until around 6 ka  BP.** The best reference for this is Strandberg et al. (2023) in Clim of  the Past and Figure 1 therein that is based on the REVEALS  reconstruction from Githumbi et al. (2022a, in ESSD).

- We will correct this.

293-294 The deglacial forest conundrium (or Holocene temperature  conundrium (HTC)) is also discussed in Strandberg et al. (2022, in  QSR).

- We will add this reference.

**References**

387-389 replace this reference by/ or add : Dallmeyer et al. (2024) in Clim Past  Discussion: Dawson, A., Williams, J. W., Gaillard, M.-J., Goring, S. J.,  Pirzamanbein, B., Lindstrom, J., Anderson, R. S., Brunelle, A., Foster,  D., Gajewski, K., Gavin, D. G., Lacourse, T., Minckley, T. A., Oswald,  W., Shuman, B., and Whitlock, C. (2024). "Holocene land cover  change in North America: continental trends, regional drivers, and  implications for vegetation-atmosphere feedbacks." Climate of the  Past Discussion [preprint]. https://doi.org/10.5194/cp-2024-6 , in  review, 2024

- We will reference Dawson et al.'s (2024) manuscript now as it was not available at the time of submission.

409-413 replace the Githumbi et al. 2021 in Clim Past Discussion by Githumbi  et al. (2022): Githumbi, E., Fyfe, R., Gaillard, M.-J., Trondman, A.-K.,  Mazier, F., Nielsen, A.-B., Poska, A., Sugita, S., Woodbridge, J.,  Azuara, J., Feurdean, A., Grindean, R., Lebreton, V., Marquer, L., Nebout - Combourieu, N., Stančikaitė, M., Tanţău, I., Tonkov, S.,  Shumilovskikh, L., and LandClimII data contributors (2022a).  "European pollen-based REVEALS land-cover reconstructions for the  Holocene: methodology, mapping and potentials." Earth System Science Data 14: 1581–1619. https://doi.org/10.5194/essd-14-1581- 2022

- We will correct this citation.

Do remember to also refer to Trondman et al. (2016) (see my major comment on the  application of REVEALS using pollen records from small sites):  Trondman, A.-K., Gaillard, M.-J., Sugita, S., Björkman, L., Greisman,
A., Hultberg, T., Lagerås, P., and Lindbladh, M. (2016). "Are pollen  records from small sites appropriate for REVEALS model-based quantitative reconstructions of past regional vegetation? An empirical  test in southern Sweden." Vegetation History and Archaeobotany 25:  131–151. https://doi.org/10.1007/s00334-015-0536-9

- We will add this reference.

---

## Author Comment (AC2)

**Reply to Mariani**

Laura Schild & Ulrike Herzschuh

**General reply**

Dear Michela Mariani,

We thank you for your careful review or our manuscript. We appreciate the valid points you have made and find them to be easily remediable. Regarding other matters you have mentioned, we see a good opportunity to provide clarification on how we follow common procedures and highlight the usability and validity of our dataset.

Firstly, the use of continental syntheses of RPP values is widely accepted in continental-scale reconstructions and has been applied in several previously published reconstructions in Europe, North America and Asia. The use of even hemispheric values, when continental ones are missing, maximizes the utility of available data while still producing improved reconstructions as our validations show. As we cannot achieve small-scale perfect reconstructions yet, we advocate for these broader and necessarily coarser insights into vegetation dynamics by generalizing reconstructions.

Furthermore, our REVEALS application using previously published synthesized RPP values is completely independent of remote sensing data. This means using remote sensing data allows for reliable validation. This shows us a clear improvement of forest cover reconstructions compared to raw pollen-data. While our optimization approach does use remote sensing data as training and testing data, we separate both in a spatial leave one out validation. This way we are effectively avoiding circularity while also considering potential spatial autocorrelation.

We do concur that uncertainties with Southern Hemispheric reconstructions are high and will exclude those from our evaluation along with samples prior to the 14 ka BP. Moreover, the few non-lake and non-peat records will be removed from the data set as they are unfit for reconstruction with REVEALS. Their removal has no effect on our spatial coverage and general reconstruction results. These adjustments are not only feasible without any problems but will improve the clarity and the quality of the manuscript.

We have added replies to your specific comments below.

Best
Laura Schild and Ulrike Herzschuh

**Specific replies**

**Major issues:**

**Original Comment**

**Inadequate regional calibrations**: The generalization of RPPEs across broad geographical scales (hemispheres) ignores crucial ecological and bioclimatic regional variations. This approach most likely leads to significant inaccuracies in the vegetation reconstructions, in spite of what the presumed 'validation' approach suggests (see below).

**Reply**

While we agree that a reconstruction using synthesized values will not reflect reality exactly, we still argue for the usability and informativeness of the result of this generalized approach. Continental syntheses of RPP values are standard practice in large-scale reconstructions as they allow an approximation of past vegetation dynamics on a large scale. Notable examples of previously used continental-scale syntheses in Europe include Serge et al. (2023), Trondman et al. (2015) and Pirzamanbein et al. (2014). Githumbi et al. (2022) also synthesize values for Northern and Central Europe and treat only mediterranean records differently. Reconstructions in North America (Dawson et al. 2024) and Northern Asia (Cao et al. 2019) synthesize values on large or continental scales as well.

While we recognize the variability of relative pollen productivity (RPP), we advocate for the use of even hemispheric averages when continental values are lacking. The direction of taxon-specific correction (over- or underproduction of pollen) will generally be correct and provide a vast improvement of REVEALS estimates to using pollen percentages alone while being able to make the most of the data currently available. We highlight that compositional reconstructions using this method come with uncertainties, but are confident that aggregates, such as reconstructed forest cover, are much closer to reality than previous pollen-based estimates. By employing this methodology, our overarching goal is to generate reconstructions that facilitate comparisons across the northern hemisphere while shedding light on general vegetation dynamics. This approach mirrors the methodology utilized in large-scale climate models, where local nuances are necessarily sacrificed for broader insights.

Importantly, this is underlined by our validation which uses independent remote sensing data and demonstrates notable improvements in reconstruction accuracy compared to reconstructions based on raw pollen data. We will expand on this in our reply to an issue below.

Githumbi, Esther, Ralph Fyfe, Marie-Jose Gaillard, Anna-Kari Trondman, Florence Mazier, Anne-Birgitte Nielsen, Anneli Poska, u. a. „European Pollen-Based REVEALS Land-Cover Reconstructions for the Holocene: Methodology, Mapping and Potentials".

*Earth System Science Data* 14, Nr. 4 (8. April 2022): 1581–1619.
https://doi.org/10.5194/essd-14-1581-2022.

Serge, M. A., F. Mazier, R. Fyfe, M.-J. Gaillard, T. Klein, A. Lagnoux, D. Galop, u. a.
„Testing the Effect of Relative Pollen Productivity on the REVEALS Model: A Validated
Reconstruction of Europe-Wide Holocene Vegetation". *Land* 12, Nr. 5 (Mai 2023): 986.
https://doi.org/10.3390/land12050986.

Dawson, Andria, John W. Williams, Marie-José Gaillard, Simon J. Goring, Behnaz
Pirzamanbein, Johan Lindstrom, R. Scott Anderson, u. a. „Holocene Land Cover
Change in North America: Continental Trends, Regional Drivers, and Implications for
Vegetation-Atmosphere Feedbacks". *Climate of the Past Discussions*, 20. Februar
2024, 1–52. https://doi.org/10.5194/cp-2024-6.

Trondman, A.-K., M.-J. Gaillard, F. Mazier, S. Sugita, R. Fyfe, A. B. Nielsen, C. Twiddle, u.
a. „Pollen-Based Quantitative Reconstructions of Holocene Regional Vegetation Cover
(Plant-Functional Types and Land-Cover Types) in Europe Suitable for Climate
Modelling". *Global Change Biology* 21, Nr. 2 (2015): 676–97.
https://doi.org/10.1111/gcb.12737.

Pirzamanbein, Behnaz, Johan Lindström, Anneli Poska, Shinya Sugita, Anna-Kari
Trondman, Ralph Fyfe, Florence Mazier, u. a. „Creating Spatially Continuous Maps of
Past Land Cover from Point Estimates: A New Statistical Approach Applied to Pollen
Data". *Ecological Complexity* 20 (1. Dezember 2014): 127–41.
https://doi.org/10.1016/j.ecocom.2014.09.005.

Cao, Xianyong, Fang Tian, Furong Li, Marie-José Gaillard, Natalia Rudaya, Qinghai Xu,
und Ulrike Herzschuh. „Pollen-Based Quantitative Land-Cover Reconstruction for
Northern Asia Covering the Last 40 Ka Cal BP". *Climate of the Past* 15, Nr. 4 (8.
August 2019): 1503–36. https://doi.org/10.5194/cp-15-1503-2019.

**Original comment**

**Questionable data assumptions and methodological gaps:** The use of northern hemisphere
RPPE values for taxa not natively present in the southern hemisphere, such as *Alnus* in
Australia, introduces substantial and confusing biases. Presumably, the authors have not
consulted the relevant scholars who worked within this field and the geographical areas
mentioned. Similarly, defaulting RPP to 1 for taxa without specific data oversimplifies
pollen-vegetation relationships. The paper does not adequately address the absence of data for
the Southern Hemisphere, leading to a misleading portrayal of global vegetation.

It is suggested >50% of RPPEs are missing for Australia and Oceanic pollen records. So, in this work a decision was made to run these records using the Northern Hemispheric RPPEs, despite very different bioclimatic and ecological contexts. This extrapolation of Northern Hemisphere RPPEs to southern locations missing PPEs without considering ecological or bioclimatic differences is particularly problematic. RPPEs empirically produced using ground truthing work (field surveys and surface pollen collection) were ignored, especially across the Southern Hemisphere (see some references below).

Duffin, K. I., & Bunting, M. J. (2008). Relative pollen productivity and fall speed estimates for southern African savanna taxa. Vegetation History and Archaeobotany, 17, 507-525.

Mariani, M., Connor, S. E., Theuerkauf, M., Kuneš, P., & Fletcher, M. S. (2016). Testing quantitative pollen dispersal models in animal-pollinated vegetation mosaics: An example from temperate Tasmania, Australia. Quaternary Science Reviews, 154, 214-225.

Mariani, M., Connor, S. E., Fletcher, M. S., Theuerkauf, M., Kuneš, P., Jacobsen, G., ... & Zawadzki, A. (2017). How old is the Tasmanian cultural landscape? A test of landscape openness using quantitative land‑cover reconstructions. Journal of Biogeography, 44(10), 2410-2420.

Mariani, M., Connor, S. E., Theuerkauf, M., Herbert, A., Kuneš, P., Bowman, D., ... & Briles, C. (2022). Disruption of cultural burning promotes shrub encroachment and unprecedented wildfires. Frontiers in Ecology and the Environment, 20(5), 292-300.

**Reply**

**We do concur that uncertainties with Southern Hemispheric reconstructions are high due to a lack of regional RPP values and will exclude those from our data set.**
We believe that including other observed taxa in the model and setting their RPP to 1, will still result in better estimates of aggregate values such as forest cover than the raw pollen data and therefore apply this standard value to include as much data as possible. Including missing RPP values by setting them to 1 or excluding taxa without RPP estimates leads to relatively similar coverage estimates as indicated by the figures below. Excluding any taxa tends to result in an overestimation of the remaining taxa as the total pollen count is reduced. By including all taxa we aim to account for this.

[Figure]

[Figure]

[Figure]

North_America

Dataset_ID
- 1522
- 3889
- 4139
- 4301
- 4421
- 22644
- 22711

Fig 1: *(newly created)* Comparison of reconstructed cover values for taxa in a reconstruction excluding taxa, for which no RPP estimates are available, and in a reconstruction where all taxa are included and unknown RPP set to 1. **The results are highly correlated, showing that including taxa with unknown RPP does not impact the reconstruction of the taxa for which RPP were already known.**

**Original Comment**

**Oversimplified and incorrect spatial and temporal settings:** The inclusion of incorrect basin types in the model without appropriate adjustments is very concerning. Why are marine records included for a model explicitly designed to work for large lakes of closed basins with wind dispersal as the only mechanism for pollen deposition?

The manuscripts states that 'all sites that were not classified as lakes were run with peatland settings' = can we consider the ocean a peatland? REVEALS cannot work with marine records and it definitely does not make sense to apply the 'peatland' settings for marine records with some random arbitrary basin radius (100m?). Further, using a deep temporal scope (50ka) without any consideration for massive climatic shifts (likely larger than the effect of regional RPPEs values vs regional bioclimate variations) are concerning oversights, making any pre-Holocene glacial REVEALS reconstructions unrealistic with current interglacial PPEs.

**Reply**

We agree with the unsuitability of non-lake and non-peat records and apologize for their inclusion. **We will remove them from our data set.** We realize that many of the peat sites used do not have basin sizes assigned to them. However, peatlands tend to be relatively small and

therefore similar in size, with the mean size of peatlands used in Trondman et al. (2016) being lower than 100m and the average peatland size in the data used by Githumbi et al. (2022) being 716 m (with a rather large standard deviation of 1901 m due to few unrealistically large peatlands). These differences of several hundred meters at most do not influence the reconstruction of REVEALS estimates considerably, which is why a standardization of peatland sizes is appropriate here. Please see Figure 2 below for an example peatland reconstruction using different basin diameters.

Trondman, Anna-Kari, Marie-José Gaillard, Shinya Sugita, Leif Björkman, Annica Greisman, Tove Hultberg, Per Lagerås, Matts Lindbladh, und Florence Mazier. „Are Pollen Records from Small Sites Appropriate for REVEALS Model-Based Quantitative Reconstructions of Past Regional Vegetation? An Empirical Test in Southern Sweden". *Vegetation History and Archaeobotany* 25, Nr. 2 (1. März 2016): 131–51. https://doi.org/10.1007/s00334-015-0536-9.

[Figure]

Fig 2: Example peatland site (Ageröds mosse, 13.42774W 55.93448N) with reconstructed vegetation at different set basin sizes (diameter in m). **The basin size has minimal impact on the reconstructed result when using peatlands and can therefore be standardized.**

**Original Comment**

**Dubious optimization and validation:** Optimizing PPEs to match remote sensing data risks validating the model based on its own assumptions rather than providing an unbiased estimation of past vegetation, which REVEALS is designed to do. This circular reasoning undermines the scientific integrity of the model's outputs. While an interesting concept this needs to be validated separately on a much smaller spatially and higher resolution scale before such a widespread application. This cannot really be called a 'validation'.

**Reply**

We apologize for any confusion that may have arisen here, but **there is no circularity associated with the validation of the REVEALS reconstruction** making use of published syntheses (titled "REVEALS (original RPP)" in our manuscript). The remotely sensed forest cover is independent of the REVEALS reconstruction and has previously been used to validate large-scale reconstruction by Serge et al. (2023) and Pirzamanbein et al. (2014). **Our validations show a clear improvement in forest cover reconstruction compared to pure pollen data.**

We do use remote sensing data both as input data and validation data in the optimization approach. The spatial leave one out validation shown in the manuscript allows us to evade any circularity and spatial autocorrelation here. The optimization is repeated several times and each time one site is left out of the optimization entirely to check the result on this site ("leave on out"). As the sites in the vicinity may be relatively similar, we decide to exclude those as well (spatial buffer) from the optimization, but do not check the result on them. As this is computationally very expensive, we limited our repetitions of this procedure ("folds") to 100 per continent, which will likely lead to a more conservative estimate of our error. Additionally, our focus lies more on the potential of this optimization method rather than the actual application of these "optimized" RPP values, as we describe in the section on data usability. We believe that this could be useful at potentially smaller scales and when more RPP values are available and are being optimized. As it stands it is more of a proof of concept and not the main contribution of our manuscript.

**Original comment**

The reconstructed forest cover for the past 500 years was compared to modern remote sensed cover. Why not a smaller and more recent age bin was considered? In the past 500 years many areas of the world have been colonised by Europeans and have experienced major shifts in vegetation structure, as management transferred from Indigenous to colonial regimes (e.g. the Americas and Australia). This means that forest cover over the whole 500 years bin is not comparable to modern remote sensing data. This highlights a Eurocentric view of the global vegetation patterns.

An example of validation of RPPEs using modern vegetation data (with surveys) has been done in the following papers:

Mariani, M., Connor, S. E., Fletcher, M. S., Theuerkauf, M., Kuneš, P., Jacobsen, G., ... & Zawadzki, A. (2017). How old is the Tasmanian cultural landscape? A test of landscape openness using quantitative land‑cover reconstructions. Journal of Biogeography, 44(10), 2410-2420.

Mariani, M., Connor, S. E., Theuerkauf, M., Herbert, A., Kuneš, P., Bowman, D., ... & Briles, C. (2022). Disruption of cultural burning promotes shrub encroachment and unprecedented wildfires. Frontiers in Ecology and the Environment, 20(5), 292-300.

**Reply**

Our choice of 500 year age bins was founded in the aim to include as many records as possible, since not all have samples as young as 100 years BP. We have, however, **tested a smaller age bin for the REVEALS reconstruction** using published synthesis values and found similar validation results. Mean absolute errors will sometimes be slightly larger or slightly smaller but generally similar (Fig 3-5 below) **highlighting the validity of the data set.**

[Figure]

Fig 3: (newly created) Site-wise validation of reconstructed forest cover at all lake and peat records in Asia, Europe, and North America. Two age intervals were used for the calculation of

a "modern" site average forest cover. The label shows the mean absolute error for each subset. The smaller modern interval changes error values only marginally. These values can be compared to the following two validations, which follow better practice, to see how the calculated MAE was potentially impacted.

[Figure]

Fig 4: Site-wise validation of reconstructed forest cover at all large lake sites (>= 50ha) in Asia, Europe, and North America. Two age intervals were used for the calculation of a "modern" site average forest cover. The label shows the mean absolute error for each subset. **This site-wise validation even shows slightly smaller errors for European sites. MAE values in Asia and North America are slightly larger. The now smaller sample size could also impact this. Error values are slightly larger in the shorter modern interval with the exception of North America. Removing small sites from the site-wise validation does not impact it significantly.**

[Figure]

Fig 5: Grid-cell wise validation of reconstructed forest cover in Asia, Europe, and North America. Two age intervals were used for the calculation of a "modern" cell average forest cover. The label shows the mean absolute error for each subset. **This gridded validation shows smaller errors compared to the original site-wise validation (Fig 1) and the corrected site-wise validation (Fig 2). Only MAE values in Asia are slightly larger. Spatial averages of our data set, therefore, do not impact the quality.**

---

## Author Comment (AC3)

Laura Schild and Ulrike Herzschuh

Dear John Williams and colleagues,

We thank you for your comments on our manuscript and appreciate your special attention towards open science and open data issues.
We apologize for the omission of a reference to Neotoma connected to the LegacyPollen dataset and will of course rectify this oversight. We add citations in the description of LegacyPollen2.0 and expand our acknowledgements. A table listing all included Neotoma records and their DOIs is appended as well. We especially thank you for the helpful inclusion of the *doi()* function in the neotoma2 R package and the example code on github.
Please see our revised passages below.

Best regards
Laura Schild and Ulrike Herzschuh

**Revised text at line 76:**

The pollen data synthesis LegacyPollen2.0 (Li et al., 2024b) includes 3728 temporally resolved records (time-series) distributed globally. Data were collected from individual publications and the Neotoma Paleoecology Database which includes data from the European Pollen Database, the QUAVIDA data base for Australasia, the Latin American Pollen Database, the African Pollen Database and the North American Pollen database (Flantua et al., 2015; Fyfe et al., 2009; Giesecke et al., 2014; Lézine et al., 2021; Rowe et al., 2007; Whitmore et al., 2005; Williams et al., 2018). An overview of Neotoma records included in LegacyPollen 2.0 can be found in Table S2.

Flantua, S.G.A., Hooghiemstra, H., Grimm, E.C., Behling, H., Bush, M.B., González-Arango, C., Gosling, W.D., Ledru, M.-P., Lozano-García, S., Maldonado, A., Prieto, A.R., Rull, V., Van Boxel, J.H., 2015. Updated site compilation of the Latin American Pollen Database. Rev. Palaeobot. Palynol. 223, 104–115. https://doi.org/10.1016/j.revpalbo.2015.09.008

Fyfe, R.M., de Beaulieu, J.-L., Binney, H., Bradshaw, R.H.W., Brewer, S., Le Flao, A., Finsinger, W., Gaillard, M.-J., Giesecke, T., Gil-Romera, G., Grimm, E.C., Huntley, B., Kunes, P., Kühl, N., Leydet, M., Lotter, A.F., Tarasov, P.E., Tonkov, S., 2009. The European Pollen Database: past efforts and current activities. Veg. Hist. Archaeobotany 18, 417–424. https://doi.org/10.1007/s00334-009-0215-9

Giesecke, T., Davis, B., Brewer, S., Finsinger, W., Wolters, S., Blaauw, M., de Beaulieu, J.-L., Binney, H., Fyfe, R.M., Gaillard, M.-J., Gil-Romera, G., van der Knaap, W.O., Kuneš, P., Kühl,

N., van Leeuwen, J.F.N., Leydet, M., Lotter, A.F., Ortu, E., Semmler, M., Bradshaw, R.H.W., 2014. Towards mapping the late Quaternary vegetation change of Europe. Veg. Hist. Archaeobotany 23, 75–86. https://doi.org/10.1007/s00334-012-0390-y

Lézine, A.-M., Ivory, S.J., Gosling, W.D., Scott, L., 2021. The African Pollen Database (APD) and tracing environmental change: State of the Art, in: Quaternary Vegetation Dynamics. CRC Press.

Rowe, C., Fraser, R., Harrison, S., Dodson, J., 2007. The QUAVIDA synergy: quaternary fire, vegetation and climate change in Australasia. Quat. Int. 167–168, 355–355. https://doi.org/10.1016/j.quaint.2007.04.001

Whitmore, J., Gajewski, K., Sawada, M., Williams, J.W., Shuman, B., Bartlein, P.J., Minckley, T., Viau, A.E., Webb, T., Shafer, S., Anderson, P., Brubaker, L., 2005. Modern pollen data from North America and Greenland for multi-scale paleoenvironmental applications. Quat. Sci. Rev. 24, 1828–1848. https://doi.org/10.1016/j.quascirev.2005.03.005

Williams, J.W., Grimm, E.C., Blois, J.L., Charles, D.F., Davis, E.B., Goring, S.J., Graham, R.W., Smith, A.J., Anderson, M., Arroyo-Cabrales, J., Ashworth, A.C., Betancourt, J.L., Bills, B.W., Booth, R.K., Buckland, P.I., Curry, B.B., Giesecke, T., Jackson, S.T., Latorre, C., Nichols, J., Purdum, T., Roth, R.E., Stryker, M., Takahara, H., 2018. The Neotoma Paleoecology Database, a multiproxy, international, community-curated data resource. Quat. Res. 89, 156–177. https://doi.org/10.1017/qua.2017.105

**Revised acknowledgements:**

We thank Thomas Böhmer for support with dataset curation and harmonization. The project was supported by the Bundesministerium für Bildung, Wissenschaft, Forschung und Technologie through the German Climate Modeling Initiative PALMOD (grant no. 01LP1510C to UH), the European Union (ERC, GlacialLegacy grant no. 772852 to UH), and the China Scholarship Council (grant no. 201908130165 to CL). Data were partly obtained from the Neotoma Paleoecology Database (http://www.neotomadb.org) and its constituent databases (European Pollen Database, QUAVIDA data base for Australasia, Latin American Pollen Database, African Pollen Database and the North American Pollen database). The work of data contributors, data stewards, and the Neotoma community is gratefully acknowledged.

Table S2: List of all Neotoma records included in LegacyPollen 2.0.

| Dataset_ID | doi |
|---|---|
| 4363 | 10.21233/00jj-rn55 |
| 22711 | 10.21233/00kz-c576 |
| 20060 | 10.21233/012s-c321 |
| 3923 | 10.21233/01cq-0v25 |
| 19784 | 10.21233/01fk-3b13 |
| 4551 | 10.21233/0471-wz80 |
| 4427 | 10.21233/04mx-dx20 |
| 19814 | 10.21233/04w0-7r18 |
| 3931 | 10.21233/054g-vs09 |
| 20012 | 10.21233/057s-wr75 |
| 26572 | 10.21233/05v5-d378 |
| 45223 | 10.21233/064r-zv51 |
| 48624 | 10.21233/0672-4W61 |
| 47864 | 10.21233/06YB-XK91 |
| 41514 | 10.21233/07ah-4r18 |
| 24330 | 10.21233/0875-yw50 |
| 4062 | 10.21233/08cj-ht50 |
| 46603 | 10.21233/0A8S-FT20 |
| 26518 | 10.21233/0ajk-6m86 |
| 24867 | 10.21233/0ax2-vj66 |
| 4053 | 10.21233/0bx3-kt48 |
| 16264 | 10.21233/0c9y-af32 |
| 45642 | 10.21233/0d1q-p773 |
| 25481 | 10.21233/0dqt-e594 |
| 17947 | 10.21233/0dt6-3493 |
| 40963 | 10.21233/0e66-gt09 |
| 40579 | 10.21233/0ebh-jc93 |
| 41602 | 10.21233/0ew2-sj35 |
| 24386 | 10.21233/0f9n-x142 |
| 24635 | 10.21233/0gh3-dt80 |
| 15211 | 10.21233/0hkg-af34 |
| 24376 | 10.21233/0jfa-j049 |
| 17354 | 10.21233/0jx5-7t71 |
| 20029 | 10.21233/0kgn-j437 |
| 15132 | 10.21233/0kpg-kx30 |
| 22672 | 10.21233/0kqz-j553 |
| 22820 | 10.21233/0kw2-2170 |
| 40572 | 10.21233/0m9v-mq69 |
| 4378 | 10.21233/0mn2-9w85 |

| Dataset_ID | doi |
|---|---|
| 14177 | 10.21233/0mss-km04 |
| 15967 | 10.21233/0n3x-5g88 |
| 22817 | 10.21233/0nty-tb40 |
| 16222 | 10.21233/0pre-hx18 |
| 41205 | 10.21233/0rb3-az79 |
| 15207 | 10.21233/0rm4-f986 |
| 46447 | 10.21233/0vhe-5b77 |
| 41448 | 10.21233/0vty-vb30 |
| 21812 | 10.21233/0w0p-my34 |
| 42417 | 10.21233/0wn3-0048 |
| 15732 | 10.21233/0wvv-hb57 |
| 3969 | 10.21233/0xa7-w271 |
| 25427 | 10.21233/0xht-hx92 |
| 24673 | 10.21233/0xky-3a80 |
| 19846 | 10.21233/0xmq-rz62 |
| 22049 | 10.21233/0yan-yj67 |
| 15309 | 10.21233/0ywb-hk24 |
| 16193 | 10.21233/0yz5-ad05 |
| 19937 | 10.21233/0zcx-jb34 |
| 45347 | 10.21233/0zgh-6r09 |
| 24207 | 10.21233/0zqa-9w57 |
| 46512 | 10.21233/102F-CT44 |
| 15586 | 10.21233/108x-ge80 |
| 25843 | 10.21233/10kw-5k34 |
| 19995 | 10.21233/11hj-eq70 |
| 41606 | 10.21233/11ph-mc56 |
| 24164 | 10.21233/13fe-7b21 |
| 3985 | 10.21233/140w-er94 |
| 10981 | 10.21233/149e-e689 |
| 25786 | 10.21233/14rr-c071 |
| 21577 | 10.21233/1627-0v53 |
| 41027 | 10.21233/16gz-mt48 |
| 25408 | 10.21233/17jd-z330 |
| 19323 | 10.21233/17ta-5252 |
| 24542 | 10.21233/19wm-rd42 |
| 42539 | 10.21233/1BDV-YC11 |
| 47669 | 10.21233/1DT1-N648 |
| 45103 | 10.21233/1bnj-jq13 |
| 15420 | 10.21233/1ca9-2k79 |
| 24159 | 10.21233/1cnb-m133 |
| 4183 | 10.21233/1csh-9q73 |

| | | | | |
|---|---|---|---|---|
| 4393 | 10.21233/1csm-st79 | | 22805 | 10.21233/2b23-jr62 |
| 22822 | 10.21233/1ef3-zj56 | | 4127 | 10.21233/2bma-0p82 |
| 4436 | 10.21233/1f47-fs20 | | 4011 | 10.21233/2c89-ct63 |
| 24086 | 10.21233/1fdm-x235 | | 26512 | 10.21233/2cxq-pb51 |
| 4239 | 10.21233/1g98-ct07 | | 40566 | 10.21233/2dyz-ac37 |
| 17617 | 10.21233/1gkg-2a65 | | 17717 | 10.21233/2ecb-3b52 |
| 45322 | 10.21233/1gmy-d414 | | 45186 | 10.21233/2fkg-zq79 |
| 18127 | 10.21233/1jd5-2935 | | 4256 | 10.21233/2gfj-a151 |
| 14612 | 10.21233/1ks0-te29 | | 24555 | 10.21233/2hdq-sr31 |
| 24084 | 10.21233/1m09-f532 | | 24950 | 10.21233/2jhm-6w05 |
| 22683 | 10.21233/1mc2-sy03 | | 42568 | 10.21233/2kj1-jp45 |
| 21675 | 10.21233/1pwg-tx22 | | 22669 | 10.21233/2ks0-7d19 |
| 20020 | 10.21233/1r8k-p831 | | 24863 | 10.21233/2kym-q693 |
| 21533 | 10.21233/1sgv-kf95 | | 17995 | 10.21233/2m53-f907 |
| 15359 | 10.21233/1t37-sc11 | | 17342 | 10.21233/2p2t-qn32 |
| 41127 | 10.21233/1tmp-8264 | | 41380 | 10.21233/2p95-3654 |
| 25273 | 10.21233/1v4w-a061 | | 14549 | 10.21233/2q3p-8t07 |
| 24044 | 10.21233/1v7j-4770 | | 17832 | 10.21233/2q7v-1454 |
| 26448 | 10.21233/1vvt-v868 | | 25514 | 10.21233/2rha-px75 |
| 4387 | 10.21233/1vxd-hm05 | | 24665 | 10.21233/2rhp-py68 |
| 24412 | 10.21233/1vz8-h207 | | 15487 | 10.21233/2s88-n175 |
| 14264 | 10.21233/1x1e-yt36 | | 22894 | 10.21233/2sa7-q544 |
| 4542 | 10.21233/1xas-kk11 | | 45385 | 10.21233/2tkh-z079 |
| 24076 | 10.21233/20j0-gk71 | | 4385 | 10.21233/2v5v-v203 |
| 4413 | 10.21233/20v9-7e75 | | 19913 | 10.21233/2vzb-bg61 |
| 4369 | 10.21233/20xs-m168 | | 42645 | 10.21233/2wvj-4383 |
| 41619 | 10.21233/216r-7y34 | | 41197 | 10.21233/2wxy-fm50 |
| 41657 | 10.21233/22yh-z232 | | 19264 | 10.21233/2x38-6j55 |
| 42542 | 10.21233/234z-6f61 | | 4188 | 10.21233/2xk9-gy52 |
| 4472 | 10.21233/26e5-b250 | | 14797 | 10.21233/2xrb-wn37 |
| 46290 | 10.21233/26yg-qw36 | | 3988 | 10.21233/2yt1-f536 |
| 24595 | 10.21233/271c-xj09 | | 24516 | 10.21233/2yv5-0c85 |
| 22732 | 10.21233/272d-vg02 | | 24513 | 10.21233/2yzn-v779 |
| 14639 | 10.21233/27j0-4960 | | 24633 | 10.21233/30sd-fp85 |
| 24195 | 10.21233/28fw-fa08 | | 25765 | 10.21233/30sw-xp94 |
| 4275 | 10.21233/28gf-4635 | | 14794 | 10.21233/314t-a768 |
| 45188 | 10.21233/28zd-fk74 | | 15724 | 10.21233/31ah-kk77 |
| 42421 | 10.21233/298g-zr75 | | 46451 | 10.21233/32mp-be87 |
| 15344 | 10.21233/29yf-f996 | | 40317 | 10.21233/32se-yq93 |
| 47618 | 10.21233/2DQ3-6153 | | 22316 | 10.21233/33pr-sh52 |
| 16209 | 10.21233/2ac6-fd67 | | 24532 | 10.21233/349p-7t96 |
| 4348 | 10.21233/2ahm-tx69 | | 3875 | 10.21233/34k1-e825 |
| 3977 | 10.21233/2arg-mj55 | | 21830 | 10.21233/34yd-xm12 |

| | |
|---|---|
| 21726 | 10.21233/3651-zp81 |
| 20163 | 10.21233/36tb-yy55 |
| 22656 | 10.21233/378b-6586 |
| 45731 | 10.21233/37a7-9h33 |
| 16105 | 10.21233/38ev-4529 |
| 40454 | 10.21233/39k5-ew03 |
| 47595 | 10.21233/3B46-2P68 |
| 46726 | 10.21233/3M1B-H276 |
| 46693 | 10.21233/3NKX-DK96 |
| 22681 | 10.21233/3aeh-hs14 |
| 4415 | 10.21233/3emp-8f59 |
| 40945 | 10.21233/3epn-hs17 |
| 41502 | 10.21233/3fmj-ne26 |
| 13079 | 10.21233/3g0t-1t98 |
| 24155 | 10.21233/3hey-1169 |
| 41589 | 10.21233/3hnm-z839 |
| 24286 | 10.21233/3j3b-qm75 |
| 4501 | 10.21233/3jez-g489 |
| 20065 | 10.21233/3jm4-0r55 |
| 21543 | 10.21233/3jx5-m665 |
| 22358 | 10.21233/3kyv-3k70 |
| 46492 | 10.21233/3mhs-jm74 |
| 40490 | 10.21233/3n0e-0283 |
| 24488 | 10.21233/3nc2-mk25 |
| 15292 | 10.21233/3p7k-v248 |
| 15383 | 10.21233/3pbp-d740 |
| 15201 | 10.21233/3pkd-8685 |
| 4168 | 10.21233/3ppz-sz86 |
| 21917 | 10.21233/3prs-zt23 |
| 45377 | 10.21233/3pzj-0y37 |
| 26319 | 10.21233/3qvt-5e15 |
| 17298 | 10.21233/3r9h-5q68 |
| 4293 | 10.21233/3s1a-0h86 |
| 4092 | 10.21233/3vvs-zb43 |
| 24373 | 10.21233/3y80-dr46 |
| 46464 | 10.21233/3yd7-jc96 |
| 46473 | 10.21233/3zmx-5r44 |
| 24524 | 10.21233/3zs9-w545 |
| 41247 | 10.21233/40c4-rg98 |
| 14948 | 10.21233/414e-k787 |
| 22747 | 10.21233/417v-rg61 |
| 15356 | 10.21233/41a2-9620 |
| 45196 | 10.21233/42f5-8f49 |

| | |
|---|---|
| 24118 | 10.21233/42qq-a716 |
| 17375 | 10.21233/42z3-5s40 |
| 41018 | 10.21233/43vz-0d54 |
| 15831 | 10.21233/44v8-7p56 |
| 45381 | 10.21233/45g4-ry85 |
| 24530 | 10.21233/46s2-ed45 |
| 15003 | 10.21233/48nv-1h83 |
| 47011 | 10.21233/4974-QC65 |
| 24624 | 10.21233/49j0-se42 |
| 4142 | 10.21233/49qd-q283 |
| 47004 | 10.21233/4BAE-XF23 |
| 48649 | 10.21233/4MXX-C134 |
| 46682 | 10.21233/4QB2-3C42 |
| 14270 | 10.21233/4b1p-c888 |
| 4484 | 10.21233/4b53-ep44 |
| 21798 | 10.21233/4b8r-yv06 |
| 4163 | 10.21233/4c5r-s872 |
| 4373 | 10.21233/4cn4-wj75 |
| 4411 | 10.21233/4cn6-3t90 |
| 20156 | 10.21233/4drc-k617 |
| 3975 | 10.21233/4dxg-aq64 |
| 3994 | 10.21233/4dz1-ny70 |
| 24618 | 10.21233/4fc7-8829 |
| 14537 | 10.21233/4fsm-w415 |
| 25767 | 10.21233/4h6f-tx74 |
| 4389 | 10.21233/4j49-be77 |
| 46457 | 10.21233/4jv3-em45 |
| 25007 | 10.21233/4jwe-5f69 |
| 15035 | 10.21233/4k5g-jx08 |
| 45343 | 10.21233/4k7v-pp08 |
| 24495 | 10.21233/4kdn-gp37 |
| 22628 | 10.21233/4m0f-6v70 |
| 4167 | 10.21233/4mp7-wm75 |
| 42722 | 10.21233/4n0x-dj49 |
| 25466 | 10.21233/4prv-bd15 |
| 41224 | 10.21233/4pxw-bm05 |
| 4445 | 10.21233/4r4q-0m97 |
| 4492 | 10.21233/4r7r-4258 |
| 4301 | 10.21233/4rk9-4w51 |
| 42675 | 10.21233/4t3y-1v17 |
| 41222 | 10.21233/4tx7-3a59 |
| 15819 | 10.21233/4vq0-4494 |
| 24356 | 10.21233/4vzb-6v35 |

| | | | | |
|---|---|---|---|---|
| 21700 | 10.21233/4ydc-n204 | | 15013 | 10.21233/5mqf-9723 |
| 4025 | 10.21233/4z1g-r833 | | 26563 | 10.21233/5mtf-w128 |
| 3951 | 10.21233/4z7r-k050 | | 4396 | 10.21233/5pbb-z224 |
| 22660 | 10.21233/4zaw-zb28 | | 22883 | 10.21233/5qw0-fa47 |
| 4045 | 10.21233/4zbw-2h35 | | 45648 | 10.21233/5rcm-aa91 |
| 41627 | 10.21233/4zww-1615 | | 25163 | 10.21233/5rh4-5f23 |
| 15408 | 10.21233/502z-2a36 | | 20027 | 10.21233/5rms-aj77 |
| 21816 | 10.21233/5076-5v40 | | 25837 | 10.21233/5rsn-hm26 |
| 4524 | 10.21233/5179-fn49 | | 4489 | 10.21233/5ry3-kt92 |
| 15580 | 10.21233/51kp-1e53 | | 14278 | 10.21233/5tbt-ws69 |
| 4198 | 10.21233/52fh-aq29 | | 46459 | 10.21233/5vs6-sq73 |
| 15140 | 10.21233/52qv-3g19 | | 3910 | 10.21233/5wm6-fj81 |
| 15781 | 10.21233/52yq-b228 | | 43525 | 10.21233/5wvt-cm23 |
| 24499 | 10.21233/536b-mm31 | | 24091 | 10.21233/5wwk-v219 |
| 24317 | 10.21233/538q-0k21 | | 46449 | 10.21233/5x6t-y404 |
| 41278 | 10.21233/53ja-ce75 | | 3930 | 10.21233/5xct-6y32 |
| 4403 | 10.21233/53ys-nt12 | | 16009 | 10.21233/5xje-xg37 |
| 14938 | 10.21233/54cn-0794 | | 15059 | 10.21233/5yf4-yx81 |
| 15886 | 10.21233/56k5-ng15 | | 15919 | 10.21233/5zt0-g990 |
| 4161 | 10.21233/56se-gy95 | | 21549 | 10.21233/6032-6p09 |
| 17363 | 10.21233/56yt-6d14 | | 24477 | 10.21233/60v1-q871 |
| 22730 | 10.21233/576f-8911 | | 214 | 10.21233/612s-t274 |
| 40486 | 10.21233/57g8-9549 | | 4186 | 10.21233/6144-2g26 |
| 22087 | 10.21233/58t7-6y73 | | 47580 | 10.21233/61K4-QF84 |
| 20281 | 10.21233/596h-wc50 | | 24053 | 10.21233/61k6-c182 |
| 42563 | 10.21233/59ks-w850 | | 16131 | 10.21233/620y-dx14 |
| 14957 | 10.21233/59wz-qx78 | | 15494 | 10.21233/626e-7k44 |
| 48614 | 10.21233/5P77-ZQ05 | | 15864 | 10.21233/62ad-2r14 |
| 47817 | 10.21233/5VZR-YH26 | | 26315 | 10.21233/62sj-zk58 |
| 48363 | 10.21233/5XXJ-GH64 | | 21794 | 10.21233/6344-nb24 |
| 15191 | 10.21233/5aes-5y61 | | 14951 | 10.21233/636g-9n86 |
| 45181 | 10.21233/5ayt-5k20 | | 13069 | 10.21233/638z-xp12 |
| 15768 | 10.21233/5ct3-qj23 | | 15306 | 10.21233/63x2-2k06 |
| 4352 | 10.21233/5d85-6d65 | | 22749 | 10.21233/64xb-g250 |
| 14911 | 10.21233/5drb-1282 | | 14674 | 10.21233/65ar-gd68 |
| 12023 | 10.21233/5f1j-ch61 | | 4170 | 10.21233/65fw-bj58 |
| 4386 | 10.21233/5gt0-z402 | | 4181 | 10.21233/66zq-3984 |
| 41098 | 10.21233/5h6b-ds48 | | 21551 | 10.21233/675d-as47 |
| 22709 | 10.21233/5hsv-fx24 | | 3932 | 10.21233/686e-3e44 |
| 41216 | 10.21233/5j3e-pn82 | | 41669 | 10.21233/69nc-5r65 |
| 40795 | 10.21233/5jx5-2077 | | 46995 | 10.21233/6G4D-TF71 |
| 24631 | 10.21233/5m52-nd84 | | 47506 | 10.21233/6M1T-DA61 |
| 15627 | 10.21233/5m7p-q097 | | 4409 | 10.21233/6ag1-k287 |

| | | | |
|---|---|---|---|
| 15740 | 10.21233/6am1-sj67 | 13055 | 10.21233/7436-ph05 |
| 14680 | 10.21233/6ans-w752 | 41426 | 10.21233/76ay-9f41 |
| 24871 | 10.21233/6av9-jx79 | 16109 | 10.21233/772k-7s80 |
| 4428 | 10.21233/6b8p-ke79 | 21557 | 10.21233/77j4-vn11 |
| 20515 | 10.21233/6beq-4462 | 22944 | 10.21233/77r6-x076 |
| 19909 | 10.21233/6bhj-jv09 | 4527 | 10.21233/793y-2a05 |
| 41214 | 10.21233/6ejf-rt19 | 46521 | 10.21233/7E9R-VP36 |
| 15678 | 10.21233/6enp-3753 | 46516 | 10.21233/7F5C-W552 |
| 14963 | 10.21233/6es0-qa61 | 46514 | 10.21233/7M90-YN39 |
| 15876 | 10.21233/6fad-4339 | 46698 | 10.21233/7VE7-FM07 |
| 16207 | 10.21233/6ftv-h112 | 47590 | 10.21233/7WF1-Z606 |
| 20022 | 10.21233/6hft-x335 | 4164 | 10.21233/7aj5-ec03 |
| 14682 | 10.21233/6hga-6436 | 32426 | 10.21233/7bhz-n612 |
| 15630 | 10.21233/6j0s-ag87 | 15130 | 10.21233/7bsm-hd67 |
| 24312 | 10.21233/6j41-er97 | 45092 | 10.21233/7btd-f613 |
| 15259 | 10.21233/6j8r-1g90 | 4225 | 10.21233/7d1m-t529 |
| 22835 | 10.21233/6kgm-zr91 | 41333 | 10.21233/7d1q-6n57 |
| 46280 | 10.21233/6n58-k786 | 4382 | 10.21233/7d6z-6j10 |
| 41273 | 10.21233/6n89-7w11 | 40584 | 10.21233/7dfq-sj65 |
| 24469 | 10.21233/6p3k-yh02 | 22876 | 10.21233/7dvy-1e52 |
| 41491 | 10.21233/6pe7-xh17 | 13097 | 10.21233/7dx1-dt81 |
| 22986 | 10.21233/6ppf-y854 | 4375 | 10.21233/7emb-xm28 |
| 15734 | 10.21233/6pqr-ch60 | 4296 | 10.21233/7gv9-gq90 |
| 17391 | 10.21233/6pz8-g423 | 17406 | 10.21233/7hjv-yz55 |
| 15944 | 10.21233/6q36-sr27 | 45184 | 10.21233/7hq9-gy29 |
| 3966 | 10.21233/6q48-7975 | 15153 | 10.21233/7j16-cj20 |
| 20001 | 10.21233/6qkv-y767 | 24354 | 10.21233/7jm9-qy50 |
| 4084 | 10.21233/6skg-fz77 | 14995 | 10.21233/7jqq-6h03 |
| 4213 | 10.21233/6ter-nj90 | 16214 | 10.21233/7k06-yk60 |
| 41909 | 10.21233/6tgg-3a45 | 45111 | 10.21233/7mpt-6z64 |
| 15410 | 10.21233/6tzm-ms82 | 42716 | 10.21233/7mz3-tp55 |
| 18108 | 10.21233/6vn0-1104 | 46477 | 10.21233/7ntn-k111 |
| 22678 | 10.21233/6w4z-z869 | 16189 | 10.21233/7ptd-jy37 |
| 15783 | 10.21233/6xsz-gv46 | 15872 | 10.21233/7q6k-xg57 |
| 24844 | 10.21233/6y4e-ym34 | 41419 | 10.21233/7q7t-v963 |
| 3990 | 10.21233/6ys9-2m10 | 4228 | 10.21233/7qh5-4032 |
| 4469 | 10.21233/6yzy-mc78 | 25370 | 10.21233/7qkc-2q02 |
| 47571 | 10.21233/70S7-C853 | 24050 | 10.21233/7qsa-xf22 |
| 4162 | 10.21233/7144-8c50 | 4143 | 10.21233/7svf-3s04 |
| 4362 | 10.21233/719p-q753 | 3956 | 10.21233/7t0m-0g70 |
| 4319 | 10.21233/71yz-am65 | 25230 | 10.21233/7t1e-5e40 |
| 20175 | 10.21233/72na-hf75 | 4028 | 10.21233/7tf3-tt41 |
| 22723 | 10.21233/72t1-ep33 | 25320 | 10.21233/7v49-sj32 |

| | |
|---|---|
| 18104 | 10.21233/7vwz-dk77 |
| 3889 | 10.21233/7xbe-9k30 |
| 15700 | 10.21233/7xwd-g592 |
| 41527 | 10.21233/7ys6-6178 |
| 41305 | 10.21233/808h-p562 |
| 43533 | 10.21233/81cx-my82 |
| 24528 | 10.21233/826g-j091 |
| 47520 | 10.21233/82XH-AC60 |
| 14535 | 10.21233/82xk-w954 |
| 42649 | 10.21233/82zt-4874 |
| 16241 | 10.21233/833d-sq44 |
| 26565 | 10.21233/833s-n396 |
| 4215 | 10.21233/8370-j585 |
| 22767 | 10.21233/84x5-sc43 |
| 3940 | 10.21233/85fe-n850 |
| 25349 | 10.21233/860r-wq86 |
| 16113 | 10.21233/86rn-w288 |
| 46490 | 10.21233/87r0-d068 |
| 25769 | 10.21233/87x4-w578 |
| 41646 | 10.21233/889v-rw31 |
| 24509 | 10.21233/88nk-cv26 |
| 22936 | 10.21233/88tp-8469 |
| 24996 | 10.21233/88vy-c298 |
| 47537 | 10.21233/89TB-QN51 |
| 41185 | 10.21233/89nd-vd67 |
| 357 | 10.21233/89x4-b040 |
| 46798 | 10.21233/8MN4-HF50 |
| 47866 | 10.21233/8P6K-2P36 |
| 47683 | 10.21233/8PFX-TC23 |
| 25414 | 10.21233/8b03-jg14 |
| 14678 | 10.21233/8bnm-7r08 |
| 45379 | 10.21233/8c8d-g988 |
| 15061 | 10.21233/8d1p-vk58 |
| 41138 | 10.21233/8d9a-qw75 |
| 24319 | 10.21233/8da7-b493 |
| 42566 | 10.21233/8djx-f954 |
| 17381 | 10.21233/8dpn-sk91 |
| 24850 | 10.21233/8ej4-q047 |
| 41345 | 10.21233/8f77-mr38 |
| 24414 | 10.21233/8gnn-8h69 |
| 15667 | 10.21233/8k2x-tx03 |
| 15702 | 10.21233/8k8v-tq87 |
| 17304 | 10.21233/8mna-yv25 |

| | |
|---|---|
| 24966 | 10.21233/8mp7-5w52 |
| 14631 | 10.21233/8n1y-1h53 |
| 22367 | 10.21233/8pbq-v525 |
| 4441 | 10.21233/8ph7-sh90 |
| 4147 | 10.21233/8q3r-xv28 |
| 15413 | 10.21233/8q9d-e745 |
| 41181 | 10.21233/8rhg-kr24 |
| 20285 | 10.21233/8s7z-k891 |
| 45298 | 10.21233/8sy1-mp41 |
| 22971 | 10.21233/8t5w-8t95 |
| 15922 | 10.21233/8thg-6w60 |
| 4192 | 10.21233/8vjb-ww55 |
| 22773 | 10.21233/8w6z-7x05 |
| 4355 | 10.21233/8w98-fk51 |
| 24906 | 10.21233/8wkf-an28 |
| 41337 | 10.21233/8xqw-a996 |
| 15750 | 10.21233/8xyz-8886 |
| 3935 | 10.21233/8yhz-ht97 |
| 15709 | 10.21233/8zgx-7708 |
| 41102 | 10.21233/9012-tw18 |
| 4115 | 10.21233/9017-sb34 |
| 22908 | 10.21233/90xr-0g13 |
| 24840 | 10.21233/910a-sq22 |
| 25472 | 10.21233/91x9-5829 |
| 4311 | 10.21233/92cn-k485 |
| 41629 | 10.21233/935p-6302 |
| 20128 | 10.21233/93dm-rp40 |
| 4494 | 10.21233/93pf-0q48 |
| 22775 | 10.21233/93vm-fw90 |
| 45105 | 10.21233/94ex-9803 |
| 16107 | 10.21233/94pt-ya63 |
| 45113 | 10.21233/94x4-yd19 |
| 3501 | 10.21233/956t-ek11 |
| 20304 | 10.21233/959q-kw55 |
| 17873 | 10.21233/95n5-xt46 |
| 17689 | 10.21233/964d-n697 |
| 4397 | 10.21233/96th-h554 |
| 24959 | 10.21233/9703-dw51 |
| 41025 | 10.21233/98b3-ep64 |
| 19842 | 10.21233/98bb-am31 |
| 24229 | 10.21233/98da-sj31 |
| 40642 | 10.21233/98xz-ex68 |
| 4040 | 10.21233/98yq-bv63 |

| | | | |
|---|---|---|---|
| 20171 | 10.21233/99e6-w772 | 46522 | 10.21233/C06D-B533 |
| 22788 | 10.21233/99pm-v778 | 48656 | 10.21233/CA6B-D485 |
| 46680 | 10.21233/9FDV-XK62 | 46730 | 10.21233/CBMY-KS16 |
| 46720 | 10.21233/9Q9F-B230 | 47606 | 10.21233/D41E-T729 |
| 3946 | 10.21233/9a5v-f927 | 46648 | 10.21233/DC6G-0117 |
| 41191 | 10.21233/9b04-ms27 | 48549 | 10.21233/DTAC-5E13 |
| 25188 | 10.21233/9dch-0a13 | 46655 | 10.21233/E28H-2K97 |
| 42161 | 10.21233/9de9-am39 | 46695 | 10.21233/E43F-HA60 |
| 24933 | 10.21233/9dw0-k479 | 46687 | 10.21233/EAP4-RF36 |
| 45713 | 10.21233/9e9x-ff23 | 48072 | 10.21233/ET35-P931 |
| 24327 | 10.21233/9efw-e650 | 47567 | 10.21233/EVQR-8065 |
| 22348 | 10.21233/9ekg-p878 | 48633 | 10.21233/F0X4-1H86 |
| 25234 | 10.21233/9esw-7x29 | 47527 | 10.21233/F7JJ-4741 |
| 14266 | 10.21233/9eyq-n046 | 46998 | 10.21233/F9QR-CV27 |
| 4107 | 10.21233/9j75-xf59 | 47673 | 10.21233/FD92-M524 |
| 4463 | 10.21233/9jtj-a557 | 48621 | 10.21233/FEPD-WS38 |
| 4247 | 10.21233/9kr8-8s34 | 48653 | 10.21233/FHH7-2056 |
| 24339 | 10.21233/9mdv-br36 | 46544 | 10.21233/FJC1-HS98 |
| 41489 | 10.21233/9n20-cc52 | 47584 | 10.21233/FXA3-JW42 |
| 40793 | 10.21233/9na2-9193 | 46645 | 10.21233/GZGE-Q874 |
| 3883 | 10.21233/9p75-y227 | 47685 | 10.21233/H8EZ-FB40 |
| 46292 | 10.21233/9pc1-0436 | 47769 | 10.21233/HBSK-FB02 |
| 19997 | 10.21233/9pdf-7072 | 46691 | 10.21233/HGKE-GM81 |
| 45331 | 10.21233/9qac-3n20 | 46841 | 10.21233/HP7D-SQ69 |
| 17336 | 10.21233/9qv2-xb92 | 46718 | 10.21233/JFQ2-0A48 |
| 24388 | 10.21233/9r0t-vm34 | 46711 | 10.21233/JK1S-VG18 |
| 24539 | 10.21233/9sm0-1589 | 47666 | 10.21233/JVAY-CW81 |
| 24441 | 10.21233/9sy2-nj92 | 47767 | 10.21233/K06E-RW51 |
| 15101 | 10.21233/9vfj-v497 | 48645 | 10.21233/KG60-S564 |
| 46471 | 10.21233/9w17-3k66 | 47557 | 10.21233/KHZZ-1827 |
| 15175 | 10.21233/9wfb-c822 | 47771 | 10.21233/KM96-3E06 |
| 24294 | 10.21233/9x8s-yg68 | 46684 | 10.21233/KSFK-9704 |
| 40448 | 10.21233/9y2h-5560 | 46650 | 10.21233/KW2V-X806 |
| 2393 | 10.21233/9yf5-e111 | 48616 | 10.21233/M5WS-VZ46 |
| 4462 | 10.21233/9yk4-5092 | 46722 | 10.21233/ME1H-RK46 |
| 46523 | 10.21233/ACHA-AZ30 | 48666 | 10.21233/MQMR-FV34 |
| 47259 | 10.21233/AEXN-ZW54 | 46630 | 10.21233/MYT8-WA75 |
| 46709 | 10.21233/ARN6-PX17 | 47597 | 10.21233/N0RB-YF14 |
| 47587 | 10.21233/B785-Q362 | 47621 | 10.21233/N4NQ-VT31 |
| 46611 | 10.21233/BHM0-3F23 | 47504 | 10.21233/N5KS-1603 |
| 46702 | 10.21233/BP0P-DN34 | 47555 | 10.21233/NQHZ-XW16 |
| 48640 | 10.21233/BS0T-C558 | 47968 | 10.21233/PKYN-3N83 |
| 46716 | 10.21233/BTKV-JS74 | 46806 | 10.21233/PTPQ-QD21 |

| | | | |
|---|---|---|---|
| 47497 | 10.21233/RC16-FQ24 | 15947 | 10.21233/a6gw-a462 |
| 47610 | 10.21233/RP9D-AX89 | 2620 | 10.21233/a7g7-e649 |
| 47494 | 10.21233/S6W8-ZW12 | 14614 | 10.21233/a7se-2c50 |
| 47663 | 10.21233/SB5J-M528 | 22880 | 10.21233/a8eg-rs67 |
| 47561 | 10.21233/SF72-GR73 | 4245 | 10.21233/a9hx-nb84 |
| 47814 | 10.21233/SGEJ-6H83 | 41189 | 10.21233/a9sm-ma36 |
| 47535 | 10.21233/SZ9R-Q288 | 26437 | 10.21233/a9x3-nr55 |
| 46689 | 10.21233/T58B-SN37 | 21921 | 10.21233/aa9j-yp34 |
| 47564 | 10.21233/TH2T-9121 | 40321 | 10.21233/abcz-q362 |
| 48668 | 10.21233/TP5S-FZ74 | 24456 | 10.21233/actv-bj33 |
| 48358 | 10.21233/TS9V-G976 | 15286 | 10.21233/ad64-ks29 |
| 47599 | 10.21233/TSWK-BC56 | 15929 | 10.21233/ad82-ja20 |
| 47027 | 10.21233/TVV5-YQ86 | 21836 | 10.21233/adjr-q523 |
| 48658 | 10.21233/V1MX-0B83 | 24089 | 10.21233/adra-c679 |
| 47604 | 10.21233/V3G5-VF48 | 4269 | 10.21233/ads2-bk31 |
| 46535 | 10.21233/V7AX-RV90 | 24620 | 10.21233/aeqn-c862 |
| 46653 | 10.21233/VC03-9R77 | 15624 | 10.21233/aetg-rg52 |
| 46724 | 10.21233/VSJM-6E28 | 45345 | 10.21233/aext-h866 |
| 47602 | 10.21233/VW7G-2J12 | 15492 | 10.21233/ajqc-2p38 |
| 48222 | 10.21233/VYYE-G931 | 16212 | 10.21233/aktj-rk17 |
| 47973 | 10.21233/WZFH-NC77 | 26514 | 10.21233/amj1-fm78 |
| 48611 | 10.21233/X95S-QX49 | 13047 | 10.21233/anc4-fp20 |
| 46672 | 10.21233/X9TF-0J97 | 45391 | 10.21233/ap4h-y067 |
| 47687 | 10.21233/XXP4-Z127 | 4257 | 10.21233/apjc-5g08 |
| 47110 | 10.21233/XZYB-6819 | 15007 | 10.21233/ar0p-hj05 |
| 46659 | 10.21233/YE3D-FJ80 | 14933 | 10.21233/arn2-5250 |
| 47671 | 10.21233/YJWS-JJ96 | 42652 | 10.21233/aryj-gv14 |
| 47502 | 10.21233/YP84-9451 | 42559 | 10.21233/asm4-3r71 |
| 47608 | 10.21233/YVFH-5B51 | 24186 | 10.21233/at68-km35 |
| 47681 | 10.21233/YWDS-WD04 | 4360 | 10.21233/atkr-cs63 |
| 46803 | 10.21233/Z852-D390 | 14923 | 10.21233/av5q-6587 |
| 48361 | 10.21233/ZF2K-PA43 | 16205 | 10.21233/awkk-cx49 |
| 46704 | 10.21233/ZJ6P-GR59 | 4251 | 10.21233/axz3-2923 |
| 46728 | 10.21233/ZNAR-CF17 | 15591 | 10.21233/az3k-5181 |
| 4031 | 10.21233/a033-7j78 | 14684 | 10.21233/azmm-9650 |
| 4326 | 10.21233/a049-xf64 | 3938 | 10.21233/b0fw-9e91 |
| 3865 | 10.21233/a0q7-m868 | 4211 | 10.21233/b18z-e808 |
| 16099 | 10.21233/a2e7-ky61 | 2899 | 10.21233/b2cd-db08 |
| 17285 | 10.21233/a40y-j676 | 22862 | 10.21233/b2sa-vs04 |
| 23015 | 10.21233/a4j6-w793 | 4479 | 10.21233/b3av-3w75 |
| 21895 | 10.21233/a56z-1027 | 24161 | 10.21233/b40a-qt66 |
| 21919 | 10.21233/a5et-h994 | 4548 | 10.21233/b54b-nw94 |
| 20160 | 10.21233/a5ym-ww41 | 45728 | 10.21233/b5h9-pm78 |

| | |
|---|---|
| 46466 | 10.21233/b62t-vg05 |
| 15793 | 10.21233/b6rr-8867 |
| 24993 | 10.21233/b706-qn93 |
| 4320 | 10.21233/b71k-3606 |
| 41310 | 10.21233/b7c6-fn54 |
| 46483 | 10.21233/b7g3-vh04 |
| 21603 | 10.21233/b9ma-cv72 |
| 15866 | 10.21233/b9nq-n244 |
| 22687 | 10.21233/bax7-sx63 |
| 4082 | 10.21233/bc5s-nr94 |
| 16275 | 10.21233/bcqk-fe25 |
| 41762 | 10.21233/bdex-mm22 |
| 4241 | 10.21233/begd-z059 |
| 15641 | 10.21233/bepp-gp08 |
| 45704 | 10.21233/bfpc-jd87 |
| 4392 | 10.21233/bgsv-qs98 |
| 15106 | 10.21233/bhxy-pk21 |
| 25839 | 10.21233/bjeh-fe46 |
| 15752 | 10.21233/bjqy-em56 |
| 45639 | 10.21233/bkjz-st84 |
| 45329 | 10.21233/bkyg-et47 |
| 20194 | 10.21233/bm1c-7055 |
| 4443 | 10.21233/bp9d-qh60 |
| 45137 | 10.21233/bpfs-3r48 |
| 22969 | 10.21233/bppw-6791 |
| 21913 | 10.21233/bqfq-4x49 |
| 15392 | 10.21233/bqja-m092 |
| 24116 | 10.21233/brk6-dd51 |
| 3945 | 10.21233/brzb-ev42 |
| 21790 | 10.21233/bsv3-k317 |
| 22630 | 10.21233/bswe-2h73 |
| 20316 | 10.21233/bt6r-4598 |
| 15952 | 10.21233/bt73-3f31 |
| 16083 | 10.21233/bt8w-b647 |
| 16115 | 10.21233/bt94-dq60 |
| 30184 | 10.21233/btw4-bh35 |
| 25221 | 10.21233/bv3d-g466 |
| 41212 | 10.21233/bv7h-dw44 |
| 21537 | 10.21233/bvws-hb34 |
| 17322 | 10.21233/bx4k-0v94 |
| 24321 | 10.21233/bx5f-9y20 |
| 24518 | 10.21233/bxa2-pc38 |
| 3984 | 10.21233/bz4w-kv74 |

| | |
|---|---|
| 42557 | 10.21233/bzbr-4h48 |
| 14134 | 10.21233/bzy2-pe09 |
| 4078 | 10.21233/c09q-sa29 |
| 46488 | 10.21233/c0s8-4k74 |
| 4418 | 10.21233/c1jz-a477 |
| 24645 | 10.21233/c2gf-2j68 |
| 15146 | 10.21233/c3aa-h411 |
| 41537 | 10.21233/c46j-ka17 |
| 16065 | 10.21233/c49c-z514 |
| 45190 | 10.21233/c52w-kn70 |
| 3876 | 10.21233/c5as-w284 |
| 4338 | 10.21233/c60e-p241 |
| 14889 | 10.21233/c69e-2m65 |
| 41308 | 10.21233/c6jy-0927 |
| 15696 | 10.21233/c6pe-5h03 |
| 17344 | 10.21233/c737-1334 |
| 20050 | 10.21233/c7eh-we69 |
| 4450 | 10.21233/c7fs-kf76 |
| 20034 | 10.21233/c8mz-0061 |
| 3900 | 10.21233/c9ps-vy90 |
| 40803 | 10.21233/ca85-bh57 |
| 3868 | 10.21233/caff-6e80 |
| 17371 | 10.21233/cbz6-xb88 |
| 21818 | 10.21233/cc4x-wz63 |
| 45393 | 10.21233/ccqb-dc70 |
| 21828 | 10.21233/cdq2-qq32 |
| 42687 | 10.21233/cecw-sb96 |
| 4283 | 10.21233/cg36-ty75 |
| 41755 | 10.21233/cg7r-v764 |
| 45202 | 10.21233/ch0p-9286 |
| 41614 | 10.21233/ch26-4s93 |
| 24481 | 10.21233/chjr-t175 |
| 20131 | 10.21233/cjmz-5m20 |
| 17352 | 10.21233/ckz2-ef80 |
| 41187 | 10.21233/cm96-6e64 |
| 20306 | 10.21233/cp60-cs35 |
| 1647 | 10.21233/cpgt-b026 |
| 41435 | 10.21233/cqkb-rx04 |
| 41382 | 10.21233/cqtf-2866 |
| 45698 | 10.21233/cr6x-be86 |
| 4308 | 10.21233/cr8f-k669 |
| 4437 | 10.21233/ct1z-ve54 |
| 15396 | 10.21233/ctpk-r067 |

| | | | |
|---|---|---|---|
| 45655 | 10.21233/cvyt-k021 | 25110 | 10.21233/dvxn-9g46 |
| 25018 | 10.21233/cwpd-7j59 | 24059 | 10.21233/dwfv-m396 |
| 4120 | 10.21233/cx8d-d024 | 4431 | 10.21233/dwrs-9h77 |
| 16224 | 10.21233/cxev-3204 | 4205 | 10.21233/dx45-we16 |
| 41351 | 10.21233/cxq0-9608 | 25285 | 10.21233/dz7x-af16 |
| 24363 | 10.21233/cxrf-mp20 | 19834 | 10.21233/dzp4-pv02 |
| 16254 | 10.21233/cznd-fg62 | 45101 | 10.21233/dztb-ga02 |
| 16133 | 10.21233/d01t-8d61 | 24641 | 10.21233/dzzf-f817 |
| 15588 | 10.21233/d06n-bw32 | 14635 | 10.21233/e00p-4q45 |
| 41207 | 10.21233/d148-6h03 | 15646 | 10.21233/e0q4-t414 |
| 22926 | 10.21233/d2tj-p590 | 4376 | 10.21233/e1h1-x270 |
| 4482 | 10.21233/d47b-gd35 | 19817 | 10.21233/e2ez-ep96 |
| 15187 | 10.21233/d4xx-zn44 | 16199 | 10.21233/e2gb-7a78 |
| 24454 | 10.21233/d6jx-vy62 | 21774 | 10.21233/e2hw-ah76 |
| 20279 | 10.21233/d73q-6954 | 17402 | 10.21233/e34c-fp95 |
| 45316 | 10.21233/d7f2-p170 | 24600 | 10.21233/e387-w812 |
| 20298 | 10.21233/d917-2x13 | 3995 | 10.21233/e3cd-bd97 |
| 41329 | 10.21233/d9jn-js49 | 45349 | 10.21233/e3mw-tc57 |
| 26433 | 10.21233/daff-1123 | 26429 | 10.21233/e4p1-1n07 |
| 41631 | 10.21233/dayz-mr34 | 14648 | 10.21233/e4z3-5t61 |
| 26608 | 10.21233/dbv5-q164 | 24547 | 10.21233/e523-zp67 |
| 15298 | 10.21233/dcm2-wn02 | 24869 | 10.21233/e5sr-kd56 |
| 41928 | 10.21233/dcqy-3516 | 4417 | 10.21233/e5sw-b991 |
| 41326 | 10.21233/ddey-hy33 | 41653 | 10.21233/e7j2-b618 |
| 18100 | 10.21233/dds5-cz13 | 22803 | 10.21233/e7v0-v862 |
| 24097 | 10.21233/dexk-db41 | 4199 | 10.21233/e8bv-7e06 |
| 3991 | 10.21233/dfsr-fr27 | 14523 | 10.21233/e93w-j223 |
| 15764 | 10.21233/dfzy-sd04 | 41760 | 10.21233/eakm-1a87 |
| 42001 | 10.21233/dge8-0c88 | 524 | 10.21233/ecm1-wf91 |
| 4367 | 10.21233/dj3h-f670 | 4402 | 10.21233/ecn2-ak50 |
| 40799 | 10.21233/djae-0e22 | 20058 | 10.21233/ef79-v217 |
| 24190 | 10.21233/djqf-s823 | 15189 | 10.21233/efy3-6z86 |
| 45117 | 10.21233/dm65-7k56 | 21665 | 10.21233/egh5-v320 |
| 21807 | 10.21233/dmvc-9j61 | 24904 | 10.21233/ehcd-fw95 |
| 24507 | 10.21233/dngj-0r02 | 26499 | 10.21233/ehet-7b80 |
| 24046 | 10.21233/dnjr-rs32 | 41525 | 10.21233/ehmm-vm11 |
| 21600 | 10.21233/dpfx-7555 | 16197 | 10.21233/ehwk-t858 |
| 14624 | 10.21233/dpjk-nw36 | 4401 | 10.21233/ehyh-vn64 |
| 45107 | 10.21233/dptw-0b10 | 41177 | 10.21233/ejfb-t464 |
| 41298 | 10.21233/dq5k-vq16 | 41864 | 10.21233/ek47-9m35 |
| 4558 | 10.21233/dqcs-v558 | 4287 | 10.21233/emr2-ry49 |
| 22988 | 10.21233/dsrz-5k22 | 22650 | 10.21233/enk7-n978 |
| 40452 | 10.21233/dtnh-n631 | 15942 | 10.21233/enmq-kb54 |

| | | | | |
|---|---|---|---|---|
| 22923 | 10.21233/enr7-wa61 | | 41764 | 10.21233/fka8-9069 |
| 4148 | 10.21233/enzj-0b81 | | 24465 | 10.21233/fkht-hr86 |
| 4442 | 10.21233/epfd-de13 | | 24341 | 10.21233/fkrs-q995 |
| 45701 | 10.21233/epv2-3m87 | | 15925 | 10.21233/fm9a-bm03 |
| 40797 | 10.21233/eqhn-8105 | | 24861 | 10.21233/fnc3-w873 |
| 21792 | 10.21233/eqv0-5067 | | 17400 | 10.21233/fnwr-vt02 |
| 25845 | 10.21233/ergd-e171 | | 45210 | 10.21233/frtz-w434 |
| 22826 | 10.21233/es4m-ew33 | | 14622 | 10.21233/fs4c-mv83 |
| 4388 | 10.21233/eye6-zs61 | | 4316 | 10.21233/fs8n-ec44 |
| 15778 | 10.21233/ez69-yw85 | | 22644 | 10.21233/fs93-kd09 |
| 4258 | 10.21233/eze7-8418 | | 25340 | 10.21233/fssx-my06 |
| 22992 | 10.21233/f06a-v762 | | 16111 | 10.21233/fvhw-s571 |
| 22910 | 10.21233/f0a8-d318 | | 16075 | 10.21233/fvkb-9e77 |
| 40951 | 10.21233/f1r6-n030 | | 41497 | 10.21233/fw3s-ft98 |
| 40443 | 10.21233/f2ee-nj62 | | 24055 | 10.21233/fw4s-2605 |
| 4095 | 10.21233/f2r9-2c53 | | 22734 | 10.21233/fwh3-s335 |
| 20289 | 10.21233/f3ba-6s09 | | 4449 | 10.21233/fwvv-4v86 |
| 4012 | 10.21233/f3pc-ac31 | | 24937 | 10.21233/fx7m-a985 |
| 24865 | 10.21233/f3pg-3x04 | | 42679 | 10.21233/fy68-3t56 |
| 15221 | 10.21233/f3va-hs11 | | 22101 | 10.21233/g0kj-nd32 |
| 25511 | 10.21233/f533-gn18 | | 16061 | 10.21233/g17d-2n36 |
| 16103 | 10.21233/f5x5-y496 | | 4259 | 10.21233/g1np-hb58 |
| 14946 | 10.21233/f630-mb35 | | 42681 | 10.21233/g1wf-0z38 |
| 22849 | 10.21233/f70k-ey12 | | 22667 | 10.21233/g20r-jy78 |
| 45395 | 10.21233/f7qr-f244 | | 15394 | 10.21233/g3pw-gj52 |
| 25776 | 10.21233/f86j-0362 | | 13051 | 10.21233/g3qv-vs64 |
| 4151 | 10.21233/f903-f525 | | 22828 | 10.21233/g4aa-8505 |
| 4364 | 10.21233/fb1t-1r29 | | 19901 | 10.21233/g546-j142 |
| 41285 | 10.21233/fb4s-ds71 | | 21580 | 10.21233/g56e-8205 |
| 16278 | 10.21233/fb6e-1p18 | | 40519 | 10.21233/g5jk-1e91 |
| 15399 | 10.21233/fbm9-6394 | | 15288 | 10.21233/g6vy-4j29 |
| 4285 | 10.21233/fbsg-8t36 | | 3898 | 10.21233/g6z1-d641 |
| 4063 | 10.21233/fbw1-p392 | | 20018 | 10.21233/g7c8-hk59 |
| 15738 | 10.21233/fcvg-km58 | | 14525 | 10.21233/g7k0-8d93 |
| 14560 | 10.21233/fd56-pw63 | | 41384 | 10.21233/g7x4-c998 |
| 41766 | 10.21233/fds1-dt10 | | 40577 | 10.21233/g8pp-n508 |
| 3879 | 10.21233/ff82-3p60 | | 40666 | 10.21233/g9hf-1t92 |
| 15136 | 10.21233/ffzs-p571 | | 25435 | 10.21233/g9v8-1z05 |
| 26431 | 10.21233/fga9-z080 | | 4279 | 10.21233/gar4-ty16 |
| 22397 | 10.21233/fgen-7074 | | 15099 | 10.21233/gb30-9e28 |
| 4423 | 10.21233/fhj7-pj76 | | 15762 | 10.21233/gd2j-5k10 |
| 15142 | 10.21233/fhxr-x596 | | 22623 | 10.21233/gdbt-zp90 |
| 41034 | 10.21233/fjnd-v956 | | 21541 | 10.21233/gdtp-j631 |

| | | | |
|---|---|---|---|
| 24216 | 10.21233/ge5b-h872 | 4136 | 10.21233/h27c-dp97 |
| 45351 | 10.21233/geb4-ch43 | 22942 | 10.21233/h3bs-6188 |
| 22706 | 10.21233/gfg1-ee16 | 24039 | 10.21233/h3rm-sh19 |
| 4137 | 10.21233/gg82-s427 | 40992 | 10.21233/h3x5-xz62 |
| 42683 | 10.21233/ggnn-mk26 | 24479 | 10.21233/h4cf-f980 |
| 25275 | 10.21233/gh60-9e82 | 40940 | 10.21233/h4gb-ka72 |
| 22636 | 10.21233/gjbn-9951 | 20295 | 10.21233/h5hf-4g09 |
| 25551 | 10.21233/gjvj-6s89 | 4485 | 10.21233/h5wh-mm89 |
| 45695 | 10.21233/gmdn-sh18 | 22890 | 10.21233/h657-1t24 |
| 46481 | 10.21233/gmfj-ka45 | 41543 | 10.21233/h686-6m80 |
| 22658 | 10.21233/gmht-f823 | 17713 | 10.21233/h72j-y982 |
| 22784 | 10.21233/gmhy-nn21 | 4473 | 10.21233/h75a-p298 |
| 15955 | 10.21233/gmm9-y553 | 4424 | 10.21233/h7g7-g111 |
| 23004 | 10.21233/gmp0-vr07 | 22797 | 10.21233/h8w0-4796 |
| 26505 | 10.21233/gn10-hw94 | 46131 | 10.21233/h94p-kc68 |
| 25470 | 10.21233/gnb0-9516 | 4317 | 10.21233/h9er-kf81 |
| 3893 | 10.21233/gp8m-6z07 | 15327 | 10.21233/h9hp-vp69 |
| 14884 | 10.21233/gpa8-an56 | 22795 | 10.21233/h9zx-zt71 |
| 43513 | 10.21233/gpyr-c840 | 15331 | 10.21233/haqy-4x05 |
| 25368 | 10.21233/gq49-ep38 | 22807 | 10.21233/hbv9-4k46 |
| 22874 | 10.21233/grf4-q917 | 4184 | 10.21233/hbws-c483 |
| 46469 | 10.21233/grh1-ap74 | 14274 | 10.21233/hca1-3g26 |
| 24080 | 10.21233/gsax-vn12 | 41842 | 10.21233/hcna-tv25 |
| 21938 | 10.21233/gt52-mt02 | 3927 | 10.21233/hd31-6942 |
| 24522 | 10.21233/gtyp-4x27 | 15081 | 10.21233/hdeg-rn31 |
| 24298 | 10.21233/gve7-qt51 | 16236 | 10.21233/hf5z-sc17 |
| 14276 | 10.21233/gvk8-r723 | 4421 | 10.21233/hfmb-2e61 |
| 15032 | 10.21233/gw2b-x179 | 24188 | 10.21233/hfzm-9410 |
| 4410 | 10.21233/gw72-9x54 | 25384 | 10.21233/hg69-1m77 |
| 41456 | 10.21233/gwdm-cz78 | 22976 | 10.21233/hga1-m504 |
| 40450 | 10.21233/gwh3-1a93 | 24305 | 10.21233/hgah-yb23 |
| 25763 | 10.21233/gwnk-ry08 | 26501 | 10.21233/hgw1-8c70 |
| 4171 | 10.21233/gwts-9s82 | 4435 | 10.21233/hhdq-x585 |
| 16124 | 10.21233/gxbr-bj21 | 24183 | 10.21233/hhxv-jy21 |
| 14521 | 10.21233/gxdp-2n47 | 4086 | 10.21233/hk80-jj60 |
| 15935 | 10.21233/gxny-ma10 | 15417 | 10.21233/hkn5-4a59 |
| 4461 | 10.21233/gy1d-0c88 | 21553 | 10.21233/hmq8-hm34 |
| 24570 | 10.21233/gy4b-s732 | 22400 | 10.21233/hnsc-1p36 |
| 17368 | 10.21233/gyh8-pb41 | 15302 | 10.21233/hpc1-5c65 |
| 4358 | 10.21233/gzqy-9x14 | 44896 | 10.21233/hpgt-nm06 |
| 24221 | 10.21233/h1k4-fg42 | 41264 | 10.21233/hqp0-6z14 |
| 20024 | 10.21233/h1s8-8h96 | 39749 | 10.21233/hr41-mn70 |
| 4531 | 10.21233/h1sz-c208 | 15776 | 10.21233/hrkc-qm74 |

| | | | |
|---:|---|---:|---|
| 20014 | 10.21233/hsdf-hx86 | 22642 | 10.21233/jm78-za93 |
| 41235 | 10.21233/htm2-cw28 | 15931 | 10.21233/jmze-9037 |
| 15157 | 10.21233/htpm-ac79 | 4156 | 10.21233/jnne-vy86 |
| 21893 | 10.21233/hvvv-sy31 | 21788 | 10.21233/jnyr-4x96 |
| 15074 | 10.21233/hwnh-8509 | 41467 | 10.21233/jpfc-1908 |
| 1578 | 10.21233/hxd2-5546 | 19967 | 10.21233/jqgd-sd88 |
| 24157 | 10.21233/hxme-qv81 | 41389 | 10.21233/jqs1-hj70 |
| 22330 | 10.21233/hyt5-1v77 | 24042 | 10.21233/jrvd-m892 |
| 17326 | 10.21233/j01p-4677 | 24526 | 10.21233/jsc2-rn41 |
| 23008 | 10.21233/j0sm-9z75 | 16177 | 10.21233/jsh7-n919 |
| 14645 | 10.21233/j0zd-6071 | 24034 | 10.21233/jsj1-4861 |
| 17396 | 10.21233/j12t-e851 | 40976 | 10.21233/jvxc-rw58 |
| 24143 | 10.21233/j1k9-2922 | 24461 | 10.21233/jvxq-h196 |
| 14410 | 10.21233/j2hz-1p75 | 22973 | 10.21233/jvze-zx40 |
| 14626 | 10.21233/j36k-8t44 | 46445 | 10.21233/jw93-c102 |
| 40441 | 10.21233/j3jg-qz14 | 42555 | 10.21233/jwt1-1r97 |
| 24848 | 10.21233/j4jk-zw12 | 22964 | 10.21233/jxea-ce51 |
| 22714 | 10.21233/j5ga-dr77 | 21606 | 10.21233/jxj8-mf14 |
| 24360 | 10.21233/j694-t641 | 20188 | 10.21233/jyp9-k385 |
| 41499 | 10.21233/j6p7-zb53 | 22698 | 10.21233/jyz6-4566 |
| 41226 | 10.21233/j71m-0095 | 4306 | 10.21233/k0vt-tc67 |
| 15290 | 10.21233/j8q3-sm07 | 4010 | 10.21233/k0xz-ct85 |
| 24390 | 10.21233/j97k-ds95 | 23000 | 10.21233/k1e4-xe95 |
| 24256 | 10.21233/j9as-rt69 | 40111 | 10.21233/k2nt-2b74 |
| 40861 | 10.21233/ja1n-6b46 | 24857 | 10.21233/k38b-1167 |
| 4555 | 10.21233/jact-n751 | 4398 | 10.21233/k41y-rh60 |
| 15789 | 10.21233/jc66-m526 | 15489 | 10.21233/k4wy-0k33 |
| 20068 | 10.21233/jcdm-8a49 | 4467 | 10.21233/k4zb-ze21 |
| 3926 | 10.21233/jdgx-9y34 | 15892 | 10.21233/k58q-9d96 |
| 24576 | 10.21233/jdtq-9666 | 42692 | 10.21233/k65h-1n80 |
| 10967 | 10.21233/je2r-dk10 | 21762 | 10.21233/k9vz-2659 |
| 25477 | 10.21233/jec9-1e73 | 14554 | 10.21233/ka3z-1h69 |
| 4017 | 10.21233/jf00-3j79 | 24842 | 10.21233/kakk-r623 |
| 4138 | 10.21233/jgh0-2062 | 34553 | 10.21233/kb77-8r67 |
| 3892 | 10.21233/jgxs-ey74 | 19266 | 10.21233/kcep-vj71 |
| 3941 | 10.21233/jhaq-tk78 | 4114 | 10.21233/kdkc-ke07 |
| 15597 | 10.21233/jhh6-7q75 | 41320 | 10.21233/ke7a-vf58 |
| 16184 | 10.21233/jhwb-y436 | 21744 | 10.21233/kefv-nq34 |
| 4493 | 10.21233/jjat-b004 | 22892 | 10.21233/keqf-m247 |
| 41283 | 10.21233/jjez-hm35 | 25468 | 10.21233/kf0z-p180 |
| 4071 | 10.21233/jkhn-6f57 | 41399 | 10.21233/kfbq-jw42 |
| 41163 | 10.21233/jky8-1c56 | 24323 | 10.21233/kfxk-5f39 |
| 40958 | 10.21233/jm6d-3395 | 19906 | 10.21233/khvf-ha85 |

| | |
|---|---|
| 15185 | 10.21233/kkjd-pf71 |
| 14527 | 10.21233/kkn8-9202 |
| 3953 | 10.21233/kmvh-v620 |
| 45170 | 10.21233/kn4a-fd55 |
| 42689 | 10.21233/kngt-v975 |
| 39364 | 10.21233/kpgn-4r23 |
| 15350 | 10.21233/kph1-ps34 |
| 15970 | 10.21233/kq0a-zm89 |
| 17419 | 10.21233/krp4-jk87 |
| 17387 | 10.21233/krw5-yg76 |
| 40484 | 10.21233/kt03-8t61 |
| 20183 | 10.21233/kts9-1316 |
| 4174 | 10.21233/kvx4-xd52 |
| 17357 | 10.21233/kw77-m233 |
| 15814 | 10.21233/kxff-0a77 |
| 24622 | 10.21233/kzqk-jj13 |
| 15379 | 10.21233/m06f-by70 |
| 41634 | 10.21233/m0yb-sp15 |
| 15242 | 10.21233/m16f-gh96 |
| 24939 | 10.21233/m1n7-n027 |
| 3869 | 10.21233/m1s8-qh32 |
| 14997 | 10.21233/m1ty-fz61 |
| 41243 | 10.21233/m24p-3135 |
| 13032 | 10.21233/m2a2-qq61 |
| 40574 | 10.21233/m2sf-vp33 |
| 25455 | 10.21233/m32v-dn19 |
| 22966 | 10.21233/m3rj-t898 |
| 41161 | 10.21233/m3wr-jd95 |
| 25186 | 10.21233/m4nq-s295 |
| 19812 | 10.21233/m57c-a803 |
| 808 | 10.21233/m5dr-hs85 |
| 22887 | 10.21233/m5mg-k090 |
| 4139 | 10.21233/m5qy-sh73 |
| 21844 | 10.21233/m6at-pw62 |
| 3971 | 10.21233/m7cb-vr77 |
| 46475 | 10.21233/m7cc-0639 |
| 21651 | 10.21233/m7v2-ks06 |
| 22720 | 10.21233/m8cg-s095 |
| 14921 | 10.21233/m8m6-rn46 |
| 26321 | 10.21233/m8qj-bz64 |
| 22726 | 10.21233/m8xc-ez05 |
| 41140 | 10.21233/ma1q-f261 |
| 3972 | 10.21233/ma29-1k87 |

| | |
|---|---|
| 22322 | 10.21233/mbbn-ff92 |
| 24551 | 10.21233/mbst-4266 |
| 15088 | 10.21233/mce3-kp31 |
| 22376 | 10.21233/mczv-f712 |
| 21903 | 10.21233/me0e-ze15 |
| 22896 | 10.21233/me4r-tx16 |
| 16231 | 10.21233/mf7y-nm83 |
| 4299 | 10.21233/mfht-2e09 |
| 41612 | 10.21233/mgtf-q479 |
| 15829 | 10.21233/mgv7-z770 |
| 14643 | 10.21233/mhdv-6n63 |
| 15005 | 10.21233/mhya-2v27 |
| 17334 | 10.21233/mj04-mx52 |
| 22751 | 10.21233/mjwh-t327 |
| 41262 | 10.21233/mjzm-jv11 |
| 4060 | 10.21233/mk13-rb17 |
| 45311 | 10.21233/mks1-mt36 |
| 40582 | 10.21233/mky9-a533 |
| 41446 | 10.21233/mmtn-dv23 |
| 4133 | 10.21233/mnay-8z91 |
| 25133 | 10.21233/mpd1-a275 |
| 22685 | 10.21233/mqcv-z280 |
| 29236 | 10.21233/mqh5-7z20 |
| 21708 | 10.21233/mqtv-9820 |
| 41199 | 10.21233/mrq9-0a25 |
| 4223 | 10.21233/mtkm-0h07 |
| 4446 | 10.21233/mtmk-d772 |
| 41314 | 10.21233/mw33-kd49 |
| 4201 | 10.21233/mybh-v067 |
| 3996 | 10.21233/mzrc-8w03 |
| 22718 | 10.21233/n0v4-yj40 |
| 25268 | 10.21233/n1n7-ya12 |
| 40634 | 10.21233/n1v6-tq67 |
| 239 | 10.21233/n3002s |
| 1694 | 10.21233/n3016n |
| 2328 | 10.21233/n3018d |
| 3132 | 10.21233/n3020q |
| 203 | 10.21233/n30302 |
| 2065 | 10.21233/n3036b |
| 260 | 10.21233/n3040d |
| 2297 | 10.21233/n30472 |
| 2905 | 10.21233/n3049t |
| 802 | 10.21233/n3061g |

| | |
|---:|:---|
| 1606 | 10.21233/n3064m |
| 1840 | 10.21233/n30650 |
| 225 | 10.21233/n30684 |
| 858 | 10.21233/n3070f |
| 1661 | 10.21233/n3073k |
| 365 | 10.21233/n30896 |
| 1805 | 10.21233/n30938 |
| 11 | 10.21233/n3097s |
| 823 | 10.21233/n3099j |
| 1442 | 10.21233/n30b1v |
| 1860 | 10.21233/n30b3m |
| 2663 | 10.21233/n30b5c |
| 1574 | 10.21233/n30d2x |
| 2054 | 10.21233/n30f3n |
| 332 | 10.21233/n30g64 |
| 1144 | 10.21233/n30g8w |
| 1543 | 10.21233/n30h06 |
| 1769 | 10.21233/n30h1k |
| 790 | 10.21233/n30h6g |
| 1595 | 10.21233/n30h9m |
| 2022 | 10.21233/n30j0j |
| 2627 | 10.21233/n30j29 |
| 298 | 10.21233/n30k65 |
| 518 | 10.21233/n30k7j |
| 1965 | 10.21233/n30m1m |
| 2364 | 10.21233/n30m3c |
| 354 | 10.21233/n30m6h |
| 1563 | 10.21233/n30n0k |
| 2596 | 10.21233/n30n3q |
| 2332 | 10.21233/n30q1n |
| 540 | 10.21233/n30q55 |
| 1987 | 10.21233/n30q9p |
| 2385 | 10.21233/n30r10 |
| 1697 | 10.21233/n30s80 |
| 507 | 10.21233/n30t3f |
| 1133 | 10.21233/n30t6k |
| 230 | 10.21233/n30w34 |
| 1664 | 10.21233/n30w7n |
| 287 | 10.21233/n30x23 |
| 688 | 10.21233/n30x3g |
| 1718 | 10.21233/n30x70 |
| 3558 | 10.21233/n30z2f |
| 1445 | 10.21233/n3106m |

| | |
|---:|:---|
| 1864 | 10.21233/n31070 |
| 2267 | 10.21233/n3108c |
| 68 | 10.21233/n3110p |
| 1632 | 10.21233/n31358 |
| 1831 | 10.21233/n3136n |
| 850 | 10.21233/n3142g |
| 1886 | 10.21233/n3145m |
| 2289 | 10.21233/n31460 |
| 1598 | 10.21233/n3164x |
| 2025 | 10.21233/n31659 |
| 2630 | 10.21233/n3168f |
| 1434 | 10.21233/n3173w |
| 1852 | 10.21233/n3175n |
| 978 | 10.21233/n3191t |
| 1795 | 10.21233/n3194z |
| 1621 | 10.21233/n31b2j |
| 2650 | 10.21233/n31b62 |
| 3050 | 10.21233/n31b7f |
| 3452 | 10.21233/n31b8t |
| 1566 | 10.21233/n31d1v |
| 2388 | 10.21233/n31d40 |
| 1001 | 10.21233/n31d9w |
| 1819 | 10.21233/n31f2k |
| 2619 | 10.21233/n31f4b |
| 324 | 10.21233/n31g7t |
| 1136 | 10.21233/n31g9k |
| 1587 | 10.21233/n31h9x |
| 510 | 10.21233/n31k6g |
| 691 | 10.21233/n31k7v |
| 1158 | 10.21233/n31m8k |
| 1979 | 10.21233/n31n0w |
| 2588 | 10.21233/n31n31 |
| 1688 | 10.21233/n31p9n |
| 2323 | 10.21233/n31q1z |
| 311 | 10.21233/n31q3q |
| 532 | 10.21233/n31q43 |
| 2955 | 10.21233/n31r32 |
| 3582 | 10.21233/n31r5t |
| 254 | 10.21233/n31s4s |
| 1890 | 10.21233/n31s9p |
| 2292 | 10.21233/n31t10 |
| 496 | 10.21233/n31t44 |
| 1522 | 10.21233/n31t78 |

| | |
|---|---|
| 219 | 10.21233/n31w22 |
| 853 | 10.21233/n31w4t |
| 1254 | 10.21233/n31w6k |
| 1656 | 10.21233/n31w7z |
| 278 | 10.21233/n31x2d |
| 2312 | 10.21233/n31x92 |
| 818 | 10.21233/n3203s |
| 1437 | 10.21233/n3205j |
| 2888 | 10.21233/n3219d |
| 2049 | 10.21233/n3236z |
| 2622 | 10.21233/n32393 |
| 841 | 10.21233/n3242s |
| 3483 | 10.21233/n3250c |
| 1590 | 10.21233/n32647 |
| 2017 | 10.21233/n3265m |
| 1844 | 10.21233/n32736 |
| 2642 | 10.21233/n3277q |
| 1161 | 10.21233/n3290r |
| 1786 | 10.21233/n3293w |
| 2591 | 10.21233/n3295n |
| 993 | 10.21233/n32995 |
| 1611 | 10.21233/n32b1g |
| 3042 | 10.21233/n32b50 |
| 314 | 10.21233/n32c8g |
| 535 | 10.21233/n32c9v |
| 1982 | 10.21233/n32d3x |
| 369 | 10.21233/n32d8t |
| 772 | 10.21233/n32d96 |
| 1810 | 10.21233/n32f38 |
| 2611 | 10.21233/n32f51 |
| 720 | 10.21233/n32g8h |
| 2348 | 10.21233/n32h3z |
| 336 | 10.21233/n32h63 |
| 1773 | 10.21233/n32j05 |
| 2579 | 10.21233/n32j4p |
| 281 | 10.21233/n32k5d |
| 1149 | 10.21233/n32m8w |
| 1547 | 10.21233/n32m98 |
| 1969 | 10.21233/n32n06 |
| 2368 | 10.21233/n32n1k |
| 1679 | 10.21233/n32p9z |
| 303 | 10.21233/n32q4d |
| 32 | 10.21233/n32s3q |

| | |
|---|---|
| 267 | 10.21233/n32t32 |
| 488 | 10.21233/n32t4f |
| 1701 | 10.21233/n32t7k |
| 210 | 10.21233/n32w2c |
| 1648 | 10.21233/n32w78 |
| 1847 | 10.21233/n32w8n |
| 672 | 10.21233/n32x33 |
| 9 | 10.21233/n3301b |
| 1615 | 10.21233/n33067 |
| 2645 | 10.21233/n33109 |
| 1450 | 10.21233/n33156 |
| 1868 | 10.21233/n3317z |
| 2614 | 10.21233/n33381 |
| 2061 | 10.21233/n33457 |
| 3468 | 10.21233/n3350p |
| 1581 | 10.21233/n33635 |
| 1776 | 10.21233/n3364j |
| 2006 | 10.21233/n3365x |
| 339 | 10.21233/n33894 |
| 525 | 10.21233/n33902 |
| 1153 | 10.21233/n3392t |
| 2371 | 10.21233/n3396b |
| 2583 | 10.21233/n3397q |
| 306 | 10.21233/n33c7d |
| 361 | 10.21233/n33d7r |
| 2603 | 10.21233/n33f5b |
| 1704 | 10.21233/n33h1h |
| 2340 | 10.21233/n33h38 |
| 328 | 10.21233/n33h51 |
| 271 | 10.21233/n33k63 |
| 675 | 10.21233/n33k8v |
| 514 | 10.21233/n33m6f |
| 3775 | 10.21233/n33n6s |
| 238 | 10.21233/n33p4c |
| 1671 | 10.21233/n33p98 |
| 294 | 10.21233/n33q4q |
| 1107 | 10.21233/n33q7v |
| 2327 | 10.21233/n33r1x |
| 835 | 10.21233/n33s5s |
| 2673 | 10.21233/n33t1m |
| 259 | 10.21233/n33t20 |
| 2904 | 10.21233/n33v1z |
| 202 | 10.21233/n33w32 |

| | |
|---:|---|
| 2637 | 10.21233/n33x21 |
| 857 | 10.21233/n33x55 |
| 1894 | 10.21233/n33x9p |
| 3500 | 10.21233/n33z3r |
| 801 | 10.21233/n3403d |
| 224 | 10.21233/n3410m |
| 1441 | 10.21233/n34144 |
| 1859 | 10.21233/n3415h |
| 364 | 10.21233/n34309 |
| 1804 | 10.21233/n34356 |
| 1010 | 10.21233/n34411 |
| 1628 | 10.21233/n3443s |
| 2053 | 10.21233/n34445 |
| 3060 | 10.21233/n34479 |
| 331 | 10.21233/n3460b |
| 1768 | 10.21233/n3464v |
| 2396 | 10.21233/n34670 |
| 788 | 10.21233/n34712 |
| 1594 | 10.21233/n34746 |
| 1826 | 10.21233/n3475k |
| 2626 | 10.21233/n3477b |
| 517 | 10.21233/n3489f |
| 1542 | 10.21233/n3493h |
| 1964 | 10.21233/n3494w |
| 2363 | 10.21233/n34958 |
| 352 | 10.21233/n3498d |
| 974 | 10.21233/n34b0q |
| 1791 | 10.21233/n34b2g |
| 1111 | 10.21233/n34c9g |
| 1733 | 10.21233/n34d25 |
| 539 | 10.21233/n34d72 |
| 1562 | 10.21233/n34f14 |
| 1986 | 10.21233/n34f2h |
| 2384 | 10.21233/n34f3w |
| 1696 | 10.21233/n34h1t |
| 318 | 10.21233/n34h6q |
| 1755 | 10.21233/n34j15 |
| 860 | 10.21233/n34k85 |
| 285 | 10.21233/n34m6r |
| 506 | 10.21233/n34m74 |
| 2352 | 10.21233/n34n2k |
| 3557 | 10.21233/n34n63 |
| 1444 | 10.21233/n34p7t |

| | |
|---:|---|
| 1663 | 10.21233/n34p86 |
| 1862 | 10.21233/n34p9k |
| 2319 | 10.21233/n34r0v |
| 1631 | 10.21233/n34s87 |
| 2266 | 10.21233/n34t0j |
| 2665 | 10.21233/n34t29 |
| 250 | 10.21233/n34t3p |
| 1684 | 10.21233/n34t76 |
| 1885 | 10.21233/n34t8k |
| 792 | 10.21233/n34w54 |
| 2629 | 10.21233/n34x1z |
| 215 | 10.21233/n34x3q |
| 849 | 10.21233/n34x5g |
| 977 | 10.21233/n3503q |
| 1794 | 10.21233/n3506v |
| 1433 | 10.21233/n3515t |
| 2649 | 10.21233/n3518z |
| 356 | 10.21233/n35310 |
| 543 | 10.21233/n3532c |
| 2598 | 10.21233/n35391 |
| 1000 | 10.21233/n3542q |
| 1818 | 10.21233/n3545v |
| 2045 | 10.21233/n35467 |
| 1135 | 10.21233/n3562d |
| 2013 | 10.21233/n3574h |
| 2618 | 10.21233/n3577n |
| 509 | 10.21233/n3589r |
| 2355 | 10.21233/n3595k |
| 343 | 10.21233/n3597b |
| 966 | 10.21233/n35993 |
| 1157 | 10.21233/n35b01 |
| 2587 | 10.21233/n35b4j |
| 531 | 10.21233/n35d8r |
| 3790 | 10.21233/n35f83 |
| 253 | 10.21233/n35g6p |
| 1687 | 10.21233/n35h14 |
| 495 | 10.21233/n35h7d |
| 715 | 10.21233/n35h8s |
| 2954 | 10.21233/n35j50 |
| 852 | 10.21233/n35k73 |
| 1655 | 10.21233/n35m0s |
| 1889 | 10.21233/n35m15 |
| 679 | 10.21233/n35m5p |

| | |
|---:|:---|
| 1709 | 10.21233/n35m96 |
| 2922 | 10.21233/n35n38 |
| 3149 | 10.21233/n35n4n |
| 1854 | 10.21233/n35q0t |
| 2652 | 10.21233/n35q4b |
| 873 | 10.21233/n35q7g |
| 2311 | 10.21233/n35r2x |
| 1003 | 10.21233/n35s6s |
| 1623 | 10.21233/n35s8j |
| 2048 | 10.21233/n35s9x |
| 27 | 10.21233/n35t17 |
| 782 | 10.21233/n35w3p |
| 2016 | 10.21233/n35w8k |
| 206 | 10.21233/n35x18 |
| 1028 | 10.21233/n35x4d |
| 2068 | 10.21233/n35x7j |
| 3482 | 10.21233/n35z20 |
| 346 | 10.21233/n3602n |
| 969 | 10.21233/n3604d |
| 2590 | 10.21233/n36099 |
| 805 | 10.21233/n36120 |
| 1610 | 10.21233/n36154 |
| 1843 | 10.21233/n3616h |
| 3041 | 10.21233/n3620k |
| 534 | 10.21233/n36319 |
| 1160 | 10.21233/n3634f |
| 1981 | 10.21233/n36366 |
| 2378 | 10.21233/n3637k |
| 1391 | 10.21233/n3643d |
| 1809 | 10.21233/n3644s |
| 2610 | 10.21233/n3647x |
| 313 | 10.21233/n36591 |
| 719 | 10.21233/n3660z |
| 1524 | 10.21233/n36633 |
| 1577 | 10.21233/n3673f |
| 2001 | 10.21233/n3674t |
| 2347 | 10.21233/n3695w |
| 335 | 10.21233/n36981 |
| 1148 | 10.21233/n36b0b |
| 1546 | 10.21233/n36b23 |
| 1772 | 10.21233/n36b3g |
| 1968 | 10.21233/n36b4v |
| 2578 | 10.21233/n36b6m |

| | |
|---:|:---|
| 682 | 10.21233/n36c93 |
| 876 | 10.21233/n36d01 |
| 2314 | 10.21233/n36d5x |
| 302 | 10.21233/n36d82 |
| 2367 | 10.21233/n36f4w |
| 245 | 10.21233/n36g60 |
| 1880 | 10.21233/n36h1f |
| 2335 | 10.21233/n36j25 |
| 31 | 10.21233/n36k5n |
| 209 | 10.21233/n36k61 |
| 843 | 10.21233/n36k8s |
| 671 | 10.21233/n36m6c |
| 1904 | 10.21233/n36n0f |
| 8 | 10.21233/n36p49 |
| 2644 | 10.21233/n36q38 |
| 234 | 10.21233/n36q51 |
| 1667 | 10.21233/n36q9j |
| 2303 | 10.21233/n36r1v |
| 995 | 10.21233/n36s5q |
| 1397 | 10.21233/n36s63 |
| 1448 | 10.21233/n36t7t |
| 1867 | 10.21233/n36t86 |
| 2669 | 10.21233/n36v0h |
| 3074 | 10.21233/n36v28 |
| 774 | 10.21233/n36w5r |
| 1580 | 10.21233/n36w7h |
| 2005 | 10.21233/n36w8w |
| 1635 | 10.21233/n36x6g |
| 1152 | 10.21233/n3704q |
| 1775 | 10.21233/n3706g |
| 796 | 10.21233/n3713p |
| 1601 | 10.21233/n3715f |
| 2633 | 10.21233/n3719z |
| 3033 | 10.21233/n3720w |
| 1549 | 10.21233/n3734r |
| 1972 | 10.21233/n3736h |
| 490 | 10.21233/n3761n |
| 547 | 10.21233/n37710 |
| 1569 | 10.21233/n37744 |
| 1763 | 10.21233/n3775h |
| 269 | 10.21233/n37880 |
| 1703 | 10.21233/n3793f |
| 327 | 10.21233/n3797z |

| | | | |
|---|---|---|---|
| 513 | 10.21233/n3798b | 2362 | 10.21233/n38376 |
| 1139 | 10.21233/n37b0n | 351 | 10.21233/n3840w |
| 1959 | 10.21233/n37b2d | 2594 | 10.21233/n3848x |
| 3774 | 10.21233/n37b92 | 317 | 10.21233/n3870x |
| 867 | 10.21233/n37c9d | 538 | 10.21233/n38719 |
| 1670 | 10.21233/n37d23 | 1754 | 10.21233/n3874f |
| 699 | 10.21233/n37d70 | 261 | 10.21233/n38889 |
| 1105 | 10.21233/n37d9r | 1695 | 10.21233/n3892c |
| 1725 | 10.21233/n37f12 | 1897 | 10.21233/n3893r |
| 3565 | 10.21233/n37f7b | 723 | 10.21233/n3898n |
| 20 | 10.21233/n37g69 | 2351 | 10.21233/n38b4g |
| 2326 | 10.21233/n37j47 | 2962 | 10.21233/n38b67 |
| 3130 | 10.21233/n37j60 | 226 | 10.21233/n38c8b |
| 834 | 10.21233/n37k7q | 1716 | 10.21233/n38f2r |
| 1639 | 10.21233/n37m0d | 1443 | 10.21233/n38h12 |
| 2063 | 10.21233/n37m1s | 2265 | 10.21233/n38h3t |
| 258 | 10.21233/n37m4x | 880 | 10.21233/n38h93 |
| 2903 | 10.21233/n37n48 | 1683 | 10.21233/n38j1d |
| 1604 | 10.21233/n37p9h | 2318 | 10.21233/n38j35 |
| 1837 | 10.21233/n37q0f | 1013 | 10.21233/n38k71 |
| 856 | 10.21233/n37q6q | 1630 | 10.21233/n38k9s |
| 1659 | 10.21233/n37q9v | 1829 | 10.21233/n38m0q |
| 3499 | 10.21233/n37r5p | 847 | 10.21233/n38m5m |
| 363 | 10.21233/n37s4n | 2287 | 10.21233/n38n02 |
| 987 | 10.21233/n37s6d | 791 | 10.21233/n38p72 |
| 1803 | 10.21233/n37s85 | 2023 | 10.21233/n38q0r |
| 821 | 10.21233/n37t40 | 1432 | 10.21233/n38q7d |
| 1440 | 10.21233/n37t6r | 1651 | 10.21233/n38q8s |
| 1627 | 10.21233/n37t74 | 1850 | 10.21233/n38q95 |
| 2262 | 10.21233/n37v0t | 355 | 10.21233/n38s3k |
| 2661 | 10.21233/n37v16 | 976 | 10.21233/n38s5b |
| 550 | 10.21233/n37w39 | 1793 | 10.21233/n38s8g |
| 1572 | 10.21233/n37w7t | 813 | 10.21233/n38t5p |
| 2395 | 10.21233/n37w9k | 3048 | 10.21233/n38v2w |
| 1008 | 10.21233/n37x41 | 1564 | 10.21233/n38w6r |
| 2625 | 10.21233/n37x9x | 1758 | 10.21233/n38w74 |
| 330 | 10.21233/n3801w | 375 | 10.21233/n38x16 |
| 1142 | 10.21233/n3803n | 778 | 10.21233/n38x2k |
| 1541 | 10.21233/n3805d | 999 | 10.21233/n38x3z |
| 973 | 10.21233/n38130 | 1817 | 10.21233/n38x63 |
| 3840 | 10.21233/n3822z | 2617 | 10.21233/n38x8v |
| 516 | 10.21233/n38329 | 321 | 10.21233/n3902k |
| 1732 | 10.21233/n3835f | 1780 | 10.21233/n3917t |

| | |
|---:|:---|
| 3611 | 10.21233/n39228 |
| 3830 | 10.21233/n3923n |
| 1719 | 10.21233/n3935r |
| 342 | 10.21233/n39398 |
| 530 | 10.21233/n39406 |
| 1156 | 10.21233/n3942z |
| 1553 | 10.21233/n3943b |
| 1976 | 10.21233/n3944q |
| 2586 | 10.21233/n3947v |
| 69 | 10.21233/n3959z |
| 309 | 10.21233/n39699 |
| 2953 | 10.21233/n3977w |
| 252 | 10.21233/n39890 |
| 1888 | 10.21233/n3994f |
| 2290 | 10.21233/n3995t |
| 1520 | 10.21233/n39b21 |
| 2343 | 10.21233/n39b55 |
| 1654 | 10.21233/n39d2q |
| 275 | 10.21233/n39d67 |
| 678 | 10.21233/n39d80 |
| 2920 | 10.21233/n39f6k |
| 816 | 10.21233/n39g8p |
| 1435 | 10.21233/n39h00 |
| 1622 | 10.21233/n39h1c |
| 2047 | 10.21233/n39h34 |
| 2651 | 10.21233/n39h5w |
| 872 | 10.21233/n39h81 |
| 1674 | 10.21233/n39j1q |
| 381 | 10.21233/n39k6z |
| 1002 | 10.21233/n39k8q |
| 1820 | 10.21233/n39m01 |
| 25 | 10.21233/n39m4j |
| 3081 | 10.21233/n39n4w |
| 968 | 10.21233/n39p7c |
| 2015 | 10.21233/n39q1f |
| 205 | 10.21233/n39q4k |
| 1842 | 10.21233/n39q9g |
| 345 | 10.21233/n39s3w |
| 1783 | 10.21233/n39s7d |
| 1980 | 10.21233/n39s8s |
| 804 | 10.21233/n39t37 |
| 1390 | 10.21233/n39t6c |
| 3040 | 10.21233/n39v1t |

| | |
|---:|:---|
| 312 | 10.21233/n39w3x |
| 2377 | 10.21233/n39w96 |
| 1807 | 10.21233/n39x6d |
| 2609 | 10.21233/n39x9j |
| 497 | 10.21233/n3b02w |
| 1523 | 10.21233/n3b07s |
| 1771 | 10.21233/n3b16r |
| 279 | 10.21233/n3b305 |
| 681 | 10.21233/n3b31j |
| 1711 | 10.21233/n3b34p |
| 1545 | 10.21233/n3b441 |
| 1967 | 10.21233/n3b45d |
| 3781 | 10.21233/n3b517 |
| 1114 | 10.21233/n3b729 |
| 1736 | 10.21233/n3b742 |
| 2945 | 10.21233/n3b78k |
| 1879 | 10.21233/n3b94r |
| 7 | 10.21233/n3bc7k |
| 208 | 10.21233/n3bc8z |
| 1030 | 10.21233/n3bd08 |
| 1646 | 10.21233/n3bd21 |
| 1845 | 10.21233/n3bd3d |
| 2643 | 10.21233/n3bh56 |
| 16 | 10.21233/n3bh6k |
| 232 | 10.21233/n3bh7z |
| 1447 | 10.21233/n3bj0n |
| 1666 | 10.21233/n3bj11 |
| 1866 | 10.21233/n3bj2d |
| 773 | 10.21233/n3bk68 |
| 994 | 10.21233/n3bk7n |
| 1393 | 10.21233/n3bk9d |
| 2612 | 10.21233/n3bm23 |
| 829 | 10.21233/n3bm6m |
| 2270 | 10.21233/n3bn0p |
| 1579 | 10.21233/n3bp82 |
| 2004 | 10.21233/n3bq0c |
| 2632 | 10.21233/n3br0q |
| 3032 | 10.21233/n3br2g |
| 337 | 10.21233/n3bs4k |
| 523 | 10.21233/n3bs5z |
| 1970 | 10.21233/n3bs9g |
| 2369 | 10.21233/n3bt1s |
| 2580 | 10.21233/n3bt25 |

| | | | |
|---:|---|---:|---|
| 982 | 10.21233/n3bt6p | 1626 | 10.21233/n3cm92 |
| 304 | 10.21233/n3bw2v | 3058 | 10.21233/n3cn34 |
| 1171 | 10.21233/n3bx4z | 329 | 10.21233/n3cp57 |
| 1568 | 10.21233/n3bx5b | 549 | 10.21233/n3cp6m |
| 2391 | 10.21233/n3bx8g | 1766 | 10.21233/n3cq0p |
| 489 | 10.21233/n3c026 | 785 | 10.21233/n3cq6z |
| 1513 | 10.21233/n3c05b | 1592 | 10.21233/n3cq8q |
| 2338 | 10.21233/n3c09v | 1824 | 10.21233/n3cq93 |
| 1138 | 10.21233/n3c149 | 2624 | 10.21233/n3cr2s |
| 1762 | 10.21233/n3c162 | 3839 | 10.21233/n3cr69 |
| 268 | 10.21233/n3c31v | 1141 | 10.21233/n3cs6n |
| 673 | 10.21233/n3c33m | 1961 | 10.21233/n3cs9s |
| 1906 | 10.21233/n3c36r | 2361 | 10.21233/n3ct0q |
| 1724 | 10.21233/n3c45q | 350 | 10.21233/n3ct3v |
| 1958 | 10.21233/n3c463 | 1789 | 10.21233/n3ct8r |
| 2358 | 10.21233/n3c48v | 2593 | 10.21233/n3cv1f |
| 236 | 10.21233/n3c59k | 701 | 10.21233/n3cw3j |
| 866 | 10.21233/n3c61w | 537 | 10.21233/n3cx14 |
| 693 | 10.21233/n3c70v | 1984 | 10.21233/n3cx61 |
| 1104 | 10.21233/n3c717 | 2382 | 10.21233/n3cx7d |
| 19 | 10.21233/n3c887 | 316 | 10.21233/n3d12v |
| 1451 | 10.21233/n3c929 | 2961 | 10.21233/n3d20f |
| 1637 | 10.21233/n3c93p | 1715 | 10.21233/n3d451 |
| 2671 | 10.21233/n3c976 | 2317 | 10.21233/n3d752 |
| 1691 | 10.21233/n3cb31 | 1883 | 10.21233/n3db4q |
| 2902 | 10.21233/n3cb65 | 1650 | 10.21233/n3df07 |
| 200 | 10.21233/n3cc88 | 2286 | 10.21233/n3df20 |
| 1836 | 10.21233/n3cd3q | 3488 | 10.21233/n3df7w |
| 2635 | 10.21233/n3cd6v | 1849 | 10.21233/n3dj08 |
| 1658 | 10.21233/n3cf2p | 2647 | 10.21233/n3dj21 |
| 1892 | 10.21233/n3cf32 | 3047 | 10.21233/n3dj3d |
| 3497 | 10.21233/n3cf8z | 998 | 10.21233/n3dm80 |
| 799 | 10.21233/n3cg89 | 1815 | 10.21233/n3dn1p |
| 986 | 10.21233/n3cg9p | 374 | 10.21233/n3dq34 |
| 1603 | 10.21233/n3ch2c | 777 | 10.21233/n3dq5w |
| 2030 | 10.21233/n3ch3r | 2011 | 10.21233/n3dq81 |
| 1439 | 10.21233/n3cj0z | 341 | 10.21233/n3dt35 |
| 1857 | 10.21233/n3cj1b | 1155 | 10.21233/n3dt69 |
| 2476 | 10.21233/n3cj33 | 1778 | 10.21233/n3dt82 |
| 2660 | 10.21233/n3cj4g | 3610 | 10.21233/n3dv4w |
| 1571 | 10.21233/n3ck8b | 527 | 10.21233/n3dx36 |
| 1802 | 10.21233/n3ck9q | 2373 | 10.21233/n3dz0d |
| 820 | 10.21233/n3cm5j | 308 | 10.21233/n3f11s |

| | |
|---|---|
| 493 | 10.21233/n3f125 |
| 2342 | 10.21233/n3f172 |
| 677 | 10.21233/n3f40f |
| 1707 | 10.21233/n3f45b |
| 2919 | 10.21233/n3f48g |
| 871 | 10.21233/n3f71v |
| 24 | 10.21233/n3f97t |
| 240 | 10.21233/n3f986 |
| 2885 | 10.21233/n3fb75 |
| 204 | 10.21233/n3fd53 |
| 1024 | 10.21233/n3fd87 |
| 1642 | 10.21233/n3ff1x |
| 3080 | 10.21233/n3ff5f |
| 803 | 10.21233/n3fh88 |
| 1607 | 10.21233/n3fj0k |
| 1841 | 10.21233/n3fj1z |
| 3039 | 10.21233/n3fj5g |
| 1806 | 10.21233/n3fn0m |
| 333 | 10.21233/n3ft4v |
| 1770 | 10.21233/n3ft8c |
| 1966 | 10.21233/n3ft9r |
| 2576 | 10.21233/n3fv12 |
| 299 | 10.21233/n3fx24 |
| 519 | 10.21233/n3fx3h |
| 1735 | 10.21233/n3fx71 |
| 2944 | 10.21233/n3g194 |
| 3138 | 10.21233/n3g202 |
| 1698 | 10.21233/n3g45n |
| 15 | 10.21233/n3g974 |
| 828 | 10.21233/n3g99w |
| 2268 | 10.21233/n3gb4b |
| 1633 | 10.21233/n3gf17 |
| 1832 | 10.21233/n3gf2m |
| 1599 | 10.21233/n3gh9z |
| 2026 | 10.21233/n3gj0w |
| 2631 | 10.21233/n3gj2n |
| 3031 | 10.21233/n3gj4d |
| 1796 | 10.21233/n3gn0x |
| 1137 | 10.21233/n3gt6x |
| 3563 | 10.21233/n3gz35 |
| 2324 | 10.21233/n3h182 |
| 3126 | 10.21233/n3h20c |
| 256 | 10.21233/n3h402 |

| | |
|---|---|
| 1891 | 10.21233/n3h45z |
| 2293 | 10.21233/n3h46b |
| 2901 | 10.21233/n3h49g |
| 221 | 10.21233/n3h68s |
| 1255 | 10.21233/n3h72v |
| 1438 | 10.21233/n3h737 |
| 1657 | 10.21233/n3h74m |
| 1856 | 10.21233/n3hb38 |
| 2050 | 10.21233/n3hb4n |
| 2654 | 10.21233/n3hb6d |
| 3056 | 10.21233/n3hb7s |
| 784 | 10.21233/n3hd97 |
| 1006 | 10.21233/n3hf05 |
| 2623 | 10.21233/n3hf52 |
| 971 | 10.21233/n3hh7h |
| 1591 | 10.21233/n3hj06 |
| 1162 | 10.21233/n3hm8x |
| 1559 | 10.21233/n3hm99 |
| 1788 | 10.21233/n3hn07 |
| 315 | 10.21233/n3hq32 |
| 536 | 10.21233/n3hq4f |
| 2381 | 10.21233/n3hr08 |
| 2960 | 10.21233/n3hr3d |
| 501 | 10.21233/n3ht33 |
| 684 | 10.21233/n3hx4h |
| 878 | 10.21233/n3j13g |
| 1680 | 10.21233/n3j157 |
| 33 | 10.21233/n3j39f |
| 1882 | 10.21233/n3j458 |
| 3487 | 10.21233/n3j50q |
| 212 | 10.21233/n3j69g |
| 1033 | 10.21233/n3j71s |
| 811 | 10.21233/n3j99h |
| 1617 | 10.21233/n3jb1t |
| 2646 | 10.21233/n3jb4z |
| 3046 | 10.21233/n3jb6q |
| 3861 | 10.21233/n3jb8g |
| 776 | 10.21233/n3jd85 |
| 997 | 10.21233/n3jd9j |
| 1814 | 10.21233/n3jf27 |
| 2615 | 10.21233/n3jf40 |
| 2007 | 10.21233/n3jj28 |
| 340 | 10.21233/n3jm53 |

| | | | |
|---|---|---|---|
| 1154 | 10.21233/n3jm87 | 3043 | 10.21233/n3xj4f |
| 1551 | 10.21233/n3jm9m | 3858 | 10.21233/n3xj8z |
| 1974 | 10.21233/n3jn0j | 2948 | 10.21233/n3z183 |
| 307 | 10.21233/n3jq3c | 2880 | 10.21233/n3zb6f |
| 2951 | 10.21233/n3jr2b | 3076 | 10.21233/n3zb7t |
| 492 | 10.21233/n3jt55 | 3035 | 10.21233/n3zj54 |
| 1705 | 10.21233/n3jt89 | 40875 | 10.21233/n4bq-p922 |
| 2341 | 10.21233/n3jv10 | 40997 | 10.21233/n4sd-es14 |
| 272 | 10.21233/n3jx1p | 23006 | 10.21233/n4x8-0k74 |
| 2884 | 10.21233/n3k18p | 41366 | 10.21233/n4xw-kd72 |
| 3079 | 10.21233/n3k493 | 21535 | 10.21233/n4y2-m886 |
| 3473 | 10.21233/n3k501 | 20495 | 10.21233/n5d1-mq43 |
| 2943 | 10.21233/n3kr31 | 26446 | 10.21233/n5nj-5q34 |
| 3067 | 10.21233/n3m49d | 4087 | 10.21233/n6bg-kc41 |
| 3561 | 10.21233/n3mn6f | 20167 | 10.21233/n7jz-0g74 |
| 3771 | 10.21233/n3mn7t | 24236 | 10.21233/n87r-7a39 |
| 3125 | 10.21233/n3mv20 | 14999 | 10.21233/n8dz-y640 |
| 3494 | 10.21233/n3mz21 | 24590 | 10.21233/n8w1-xz08 |
| 3054 | 10.21233/n3n48b | 4347 | 10.21233/n8z7-nx69 |
| 2959 | 10.21233/n3nj5b | 4216 | 10.21233/n9a0-fk75 |
| 2892 | 10.21233/n3nv29 | 24467 | 10.21233/n9vy-3c22 |
| 3486 | 10.21233/n3nz2b | 15415 | 10.21233/na70-nc19 |
| 2882 | 10.21233/n3pv17 | 4377 | 10.21233/nbg7-rk14 |
| 3078 | 10.21233/n3pv2m | 26507 | 10.21233/nbnz-tz48 |
| 3471 | 10.21233/n3pz18 | 14637 | 10.21233/nc0w-mk91 |
| 2907 | 10.21233/n3qn37 | 4081 | 10.21233/nc7w-an65 |
| 3029 | 10.21233/n3r20j | 2069 | 10.21233/nda9-aw65 |
| 3493 | 10.21233/n3rr5b | 4173 | 10.21233/ndhk-ws14 |
| 3454 | 10.21233/n3rv3m | 24325 | 10.21233/nep1-3z35 |
| 2925 | 10.21233/n3sf5j | 24900 | 10.21233/nf08-bn61 |
| 2890 | 10.21233/n3sj5k | 24534 | 10.21233/nfah-4t91 |
| 3485 | 10.21233/n3sn5m | 41146 | 10.21233/ngxf-6p79 |
| 3859 | 10.21233/n3sv49 | 4057 | 10.21233/nkzg-w156 |
| 2949 | 10.21233/n3t77j | 15115 | 10.21233/nm9j-w372 |
| 3077 | 10.21233/n3tn4j | 24559 | 10.21233/nmwt-fc53 |
| 3568 | 10.21233/n3v51w | 15366 | 10.21233/nn1h-xp19 |
| 2906 | 10.21233/n3vb7w | 16269 | 10.21233/nn2g-qt23 |
| 3559 | 10.21233/n3w516 | 4169 | 10.21233/nnyx-kj31 |
| 3492 | 10.21233/n3wf8m | 14494 | 10.21233/npm2-8d68 |
| 3051 | 10.21233/n3wj5h | 17859 | 10.21233/npmk-6h16 |
| 3583 | 10.21233/n3wz3w | 4481 | 10.21233/nq7x-mx79 |
| 2956 | 10.21233/n3x203 | 22373 | 10.21233/nqb2-na71 |
| 3484 | 10.21233/n3xf7j | 22693 | 10.21233/nrd3-gj67 |

| | | | | |
|---|---|---|---|---|
| 15772 | 10.21233/nre4-q408 | | 32258 | 10.21233/pgwj-9f54 |
| 16086 | 10.21233/nrq0-e757 | | 24348 | 10.21233/ph7f-8b78 |
| 41201 | 10.21233/nrvm-5m89 | | 25078 | 10.21233/ph7n-8m49 |
| 21838 | 10.21233/nryw-9765 | | 16217 | 10.21233/phdj-h055 |
| 3890 | 10.21233/nsm3-eh40 | | 4103 | 10.21233/phdz-hs08 |
| 25217 | 10.21233/nssh-az33 | | 44941 | 10.21233/phqc-me73 |
| 40999 | 10.21233/nt4c-9462 | | 25344 | 10.21233/pht8-mj95 |
| 22982 | 10.21233/nt73-cr73 | | 4504 | 10.21233/pj9k-qg05 |
| 4502 | 10.21233/nvn7-f141 | | 45097 | 10.21233/pkhx-pf62 |
| 41604 | 10.21233/nwfm-st50 | | 16179 | 10.21233/pkt4-am12 |
| 45629 | 10.21233/nwgq-s472 | | 15371 | 10.21233/pn5c-9x40 |
| 45281 | 10.21233/ny4b-e654 | | 18123 | 10.21233/pnbx-1e72 |
| 15635 | 10.21233/ny7s-f780 | | 14928 | 10.21233/pps9-sn76 |
| 45383 | 10.21233/p0mn-7534 | | 45636 | 10.21233/pr2m-cc37 |
| 41230 | 10.21233/p1h4-4v17 | | 14556 | 10.21233/ps1x-es22 |
| 22786 | 10.21233/p1tb-df93 | | 15811 | 10.21233/psnq-xv46 |
| 4021 | 10.21233/p28f-qv78 | | 14446 | 10.21233/ptaw-4d96 |
| 15250 | 10.21233/p37m-jn82 | | 14768 | 10.21233/ptjw-d867 |
| 41252 | 10.21233/p39d-rd93 | | 40439 | 10.21233/ptq6-q055 |
| 25213 | 10.21233/p3md-9032 | | 14930 | 10.21233/ptx4-dz18 |
| 14529 | 10.21233/p4mc-9f34 | | 4286 | 10.21233/pvn2-a273 |
| 20180 | 10.21233/p4nw-2d17 | | 25386 | 10.21233/pvpd-tm53 |
| 4549 | 10.21233/p4zh-1534 | | 45219 | 10.21233/pwgs-4g85 |
| 24281 | 10.21233/p5jz-zk41 | | 22851 | 10.21233/pxem-ce07 |
| 4055 | 10.21233/p76r-3q63 | | 15342 | 10.21233/pyjt-xa38 |
| 16081 | 10.21233/p7fp-g179 | | 41280 | 10.21233/pys7-k313 |
| 22810 | 10.21233/p83k-v204 | | 4047 | 10.21233/q057-d355 |
| 45200 | 10.21233/p8ac-sf20 | | 19993 | 10.21233/q20r-m406 |
| 24218 | 10.21233/p8jv-nx13 | | 22847 | 10.21233/q2tc-z758 |
| 17340 | 10.21233/p96n-6v19 | | 4523 | 10.21233/q36c-4q49 |
| 24846 | 10.21233/p9bs-0686 | | 21545 | 10.21233/q6p2-kw54 |
| 15649 | 10.21233/paxn-mh26 | | 17292 | 10.21233/q744-pp22 |
| 15746 | 10.21233/pbbn-5h88 | | 42570 | 10.21233/q95g-2x92 |
| 24929 | 10.21233/pbc8-c366 | | 41421 | 10.21233/q9sk-t894 |
| 25287 | 10.21233/pbmw-9s02 | | 45192 | 10.21233/q9wh-hq41 |
| 24626 | 10.21233/pbq8-yy15 | | 14104 | 10.21233/qaht-6745 |
| 16251 | 10.21233/pbwt-b156 | | 14935 | 10.21233/qamn-c462 |
| 15296 | 10.21233/pck3-ct03 | | 25400 | 10.21233/qbf7-hn56 |
| 16271 | 10.21233/pcmd-m560 | | 21770 | 10.21233/qbkz-hz40 |
| 25283 | 10.21233/pe6c-pb93 | | 32260 | 10.21233/qd2q-5c39 |
| 24277 | 10.21233/pegh-gp75 | | 24553 | 10.21233/qdx7-af11 |
| 16063 | 10.21233/pfsb-bc85 | | 4135 | 10.21233/qem1-6q63 |
| 24671 | 10.21233/pgj9-5v59 | | 13029 | 10.21233/qf6p-8j75 |

| | | | |
|---|---|---|---|
| 4209 | 10.21233/qf8f-me80 | 22842 | 10.21233/r8zp-ww92 |
| 22648 | 10.21233/qf8f-wb97 | 4102 | 10.21233/r9pn-tx36 |
| 4470 | 10.21233/qfrx-zf42 | 17338 | 10.21233/r9wb-gz24 |
| 42547 | 10.21233/qg8e-np07 | 22646 | 10.21233/rahy-8590 |
| 45173 | 10.21233/qgv1-sv24 | 4203 | 10.21233/raph-rb39 |
| 45164 | 10.21233/qh50-8208 | 41753 | 10.21233/rarx-fe28 |
| 15377 | 10.21233/qhtr-fk23 | 26425 | 10.21233/rb09-6144 |
| 41183 | 10.21233/qjda-f561 | 15325 | 10.21233/rb6f-1g25 |
| 4085 | 10.21233/qjjs-h634 | 15368 | 10.21233/rc8q-je06 |
| 4116 | 10.21233/qjxn-st60 | 4154 | 10.21233/rce2-en21 |
| 44723 | 10.21233/qk2b-8b38 | 41746 | 10.21233/rd01-0v56 |
| 22904 | 10.21233/qka4-j483 | 22757 | 10.21233/rd1y-0x62 |
| 46479 | 10.21233/qkmf-dk52 | 45716 | 10.21233/rfek-5a50 |
| 41417 | 10.21233/qmke-wg69 | 14650 | 10.21233/rg17-hr80 |
| 14558 | 10.21233/qmy5-xr61 | 42663 | 10.21233/rgr7-f616 |
| 4483 | 10.21233/qp8c-4854 | 15020 | 10.21233/rgvz-gj38 |
| 22344 | 10.21233/qphx-cn83 | 42677 | 10.21233/rhm6-pc30 |
| 4197 | 10.21233/qq1x-zk02 | 15009 | 10.21233/rhs1-wt07 |
| 41476 | 10.21233/qqk2-fy61 | 45324 | 10.21233/rj4f-fn81 |
| 15612 | 10.21233/qrjk-gj85 | 13060 | 10.21233/rja4-q512 |
| 24100 | 10.21233/qrwc-m362 | 4420 | 10.21233/rmkd-2162 |
| 22814 | 10.21233/qtf9-zf76 | 41411 | 10.21233/rmpn-9878 |
| 24300 | 10.21233/qtfz-0018 | 24931 | 10.21233/rn1j-tb02 |
| 41237 | 10.21233/qx7x-1245 | 25404 | 10.21233/rn65-2k02 |
| 24250 | 10.21233/qx94-hx64 | 4490 | 10.21233/rnkp-2723 |
| 4330 | 10.21233/qxv2-w585 | 15680 | 10.21233/rp2w-0753 |
| 40949 | 10.21233/qyrx-tz10 | 15785 | 10.21233/rp8s-2k53 |
| 42685 | 10.21233/qzek-wp34 | 41750 | 10.21233/rpzt-3p58 |
| 24956 | 10.21233/r08t-2829 | 15909 | 10.21233/rqzm-ma95 |
| 15294 | 10.21233/r0cq-9083 | 24574 | 10.21233/rrws-7771 |
| 20397 | 10.21233/r0kz-yt78 | 41521 | 10.21233/rrzj-sp79 |
| 20311 | 10.21233/r0mg-q111 | 24944 | 10.21233/rsdg-ep81 |
| 41339 | 10.21233/r11j-mj44 | 20042 | 10.21233/rt31-zw12 |
| 4058 | 10.21233/r27w-7c93 | 45207 | 10.21233/rt3s-g752 |
| 4532 | 10.21233/r2nw-6c72 | 45327 | 10.21233/rtcq-kn47 |
| 23019 | 10.21233/r37v-gx98 | 22761 | 10.21233/rw25-2v24 |
| 24193 | 10.21233/r3mj-5588 | 25372 | 10.21233/rwdw-1852 |
| 15704 | 10.21233/r4mc-tc72 | 21418 | 10.21233/rwpj-e158 |
| 25127 | 10.21233/r4xz-v815 | 24952 | 10.21233/ryar-ev70 |
| 16238 | 10.21233/r5wm-z686 | 22689 | 10.21233/rybb-mp20 |
| 26516 | 10.21233/r6kr-hg54 | 4134 | 10.21233/ryfa-y280 |
| 4351 | 10.21233/r749-6141 | 45725 | 10.21233/s0pq-6335 |
| 25318 | 10.21233/r7dr-q154 | 4545 | 10.21233/s1ae-ar89 |

| | | | |
|---|---|---|---|
| 40627 | 10.21233/s1va-qn28 | 24110 | 10.21233/sq7c-9a90 |
| 3913 | 10.21233/s1x4-mq32 | 25232 | 10.21233/sqtj-fr11 |
| 46485 | 10.21233/s2j6-5w36 | 15381 | 10.21233/ss13-7n86 |
| 17597 | 10.21233/s3er-nf45 | 17880 | 10.21233/ssg9-gk36 |
| 15284 | 10.21233/s3ma-a925 | 40955 | 10.21233/stpw-ta17 |
| 18106 | 10.21233/s4jg-yh87 | 24497 | 10.21233/sxfw-2b43 |
| 22119 | 10.21233/s4k3-4429 | 24181 | 10.21233/sxhe-sa20 |
| 14370 | 10.21233/s4yh-p034 | 24227 | 10.21233/sxhx-j135 |
| 22738 | 10.21233/s54m-1k85 | 22824 | 10.21233/sy1b-py46 |
| 16073 | 10.21233/s6jx-f115 | 41245 | 10.21233/syjt-3739 |
| 42673 | 10.21233/s6my-6052 | 4002 | 10.21233/szbb-8k05 |
| 45387 | 10.21233/s6wg-4d92 | 18159 | 10.21233/t00e-b616 |
| 4288 | 10.21233/s7rb-ap65 | 22342 | 10.21233/t052-s807 |
| 41220 | 10.21233/s7t1-hx33 | 22674 | 10.21233/t0ea-cd42 |
| 24520 | 10.21233/s85f-3f36 | 4383 | 10.21233/t0t5-0w85 |
| 24837 | 10.21233/s8ht-3z18 | 24108 | 10.21233/t1ba-1q15 |
| 4517 | 10.21233/s8kk-y022 | 4032 | 10.21233/t1n7-cq82 |
| 17383 | 10.21233/s9wz-1g21 | 45406 | 10.21233/t1p9-n569 |
| 25013 | 10.21233/sa5v-5h89 | 41030 | 10.21233/t2fv-s268 |
| 24102 | 10.21233/sa88-mb64 | 41453 | 10.21233/t318-2y34 |
| 14972 | 10.21233/sam2-cd21 | 42712 | 10.21233/t318-n046 |
| 45314 | 10.21233/sb36-c222 | 20223 | 10.21233/t3c4-7x23 |
| 24284 | 10.21233/sbes-jp06 | 45221 | 10.21233/t3tp-6180 |
| 46462 | 10.21233/sc2f-5d04 | 16129 | 10.21233/t4b6-ax18 |
| 24197 | 10.21233/sc9n-2r53 | 24669 | 10.21233/t4ct-mw46 |
| 24578 | 10.21233/scew-1h82 | 24493 | 10.21233/t4n0-6s83 |
| 45168 | 10.21233/scrs-nq43 | 41210 | 10.21233/t55e-8g66 |
| 4118 | 10.21233/sczz-2g33 | 3973 | 10.21233/t57v-4b19 |
| 22763 | 10.21233/sd3z-7v27 | 45144 | 10.21233/t6dq-4b03 |
| 4554 | 10.21233/sdqe-ky81 | 24753 | 10.21233/t7ne-1h11 |
| 3884 | 10.21233/sewp-vp39 | 4117 | 10.21233/tab2-ts04 |
| 25027 | 10.21233/sf2s-x915 | 15462 | 10.21233/tadv-5c57 |
| 4543 | 10.21233/sfsk-ej53 | 24145 | 10.21233/tbav-4v25 |
| 21422 | 10.21233/sg70-ew74 | 21415 | 10.21233/tbhk-kw62 |
| 4150 | 10.21233/shhz-0f61 | 3866 | 10.21233/tcav-fn36 |
| 4005 | 10.21233/sj04-8c24 | 20046 | 10.21233/td9p-m745 |
| 22793 | 10.21233/sjqw-qa18 | 14676 | 10.21233/te1p-he56 |
| 15511 | 10.21233/sk02-jh34 | 24859 | 10.21233/te66-cy52 |
| 17672 | 10.21233/skp4-td45 | 20512 | 10.21233/tegf-sg62 |
| 15713 | 10.21233/sn81-zc38 | 42745 | 10.21233/tekc-rz82 |
| 22753 | 10.21233/snt9-p808 | 17398 | 10.21233/teyv-5z70 |
| 20293 | 10.21233/snyf-b526 | 14510 | 10.21233/tfq7-s408 |
| 4030 | 10.21233/sq17-8k63 | 22955 | 10.21233/tgd4-nk81 |

| | |
|---|---|
| 3872 | 10.21233/tk7w-th94 |
| 26435 | 10.21233/tktg-sh56 |
| 4518 | 10.21233/tkzz-2t77 |
| 14641 | 10.21233/tmdc-mj73 |
| 1147 | 10.21233/tp4a-h605 |
| 4529 | 10.21233/tphv-z798 |
| 45722 | 10.21233/tpvk-wa79 |
| 24112 | 10.21233/tq8x-5z45 |
| 17328 | 10.21233/tr2m-fm40 |
| 25410 | 10.21233/ts6e-h593 |
| 22938 | 10.21233/tt6f-9c78 |
| 15825 | 10.21233/ttaf-pc16 |
| 22771 | 10.21233/ttjd-w232 |
| 15665 | 10.21233/twkp-hz46 |
| 45204 | 10.21233/tx0m-hk41 |
| 15323 | 10.21233/txns-sv58 |
| 19931 | 10.21233/txpr-cb18 |
| 25131 | 10.21233/tz5g-8d59 |
| 4294 | 10.21233/v088-1d37 |
| 17359 | 10.21233/v0wk-b742 |
| 24048 | 10.21233/v1dw-9x16 |
| 15363 | 10.21233/v2kw-6697 |
| 24061 | 10.21233/v3yx-9c11 |
| 22640 | 10.21233/v4mx-q473 |
| 22662 | 10.21233/v7ky-p784 |
| 17711 | 10.21233/v7nq-4w81 |
| 24343 | 10.21233/v8jm-v785 |
| 1582 | 10.21233/v98s-4073 |
| 4405 | 10.21233/v9em-3b09 |
| 25406 | 10.21233/vbwz-tg22 |
| 41401 | 10.21233/vc1n-v894 |
| 14955 | 10.21233/vcma-8k37 |
| 4533 | 10.21233/vdpe-dp57 |
| 15403 | 10.21233/vds5-pf58 |
| 14803 | 10.21233/ve01-y167 |
| 40990 | 10.21233/ve15-mn82 |
| 24279 | 10.21233/veqa-0n52 |
| 14619 | 10.21233/vf6b-8k35 |
| 16016 | 10.21233/vffp-x175 |
| 41460 | 10.21233/vg3e-tx36 |
| 41485 | 10.21233/vgxh-qv84 |
| 4207 | 10.21233/vh8n-2090 |
| 22664 | 10.21233/vhkm-se51 |

| | |
|---|---|
| 4468 | 10.21233/vhxz-9969 |
| 24032 | 10.21233/vj9n-bt57 |
| 22960 | 10.21233/vjdb-ec95 |
| 22845 | 10.21233/vkd7-vh44 |
| 22652 | 10.21233/vmeb-gz58 |
| 15766 | 10.21233/vmwe-g130 |
| 15276 | 10.21233/vmxy-3004 |
| 22745 | 10.21233/vn5f-tz80 |
| 4200 | 10.21233/vpnw-pv89 |
| 24410 | 10.21233/vq9d-x874 |
| 45410 | 10.21233/vqfq-2n70 |
| 16054 | 10.21233/vqnc-sx41 |
| 4020 | 10.21233/vqqw-nc29 |
| 15633 | 10.21233/vr5d-4130 |
| 1782 | 10.21233/vrv9-4t45 |
| 45115 | 10.21233/vs9g-8428 |
| 41770 | 10.21233/vsfm-xz32 |
| 24902 | 10.21233/vtmh-9d28 |
| 15882 | 10.21233/vx1h-dj04 |
| 4056 | 10.21233/vxzv-9k94 |
| 4452 | 10.21233/vy6k-x487 |
| 15022 | 10.21233/w03f-sr85 |
| 14498 | 10.21233/w0ab-4w11 |
| 42671 | 10.21233/w0c2-k512 |
| 45719 | 10.21233/w0j3-9h31 |
| 17296 | 10.21233/w2gd-mz79 |
| 43295 | 10.21233/w4n6-qv07 |
| 3999 | 10.21233/w4z8-3q21 |
| 14564 | 10.21233/w4zt-yj57 |
| 25308 | 10.21233/w5qv-sv92 |
| 15214 | 10.21233/w5rw-qd24 |
| 25157 | 10.21233/w61s-9827 |
| 4069 | 10.21233/w63t-nf61 |
| 3896 | 10.21233/w6j3-gn15 |
| 17324 | 10.21233/w72d-bx06 |
| 41469 | 10.21233/w7bt-xr68 |
| 45194 | 10.21233/w88p-xj69 |
| 41532 | 10.21233/w9z1-fn16 |
| 19933 | 10.21233/watr-af89 |
| 24637 | 10.21233/wb8q-wk43 |
| 4476 | 10.21233/wbfp-6424 |
| 22638 | 10.21233/wc9m-v912 |
| 41233 | 10.21233/wefk-sp33 |

| | | | | |
|---|---|---|---|---|
| 4097 | 10.21233/weqb-ef25 | | 24463 | 10.21233/x7a5-2c36 |
| 4380 | 10.21233/whjz-0k24 | | 45816 | 10.21233/x7et-qe66 |
| 45710 | 10.21233/wjmk-7376 | | 15719 | 10.21233/x7hq-ts05 |
| 4510 | 10.21233/wkz1-sd64 | | 41758 | 10.21233/x8p5-xp81 |
| 41323 | 10.21233/wm29-be77 | | 19820 | 10.21233/x8y7-re14 |
| 15103 | 10.21233/wn4z-7521 | | 24927 | 10.21233/x90a-8v66 |
| 26510 | 10.21233/wnfe-f385 | | 41171 | 10.21233/x91r-1a30 |
| 41587 | 10.21233/wp2z-kt38 | | 4145 | 10.21233/x9b0-z153 |
| 17320 | 10.21233/wq2v-5053 | | 24616 | 10.21233/x9jq-my02 |
| 3882 | 10.21233/wq2x-c027 | | 4332 | 10.21233/xa3s-kj49 |
| 26520 | 10.21233/wqej-0r45 | | 40914 | 10.21233/xa6z-jh17 |
| 14897 | 10.21233/wrb3-qd23 | | 24639 | 10.21233/xadk-1x55 |
| 14959 | 10.21233/wrt5-qc27 | | 24941 | 10.21233/xb5c-wb44 |
| 45389 | 10.21233/ws4c-an72 | | 14512 | 10.21233/xc0y-6b19 |
| 22962 | 10.21233/wvsb-6130 | | 4314 | 10.21233/xe1b-tf19 |
| 21796 | 10.21233/wwa4-0236 | | 32262 | 10.21233/xe3h-wb42 |
| 13073 | 10.21233/wwc5-6x04 | | 22953 | 10.21233/xeew-fr63 |
| 4466 | 10.21233/wwj8-s239 | | 41203 | 10.21233/xeke-2h87 |
| 25382 | 10.21233/wxyd-q154 | | 41506 | 10.21233/xett-n046 |
| 16181 | 10.21233/wy62-rm78 | | 41911 | 10.21233/xfh1-6085 |
| 15385 | 10.21233/wy8b-3e83 | | 41582 | 10.21233/xfrs-q084 |
| 3934 | 10.21233/wyaq-1b97 | | 4391 | 10.21233/xj85-7c26 |
| 25003 | 10.21233/x0ba-j040 | | 22801 | 10.21233/xm08-vp32 |
| 15594 | 10.21233/x0c3-ed21 | | 26439 | 10.21233/xmbg-mz72 |
| 41228 | 10.21233/x0cd-0q06 | | 41269 | 10.21233/xn7x-eb17 |
| 17404 | 10.21233/x0jy-tp30 | | 3870 | 10.21233/xnjz-z143 |
| 24259 | 10.21233/x0z2-py02 | | 21702 | 10.21233/xp8e-bd49 |
| 4304 | 10.21233/x11h-5275 | | 40844 | 10.21233/xpy2-m222 |
| 4059 | 10.21233/x17y-bb18 | | 4042 | 10.21233/xq55-z804 |
| 21706 | 10.21233/x19k-0e45 | | 15904 | 10.21233/xqhj-c782 |
| 15878 | 10.21233/x1bk-ra92 | | 22791 | 10.21233/xraa-re03 |
| 3894 | 10.21233/x1s2-s723 | | 17417 | 10.21233/xrzb-zq82 |
| 15138 | 10.21233/x1ts-qh05 | | 22691 | 10.21233/xsdn-cx54 |
| 18110 | 10.21233/x2md-aj39 | | 41479 | 10.21233/xsk9-bx90 |
| 4024 | 10.21233/x2v2-4v32 | | 24964 | 10.21233/xss2-e959 |
| 14110 | 10.21233/x34r-6x54 | | 15406 | 10.21233/xstm-aq61 |
| 14633 | 10.21233/x35d-q248 | | 24572 | 10.21233/xsxr-ky48 |
| 4041 | 10.21233/x43j-z193 | | 4004 | 10.21233/xw05-mj88 |
| 4083 | 10.21233/x49t-wj37 | | 25402 | 10.21233/xw8c-gq46 |
| 42655 | 10.21233/x4x4-tg29 | | 3937 | 10.21233/xwhx-v671 |
| 3986 | 10.21233/x4xa-1k90 | | 3874 | 10.21233/xwk9-q426 |
| 24593 | 10.21233/x6et-gw35 | | 15244 | 10.21233/xyf7-9v15 |
| 26442 | 10.21233/x6sj-rv93 | | 14491 | 10.21233/xyvh-px20 |

| | |
|---|---|
| 22736 | 10.21233/xyxq-gr38 |
| 15950 | 10.21233/xz5z-pr53 |
| 3928 | 10.21233/y0cr-gc25 |
| 20308 | 10.21233/y0pj-8d71 |
| 22352 | 10.21233/y1ej-xh18 |
| 24563 | 10.21233/y1f4-bz49 |
| 14125 | 10.21233/y28m-nq05 |
| 18051 | 10.21233/y3mq-m120 |
| 24106 | 10.21233/y42f-6218 |
| 3878 | 10.21233/y4e7-me97 |
| 16186 | 10.21233/y4mj-v434 |
| 44951 | 10.21233/y4ss-7811 |
| 41276 | 10.21233/y554-yd92 |
| 25835 | 10.21233/y5bx-3n98 |
| 22238 | 10.21233/y6je-9a17 |
| 4189 | 10.21233/y7b7-5a72 |
| 24759 | 10.21233/ya3m-xh05 |
| 24057 | 10.21233/ya53-0561 |
| 22346 | 10.21233/yb97-px46 |
| 24238 | 10.21233/ycfn-p887 |
| 14289 | 10.21233/yczc-8b58 |
| 4541 | 10.21233/yd8b-wb47 |
| 14925 | 10.21233/ydnc-0495 |
| 4249 | 10.21233/ye9x-z889 |
| 4539 | 10.21233/yeg0-6p54 |
| 24852 | 10.21233/yfcd-n026 |
| 25083 | 10.21233/yfwj-f753 |
| 40569 | 10.21233/ygbz-tf05 |
| 15939 | 10.21233/yhre-qg42 |
| 4187 | 10.21233/yj5d-ej18 |
| 24954 | 10.21233/yjbz-h684 |
| 15209 | 10.21233/yjhg-b425 |
| 41397 | 10.21233/yktz-5q37 |
| 14944 | 10.21233/ym56-bk43 |
| 21915 | 10.21233/ymye-gn96 |
| 41218 | 10.21233/yp3p-5z78 |
| 22931 | 10.21233/yp88-eq30 |
| 24166 | 10.21233/yr7f-7517 |
| 41562 | 10.21233/ysrz-ve64 |
| 15223 | 10.21233/yt10-kv93 |
| 24491 | 10.21233/ytdh-rq41 |
| 24358 | 10.21233/yvzf-7m50 |
| 15205 | 10.21233/yw2g-1t26 |

| | |
|---|---|
| 16175 | 10.21233/ywa8-4h38 |
| 15976 | 10.21233/ywkn-vk84 |
| 41580 | 10.21233/ywn4-f081 |
| 21772 | 10.21233/yxa5-7p09 |
| 14291 | 10.21233/yxsc-gf06 |
| 41121 | 10.21233/yxxp-as21 |
| 45707 | 10.21233/yxzd-xn66 |
| 22812 | 10.21233/yy4y-0p72 |
| 4340 | 10.21233/yys3-qz47 |
| 14792 | 10.21233/yzwy-0p89 |
| 17945 | 10.21233/yzzz-aa38 |
| 17377 | 10.21233/z08c-rb24 |
| 21766 | 10.21233/z256-fh92 |
| 15197 | 10.21233/z377-nz81 |
| 24611 | 10.21233/z3vf-sk02 |
| 17385 | 10.21233/z5j4-8v57 |
| 17366 | 10.21233/z627-yk93 |
| 41159 | 10.21233/z7gy-mw75 |
| 15247 | 10.21233/z7yn-c813 |
| 4323 | 10.21233/z80c-sf19 |
| 17347 | 10.21233/z81b-ch22 |
| 25289 | 10.21233/z8a8-vz94 |
| 4272 | 10.21233/z913-g676 |
| 3887 | 10.21233/z9wq-cc18 |
| 24030 | 10.21233/za1e-fb06 |
| 15730 | 10.21233/zakg-m595 |
| 20287 | 10.21233/zc0w-ge65 |
| 14539 | 10.21233/zcrn-2h47 |
| 4497 | 10.21233/zcst-x145 |
| 15660 | 10.21233/zd0z-ve57 |
| 24854 | 10.21233/zd47-af67 |
| 14941 | 10.21233/zdh7-r634 |
| 45398 | 10.21233/zdqt-kr34 |
| 41926 | 10.21233/zdyp-3t68 |
| 15796 | 10.21233/ze2y-2e90 |
| 15715 | 10.21233/zedd-cm35 |
| 2028 | 10.21233/zesr-6507 |
| 4309 | 10.21233/zf76-3490 |
| 16079 | 10.21233/zh76-mx93 |
| 14799 | 10.21233/zhp9-cm48 |
| 4359 | 10.21233/zhsh-zr85 |
| 14629 | 10.21233/zjqh-j697 |
| 22831 | 10.21233/zjr5-v667 |

| | |
|---|---|
| 4374 | 10.21233/zjtz-k726 |
| 20165 | 10.21233/zk15-wq18 |
| 40791 | 10.21233/zkeb-rj50 |
| 15933 | 10.21233/zkg7-4840 |
| 4505 | 10.21233/zkna-r256 |
| 15902 | 10.21233/zn7m-gc31 |
| 12 | 10.21233/znex-sp94 |
| 15682 | 10.21233/zpe5-s053 |
| 40168 | 10.21233/zq7z-7471 |
| 24078 | 10.21233/zqez-kr13 |
| 41335 | 10.21233/zqj0-ma56 |
| 24149 | 10.21233/zqk8-aa65 |
| 4372 | 10.21233/zqse-ep64 |
| 4507 | 10.21233/zr55-9j43 |
| 42714 | 10.21233/zspv-ta39 |
| 24243 | 10.21233/zvcd-yn53 |
| 13053 | 10.21233/zvn7-m788 |
| 24302 | 10.21233/zw4f-ce58 |
| 13071 | 10.21233/zw81-nc35 |
| 20372 | 10.21233/zwdj-4e50 |
| 4404 | 10.21233/zwpy-vk55 |
| 4498 | 10.21233/zx8x-vj28 |
| 4050 | 10.21233/zxz5-q261 |

---

## Author Comment (AC4)

**Reply to Giesecke**

Laura Schild and Ulrike Herzschuh

**General reply**

Dear Thomas Giesecke,

Thank you for your careful review of our manuscript. While you welcome our effort to conduct a global pollen-based vegetation reconstruction you raise some concerns. We are confident that these can be resolved quickly and easily and will ultimately underline the validity and usability of our reconstruction.

We addressed connectivity of published data to original Neotoma records by adding citations and DOIs and we are keen to add revised chronologies and taxonomic harmonizations to existing Neotoma records. We acknowledge that manuscripts such as ours would not be possible without the considerable effort toward the Neotoma Paleoecological Database and include this in our acknowledgments. Added clarification of how we envision the data set to be used is indeed needed and we will expand on this in the manuscript and provide an additional R script for dynamic rasterization. We agree with the high uncertainty connected to the reconstructions in the Southern Hemisphere and have decided to omit these reconstructions from our data set. Additionally, we will change the name of our calculated source area and will include only reconstructed forest cover in our data set connected to the optimized reconstruction.

We are certain that these adjustments are feasible and have already been able to implement several of them. In our detailed response below, we have addressed your general and specific comments in more detail.

Best regards
Laura Schild and Ulrike Herzschuh

**Detailed replies**

**General comments**

**Original comment**

Before looking more closely at the manuscript I like to extend on the comment by Williams et al. of developing the underlying Legacy Pollen Dataset. Branching of a large dataset from Neotoma

into the Legacy Pollen Dataset with vetting and adding metadata, results in the additional work not being linked back to Neotoma. In the spirit of open science it would be better practice to contribute to Neotoma by uploading additional datasets and correcting or adding metadata. Look up tables for taxonomic harmonization or new chronologies could then be linked to the data in Neotoma. Republishing Neotoma derived data makes scientists using that data ignore the original data source. This also means that Neotoma loses recognition for the work of the data stewards and support for acquiring funding to maintain and develop the database.

**Reply**

We have added citations to Neotoma and constituent data bases and revised our acknowledgements. Additionally we provide a list of Neotoma records included in the LegacyPollen2.0 data set and their DOIs (see attached at the end of this reply). We are also open to expanding Neotoma's records by adding revised chronologies and taxonomic harmonizations, but would require assistance from the Neotoma team to train one of our team members.

Revised text at line 76:

The pollen data synthesis LegacyPollen2.0 (Li et al., 2024b) includes 3728 temporally resolved records (time-series) distributed globally. Data were collected from individual publications and the Neotoma Paleoecology Database which includes data from the European Pollen Database, the QUAVIDA data base for Australasia, the Latin American Pollen Database, the African Pollen Database and the North American Pollen database (Flantua et al., 2015; Fyfe et al., 2009; Giesecke et al., 2014; Lézine et al., 2021; Rowe et al., 2007; Whitmore et al., 2005; Williams et al., 2018). An overview of Neotoma records included in LegacyPollen 2.0 can be found in Table S2.

Flantua, S.G.A., Hooghiemstra, H., Grimm, E.C., Behling, H., Bush, M.B., González-Arango, C., Gosling, W.D., Ledru, M.-P., Lozano-García, S., Maldonado, A., Prieto, A.R., Rull, V., Van Boxel, J.H., 2015. Updated site compilation of the Latin American Pollen Database. Rev. Palaeobot. Palynol. 223, 104–115. https://doi.org/10.1016/j.revpalbo.2015.09.008

Fyfe, R.M., de Beaulieu, J.-L., Binney, H., Bradshaw, R.H.W., Brewer, S., Le Flao, A., Finsinger, W., Gaillard, M.-J., Giesecke, T., Gil-Romera, G., Grimm, E.C., Huntley, B., Kunes, P., Kühl, N., Leydet, M., Lotter, A.F., Tarasov, P.E., Tonkov, S., 2009. The European Pollen Database: past efforts and current activities. Veg. Hist. Archaeobotany 18, 417–424. https://doi.org/10.1007/s00334-009-0215-9

Giesecke, T., Davis, B., Brewer, S., Finsinger, W., Wolters, S., Blaauw, M., de Beaulieu, J.-L., Binney, H., Fyfe, R.M., Gaillard, M.-J., Gil-Romera, G., van der Knaap, W.O., Kuneš, P., Kühl, N., van Leeuwen, J.F.N., Leydet, M., Lotter, A.F., Ortu, E., Semmler, M., Bradshaw, R.H.W., 2014. Towards mapping the late Quaternary vegetation change of Europe. Veg. Hist. Archaeobotany 23, 75–86. https://doi.org/10.1007/s00334-012-0390-y

Lézine, A.-M., Ivory, S.J., Gosling, W.D., Scott, L., 2021. The African Pollen Database (APD) and tracing environmental change: State of the Art, in: Quaternary Vegetation Dynamics. CRC Press.

Rowe, C., Fraser, R., Harrison, S., Dodson, J., 2007. The QUAVIDA synergy: quaternary fire, vegetation and climate change in Australasia. Quat. Int. 167–168, 355–355. https://doi.org/10.1016/j.quaint.2007.04.001

Whitmore, J., Gajewski, K., Sawada, M., Williams, J.W., Shuman, B., Bartlein, P.J., Minckley, T., Viau, A.E., Webb, T., Shafer, S., Anderson, P., Brubaker, L., 2005. Modern pollen data from North America and Greenland for multi-scale paleoenvironmental applications. Quat. Sci. Rev. 24, 1828–1848. https://doi.org/10.1016/j.quascirev.2005.03.005

Williams, J.W., Grimm, E.C., Blois, J.L., Charles, D.F., Davis, E.B., Goring, S.J., Graham, R.W., Smith, A.J., Anderson, M., Arroyo-Cabrales, J., Ashworth, A.C., Betancourt, J.L., Bills, B.W., Booth, R.K., Buckland, P.I., Curry, B.B., Giesecke, T., Jackson, S.T., Latorre, C., Nichols, J., Purdum, T., Roth, R.E., Stryker, M., Takahara, H., 2018. The Neotoma Paleoecology Database, a multiproxy, international, community-curated data resource. Quat. Res. 89, 156–177. https://doi.org/10.1017/qua.2017.105

**Original comment**

Regarding the here presented manuscript by Shild et al. I see several problems and directions of how to address them. I generally agree with the comments by Marie-Jose Gaillard and Michela Mariani regarding technical shortcomings, definition of the source area of pollen (NOT "relevant source area") and the recommendation to restrict a REVEALS application to the northern hemisphere. If attempting to include the southern hemisphere the authors should reduce the unrealistic assumptions as some more information could be gained. Fall speeds could be estimated using the size of pollen grains and initial guesses of RPPEs could have been made by inviting experts working on the different continents and including recent publications on RPPEs.

**Reply**

We will adjust our terminology to refer to the parameter calculated by us as the "80% pollen source area." This term describes the area from which the median relative influx of all taxa reaches 80%. This calculation uses the lake deposition model described in Theuerkauf et al.'s REVEALSinR. Initially, pollen deposition is calculated per taxon from zmax, representing the maximum depth. While this assumes that each taxon deposits all its pollen, it simplifies the reality where pollen can originate from farther distances, and fluvial inputs into lakes are inevitable. Nonetheless, this assumption aligns with REVEALS. Through a stepwise process, the radius around the basin is incrementally expanded, and the relative influx of deposited pollen for each taxon is calculated relative to the total influx at zmax. We define our 80% pollen source radius as the radius at which the median relative influx of all taxa reaches 80%. This calculation primarily serves to provide a sense of the source area's scale to users unfamiliar with pollen data. It underscores the regional nature of lacustrine pollen data and illustrates how lake size influences this source area. We will include this detailed explanation in the manuscript.

We agree that uncertainties in Southern Hemispheric reconstructions are considerable due to limited regional RPP values and have decided to exclude them from our dataset.

We thank you for the suggestion to calculate missing fall speed values. We will implement this for taxa with known pollen grain sizes in the Northern Hemisphere, where other fall speed values are not available.

**Original comment**

My concerns are particularly related to the aim of the authors to publish the data resulting from the analysis. REVEALS results not only provide information on past woodland cover, but also on bias reduced abundance of the major plant genera or families. Given the way the authors used RPPEs and fall speeds for the southern hemisphere such estimates are unlikely to improve the bias in pollen percentage data. However, they may invite researchers not understanding the limitations to misuse such data. This is particularly the case where the authors adjusted RPPEs to obtain an overall better fit with modern tree cover globally. Here the authors admit that the adjusted RPPEs are ecologically meaningless, and I therefore urge the authors not to publish the resulting vegetation reconstructions as they will also be meaningless. If the authors are convinced that the resulting tree cover is meaningful, they could restrict the data publication to that.

**Reply**

We consider the optimized RPP and the related reconstruction not as the most relevant outcome from this manuscript but rather an addition to the reconstruction using a synthesis of published RPP. Therefore, we will remove taxonomic reconstructions and only include forest cover for the optimized RPP data set and move optimized RPP from the manuscript body to the appendix. In the reconstruction using synthesized RPP values, we will highlight which taxa had RPP available and which used standardized RPP values.

As stated above, we intend to remove the Southern Hemisphere from the data set as we agree with the notable uncertainties.

**Original comment**

If a data publication is pursued, the authors should explain in which way they envision the data to be used. Is the attempt to estimate the source area of 80% of pollen to then relate the information on tree cover to that area, and if so how shall that be implemented? Also, how will overlapping areas be treated? Even on the northern hemisphere the gained information is not continues and I therefore wonder how the results will be used in climate modelling. Moreover, I did not find anything in the manuscript of how the obtained data informs on the position of northern or southern forest or tree limits, that are difficult to estimate from percentage pollen data.

**Reply**

We mainly include the calculation of the 80% source area to give an idea of the scale of source area to users not familiar with pollen data. This emphasizes the regional scale of pollen data from large basins, when reconstructions are used at site-level.

Site-wise reconstructions using large lakes are valid alone and their information can be used in gridded versions of this data set as well. We recognize that reconstructions from small lakes and peatland sites should not be used alone as site-wise reconstructions. Our aim is for the data set to be used flexibly, meaning that users can set their own temporal and spatial resolution for rasterization.This is why we did not prepare a set rasterization. To highlight this use case we will provide a script to rasterize the dataset dynamically and classify grid cell reliability by record availability (https://github.com/lauraschild/rasterization/tree/main). Additionally, we will expand on this in our data usability section and clarify how we intend the data set to be used reliably. Small sites and peatland will also receive an additional flag in the data set as "unfit to be used on site-level" .

Even though using REVEALS improves the reconstruction of vegetation compared to pollen data, we will highlight the difficulty of detecting tree lines with compositional data in our manuscript.

**Original comment**

Retaining all the current aspects of the manuscript I would recommend to revise the manuscript and publish it in a disciplinary journal to discuss aspects of this analysis which yield new insights. If continuing with the modern comparison I would suggest to use available surface sample data or core tops marked as modern. Using top samples as old as 500 years in the comparison with modern woodland cover introduces a huge bias.

**Reply**

Thank you for your suggestion. We do believe that it is more useful to validate with the data set instead of using independent surface samples. We agree that 500 years constitute a large age bin, we decided to use this in order to include as many records as possible. We are happy to include a validation using a smaller age bin in the manuscript. You can find a comparison of validations using the different age bins in the figures below. Using smaller bins changes mean absolute errors only marginally.

[Figure]

Fig 1: Site-wise validation of REVEALS reconstructed forest cover using only large lakes for two different age bins as modern forest cover.

[Figure]

Fig 2: Grid-cell based validation of REVEALS reconstructed forest based on aggregated grid cell values for two different age bins as modern forest cover. Errors are smaller compared to site-wise validations (Fig 1) with the exception of Asia. The smaller age bin has virtually no effect on MAE.

**Original comment**

In summary, I don't recommend the publication of the data from the current analysis and suggest restricting the analysis to the northern hemisphere with a publication of the findings in a topical journal.

**Reply**

The reviewers and commenters unanimously express interest in our dataset and acknowledge its potential usefulness. They also recognize the inherent challenges and uncertainties. A data publication allows for the thorough exploration and discussion of these limitations. This accurate documentation is especially necessary in the context of a PhD thesis. Publishing in a topical journal would be counterproductive as it might not allow the space or focus needed for comprehensive discussion of the data. The data have also significantly been altered compared to the original pollen data which justifies the new dataset.

**Detailed comments**

17: The uncertainties introduced here will not make the results invaluable for the investigation of past vegetation dynamics.

We'll rephrase this statement. But we do believe that usefulness is a given as we exclude the Southern Hemisphere and show a clear improvement of forest cover reconstructions compared to Pollen data in our validations.

28: The study is on vegetation cover not climate.

We will clarify that we are not providing past climate data.

34: Fyfe et al. 2009 is not in the reference list.

We added Fyfe et al. 2009 to the reference list and cite it at line 76 (revised).

> The pollen data synthesis LegacyPollen2.0 (Li et al., 2024b) includes 3728 temporally resolved records (time-series) distributed globally. Data were collected from individual publications and the Neotoma Paleoecology Database which includes data from the European Pollen Database, the QUAVIDA data base for Australasia, the Latin American Pollen Database, the African Pollen Database and the North American Pollen database (Flantua et al., 2015; Fyfe et al., 2009; Giesecke et al., 2014; Lézine et al., 2021; Rowe et al., 2007; Whitmore et al., 2005; Williams et al., 2018). An overview of Neotoma records included in LegacyPollen 2.0 can be found in Table S2.

36-38: This is not allowing for a broader but a more restricted application of pollen data as some aspects may not be possible to investigate with a reduced taxonomic depth. Please reword.

We will reword this section to highlight the improved comparability as a trade-off with taxonomic depth.

49: "relevant" pollen source area is well defined, but this is not meant here.

As described above we will change this to be named "80% pollen source area".

50: Nielsen and Odgaard 2010 are a good example of applying the full Landscape Reconstruction Algorithm with a focus on LOVE not so much REVEALS.

We will remove Nielsen and Odgaard from the citations here.

51: "ability" rather than "accuracy"

We will change "ability" to "accuracy".

105: No information is given where the RPPEs are coming from for the southern hemisphere if they are not set to 1 and not used from the northern Hemisphere. Please provide references.

We have decided to omit reconstructions for the Southern Hemisphere from the data set.

106: Fall speeds can be easily estimated based on the size of pollen grains.

Thank you for this suggestion. We will calculate missing fall speed for Northern Hemispheric taxa when grain size data is available.

111: Not "relevant source area"

We changed the name of this value to "80% pollen source area".

118: How was that latitudinal limit used or derived for the past?

It is constant in time.

120: In most situations the forest cover has changed dramatically over the last 500 years. Why did you not use surface sample datasets for this exercise or sites with the top sample marked as modern?

We reduced the size of the age bin for validation to the past 100 years as illustrated in Fig. 1 and 2 above.

168: Chenopodiaceae are an old classification and now included in Amaranthaceae. Why do they have such different RPPEs in Europe?

Chenopodiaceae are included in Amaranthaceae in our input data. They have the same RPP in Europe. The next RPP in this sentence refers to Australia and Oceania (now void). Rephrasing of this sentence will hopefully resolve this confusion.

189: Particularly in the southern hemisphere one would expect that the adjustments lead to improvements. It would be interesting to explore the reasons why that is not the case e.g. the low pollen productivity of many tropical trees. I cannot see that low data availability is a reason here.

We decided to exclude reconstructions from the Southern Hemisphere in our data set.

195: Starting with initial values different from 1 may also be a way to explore this further.

Thank you for this interesting idea. We will try this an include results in the manuscript.

201: It would be interesting to look at these trends for different continents. If a first approximation of forest cover could be made by adjusting arboreal pollen percentages on a continental scale that could be used by modelers as a first order estimate. Again such comparisons should better be done using the modern analogue approach.

We can also supply this trend on a continental scale. That this could provide a rough estimate of potential error and its direction is true and could indeed be useful for modelers. We agree that the modern analogue technique is also a valid reconstruction method for past vegetation. However, REVEALS has been used for several large-scale and continental scale reconstructions in the past, which is why we think it is valid as well (see for example Githumbi et al. 2021, Dawson et al. 2024, Serge et al. 2023).

Dawson, Andria, John W. Williams, Marie-José Gaillard, Simon J. Goring, Behnaz Pirzamanbein, Johan Lindstrom, R. Scott Anderson, u. a. „Holocene Land Cover Change in North America: Continental Trends, Regional Drivers, and Implications for Vegetation-Atmosphere Feedbacks". *Climate of the Past Discussions*, 20. Februar 2024, 1–52. https://doi.org/10.5194/cp-2024-6.

Githumbi, Esther, Ralph Fyfe, Marie-Jose Gaillard, Anna-Kari Trondman, Florence Mazier, Anne-Birgitte Nielsen, Anneli Poska, u. a. „European Pollen-Based REVEALS Land-Cover Reconstructions for the Holocene: Methodology, Mapping and Potentials". *Earth System Science Data* 14, Nr. 4 (8. April 2022): 1581–1619. https://doi.org/10.5194/essd-14-1581-2022.

Serge, M. A., F. Mazier, R. Fyfe, M.-J. Gaillard, T. Klein, A. Lagnoux, D. Galop, u. a. „Testing the Effect of Relative Pollen Productivity on the REVEALS Model: A Validated Reconstruction of Europe-Wide Holocene Vegetation". *Land* 12, Nr. 5 (Mai 2023): 986. https://doi.org/10.3390/land12050986.

289-296: I strongly disagree with these statements. Particularly regarding the northern and southern tree limits the manuscript is not demonstrating how their detection has improved.

We will relativize these statements. However, the validations show a clear improvement of forest cover reconstruction compared to pollen data which, at the very least, now constitutes a **better** reconstruction product. Due to the potentially high temporal and spatial variability of pollen data we cannot (and may not ever) be able to reconstruct true past vegetation (something that a modern analogue is also not able to do). Still we see that we reconstruct modern vegetation better than before.

Data: I looked at the resulting data for a few sites in northern Patagonia where I am familiar with the vegetation. First of all I could not find where the RPPE for Nothofagus comes from. For a site in the steppe (Lago Mosquito) with Austrocedrus on its western shore, REVEALS estimated Holocene values around 60 % forest cover, which is too high. The adjusted run did indeed lower the forest cover to between 40 and 20 %. However, this reconstruction returned the lowest forest cover for the time that Austrocedrus woodlands were present on its western shore and tree cover was highest. This may be due to the fact that Cupressaceae was not part of the taxa used for optimization. Thus, while the amount of forest cover is more realistic in the adjusted run the reconstructed Holocene trend is the reverse of what was interpreted by the authors of the site based data. Moreover, the only tree growing abundantly near the site for the last 3000 years is not part of the reconstructed vegetation in the adjusted run. This is of cause just a single example, but it does illustrate my earlier points.

We have decided to omit reconstructions from the Southern Hemisphere from our data set.

Table S2: List of all Neotoma records included in LegacyPollen 2.0.

| Dataset_ID | doi |
|---|---|
| 4363 | 10.21233/00jj-rn55 |
| 22711 | 10.21233/00kz-c576 |
| 20060 | 10.21233/012s-c321 |
| 3923 | 10.21233/01cq-0v25 |
| 19784 | 10.21233/01fk-3b13 |
| 4551 | 10.21233/0471-wz80 |
| 4427 | 10.21233/04mx-dx20 |
| 19814 | 10.21233/04w0-7r18 |
| 3931 | 10.21233/054g-vs09 |
| 20012 | 10.21233/057s-wr75 |
| 26572 | 10.21233/05v5-d378 |
| 45223 | 10.21233/064r-zv51 |
| 48624 | 10.21233/0672-4W61 |
| 47864 | 10.21233/06YB-XK91 |
| 41514 | 10.21233/07ah-4r18 |
| 24330 | 10.21233/0875-yw50 |
| 4062 | 10.21233/08cj-ht50 |
| 46603 | 10.21233/0A8S-FT20 |
| 26518 | 10.21233/0ajk-6m86 |
| 24867 | 10.21233/0ax2-vj66 |
| 4053 | 10.21233/0bx3-kt48 |
| 16264 | 10.21233/0c9y-af32 |
| 45642 | 10.21233/0d1q-p773 |
| 25481 | 10.21233/0dqt-e594 |
| 17947 | 10.21233/0dt6-3493 |
| 40963 | 10.21233/0e66-gt09 |
| 40579 | 10.21233/0ebh-jc93 |
| 41602 | 10.21233/0ew2-sj35 |
| 24386 | 10.21233/0f9n-x142 |
| 24635 | 10.21233/0gh3-dt80 |
| 15211 | 10.21233/0hkg-af34 |
| 24376 | 10.21233/0jfa-j049 |
| 17354 | 10.21233/0jx5-7t71 |
| 20029 | 10.21233/0kgn-j437 |
| 15132 | 10.21233/0kpg-kx30 |
| 22672 | 10.21233/0kqz-j553 |
| 22820 | 10.21233/0kw2-2170 |
| 40572 | 10.21233/0m9v-mq69 |
| 4378 | 10.21233/0mn2-9w85 |

| | |
|---|---|
| 14177 | 10.21233/0mss-km04 |
| 15967 | 10.21233/0n3x-5g88 |
| 22817 | 10.21233/0nty-tb40 |
| 16222 | 10.21233/0pre-hx18 |
| 41205 | 10.21233/0rb3-az79 |
| 15207 | 10.21233/0rm4-f986 |
| 46447 | 10.21233/0vhe-5b77 |
| 41448 | 10.21233/0vty-vb30 |
| 21812 | 10.21233/0w0p-my34 |
| 42417 | 10.21233/0wn3-0048 |
| 15732 | 10.21233/0wvv-hb57 |
| 3969 | 10.21233/0xa7-w271 |
| 25427 | 10.21233/0xht-hx92 |
| 24673 | 10.21233/0xky-3a80 |
| 19846 | 10.21233/0xmq-rz62 |
| 22049 | 10.21233/0yan-yj67 |
| 15309 | 10.21233/0ywb-hk24 |
| 16193 | 10.21233/0yz5-ad05 |
| 19937 | 10.21233/0zcx-jb34 |
| 45347 | 10.21233/0zgh-6r09 |
| 24207 | 10.21233/0zqa-9w57 |
| 46512 | 10.21233/102F-CT44 |
| 15586 | 10.21233/108x-ge80 |
| 25843 | 10.21233/10kw-5k34 |
| 19995 | 10.21233/11hj-eq70 |
| 41606 | 10.21233/11ph-mc56 |
| 24164 | 10.21233/13fe-7b21 |
| 3985 | 10.21233/140w-er94 |
| 10981 | 10.21233/149e-e689 |
| 25786 | 10.21233/14rr-c071 |
| 21577 | 10.21233/1627-0v53 |
| 41027 | 10.21233/16gz-mt48 |
| 25408 | 10.21233/17jd-z330 |
| 19323 | 10.21233/17ta-5252 |
| 24542 | 10.21233/19wm-rd42 |
| 42539 | 10.21233/1BDV-YC11 |
| 47669 | 10.21233/1DT1-N648 |
| 45103 | 10.21233/1bnj-jq13 |
| 15420 | 10.21233/1ca9-2k79 |
| 24159 | 10.21233/1cnb-m133 |
| 4183 | 10.21233/1csh-9q73 |

| | | | | |
|---|---|---|---|---|
| 4393 | 10.21233/1csm-st79 | | 22805 | 10.21233/2b23-jr62 |
| 22822 | 10.21233/1ef3-zj56 | | 4127 | 10.21233/2bma-0p82 |
| 4436 | 10.21233/1f47-fs20 | | 4011 | 10.21233/2c89-ct63 |
| 24086 | 10.21233/1fdm-x235 | | 26512 | 10.21233/2cxq-pb51 |
| 4239 | 10.21233/1g98-ct07 | | 40566 | 10.21233/2dyz-ac37 |
| 17617 | 10.21233/1gkg-2a65 | | 17717 | 10.21233/2ecb-3b52 |
| 45322 | 10.21233/1gmy-d414 | | 45186 | 10.21233/2fkg-zq79 |
| 18127 | 10.21233/1jd5-2935 | | 4256 | 10.21233/2gfj-a151 |
| 14612 | 10.21233/1ks0-te29 | | 24555 | 10.21233/2hdq-sr31 |
| 24084 | 10.21233/1m09-f532 | | 24950 | 10.21233/2jhm-6w05 |
| 22683 | 10.21233/1mc2-sy03 | | 42568 | 10.21233/2kj1-jp45 |
| 21675 | 10.21233/1pwg-tx22 | | 22669 | 10.21233/2ks0-7d19 |
| 20020 | 10.21233/1r8k-p831 | | 24863 | 10.21233/2kym-q693 |
| 21533 | 10.21233/1sgv-kf95 | | 17995 | 10.21233/2m53-f907 |
| 15359 | 10.21233/1t37-sc11 | | 17342 | 10.21233/2p2t-qn32 |
| 41127 | 10.21233/1tmp-8264 | | 41380 | 10.21233/2p95-3654 |
| 25273 | 10.21233/1v4w-a061 | | 14549 | 10.21233/2q3p-8t07 |
| 24044 | 10.21233/1v7j-4770 | | 17832 | 10.21233/2q7v-1454 |
| 26448 | 10.21233/1vvt-v868 | | 25514 | 10.21233/2rha-px75 |
| 4387 | 10.21233/1vxd-hm05 | | 24665 | 10.21233/2rhp-py68 |
| 24412 | 10.21233/1vz8-h207 | | 15487 | 10.21233/2s88-n175 |
| 14264 | 10.21233/1x1e-yt36 | | 22894 | 10.21233/2sa7-q544 |
| 4542 | 10.21233/1xas-kk11 | | 45385 | 10.21233/2tkh-z079 |
| 24076 | 10.21233/20j0-gk71 | | 4385 | 10.21233/2v5v-v203 |
| 4413 | 10.21233/20v9-7e75 | | 19913 | 10.21233/2vzb-bg61 |
| 4369 | 10.21233/20xs-m168 | | 42645 | 10.21233/2wvj-4383 |
| 41619 | 10.21233/216r-7y34 | | 41197 | 10.21233/2wxy-fm50 |
| 41657 | 10.21233/22yh-z232 | | 19264 | 10.21233/2x38-6j55 |
| 42542 | 10.21233/234z-6f61 | | 4188 | 10.21233/2xk9-gy52 |
| 4472 | 10.21233/26e5-b250 | | 14797 | 10.21233/2xrb-wn37 |
| 46290 | 10.21233/26yg-qw36 | | 3988 | 10.21233/2yt1-f536 |
| 24595 | 10.21233/271c-xj09 | | 24516 | 10.21233/2yv5-0c85 |
| 22732 | 10.21233/272d-vg02 | | 24513 | 10.21233/2yzn-v779 |
| 14639 | 10.21233/27j0-4960 | | 24633 | 10.21233/30sd-fp85 |
| 24195 | 10.21233/28fw-fa08 | | 25765 | 10.21233/30sw-xp94 |
| 4275 | 10.21233/28gf-4635 | | 14794 | 10.21233/314t-a768 |
| 45188 | 10.21233/28zd-fk74 | | 15724 | 10.21233/31ah-kk77 |
| 42421 | 10.21233/298g-zr75 | | 46451 | 10.21233/32mp-be87 |
| 15344 | 10.21233/29yf-f996 | | 40317 | 10.21233/32se-yq93 |
| 47618 | 10.21233/2DQ3-6153 | | 22316 | 10.21233/33pr-sh52 |
| 16209 | 10.21233/2ac6-fd67 | | 24532 | 10.21233/349p-7t96 |
| 4348 | 10.21233/2ahm-tx69 | | 3875 | 10.21233/34k1-e825 |
| 3977 | 10.21233/2arg-mj55 | | 21830 | 10.21233/34yd-xm12 |

| | |
|---|---|
| 21726 | 10.21233/3651-zp81 |
| 20163 | 10.21233/36tb-yy55 |
| 22656 | 10.21233/378b-6586 |
| 45731 | 10.21233/37a7-9h33 |
| 16105 | 10.21233/38ev-4529 |
| 40454 | 10.21233/39k5-ew03 |
| 47595 | 10.21233/3B46-2P68 |
| 46726 | 10.21233/3M1B-H276 |
| 46693 | 10.21233/3NKX-DK96 |
| 22681 | 10.21233/3aeh-hs14 |
| 4415 | 10.21233/3emp-8f59 |
| 40945 | 10.21233/3epn-hs17 |
| 41502 | 10.21233/3fmj-ne26 |
| 13079 | 10.21233/3g0t-1t98 |
| 24155 | 10.21233/3hey-1169 |
| 41589 | 10.21233/3hnm-z839 |
| 24286 | 10.21233/3j3b-qm75 |
| 4501 | 10.21233/3jez-g489 |
| 20065 | 10.21233/3jm4-0r55 |
| 21543 | 10.21233/3jx5-m665 |
| 22358 | 10.21233/3kyv-3k70 |
| 46492 | 10.21233/3mhs-jm74 |
| 40490 | 10.21233/3n0e-0283 |
| 24488 | 10.21233/3nc2-mk25 |
| 15292 | 10.21233/3p7k-v248 |
| 15383 | 10.21233/3pbp-d740 |
| 15201 | 10.21233/3pkd-8685 |
| 4168 | 10.21233/3ppz-sz86 |
| 21917 | 10.21233/3prs-zt23 |
| 45377 | 10.21233/3pzj-0y37 |
| 26319 | 10.21233/3qvt-5e15 |
| 17298 | 10.21233/3r9h-5q68 |
| 4293 | 10.21233/3s1a-0h86 |
| 4092 | 10.21233/3vvs-zb43 |
| 24373 | 10.21233/3y80-dr46 |
| 46464 | 10.21233/3yd7-jc96 |
| 46473 | 10.21233/3zmx-5r44 |
| 24524 | 10.21233/3zs9-w545 |
| 41247 | 10.21233/40c4-rg98 |
| 14948 | 10.21233/414e-k787 |
| 22747 | 10.21233/417v-rg61 |
| 15356 | 10.21233/41a2-9620 |
| 45196 | 10.21233/42f5-8f49 |

| | |
|---|---|
| 24118 | 10.21233/42qq-a716 |
| 17375 | 10.21233/42z3-5s40 |
| 41018 | 10.21233/43vz-0d54 |
| 15831 | 10.21233/44v8-7p56 |
| 45381 | 10.21233/45g4-ry85 |
| 24530 | 10.21233/46s2-ed45 |
| 15003 | 10.21233/48nv-1h83 |
| 47011 | 10.21233/4974-QC65 |
| 24624 | 10.21233/49j0-se42 |
| 4142 | 10.21233/49qd-q283 |
| 47004 | 10.21233/4BAE-XF23 |
| 48649 | 10.21233/4MXX-C134 |
| 46682 | 10.21233/4QB2-3C42 |
| 14270 | 10.21233/4b1p-c888 |
| 4484 | 10.21233/4b53-ep44 |
| 21798 | 10.21233/4b8r-yv06 |
| 4163 | 10.21233/4c5r-s872 |
| 4373 | 10.21233/4cn4-wj75 |
| 4411 | 10.21233/4cn6-3t90 |
| 20156 | 10.21233/4drc-k617 |
| 3975 | 10.21233/4dxg-aq64 |
| 3994 | 10.21233/4dz1-ny70 |
| 24618 | 10.21233/4fc7-8829 |
| 14537 | 10.21233/4fsm-w415 |
| 25767 | 10.21233/4h6f-tx74 |
| 4389 | 10.21233/4j49-be77 |
| 46457 | 10.21233/4jv3-em45 |
| 25007 | 10.21233/4jwe-5f69 |
| 15035 | 10.21233/4k5g-jx08 |
| 45343 | 10.21233/4k7v-pp08 |
| 24495 | 10.21233/4kdn-gp37 |
| 22628 | 10.21233/4m0f-6v70 |
| 4167 | 10.21233/4mp7-wm75 |
| 42722 | 10.21233/4n0x-dj49 |
| 25466 | 10.21233/4prv-bd15 |
| 41224 | 10.21233/4pxw-bm05 |
| 4445 | 10.21233/4r4q-0m97 |
| 4492 | 10.21233/4r7r-4258 |
| 4301 | 10.21233/4rk9-4w51 |
| 42675 | 10.21233/4t3y-1v17 |
| 41222 | 10.21233/4tx7-3a59 |
| 15819 | 10.21233/4vq0-4494 |
| 24356 | 10.21233/4vzb-6v35 |

| | | | |
|---|---|---|---|
| 21700 | 10.21233/4ydc-n204 | 15013 | 10.21233/5mqf-9723 |
| 4025 | 10.21233/4z1g-r833 | 26563 | 10.21233/5mtf-w128 |
| 3951 | 10.21233/4z7r-k050 | 4396 | 10.21233/5pbb-z224 |
| 22660 | 10.21233/4zaw-zb28 | 22883 | 10.21233/5qw0-fa47 |
| 4045 | 10.21233/4zbw-2h35 | 45648 | 10.21233/5rcm-aa91 |
| 41627 | 10.21233/4zww-1615 | 25163 | 10.21233/5rh4-5f23 |
| 15408 | 10.21233/502z-2a36 | 20027 | 10.21233/5rms-aj77 |
| 21816 | 10.21233/5076-5v40 | 25837 | 10.21233/5rsn-hm26 |
| 4524 | 10.21233/5179-fn49 | 4489 | 10.21233/5ry3-kt92 |
| 15580 | 10.21233/51kp-1e53 | 14278 | 10.21233/5tbt-ws69 |
| 4198 | 10.21233/52fh-aq29 | 46459 | 10.21233/5vs6-sq73 |
| 15140 | 10.21233/52qv-3g19 | 3910 | 10.21233/5wm6-fj81 |
| 15781 | 10.21233/52yq-b228 | 43525 | 10.21233/5wvt-cm23 |
| 24499 | 10.21233/536b-mm31 | 24091 | 10.21233/5wwk-v219 |
| 24317 | 10.21233/538q-0k21 | 46449 | 10.21233/5x6t-y404 |
| 41278 | 10.21233/53ja-ce75 | 3930 | 10.21233/5xct-6y32 |
| 4403 | 10.21233/53ys-nt12 | 16009 | 10.21233/5xje-xg37 |
| 14938 | 10.21233/54cn-0794 | 15059 | 10.21233/5yf4-yx81 |
| 15886 | 10.21233/56k5-ng15 | 15919 | 10.21233/5zt0-g990 |
| 4161 | 10.21233/56se-gy95 | 21549 | 10.21233/6032-6p09 |
| 17363 | 10.21233/56yt-6d14 | 24477 | 10.21233/60v1-q871 |
| 22730 | 10.21233/576f-8911 | 214 | 10.21233/612s-t274 |
| 40486 | 10.21233/57g8-9549 | 4186 | 10.21233/6144-2g26 |
| 22087 | 10.21233/58t7-6y73 | 47580 | 10.21233/61K4-QF84 |
| 20281 | 10.21233/596h-wc50 | 24053 | 10.21233/61k6-c182 |
| 42563 | 10.21233/59ks-w850 | 16131 | 10.21233/620y-dx14 |
| 14957 | 10.21233/59wz-qx78 | 15494 | 10.21233/626e-7k44 |
| 48614 | 10.21233/5P77-ZQ05 | 15864 | 10.21233/62ad-2r14 |
| 47817 | 10.21233/5VZR-YH26 | 26315 | 10.21233/62sj-zk58 |
| 48363 | 10.21233/5XXJ-GH64 | 21794 | 10.21233/6344-nb24 |
| 15191 | 10.21233/5aes-5y61 | 14951 | 10.21233/636g-9n86 |
| 45181 | 10.21233/5ayt-5k20 | 13069 | 10.21233/638z-xp12 |
| 15768 | 10.21233/5ct3-qj23 | 15306 | 10.21233/63x2-2k06 |
| 4352 | 10.21233/5d85-6d65 | 22749 | 10.21233/64xb-g250 |
| 14911 | 10.21233/5drb-1282 | 14674 | 10.21233/65ar-gd68 |
| 12023 | 10.21233/5f1j-ch61 | 4170 | 10.21233/65fw-bj58 |
| 4386 | 10.21233/5gt0-z402 | 4181 | 10.21233/66zq-3984 |
| 41098 | 10.21233/5h6b-ds48 | 21551 | 10.21233/675d-as47 |
| 22709 | 10.21233/5hsv-fx24 | 3932 | 10.21233/686e-3e44 |
| 41216 | 10.21233/5j3e-pn82 | 41669 | 10.21233/69nc-5r65 |
| 40795 | 10.21233/5jx5-2077 | 46995 | 10.21233/6G4D-TF71 |
| 24631 | 10.21233/5m52-nd84 | 47506 | 10.21233/6M1T-DA61 |
| 15627 | 10.21233/5m7p-q097 | 4409 | 10.21233/6ag1-k287 |

| | |
|---:|:---|
| 15740 | 10.21233/6am1-sj67 |
| 14680 | 10.21233/6ans-w752 |
| 24871 | 10.21233/6av9-jx79 |
| 4428 | 10.21233/6b8p-ke79 |
| 20515 | 10.21233/6beq-4462 |
| 19909 | 10.21233/6bhj-jv09 |
| 41214 | 10.21233/6ejf-rt19 |
| 15678 | 10.21233/6enp-3753 |
| 14963 | 10.21233/6es0-qa61 |
| 15876 | 10.21233/6fad-4339 |
| 16207 | 10.21233/6ftv-h112 |
| 20022 | 10.21233/6hft-x335 |
| 14682 | 10.21233/6hga-6436 |
| 15630 | 10.21233/6j0s-ag87 |
| 24312 | 10.21233/6j41-er97 |
| 15259 | 10.21233/6j8r-1g90 |
| 22835 | 10.21233/6kgm-zr91 |
| 46280 | 10.21233/6n58-k786 |
| 41273 | 10.21233/6n89-7w11 |
| 24469 | 10.21233/6p3k-yh02 |
| 41491 | 10.21233/6pe7-xh17 |
| 22986 | 10.21233/6ppf-y854 |
| 15734 | 10.21233/6pqr-ch60 |
| 17391 | 10.21233/6pz8-g423 |
| 15944 | 10.21233/6q36-sr27 |
| 3966 | 10.21233/6q48-7975 |
| 20001 | 10.21233/6qkv-y767 |
| 4084 | 10.21233/6skg-fz77 |
| 4213 | 10.21233/6ter-nj90 |
| 41909 | 10.21233/6tgg-3a45 |
| 15410 | 10.21233/6tzm-ms82 |
| 18108 | 10.21233/6vn0-1104 |
| 22678 | 10.21233/6w4z-z869 |
| 15783 | 10.21233/6xsz-gv46 |
| 24844 | 10.21233/6y4e-ym34 |
| 3990 | 10.21233/6ys9-2m10 |
| 4469 | 10.21233/6yzy-mc78 |
| 47571 | 10.21233/70S7-C853 |
| 4162 | 10.21233/7144-8c50 |
| 4362 | 10.21233/719p-q753 |
| 4319 | 10.21233/71yz-am65 |
| 20175 | 10.21233/72na-hf75 |
| 22723 | 10.21233/72t1-ep33 |

| | |
|---:|:---|
| 13055 | 10.21233/7436-ph05 |
| 41426 | 10.21233/76ay-9f41 |
| 16109 | 10.21233/772k-7s80 |
| 21557 | 10.21233/77j4-vn11 |
| 22944 | 10.21233/77r6-x076 |
| 4527 | 10.21233/793y-2a05 |
| 46521 | 10.21233/7E9R-VP36 |
| 46516 | 10.21233/7F5C-W552 |
| 46514 | 10.21233/7M90-YN39 |
| 46698 | 10.21233/7VE7-FM07 |
| 47590 | 10.21233/7WF1-Z606 |
| 4164 | 10.21233/7aj5-ec03 |
| 32426 | 10.21233/7bhz-n612 |
| 15130 | 10.21233/7bsm-hd67 |
| 45092 | 10.21233/7btd-f613 |
| 4225 | 10.21233/7d1m-t529 |
| 41333 | 10.21233/7d1q-6n57 |
| 4382 | 10.21233/7d6z-6j10 |
| 40584 | 10.21233/7dfq-sj65 |
| 22876 | 10.21233/7dvy-1e52 |
| 13097 | 10.21233/7dx1-dt81 |
| 4375 | 10.21233/7emb-xm28 |
| 4296 | 10.21233/7gv9-gq90 |
| 17406 | 10.21233/7hjv-yz55 |
| 45184 | 10.21233/7hq9-gy29 |
| 15153 | 10.21233/7j16-cj20 |
| 24354 | 10.21233/7jm9-qy50 |
| 14995 | 10.21233/7jqq-6h03 |
| 16214 | 10.21233/7k06-yk60 |
| 45111 | 10.21233/7mpt-6z64 |
| 42716 | 10.21233/7mz3-tp55 |
| 46477 | 10.21233/7ntn-k111 |
| 16189 | 10.21233/7ptd-jy37 |
| 15872 | 10.21233/7q6k-xg57 |
| 41419 | 10.21233/7q7t-v963 |
| 4228 | 10.21233/7qh5-4032 |
| 25370 | 10.21233/7qkc-2q02 |
| 24050 | 10.21233/7qsa-xf22 |
| 4143 | 10.21233/7svf-3s04 |
| 3956 | 10.21233/7t0m-0g70 |
| 25230 | 10.21233/7t1e-5e40 |
| 4028 | 10.21233/7tf3-tt41 |
| 25320 | 10.21233/7v49-sj32 |

| | |
|---|---|
| 18104 | 10.21233/7vwz-dk77 |
| 3889 | 10.21233/7xbe-9k30 |
| 15700 | 10.21233/7xwd-g592 |
| 41527 | 10.21233/7ys6-6178 |
| 41305 | 10.21233/808h-p562 |
| 43533 | 10.21233/81cx-my82 |
| 24528 | 10.21233/826g-j091 |
| 47520 | 10.21233/82XH-AC60 |
| 14535 | 10.21233/82xk-w954 |
| 42649 | 10.21233/82zt-4874 |
| 16241 | 10.21233/833d-sq44 |
| 26565 | 10.21233/833s-n396 |
| 4215 | 10.21233/8370-j585 |
| 22767 | 10.21233/84x5-sc43 |
| 3940 | 10.21233/85fe-n850 |
| 25349 | 10.21233/860r-wq86 |
| 16113 | 10.21233/86rn-w288 |
| 46490 | 10.21233/87r0-d068 |
| 25769 | 10.21233/87x4-w578 |
| 41646 | 10.21233/889v-rw31 |
| 24509 | 10.21233/88nk-cv26 |
| 22936 | 10.21233/88tp-8469 |
| 24996 | 10.21233/88vy-c298 |
| 47537 | 10.21233/89TB-QN51 |
| 41185 | 10.21233/89nd-vd67 |
| 357 | 10.21233/89x4-b040 |
| 46798 | 10.21233/8MN4-HF50 |
| 47866 | 10.21233/8P6K-2P36 |
| 47683 | 10.21233/8PFX-TC23 |
| 25414 | 10.21233/8b03-jg14 |
| 14678 | 10.21233/8bnm-7r08 |
| 45379 | 10.21233/8c8d-g988 |
| 15061 | 10.21233/8d1p-vk58 |
| 41138 | 10.21233/8d9a-qw75 |
| 24319 | 10.21233/8da7-b493 |
| 42566 | 10.21233/8djx-f954 |
| 17381 | 10.21233/8dpn-sk91 |
| 24850 | 10.21233/8ej4-q047 |
| 41345 | 10.21233/8f77-mr38 |
| 24414 | 10.21233/8gnn-8h69 |
| 15667 | 10.21233/8k2x-tx03 |
| 15702 | 10.21233/8k8v-tq87 |
| 17304 | 10.21233/8mna-yv25 |

| | |
|---|---|
| 24966 | 10.21233/8mp7-5w52 |
| 14631 | 10.21233/8n1y-1h53 |
| 22367 | 10.21233/8pbq-v525 |
| 4441 | 10.21233/8ph7-sh90 |
| 4147 | 10.21233/8q3r-xv28 |
| 15413 | 10.21233/8q9d-e745 |
| 41181 | 10.21233/8rhg-kr24 |
| 20285 | 10.21233/8s7z-k891 |
| 45298 | 10.21233/8sy1-mp41 |
| 22971 | 10.21233/8t5w-8t95 |
| 15922 | 10.21233/8thg-6w60 |
| 4192 | 10.21233/8vjb-ww55 |
| 22773 | 10.21233/8w6z-7x05 |
| 4355 | 10.21233/8w98-fk51 |
| 24906 | 10.21233/8wkf-an28 |
| 41337 | 10.21233/8xqw-a996 |
| 15750 | 10.21233/8xyz-8886 |
| 3935 | 10.21233/8yhz-ht97 |
| 15709 | 10.21233/8zgx-7708 |
| 41102 | 10.21233/9012-tw18 |
| 4115 | 10.21233/9017-sb34 |
| 22908 | 10.21233/90xr-0g13 |
| 24840 | 10.21233/910a-sq22 |
| 25472 | 10.21233/91x9-5829 |
| 4311 | 10.21233/92cn-k485 |
| 41629 | 10.21233/935p-6302 |
| 20128 | 10.21233/93dm-rp40 |
| 4494 | 10.21233/93pf-0q48 |
| 22775 | 10.21233/93vm-fw90 |
| 45105 | 10.21233/94ex-9803 |
| 16107 | 10.21233/94pt-ya63 |
| 45113 | 10.21233/94x4-yd19 |
| 3501 | 10.21233/956t-ek11 |
| 20304 | 10.21233/959q-kw55 |
| 17873 | 10.21233/95n5-xt46 |
| 17689 | 10.21233/964d-n697 |
| 4397 | 10.21233/96th-h554 |
| 24959 | 10.21233/9703-dw51 |
| 41025 | 10.21233/98b3-ep64 |
| 19842 | 10.21233/98bb-am31 |
| 24229 | 10.21233/98da-sj31 |
| 40642 | 10.21233/98xz-ex68 |
| 4040 | 10.21233/98yq-bv63 |

| | | | |
|---|---|---|---|
| 20171 | 10.21233/99e6-w772 | 46522 | 10.21233/C06D-B533 |
| 22788 | 10.21233/99pm-v778 | 48656 | 10.21233/CA6B-D485 |
| 46680 | 10.21233/9FDV-XK62 | 46730 | 10.21233/CBMY-KS16 |
| 46720 | 10.21233/9Q9F-B230 | 47606 | 10.21233/D41E-T729 |
| 3946 | 10.21233/9a5v-f927 | 46648 | 10.21233/DC6G-0117 |
| 41191 | 10.21233/9b04-ms27 | 48549 | 10.21233/DTAC-5E13 |
| 25188 | 10.21233/9dch-0a13 | 46655 | 10.21233/E28H-2K97 |
| 42161 | 10.21233/9de9-am39 | 46695 | 10.21233/E43F-HA60 |
| 24933 | 10.21233/9dw0-k479 | 46687 | 10.21233/EAP4-RF36 |
| 45713 | 10.21233/9e9x-ff23 | 48072 | 10.21233/ET35-P931 |
| 24327 | 10.21233/9efw-e650 | 47567 | 10.21233/EVQR-8065 |
| 22348 | 10.21233/9ekg-p878 | 48633 | 10.21233/F0X4-1H86 |
| 25234 | 10.21233/9esw-7x29 | 47527 | 10.21233/F7JJ-4741 |
| 14266 | 10.21233/9eyq-n046 | 46998 | 10.21233/F9QR-CV27 |
| 4107 | 10.21233/9j75-xf59 | 47673 | 10.21233/FD92-M524 |
| 4463 | 10.21233/9jtj-a557 | 48621 | 10.21233/FEPD-WS38 |
| 4247 | 10.21233/9kr8-8s34 | 48653 | 10.21233/FHH7-2056 |
| 24339 | 10.21233/9mdv-br36 | 46544 | 10.21233/FJC1-HS98 |
| 41489 | 10.21233/9n20-cc52 | 47584 | 10.21233/FXA3-JW42 |
| 40793 | 10.21233/9na2-9193 | 46645 | 10.21233/GZGE-Q874 |
| 3883 | 10.21233/9p75-y227 | 47685 | 10.21233/H8EZ-FB40 |
| 46292 | 10.21233/9pc1-0436 | 47769 | 10.21233/HBSK-FB02 |
| 19997 | 10.21233/9pdf-7072 | 46691 | 10.21233/HGKE-GM81 |
| 45331 | 10.21233/9qac-3n20 | 46841 | 10.21233/HP7D-SQ69 |
| 17336 | 10.21233/9qv2-xb92 | 46718 | 10.21233/JFQ2-0A48 |
| 24388 | 10.21233/9r0t-vm34 | 46711 | 10.21233/JK1S-VG18 |
| 24539 | 10.21233/9sm0-1589 | 47666 | 10.21233/JVAY-CW81 |
| 24441 | 10.21233/9sy2-nj92 | 47767 | 10.21233/K06E-RW51 |
| 15101 | 10.21233/9vfj-v497 | 48645 | 10.21233/KG60-S564 |
| 46471 | 10.21233/9w17-3k66 | 47557 | 10.21233/KHZZ-1827 |
| 15175 | 10.21233/9wfb-c822 | 47771 | 10.21233/KM96-3E06 |
| 24294 | 10.21233/9x8s-yg68 | 46684 | 10.21233/KSFK-9704 |
| 40448 | 10.21233/9y2h-5560 | 46650 | 10.21233/KW2V-X806 |
| 2393 | 10.21233/9yf5-e111 | 48616 | 10.21233/M5WS-VZ46 |
| 4462 | 10.21233/9yk4-5092 | 46722 | 10.21233/ME1H-RK46 |
| 46523 | 10.21233/ACHA-AZ30 | 48666 | 10.21233/MQMR-FV34 |
| 47259 | 10.21233/AEXN-ZW54 | 46630 | 10.21233/MYT8-WA75 |
| 46709 | 10.21233/ARN6-PX17 | 47597 | 10.21233/N0RB-YF14 |
| 47587 | 10.21233/B785-Q362 | 47621 | 10.21233/N4NQ-VT31 |
| 46611 | 10.21233/BHM0-3F23 | 47504 | 10.21233/N5KS-1603 |
| 46702 | 10.21233/BP0P-DN34 | 47555 | 10.21233/NQHZ-XW16 |
| 48640 | 10.21233/BS0T-C558 | 47968 | 10.21233/PKYN-3N83 |
| 46716 | 10.21233/BTKV-JS74 | 46806 | 10.21233/PTPQ-QD21 |

| | | | |
|---|---|---|---|
| 47497 | 10.21233/RC16-FQ24 | 15947 | 10.21233/a6gw-a462 |
| 47610 | 10.21233/RP9D-AX89 | 2620 | 10.21233/a7g7-e649 |
| 47494 | 10.21233/S6W8-ZW12 | 14614 | 10.21233/a7se-2c50 |
| 47663 | 10.21233/SB5J-M528 | 22880 | 10.21233/a8eg-rs67 |
| 47561 | 10.21233/SF72-GR73 | 4245 | 10.21233/a9hx-nb84 |
| 47814 | 10.21233/SGEJ-6H83 | 41189 | 10.21233/a9sm-ma36 |
| 47535 | 10.21233/SZ9R-Q288 | 26437 | 10.21233/a9x3-nr55 |
| 46689 | 10.21233/T58B-SN37 | 21921 | 10.21233/aa9j-yp34 |
| 47564 | 10.21233/TH2T-9121 | 40321 | 10.21233/abcz-q362 |
| 48668 | 10.21233/TP5S-FZ74 | 24456 | 10.21233/actv-bj33 |
| 48358 | 10.21233/TS9V-G976 | 15286 | 10.21233/ad64-ks29 |
| 47599 | 10.21233/TSWK-BC56 | 15929 | 10.21233/ad82-ja20 |
| 47027 | 10.21233/TVV5-YQ86 | 21836 | 10.21233/adjr-q523 |
| 48658 | 10.21233/V1MX-0B83 | 24089 | 10.21233/adra-c679 |
| 47604 | 10.21233/V3G5-VF48 | 4269 | 10.21233/ads2-bk31 |
| 46535 | 10.21233/V7AX-RV90 | 24620 | 10.21233/aeqn-c862 |
| 46653 | 10.21233/VC03-9R77 | 15624 | 10.21233/aetg-rg52 |
| 46724 | 10.21233/VSJM-6E28 | 45345 | 10.21233/aext-h866 |
| 47602 | 10.21233/VW7G-2J12 | 15492 | 10.21233/ajqc-2p38 |
| 48222 | 10.21233/VYYE-G931 | 16212 | 10.21233/aktj-rk17 |
| 47973 | 10.21233/WZFH-NC77 | 26514 | 10.21233/amj1-fm78 |
| 48611 | 10.21233/X95S-QX49 | 13047 | 10.21233/anc4-fp20 |
| 46672 | 10.21233/X9TF-0J97 | 45391 | 10.21233/ap4h-y067 |
| 47687 | 10.21233/XXP4-Z127 | 4257 | 10.21233/apjc-5g08 |
| 47110 | 10.21233/XZYB-6819 | 15007 | 10.21233/ar0p-hj05 |
| 46659 | 10.21233/YE3D-FJ80 | 14933 | 10.21233/arn2-5250 |
| 47671 | 10.21233/YJWS-JJ96 | 42652 | 10.21233/aryj-gv14 |
| 47502 | 10.21233/YP84-9451 | 42559 | 10.21233/asm4-3r71 |
| 47608 | 10.21233/YVFH-5B51 | 24186 | 10.21233/at68-km35 |
| 47681 | 10.21233/YWDS-WD04 | 4360 | 10.21233/atkr-cs63 |
| 46803 | 10.21233/Z852-D390 | 14923 | 10.21233/av5q-6587 |
| 48361 | 10.21233/ZF2K-PA43 | 16205 | 10.21233/awkk-cx49 |
| 46704 | 10.21233/ZJ6P-GR59 | 4251 | 10.21233/axz3-2923 |
| 46728 | 10.21233/ZNAR-CF17 | 15591 | 10.21233/az3k-5181 |
| 4031 | 10.21233/a033-7j78 | 14684 | 10.21233/azmm-9650 |
| 4326 | 10.21233/a049-xf64 | 3938 | 10.21233/b0fw-9e91 |
| 3865 | 10.21233/a0q7-m868 | 4211 | 10.21233/b18z-e808 |
| 16099 | 10.21233/a2e7-ky61 | 2899 | 10.21233/b2cd-db08 |
| 17285 | 10.21233/a40y-j676 | 22862 | 10.21233/b2sa-vs04 |
| 23015 | 10.21233/a4j6-w793 | 4479 | 10.21233/b3av-3w75 |
| 21895 | 10.21233/a56z-1027 | 24161 | 10.21233/b40a-qt66 |
| 21919 | 10.21233/a5et-h994 | 4548 | 10.21233/b54b-nw94 |
| 20160 | 10.21233/a5ym-ww41 | 45728 | 10.21233/b5h9-pm78 |

| | |
|---|---|
| 46466 | 10.21233/b62t-vg05 |
| 15793 | 10.21233/b6rr-8867 |
| 24993 | 10.21233/b706-qn93 |
| 4320 | 10.21233/b71k-3606 |
| 41310 | 10.21233/b7c6-fn54 |
| 46483 | 10.21233/b7g3-vh04 |
| 21603 | 10.21233/b9ma-cv72 |
| 15866 | 10.21233/b9nq-n244 |
| 22687 | 10.21233/bax7-sx63 |
| 4082 | 10.21233/bc5s-nr94 |
| 16275 | 10.21233/bcqk-fe25 |
| 41762 | 10.21233/bdex-mm22 |
| 4241 | 10.21233/begd-z059 |
| 15641 | 10.21233/bepp-gp08 |
| 45704 | 10.21233/bfpc-jd87 |
| 4392 | 10.21233/bgsv-qs98 |
| 15106 | 10.21233/bhxy-pk21 |
| 25839 | 10.21233/bjeh-fe46 |
| 15752 | 10.21233/bjqy-em56 |
| 45639 | 10.21233/bkjz-st84 |
| 45329 | 10.21233/bkyg-et47 |
| 20194 | 10.21233/bm1c-7055 |
| 4443 | 10.21233/bp9d-qh60 |
| 45137 | 10.21233/bpfs-3r48 |
| 22969 | 10.21233/bppw-6791 |
| 21913 | 10.21233/bqfq-4x49 |
| 15392 | 10.21233/bqja-m092 |
| 24116 | 10.21233/brk6-dd51 |
| 3945 | 10.21233/brzb-ev42 |
| 21790 | 10.21233/bsv3-k317 |
| 22630 | 10.21233/bswe-2h73 |
| 20316 | 10.21233/bt6r-4598 |
| 15952 | 10.21233/bt73-3f31 |
| 16083 | 10.21233/bt8w-b647 |
| 16115 | 10.21233/bt94-dq60 |
| 30184 | 10.21233/btw4-bh35 |
| 25221 | 10.21233/bv3d-g466 |
| 41212 | 10.21233/bv7h-dw44 |
| 21537 | 10.21233/bvws-hb34 |
| 17322 | 10.21233/bx4k-0v94 |
| 24321 | 10.21233/bx5f-9y20 |
| 24518 | 10.21233/bxa2-pc38 |
| 3984 | 10.21233/bz4w-kv74 |

| | |
|---|---|
| 42557 | 10.21233/bzbr-4h48 |
| 14134 | 10.21233/bzy2-pe09 |
| 4078 | 10.21233/c09q-sa29 |
| 46488 | 10.21233/c0s8-4k74 |
| 4418 | 10.21233/c1jz-a477 |
| 24645 | 10.21233/c2gf-2j68 |
| 15146 | 10.21233/c3aa-h411 |
| 41537 | 10.21233/c46j-ka17 |
| 16065 | 10.21233/c49c-z514 |
| 45190 | 10.21233/c52w-kn70 |
| 3876 | 10.21233/c5as-w284 |
| 4338 | 10.21233/c60e-p241 |
| 14889 | 10.21233/c69e-2m65 |
| 41308 | 10.21233/c6jy-0927 |
| 15696 | 10.21233/c6pe-5h03 |
| 17344 | 10.21233/c737-1334 |
| 20050 | 10.21233/c7eh-we69 |
| 4450 | 10.21233/c7fs-kf76 |
| 20034 | 10.21233/c8mz-0061 |
| 3900 | 10.21233/c9ps-vy90 |
| 40803 | 10.21233/ca85-bh57 |
| 3868 | 10.21233/caff-6e80 |
| 17371 | 10.21233/cbz6-xb88 |
| 21818 | 10.21233/cc4x-wz63 |
| 45393 | 10.21233/ccqb-dc70 |
| 21828 | 10.21233/cdq2-qq32 |
| 42687 | 10.21233/cecw-sb96 |
| 4283 | 10.21233/cg36-ty75 |
| 41755 | 10.21233/cg7r-v764 |
| 45202 | 10.21233/ch0p-9286 |
| 41614 | 10.21233/ch26-4s93 |
| 24481 | 10.21233/chjr-t175 |
| 20131 | 10.21233/cjmz-5m20 |
| 17352 | 10.21233/ckz2-ef80 |
| 41187 | 10.21233/cm96-6e64 |
| 20306 | 10.21233/cp60-cs35 |
| 1647 | 10.21233/cpgt-b026 |
| 41435 | 10.21233/cqkb-rx04 |
| 41382 | 10.21233/cqtf-2866 |
| 45698 | 10.21233/cr6x-be86 |
| 4308 | 10.21233/cr8f-k669 |
| 4437 | 10.21233/ct1z-ve54 |
| 15396 | 10.21233/ctpk-r067 |

| | | | |
|---|---|---|---|
| 45655 | 10.21233/cvyt-k021 | 25110 | 10.21233/dvxn-9g46 |
| 25018 | 10.21233/cwpd-7j59 | 24059 | 10.21233/dwfv-m396 |
| 4120 | 10.21233/cx8d-d024 | 4431 | 10.21233/dwrs-9h77 |
| 16224 | 10.21233/cxev-3204 | 4205 | 10.21233/dx45-we16 |
| 41351 | 10.21233/cxq0-9608 | 25285 | 10.21233/dz7x-af16 |
| 24363 | 10.21233/cxrf-mp20 | 19834 | 10.21233/dzp4-pv02 |
| 16254 | 10.21233/cznd-fg62 | 45101 | 10.21233/dztb-ga02 |
| 16133 | 10.21233/d01t-8d61 | 24641 | 10.21233/dzzf-f817 |
| 15588 | 10.21233/d06n-bw32 | 14635 | 10.21233/e00p-4q45 |
| 41207 | 10.21233/d148-6h03 | 15646 | 10.21233/e0q4-t414 |
| 22926 | 10.21233/d2tj-p590 | 4376 | 10.21233/e1h1-x270 |
| 4482 | 10.21233/d47b-gd35 | 19817 | 10.21233/e2ez-ep96 |
| 15187 | 10.21233/d4xx-zn44 | 16199 | 10.21233/e2gb-7a78 |
| 24454 | 10.21233/d6jx-vy62 | 21774 | 10.21233/e2hw-ah76 |
| 20279 | 10.21233/d73q-6954 | 17402 | 10.21233/e34c-fp95 |
| 45316 | 10.21233/d7f2-p170 | 24600 | 10.21233/e387-w812 |
| 20298 | 10.21233/d917-2x13 | 3995 | 10.21233/e3cd-bd97 |
| 41329 | 10.21233/d9jn-js49 | 45349 | 10.21233/e3mw-tc57 |
| 26433 | 10.21233/daff-1123 | 26429 | 10.21233/e4p1-1n07 |
| 41631 | 10.21233/dayz-mr34 | 14648 | 10.21233/e4z3-5t61 |
| 26608 | 10.21233/dbv5-q164 | 24547 | 10.21233/e523-zp67 |
| 15298 | 10.21233/dcm2-wn02 | 24869 | 10.21233/e5sr-kd56 |
| 41928 | 10.21233/dcqy-3516 | 4417 | 10.21233/e5sw-b991 |
| 41326 | 10.21233/ddey-hy33 | 41653 | 10.21233/e7j2-b618 |
| 18100 | 10.21233/dds5-cz13 | 22803 | 10.21233/e7v0-v862 |
| 24097 | 10.21233/dexk-db41 | 4199 | 10.21233/e8bv-7e06 |
| 3991 | 10.21233/dfsr-fr27 | 14523 | 10.21233/e93w-j223 |
| 15764 | 10.21233/dfzy-sd04 | 41760 | 10.21233/eakm-1a87 |
| 42001 | 10.21233/dge8-0c88 | 524 | 10.21233/ecm1-wf91 |
| 4367 | 10.21233/dj3h-f670 | 4402 | 10.21233/ecn2-ak50 |
| 40799 | 10.21233/djae-0e22 | 20058 | 10.21233/ef79-v217 |
| 24190 | 10.21233/djqf-s823 | 15189 | 10.21233/efy3-6z86 |
| 45117 | 10.21233/dm65-7k56 | 21665 | 10.21233/egh5-v320 |
| 21807 | 10.21233/dmvc-9j61 | 24904 | 10.21233/ehcd-fw95 |
| 24507 | 10.21233/dngj-0r02 | 26499 | 10.21233/ehet-7b80 |
| 24046 | 10.21233/dnjr-rs32 | 41525 | 10.21233/ehmm-vm11 |
| 21600 | 10.21233/dpfx-7555 | 16197 | 10.21233/ehwk-t858 |
| 14624 | 10.21233/dpjk-nw36 | 4401 | 10.21233/ehyh-vn64 |
| 45107 | 10.21233/dptw-0b10 | 41177 | 10.21233/ejfb-t464 |
| 41298 | 10.21233/dq5k-vq16 | 41864 | 10.21233/ek47-9m35 |
| 4558 | 10.21233/dqcs-v558 | 4287 | 10.21233/emr2-ry49 |
| 22988 | 10.21233/dsrz-5k22 | 22650 | 10.21233/enk7-n978 |
| 40452 | 10.21233/dtnh-n631 | 15942 | 10.21233/enmq-kb54 |

| | |
|---|---|
| 22923 | 10.21233/enr7-wa61 |
| 4148 | 10.21233/enzj-0b81 |
| 4442 | 10.21233/epfd-de13 |
| 45701 | 10.21233/epv2-3m87 |
| 40797 | 10.21233/eqhn-8105 |
| 21792 | 10.21233/eqv0-5067 |
| 25845 | 10.21233/ergd-e171 |
| 22826 | 10.21233/es4m-ew33 |
| 4388 | 10.21233/eye6-zs61 |
| 15778 | 10.21233/ez69-yw85 |
| 4258 | 10.21233/eze7-8418 |
| 22992 | 10.21233/f06a-v762 |
| 22910 | 10.21233/f0a8-d318 |
| 40951 | 10.21233/f1r6-n030 |
| 40443 | 10.21233/f2ee-nj62 |
| 4095 | 10.21233/f2r9-2c53 |
| 20289 | 10.21233/f3ba-6s09 |
| 4012 | 10.21233/f3pc-ac31 |
| 24865 | 10.21233/f3pg-3x04 |
| 15221 | 10.21233/f3va-hs11 |
| 25511 | 10.21233/f533-gn18 |
| 16103 | 10.21233/f5x5-y496 |
| 14946 | 10.21233/f630-mb35 |
| 22849 | 10.21233/f70k-ey12 |
| 45395 | 10.21233/f7qr-f244 |
| 25776 | 10.21233/f86j-0362 |
| 4151 | 10.21233/f903-f525 |
| 4364 | 10.21233/fb1t-1r29 |
| 41285 | 10.21233/fb4s-ds71 |
| 16278 | 10.21233/fb6e-1p18 |
| 15399 | 10.21233/fbm9-6394 |
| 4285 | 10.21233/fbsg-8t36 |
| 4063 | 10.21233/fbw1-p392 |
| 15738 | 10.21233/fcvg-km58 |
| 14560 | 10.21233/fd56-pw63 |
| 41766 | 10.21233/fds1-dt10 |
| 3879 | 10.21233/ff82-3p60 |
| 15136 | 10.21233/ffzs-p571 |
| 26431 | 10.21233/fga9-z080 |
| 22397 | 10.21233/fgen-7074 |
| 4423 | 10.21233/fhj7-pj76 |
| 15142 | 10.21233/fhxr-x596 |
| 41034 | 10.21233/fjnd-v956 |

| | |
|---|---|
| 41764 | 10.21233/fka8-9069 |
| 24465 | 10.21233/fkht-hr86 |
| 24341 | 10.21233/fkrs-q995 |
| 15925 | 10.21233/fm9a-bm03 |
| 24861 | 10.21233/fnc3-w873 |
| 17400 | 10.21233/fnwr-vt02 |
| 45210 | 10.21233/frtz-w434 |
| 14622 | 10.21233/fs4c-mv83 |
| 4316 | 10.21233/fs8n-ec44 |
| 22644 | 10.21233/fs93-kd09 |
| 25340 | 10.21233/fssx-my06 |
| 16111 | 10.21233/fvhw-s571 |
| 16075 | 10.21233/fvkb-9e77 |
| 41497 | 10.21233/fw3s-ft98 |
| 24055 | 10.21233/fw4s-2605 |
| 22734 | 10.21233/fwh3-s335 |
| 4449 | 10.21233/fwvv-4v86 |
| 24937 | 10.21233/fx7m-a985 |
| 42679 | 10.21233/fy68-3t56 |
| 22101 | 10.21233/g0kj-nd32 |
| 16061 | 10.21233/g17d-2n36 |
| 4259 | 10.21233/g1np-hb58 |
| 42681 | 10.21233/g1wf-0z38 |
| 22667 | 10.21233/g20r-jy78 |
| 15394 | 10.21233/g3pw-gj52 |
| 13051 | 10.21233/g3qv-vs64 |
| 22828 | 10.21233/g4aa-8505 |
| 19901 | 10.21233/g546-j142 |
| 21580 | 10.21233/g56e-8205 |
| 40519 | 10.21233/g5jk-1e91 |
| 15288 | 10.21233/g6vy-4j29 |
| 3898 | 10.21233/g6z1-d641 |
| 20018 | 10.21233/g7c8-hk59 |
| 14525 | 10.21233/g7k0-8d93 |
| 41384 | 10.21233/g7x4-c998 |
| 40577 | 10.21233/g8pp-n508 |
| 40666 | 10.21233/g9hf-1t92 |
| 25435 | 10.21233/g9v8-1z05 |
| 4279 | 10.21233/gar4-ty16 |
| 15099 | 10.21233/gb30-9e28 |
| 15762 | 10.21233/gd2j-5k10 |
| 22623 | 10.21233/gdbt-zp90 |
| 21541 | 10.21233/gdtp-j631 |

| | |
|---|---|
| 24216 | 10.21233/ge5b-h872 |
| 45351 | 10.21233/geb4-ch43 |
| 22706 | 10.21233/gfg1-ee16 |
| 4137 | 10.21233/gg82-s427 |
| 42683 | 10.21233/ggnn-mk26 |
| 25275 | 10.21233/gh60-9e82 |
| 22636 | 10.21233/gjbn-9951 |
| 25551 | 10.21233/gjvj-6s89 |
| 45695 | 10.21233/gmdn-sh18 |
| 46481 | 10.21233/gmfj-ka45 |
| 22658 | 10.21233/gmht-f823 |
| 22784 | 10.21233/gmhy-nn21 |
| 15955 | 10.21233/gmm9-y553 |
| 23004 | 10.21233/gmp0-vr07 |
| 26505 | 10.21233/gn10-hw94 |
| 25470 | 10.21233/gnb0-9516 |
| 3893 | 10.21233/gp8m-6z07 |
| 14884 | 10.21233/gpa8-an56 |
| 43513 | 10.21233/gpyr-c840 |
| 25368 | 10.21233/gq49-ep38 |
| 22874 | 10.21233/grf4-q917 |
| 46469 | 10.21233/grh1-ap74 |
| 24080 | 10.21233/gsax-vn12 |
| 21938 | 10.21233/gt52-mt02 |
| 24522 | 10.21233/gtyp-4x27 |
| 24298 | 10.21233/gve7-qt51 |
| 14276 | 10.21233/gvk8-r723 |
| 15032 | 10.21233/gw2b-x179 |
| 4410 | 10.21233/gw72-9x54 |
| 41456 | 10.21233/gwdm-cz78 |
| 40450 | 10.21233/gwh3-1a93 |
| 25763 | 10.21233/gwnk-ry08 |
| 4171 | 10.21233/gwts-9s82 |
| 16124 | 10.21233/gxbr-bj21 |
| 14521 | 10.21233/gxdp-2n47 |
| 15935 | 10.21233/gxny-ma10 |
| 4461 | 10.21233/gy1d-0c88 |
| 24570 | 10.21233/gy4b-s732 |
| 17368 | 10.21233/gyh8-pb41 |
| 4358 | 10.21233/gzqy-9x14 |
| 24221 | 10.21233/h1k4-fg42 |
| 20024 | 10.21233/h1s8-8h96 |
| 4531 | 10.21233/h1sz-c208 |

| | |
|---|---|
| 4136 | 10.21233/h27c-dp97 |
| 22942 | 10.21233/h3bs-6188 |
| 24039 | 10.21233/h3rm-sh19 |
| 40992 | 10.21233/h3x5-xz62 |
| 24479 | 10.21233/h4cf-f980 |
| 40940 | 10.21233/h4gb-ka72 |
| 20295 | 10.21233/h5hf-4g09 |
| 4485 | 10.21233/h5wh-mm89 |
| 22890 | 10.21233/h657-1t24 |
| 41543 | 10.21233/h686-6m80 |
| 17713 | 10.21233/h72j-y982 |
| 4473 | 10.21233/h75a-p298 |
| 4424 | 10.21233/h7g7-g111 |
| 22797 | 10.21233/h8w0-4796 |
| 46131 | 10.21233/h94p-kc68 |
| 4317 | 10.21233/h9er-kf81 |
| 15327 | 10.21233/h9hp-vp69 |
| 22795 | 10.21233/h9zx-zt71 |
| 15331 | 10.21233/haqy-4x05 |
| 22807 | 10.21233/hbv9-4k46 |
| 4184 | 10.21233/hbws-c483 |
| 14274 | 10.21233/hca1-3g26 |
| 41842 | 10.21233/hcna-tv25 |
| 3927 | 10.21233/hd31-6942 |
| 15081 | 10.21233/hdeg-rn31 |
| 16236 | 10.21233/hf5z-sc17 |
| 4421 | 10.21233/hfmb-2e61 |
| 24188 | 10.21233/hfzm-9410 |
| 25384 | 10.21233/hg69-1m77 |
| 22976 | 10.21233/hga1-m504 |
| 24305 | 10.21233/hgah-yb23 |
| 26501 | 10.21233/hgw1-8c70 |
| 4435 | 10.21233/hhdq-x585 |
| 24183 | 10.21233/hhxv-jy21 |
| 4086 | 10.21233/hk80-jj60 |
| 15417 | 10.21233/hkn5-4a59 |
| 21553 | 10.21233/hmq8-hm34 |
| 22400 | 10.21233/hnsc-1p36 |
| 15302 | 10.21233/hpc1-5c65 |
| 44896 | 10.21233/hpgt-nm06 |
| 41264 | 10.21233/hqp0-6z14 |
| 39749 | 10.21233/hr41-mn70 |
| 15776 | 10.21233/hrkc-qm74 |

| | | | |
|---|---|---|---|
| 20014 | 10.21233/hsdf-hx86 | 22642 | 10.21233/jm78-za93 |
| 41235 | 10.21233/htm2-cw28 | 15931 | 10.21233/jmze-9037 |
| 15157 | 10.21233/htpm-ac79 | 4156 | 10.21233/jnne-vy86 |
| 21893 | 10.21233/hvvv-sy31 | 21788 | 10.21233/jnyr-4x96 |
| 15074 | 10.21233/hwnh-8509 | 41467 | 10.21233/jpfc-1908 |
| 1578 | 10.21233/hxd2-5546 | 19967 | 10.21233/jqgd-sd88 |
| 24157 | 10.21233/hxme-qv81 | 41389 | 10.21233/jqs1-hj70 |
| 22330 | 10.21233/hyt5-1v77 | 24042 | 10.21233/jrvd-m892 |
| 17326 | 10.21233/j01p-4677 | 24526 | 10.21233/jsc2-rn41 |
| 23008 | 10.21233/j0sm-9z75 | 16177 | 10.21233/jsh7-n919 |
| 14645 | 10.21233/j0zd-6071 | 24034 | 10.21233/jsj1-4861 |
| 17396 | 10.21233/j12t-e851 | 40976 | 10.21233/jvxc-rw58 |
| 24143 | 10.21233/j1k9-2922 | 24461 | 10.21233/jvxq-h196 |
| 14410 | 10.21233/j2hz-1p75 | 22973 | 10.21233/jvze-zx40 |
| 14626 | 10.21233/j36k-8t44 | 46445 | 10.21233/jw93-c102 |
| 40441 | 10.21233/j3jg-qz14 | 42555 | 10.21233/jwt1-1r97 |
| 24848 | 10.21233/j4jk-zw12 | 22964 | 10.21233/jxea-ce51 |
| 22714 | 10.21233/j5ga-dr77 | 21606 | 10.21233/jxj8-mf14 |
| 24360 | 10.21233/j694-t641 | 20188 | 10.21233/jyp9-k385 |
| 41499 | 10.21233/j6p7-zb53 | 22698 | 10.21233/jyz6-4566 |
| 41226 | 10.21233/j71m-0095 | 4306 | 10.21233/k0vt-tc67 |
| 15290 | 10.21233/j8q3-sm07 | 4010 | 10.21233/k0xz-ct85 |
| 24390 | 10.21233/j97k-ds95 | 23000 | 10.21233/k1e4-xe95 |
| 24256 | 10.21233/j9as-rt69 | 40111 | 10.21233/k2nt-2b74 |
| 40861 | 10.21233/ja1n-6b46 | 24857 | 10.21233/k38b-1167 |
| 4555 | 10.21233/jact-n751 | 4398 | 10.21233/k41y-rh60 |
| 15789 | 10.21233/jc66-m526 | 15489 | 10.21233/k4wy-0k33 |
| 20068 | 10.21233/jcdm-8a49 | 4467 | 10.21233/k4zb-ze21 |
| 3926 | 10.21233/jdgx-9y34 | 15892 | 10.21233/k58q-9d96 |
| 24576 | 10.21233/jdtq-9666 | 42692 | 10.21233/k65h-1n80 |
| 10967 | 10.21233/je2r-dk10 | 21762 | 10.21233/k9vz-2659 |
| 25477 | 10.21233/jec9-1e73 | 14554 | 10.21233/ka3z-1h69 |
| 4017 | 10.21233/jf00-3j79 | 24842 | 10.21233/kakk-r623 |
| 4138 | 10.21233/jgh0-2062 | 34553 | 10.21233/kb77-8r67 |
| 3892 | 10.21233/jgxs-ey74 | 19266 | 10.21233/kcep-vj71 |
| 3941 | 10.21233/jhaq-tk78 | 4114 | 10.21233/kdkc-ke07 |
| 15597 | 10.21233/jhh6-7q75 | 41320 | 10.21233/ke7a-vf58 |
| 16184 | 10.21233/jhwb-y436 | 21744 | 10.21233/kefv-nq34 |
| 4493 | 10.21233/jjat-b004 | 22892 | 10.21233/keqf-m247 |
| 41283 | 10.21233/jjez-hm35 | 25468 | 10.21233/kf0z-p180 |
| 4071 | 10.21233/jkhn-6f57 | 41399 | 10.21233/kfbq-jw42 |
| 41163 | 10.21233/jky8-1c56 | 24323 | 10.21233/kfxk-5f39 |
| 40958 | 10.21233/jm6d-3395 | 19906 | 10.21233/khvf-ha85 |

| | |
|---|---|
| 15185 | 10.21233/kkjd-pf71 |
| 14527 | 10.21233/kkn8-9202 |
| 3953 | 10.21233/kmvh-v620 |
| 45170 | 10.21233/kn4a-fd55 |
| 42689 | 10.21233/kngt-v975 |
| 39364 | 10.21233/kpgn-4r23 |
| 15350 | 10.21233/kph1-ps34 |
| 15970 | 10.21233/kq0a-zm89 |
| 17419 | 10.21233/krp4-jk87 |
| 17387 | 10.21233/krw5-yg76 |
| 40484 | 10.21233/kt03-8t61 |
| 20183 | 10.21233/kts9-1316 |
| 4174 | 10.21233/kvx4-xd52 |
| 17357 | 10.21233/kw77-m233 |
| 15814 | 10.21233/kxff-0a77 |
| 24622 | 10.21233/kzqk-jj13 |
| 15379 | 10.21233/m06f-by70 |
| 41634 | 10.21233/m0yb-sp15 |
| 15242 | 10.21233/m16f-gh96 |
| 24939 | 10.21233/m1n7-n027 |
| 3869 | 10.21233/m1s8-qh32 |
| 14997 | 10.21233/m1ty-fz61 |
| 41243 | 10.21233/m24p-3135 |
| 13032 | 10.21233/m2a2-qq61 |
| 40574 | 10.21233/m2sf-vp33 |
| 25455 | 10.21233/m32v-dn19 |
| 22966 | 10.21233/m3rj-t898 |
| 41161 | 10.21233/m3wr-jd95 |
| 25186 | 10.21233/m4nq-s295 |
| 19812 | 10.21233/m57c-a803 |
| 808 | 10.21233/m5dr-hs85 |
| 22887 | 10.21233/m5mg-k090 |
| 4139 | 10.21233/m5qy-sh73 |
| 21844 | 10.21233/m6at-pw62 |
| 3971 | 10.21233/m7cb-vr77 |
| 46475 | 10.21233/m7cc-0639 |
| 21651 | 10.21233/m7v2-ks06 |
| 22720 | 10.21233/m8cg-s095 |
| 14921 | 10.21233/m8m6-rn46 |
| 26321 | 10.21233/m8qj-bz64 |
| 22726 | 10.21233/m8xc-ez05 |
| 41140 | 10.21233/ma1q-f261 |
| 3972 | 10.21233/ma29-1k87 |

| | |
|---|---|
| 22322 | 10.21233/mbbn-ff92 |
| 24551 | 10.21233/mbst-4266 |
| 15088 | 10.21233/mce3-kp31 |
| 22376 | 10.21233/mczv-f712 |
| 21903 | 10.21233/me0e-ze15 |
| 22896 | 10.21233/me4r-tx16 |
| 16231 | 10.21233/mf7y-nm83 |
| 4299 | 10.21233/mfht-2e09 |
| 41612 | 10.21233/mgtf-q479 |
| 15829 | 10.21233/mgv7-z770 |
| 14643 | 10.21233/mhdv-6n63 |
| 15005 | 10.21233/mhya-2v27 |
| 17334 | 10.21233/mj04-mx52 |
| 22751 | 10.21233/mjwh-t327 |
| 41262 | 10.21233/mjzm-jv11 |
| 4060 | 10.21233/mk13-rb17 |
| 45311 | 10.21233/mks1-mt36 |
| 40582 | 10.21233/mky9-a533 |
| 41446 | 10.21233/mmtn-dv23 |
| 4133 | 10.21233/mnay-8z91 |
| 25133 | 10.21233/mpd1-a275 |
| 22685 | 10.21233/mqcv-z280 |
| 29236 | 10.21233/mqh5-7z20 |
| 21708 | 10.21233/mqtv-9820 |
| 41199 | 10.21233/mrq9-0a25 |
| 4223 | 10.21233/mtkm-0h07 |
| 4446 | 10.21233/mtmk-d772 |
| 41314 | 10.21233/mw33-kd49 |
| 4201 | 10.21233/mybh-v067 |
| 3996 | 10.21233/mzrc-8w03 |
| 22718 | 10.21233/n0v4-yj40 |
| 25268 | 10.21233/n1n7-ya12 |
| 40634 | 10.21233/n1v6-tq67 |
| 239 | 10.21233/n3002s |
| 1694 | 10.21233/n3016n |
| 2328 | 10.21233/n3018d |
| 3132 | 10.21233/n3020q |
| 203 | 10.21233/n30302 |
| 2065 | 10.21233/n3036b |
| 260 | 10.21233/n3040d |
| 2297 | 10.21233/n30472 |
| 2905 | 10.21233/n3049t |
| 802 | 10.21233/n3061g |

| | |
|---|---|
| 1606 | 10.21233/n3064m |
| 1840 | 10.21233/n30650 |
| 225 | 10.21233/n30684 |
| 858 | 10.21233/n3070f |
| 1661 | 10.21233/n3073k |
| 365 | 10.21233/n30896 |
| 1805 | 10.21233/n30938 |
| 11 | 10.21233/n3097s |
| 823 | 10.21233/n3099j |
| 1442 | 10.21233/n30b1v |
| 1860 | 10.21233/n30b3m |
| 2663 | 10.21233/n30b5c |
| 1574 | 10.21233/n30d2x |
| 2054 | 10.21233/n30f3n |
| 332 | 10.21233/n30g64 |
| 1144 | 10.21233/n30g8w |
| 1543 | 10.21233/n30h06 |
| 1769 | 10.21233/n30h1k |
| 790 | 10.21233/n30h6g |
| 1595 | 10.21233/n30h9m |
| 2022 | 10.21233/n30j0j |
| 2627 | 10.21233/n30j29 |
| 298 | 10.21233/n30k65 |
| 518 | 10.21233/n30k7j |
| 1965 | 10.21233/n30m1m |
| 2364 | 10.21233/n30m3c |
| 354 | 10.21233/n30m6h |
| 1563 | 10.21233/n30n0k |
| 2596 | 10.21233/n30n3q |
| 2332 | 10.21233/n30q1n |
| 540 | 10.21233/n30q55 |
| 1987 | 10.21233/n30q9p |
| 2385 | 10.21233/n30r10 |
| 1697 | 10.21233/n30s80 |
| 507 | 10.21233/n30t3f |
| 1133 | 10.21233/n30t6k |
| 230 | 10.21233/n30w34 |
| 1664 | 10.21233/n30w7n |
| 287 | 10.21233/n30x23 |
| 688 | 10.21233/n30x3g |
| 1718 | 10.21233/n30x70 |
| 3558 | 10.21233/n30z2f |
| 1445 | 10.21233/n3106m |

| | |
|---|---|
| 1864 | 10.21233/n31070 |
| 2267 | 10.21233/n3108c |
| 68 | 10.21233/n3110p |
| 1632 | 10.21233/n31358 |
| 1831 | 10.21233/n3136n |
| 850 | 10.21233/n3142g |
| 1886 | 10.21233/n3145m |
| 2289 | 10.21233/n31460 |
| 1598 | 10.21233/n3164x |
| 2025 | 10.21233/n31659 |
| 2630 | 10.21233/n3168f |
| 1434 | 10.21233/n3173w |
| 1852 | 10.21233/n3175n |
| 978 | 10.21233/n3191t |
| 1795 | 10.21233/n3194z |
| 1621 | 10.21233/n31b2j |
| 2650 | 10.21233/n31b62 |
| 3050 | 10.21233/n31b7f |
| 3452 | 10.21233/n31b8t |
| 1566 | 10.21233/n31d1v |
| 2388 | 10.21233/n31d40 |
| 1001 | 10.21233/n31d9w |
| 1819 | 10.21233/n31f2k |
| 2619 | 10.21233/n31f4b |
| 324 | 10.21233/n31g7t |
| 1136 | 10.21233/n31g9k |
| 1587 | 10.21233/n31h9x |
| 510 | 10.21233/n31k6g |
| 691 | 10.21233/n31k7v |
| 1158 | 10.21233/n31m8k |
| 1979 | 10.21233/n31n0w |
| 2588 | 10.21233/n31n31 |
| 1688 | 10.21233/n31p9n |
| 2323 | 10.21233/n31q1z |
| 311 | 10.21233/n31q3q |
| 532 | 10.21233/n31q43 |
| 2955 | 10.21233/n31r32 |
| 3582 | 10.21233/n31r5t |
| 254 | 10.21233/n31s4s |
| 1890 | 10.21233/n31s9p |
| 2292 | 10.21233/n31t10 |
| 496 | 10.21233/n31t44 |
| 1522 | 10.21233/n31t78 |

| | | | |
|---|---|---|---|
| 219 | 10.21233/n31w22 | 267 | 10.21233/n32t32 |
| 853 | 10.21233/n31w4t | 488 | 10.21233/n32t4f |
| 1254 | 10.21233/n31w6k | 1701 | 10.21233/n32t7k |
| 1656 | 10.21233/n31w7z | 210 | 10.21233/n32w2c |
| 278 | 10.21233/n31x2d | 1648 | 10.21233/n32w78 |
| 2312 | 10.21233/n31x92 | 1847 | 10.21233/n32w8n |
| 818 | 10.21233/n3203s | 672 | 10.21233/n32x33 |
| 1437 | 10.21233/n3205j | 9 | 10.21233/n3301b |
| 2888 | 10.21233/n3219d | 1615 | 10.21233/n33067 |
| 2049 | 10.21233/n3236z | 2645 | 10.21233/n33109 |
| 2622 | 10.21233/n32393 | 1450 | 10.21233/n33156 |
| 841 | 10.21233/n3242s | 1868 | 10.21233/n3317z |
| 3483 | 10.21233/n3250c | 2614 | 10.21233/n33381 |
| 1590 | 10.21233/n32647 | 2061 | 10.21233/n33457 |
| 2017 | 10.21233/n3265m | 3468 | 10.21233/n3350p |
| 1844 | 10.21233/n32736 | 1581 | 10.21233/n33635 |
| 2642 | 10.21233/n3277q | 1776 | 10.21233/n3364j |
| 1161 | 10.21233/n3290r | 2006 | 10.21233/n3365x |
| 1786 | 10.21233/n3293w | 339 | 10.21233/n33894 |
| 2591 | 10.21233/n3295n | 525 | 10.21233/n33902 |
| 993 | 10.21233/n32995 | 1153 | 10.21233/n3392t |
| 1611 | 10.21233/n32b1g | 2371 | 10.21233/n3396b |
| 3042 | 10.21233/n32b50 | 2583 | 10.21233/n3397q |
| 314 | 10.21233/n32c8g | 306 | 10.21233/n33c7d |
| 535 | 10.21233/n32c9v | 361 | 10.21233/n33d7r |
| 1982 | 10.21233/n32d3x | 2603 | 10.21233/n33f5b |
| 369 | 10.21233/n32d8t | 1704 | 10.21233/n33h1h |
| 772 | 10.21233/n32d96 | 2340 | 10.21233/n33h38 |
| 1810 | 10.21233/n32f38 | 328 | 10.21233/n33h51 |
| 2611 | 10.21233/n32f51 | 271 | 10.21233/n33k63 |
| 720 | 10.21233/n32g8h | 675 | 10.21233/n33k8v |
| 2348 | 10.21233/n32h3z | 514 | 10.21233/n33m6f |
| 336 | 10.21233/n32h63 | 3775 | 10.21233/n33n6s |
| 1773 | 10.21233/n32j05 | 238 | 10.21233/n33p4c |
| 2579 | 10.21233/n32j4p | 1671 | 10.21233/n33p98 |
| 281 | 10.21233/n32k5d | 294 | 10.21233/n33q4q |
| 1149 | 10.21233/n32m8w | 1107 | 10.21233/n33q7v |
| 1547 | 10.21233/n32m98 | 2327 | 10.21233/n33r1x |
| 1969 | 10.21233/n32n06 | 835 | 10.21233/n33s5s |
| 2368 | 10.21233/n32n1k | 2673 | 10.21233/n33t1m |
| 1679 | 10.21233/n32p9z | 259 | 10.21233/n33t20 |
| 303 | 10.21233/n32q4d | 2904 | 10.21233/n33v1z |
| 32 | 10.21233/n32s3q | 202 | 10.21233/n33w32 |

| | | | | |
|---|---|---|---|---|
| 2637 | 10.21233/n33x21 | | 1663 | 10.21233/n34p86 |
| 857 | 10.21233/n33x55 | | 1862 | 10.21233/n34p9k |
| 1894 | 10.21233/n33x9p | | 2319 | 10.21233/n34r0v |
| 3500 | 10.21233/n33z3r | | 1631 | 10.21233/n34s87 |
| 801 | 10.21233/n3403d | | 2266 | 10.21233/n34t0j |
| 224 | 10.21233/n3410m | | 2665 | 10.21233/n34t29 |
| 1441 | 10.21233/n34144 | | 250 | 10.21233/n34t3p |
| 1859 | 10.21233/n3415h | | 1684 | 10.21233/n34t76 |
| 364 | 10.21233/n34309 | | 1885 | 10.21233/n34t8k |
| 1804 | 10.21233/n34356 | | 792 | 10.21233/n34w54 |
| 1010 | 10.21233/n34411 | | 2629 | 10.21233/n34x1z |
| 1628 | 10.21233/n3443s | | 215 | 10.21233/n34x3q |
| 2053 | 10.21233/n34445 | | 849 | 10.21233/n34x5g |
| 3060 | 10.21233/n34479 | | 977 | 10.21233/n3503q |
| 331 | 10.21233/n3460b | | 1794 | 10.21233/n3506v |
| 1768 | 10.21233/n3464v | | 1433 | 10.21233/n3515t |
| 2396 | 10.21233/n34670 | | 2649 | 10.21233/n3518z |
| 788 | 10.21233/n34712 | | 356 | 10.21233/n35310 |
| 1594 | 10.21233/n34746 | | 543 | 10.21233/n3532c |
| 1826 | 10.21233/n3475k | | 2598 | 10.21233/n35391 |
| 2626 | 10.21233/n3477b | | 1000 | 10.21233/n3542q |
| 517 | 10.21233/n3489f | | 1818 | 10.21233/n3545v |
| 1542 | 10.21233/n3493h | | 2045 | 10.21233/n35467 |
| 1964 | 10.21233/n3494w | | 1135 | 10.21233/n3562d |
| 2363 | 10.21233/n34958 | | 2013 | 10.21233/n3574h |
| 352 | 10.21233/n3498d | | 2618 | 10.21233/n3577n |
| 974 | 10.21233/n34b0q | | 509 | 10.21233/n3589r |
| 1791 | 10.21233/n34b2g | | 2355 | 10.21233/n3595k |
| 1111 | 10.21233/n34c9g | | 343 | 10.21233/n3597b |
| 1733 | 10.21233/n34d25 | | 966 | 10.21233/n35993 |
| 539 | 10.21233/n34d72 | | 1157 | 10.21233/n35b01 |
| 1562 | 10.21233/n34f14 | | 2587 | 10.21233/n35b4j |
| 1986 | 10.21233/n34f2h | | 531 | 10.21233/n35d8r |
| 2384 | 10.21233/n34f3w | | 3790 | 10.21233/n35f83 |
| 1696 | 10.21233/n34h1t | | 253 | 10.21233/n35g6p |
| 318 | 10.21233/n34h6q | | 1687 | 10.21233/n35h14 |
| 1755 | 10.21233/n34j15 | | 495 | 10.21233/n35h7d |
| 860 | 10.21233/n34k85 | | 715 | 10.21233/n35h8s |
| 285 | 10.21233/n34m6r | | 2954 | 10.21233/n35j50 |
| 506 | 10.21233/n34m74 | | 852 | 10.21233/n35k73 |
| 2352 | 10.21233/n34n2k | | 1655 | 10.21233/n35m0s |
| 3557 | 10.21233/n34n63 | | 1889 | 10.21233/n35m15 |
| 1444 | 10.21233/n34p7t | | 679 | 10.21233/n35m5p |

| | | | |
|---|---|---|---|
| 1709 | 10.21233/n35m96 | 682 | 10.21233/n36c93 |
| 2922 | 10.21233/n35n38 | 876 | 10.21233/n36d01 |
| 3149 | 10.21233/n35n4n | 2314 | 10.21233/n36d5x |
| 1854 | 10.21233/n35q0t | 302 | 10.21233/n36d82 |
| 2652 | 10.21233/n35q4b | 2367 | 10.21233/n36f4w |
| 873 | 10.21233/n35q7g | 245 | 10.21233/n36g60 |
| 2311 | 10.21233/n35r2x | 1880 | 10.21233/n36h1f |
| 1003 | 10.21233/n35s6s | 2335 | 10.21233/n36j25 |
| 1623 | 10.21233/n35s8j | 31 | 10.21233/n36k5n |
| 2048 | 10.21233/n35s9x | 209 | 10.21233/n36k61 |
| 27 | 10.21233/n35t17 | 843 | 10.21233/n36k8s |
| 782 | 10.21233/n35w3p | 671 | 10.21233/n36m6c |
| 2016 | 10.21233/n35w8k | 1904 | 10.21233/n36n0f |
| 206 | 10.21233/n35x18 | 8 | 10.21233/n36p49 |
| 1028 | 10.21233/n35x4d | 2644 | 10.21233/n36q38 |
| 2068 | 10.21233/n35x7j | 234 | 10.21233/n36q51 |
| 3482 | 10.21233/n35z20 | 1667 | 10.21233/n36q9j |
| 346 | 10.21233/n3602n | 2303 | 10.21233/n36r1v |
| 969 | 10.21233/n3604d | 995 | 10.21233/n36s5q |
| 2590 | 10.21233/n36099 | 1397 | 10.21233/n36s63 |
| 805 | 10.21233/n36120 | 1448 | 10.21233/n36t7t |
| 1610 | 10.21233/n36154 | 1867 | 10.21233/n36t86 |
| 1843 | 10.21233/n3616h | 2669 | 10.21233/n36v0h |
| 3041 | 10.21233/n3620k | 3074 | 10.21233/n36v28 |
| 534 | 10.21233/n36319 | 774 | 10.21233/n36w5r |
| 1160 | 10.21233/n3634f | 1580 | 10.21233/n36w7h |
| 1981 | 10.21233/n36366 | 2005 | 10.21233/n36w8w |
| 2378 | 10.21233/n3637k | 1635 | 10.21233/n36x6g |
| 1391 | 10.21233/n3643d | 1152 | 10.21233/n3704q |
| 1809 | 10.21233/n3644s | 1775 | 10.21233/n3706g |
| 2610 | 10.21233/n3647x | 796 | 10.21233/n3713p |
| 313 | 10.21233/n36591 | 1601 | 10.21233/n3715f |
| 719 | 10.21233/n3660z | 2633 | 10.21233/n3719z |
| 1524 | 10.21233/n36633 | 3033 | 10.21233/n3720w |
| 1577 | 10.21233/n3673f | 1549 | 10.21233/n3734r |
| 2001 | 10.21233/n3674t | 1972 | 10.21233/n3736h |
| 2347 | 10.21233/n3695w | 490 | 10.21233/n3761n |
| 335 | 10.21233/n36981 | 547 | 10.21233/n37710 |
| 1148 | 10.21233/n36b0b | 1569 | 10.21233/n37744 |
| 1546 | 10.21233/n36b23 | 1763 | 10.21233/n3775h |
| 1772 | 10.21233/n36b3g | 269 | 10.21233/n37880 |
| 1968 | 10.21233/n36b4v | 1703 | 10.21233/n3793f |
| 2578 | 10.21233/n36b6m | 327 | 10.21233/n3797z |

| | | | | |
|---|---|---|---|---|
| 513 | 10.21233/n3798b | | 2362 | 10.21233/n38376 |
| 1139 | 10.21233/n37b0n | | 351 | 10.21233/n3840w |
| 1959 | 10.21233/n37b2d | | 2594 | 10.21233/n3848x |
| 3774 | 10.21233/n37b92 | | 317 | 10.21233/n3870x |
| 867 | 10.21233/n37c9d | | 538 | 10.21233/n38719 |
| 1670 | 10.21233/n37d23 | | 1754 | 10.21233/n3874f |
| 699 | 10.21233/n37d70 | | 261 | 10.21233/n38889 |
| 1105 | 10.21233/n37d9r | | 1695 | 10.21233/n3892c |
| 1725 | 10.21233/n37f12 | | 1897 | 10.21233/n3893r |
| 3565 | 10.21233/n37f7b | | 723 | 10.21233/n3898n |
| 20 | 10.21233/n37g69 | | 2351 | 10.21233/n38b4g |
| 2326 | 10.21233/n37j47 | | 2962 | 10.21233/n38b67 |
| 3130 | 10.21233/n37j60 | | 226 | 10.21233/n38c8b |
| 834 | 10.21233/n37k7q | | 1716 | 10.21233/n38f2r |
| 1639 | 10.21233/n37m0d | | 1443 | 10.21233/n38h12 |
| 2063 | 10.21233/n37m1s | | 2265 | 10.21233/n38h3t |
| 258 | 10.21233/n37m4x | | 880 | 10.21233/n38h93 |
| 2903 | 10.21233/n37n48 | | 1683 | 10.21233/n38j1d |
| 1604 | 10.21233/n37p9h | | 2318 | 10.21233/n38j35 |
| 1837 | 10.21233/n37q0f | | 1013 | 10.21233/n38k71 |
| 856 | 10.21233/n37q6q | | 1630 | 10.21233/n38k9s |
| 1659 | 10.21233/n37q9v | | 1829 | 10.21233/n38m0q |
| 3499 | 10.21233/n37r5p | | 847 | 10.21233/n38m5m |
| 363 | 10.21233/n37s4n | | 2287 | 10.21233/n38n02 |
| 987 | 10.21233/n37s6d | | 791 | 10.21233/n38p72 |
| 1803 | 10.21233/n37s85 | | 2023 | 10.21233/n38q0r |
| 821 | 10.21233/n37t40 | | 1432 | 10.21233/n38q7d |
| 1440 | 10.21233/n37t6r | | 1651 | 10.21233/n38q8s |
| 1627 | 10.21233/n37t74 | | 1850 | 10.21233/n38q95 |
| 2262 | 10.21233/n37v0t | | 355 | 10.21233/n38s3k |
| 2661 | 10.21233/n37v16 | | 976 | 10.21233/n38s5b |
| 550 | 10.21233/n37w39 | | 1793 | 10.21233/n38s8g |
| 1572 | 10.21233/n37w7t | | 813 | 10.21233/n38t5p |
| 2395 | 10.21233/n37w9k | | 3048 | 10.21233/n38v2w |
| 1008 | 10.21233/n37x41 | | 1564 | 10.21233/n38w6r |
| 2625 | 10.21233/n37x9x | | 1758 | 10.21233/n38w74 |
| 330 | 10.21233/n3801w | | 375 | 10.21233/n38x16 |
| 1142 | 10.21233/n3803n | | 778 | 10.21233/n38x2k |
| 1541 | 10.21233/n3805d | | 999 | 10.21233/n38x3z |
| 973 | 10.21233/n38130 | | 1817 | 10.21233/n38x63 |
| 3840 | 10.21233/n3822z | | 2617 | 10.21233/n38x8v |
| 516 | 10.21233/n38329 | | 321 | 10.21233/n3902k |
| 1732 | 10.21233/n3835f | | 1780 | 10.21233/n3917t |

| | |
|---|---|
| 3611 | 10.21233/n39228 |
| 3830 | 10.21233/n3923n |
| 1719 | 10.21233/n3935r |
| 342 | 10.21233/n39398 |
| 530 | 10.21233/n39406 |
| 1156 | 10.21233/n3942z |
| 1553 | 10.21233/n3943b |
| 1976 | 10.21233/n3944q |
| 2586 | 10.21233/n3947v |
| 69 | 10.21233/n3959z |
| 309 | 10.21233/n39699 |
| 2953 | 10.21233/n3977w |
| 252 | 10.21233/n39890 |
| 1888 | 10.21233/n3994f |
| 2290 | 10.21233/n3995t |
| 1520 | 10.21233/n39b21 |
| 2343 | 10.21233/n39b55 |
| 1654 | 10.21233/n39d2q |
| 275 | 10.21233/n39d67 |
| 678 | 10.21233/n39d80 |
| 2920 | 10.21233/n39f6k |
| 816 | 10.21233/n39g8p |
| 1435 | 10.21233/n39h00 |
| 1622 | 10.21233/n39h1c |
| 2047 | 10.21233/n39h34 |
| 2651 | 10.21233/n39h5w |
| 872 | 10.21233/n39h81 |
| 1674 | 10.21233/n39j1q |
| 381 | 10.21233/n39k6z |
| 1002 | 10.21233/n39k8q |
| 1820 | 10.21233/n39m01 |
| 25 | 10.21233/n39m4j |
| 3081 | 10.21233/n39n4w |
| 968 | 10.21233/n39p7c |
| 2015 | 10.21233/n39q1f |
| 205 | 10.21233/n39q4k |
| 1842 | 10.21233/n39q9g |
| 345 | 10.21233/n39s3w |
| 1783 | 10.21233/n39s7d |
| 1980 | 10.21233/n39s8s |
| 804 | 10.21233/n39t37 |
| 1390 | 10.21233/n39t6c |
| 3040 | 10.21233/n39v1t |

| | |
|---|---|
| 312 | 10.21233/n39w3x |
| 2377 | 10.21233/n39w96 |
| 1807 | 10.21233/n39x6d |
| 2609 | 10.21233/n39x9j |
| 497 | 10.21233/n3b02w |
| 1523 | 10.21233/n3b07s |
| 1771 | 10.21233/n3b16r |
| 279 | 10.21233/n3b305 |
| 681 | 10.21233/n3b31j |
| 1711 | 10.21233/n3b34p |
| 1545 | 10.21233/n3b441 |
| 1967 | 10.21233/n3b45d |
| 3781 | 10.21233/n3b517 |
| 1114 | 10.21233/n3b729 |
| 1736 | 10.21233/n3b742 |
| 2945 | 10.21233/n3b78k |
| 1879 | 10.21233/n3b94r |
| 7 | 10.21233/n3bc7k |
| 208 | 10.21233/n3bc8z |
| 1030 | 10.21233/n3bd08 |
| 1646 | 10.21233/n3bd21 |
| 1845 | 10.21233/n3bd3d |
| 2643 | 10.21233/n3bh56 |
| 16 | 10.21233/n3bh6k |
| 232 | 10.21233/n3bh7z |
| 1447 | 10.21233/n3bj0n |
| 1666 | 10.21233/n3bj11 |
| 1866 | 10.21233/n3bj2d |
| 773 | 10.21233/n3bk68 |
| 994 | 10.21233/n3bk7n |
| 1393 | 10.21233/n3bk9d |
| 2612 | 10.21233/n3bm23 |
| 829 | 10.21233/n3bm6m |
| 2270 | 10.21233/n3bn0p |
| 1579 | 10.21233/n3bp82 |
| 2004 | 10.21233/n3bq0c |
| 2632 | 10.21233/n3br0q |
| 3032 | 10.21233/n3br2g |
| 337 | 10.21233/n3bs4k |
| 523 | 10.21233/n3bs5z |
| 1970 | 10.21233/n3bs9g |
| 2369 | 10.21233/n3bt1s |
| 2580 | 10.21233/n3bt25 |

| | |
|---:|:---|
| 982 | 10.21233/n3bt6p |
| 304 | 10.21233/n3bw2v |
| 1171 | 10.21233/n3bx4z |
| 1568 | 10.21233/n3bx5b |
| 2391 | 10.21233/n3bx8g |
| 489 | 10.21233/n3c026 |
| 1513 | 10.21233/n3c05b |
| 2338 | 10.21233/n3c09v |
| 1138 | 10.21233/n3c149 |
| 1762 | 10.21233/n3c162 |
| 268 | 10.21233/n3c31v |
| 673 | 10.21233/n3c33m |
| 1906 | 10.21233/n3c36r |
| 1724 | 10.21233/n3c45q |
| 1958 | 10.21233/n3c463 |
| 2358 | 10.21233/n3c48v |
| 236 | 10.21233/n3c59k |
| 866 | 10.21233/n3c61w |
| 693 | 10.21233/n3c70v |
| 1104 | 10.21233/n3c717 |
| 19 | 10.21233/n3c887 |
| 1451 | 10.21233/n3c929 |
| 1637 | 10.21233/n3c93p |
| 2671 | 10.21233/n3c976 |
| 1691 | 10.21233/n3cb31 |
| 2902 | 10.21233/n3cb65 |
| 200 | 10.21233/n3cc88 |
| 1836 | 10.21233/n3cd3q |
| 2635 | 10.21233/n3cd6v |
| 1658 | 10.21233/n3cf2p |
| 1892 | 10.21233/n3cf32 |
| 3497 | 10.21233/n3cf8z |
| 799 | 10.21233/n3cg89 |
| 986 | 10.21233/n3cg9p |
| 1603 | 10.21233/n3ch2c |
| 2030 | 10.21233/n3ch3r |
| 1439 | 10.21233/n3cj0z |
| 1857 | 10.21233/n3cj1b |
| 2476 | 10.21233/n3cj33 |
| 2660 | 10.21233/n3cj4g |
| 1571 | 10.21233/n3ck8b |
| 1802 | 10.21233/n3ck9q |
| 820 | 10.21233/n3cm5j |

| | |
|---:|:---|
| 1626 | 10.21233/n3cm92 |
| 3058 | 10.21233/n3cn34 |
| 329 | 10.21233/n3cp57 |
| 549 | 10.21233/n3cp6m |
| 1766 | 10.21233/n3cq0p |
| 785 | 10.21233/n3cq6z |
| 1592 | 10.21233/n3cq8q |
| 1824 | 10.21233/n3cq93 |
| 2624 | 10.21233/n3cr2s |
| 3839 | 10.21233/n3cr69 |
| 1141 | 10.21233/n3cs6n |
| 1961 | 10.21233/n3cs9s |
| 2361 | 10.21233/n3ct0q |
| 350 | 10.21233/n3ct3v |
| 1789 | 10.21233/n3ct8r |
| 2593 | 10.21233/n3cv1f |
| 701 | 10.21233/n3cw3j |
| 537 | 10.21233/n3cx14 |
| 1984 | 10.21233/n3cx61 |
| 2382 | 10.21233/n3cx7d |
| 316 | 10.21233/n3d12v |
| 2961 | 10.21233/n3d20f |
| 1715 | 10.21233/n3d451 |
| 2317 | 10.21233/n3d752 |
| 1883 | 10.21233/n3db4q |
| 1650 | 10.21233/n3df07 |
| 2286 | 10.21233/n3df20 |
| 3488 | 10.21233/n3df7w |
| 1849 | 10.21233/n3dj08 |
| 2647 | 10.21233/n3dj21 |
| 3047 | 10.21233/n3dj3d |
| 998 | 10.21233/n3dm80 |
| 1815 | 10.21233/n3dn1p |
| 374 | 10.21233/n3dq34 |
| 777 | 10.21233/n3dq5w |
| 2011 | 10.21233/n3dq81 |
| 341 | 10.21233/n3dt35 |
| 1155 | 10.21233/n3dt69 |
| 1778 | 10.21233/n3dt82 |
| 3610 | 10.21233/n3dv4w |
| 527 | 10.21233/n3dx36 |
| 2373 | 10.21233/n3dz0d |
| 308 | 10.21233/n3f11s |

| | | | | |
|---:|---|---|---:|---|
| 493 | 10.21233/n3f125 | | 1891 | 10.21233/n3h45z |
| 2342 | 10.21233/n3f172 | | 2293 | 10.21233/n3h46b |
| 677 | 10.21233/n3f40f | | 2901 | 10.21233/n3h49g |
| 1707 | 10.21233/n3f45b | | 221 | 10.21233/n3h68s |
| 2919 | 10.21233/n3f48g | | 1255 | 10.21233/n3h72v |
| 871 | 10.21233/n3f71v | | 1438 | 10.21233/n3h737 |
| 24 | 10.21233/n3f97t | | 1657 | 10.21233/n3h74m |
| 240 | 10.21233/n3f986 | | 1856 | 10.21233/n3hb38 |
| 2885 | 10.21233/n3fb75 | | 2050 | 10.21233/n3hb4n |
| 204 | 10.21233/n3fd53 | | 2654 | 10.21233/n3hb6d |
| 1024 | 10.21233/n3fd87 | | 3056 | 10.21233/n3hb7s |
| 1642 | 10.21233/n3ff1x | | 784 | 10.21233/n3hd97 |
| 3080 | 10.21233/n3ff5f | | 1006 | 10.21233/n3hf05 |
| 803 | 10.21233/n3fh88 | | 2623 | 10.21233/n3hf52 |
| 1607 | 10.21233/n3fj0k | | 971 | 10.21233/n3hh7h |
| 1841 | 10.21233/n3fj1z | | 1591 | 10.21233/n3hj06 |
| 3039 | 10.21233/n3fj5g | | 1162 | 10.21233/n3hm8x |
| 1806 | 10.21233/n3fn0m | | 1559 | 10.21233/n3hm99 |
| 333 | 10.21233/n3ft4v | | 1788 | 10.21233/n3hn07 |
| 1770 | 10.21233/n3ft8c | | 315 | 10.21233/n3hq32 |
| 1966 | 10.21233/n3ft9r | | 536 | 10.21233/n3hq4f |
| 2576 | 10.21233/n3fv12 | | 2381 | 10.21233/n3hr08 |
| 299 | 10.21233/n3fx24 | | 2960 | 10.21233/n3hr3d |
| 519 | 10.21233/n3fx3h | | 501 | 10.21233/n3ht33 |
| 1735 | 10.21233/n3fx71 | | 684 | 10.21233/n3hx4h |
| 2944 | 10.21233/n3g194 | | 878 | 10.21233/n3j13g |
| 3138 | 10.21233/n3g202 | | 1680 | 10.21233/n3j157 |
| 1698 | 10.21233/n3g45n | | 33 | 10.21233/n3j39f |
| 15 | 10.21233/n3g974 | | 1882 | 10.21233/n3j458 |
| 828 | 10.21233/n3g99w | | 3487 | 10.21233/n3j50q |
| 2268 | 10.21233/n3gb4b | | 212 | 10.21233/n3j69g |
| 1633 | 10.21233/n3gf17 | | 1033 | 10.21233/n3j71s |
| 1832 | 10.21233/n3gf2m | | 811 | 10.21233/n3j99h |
| 1599 | 10.21233/n3gh9z | | 1617 | 10.21233/n3jb1t |
| 2026 | 10.21233/n3gj0w | | 2646 | 10.21233/n3jb4z |
| 2631 | 10.21233/n3gj2n | | 3046 | 10.21233/n3jb6q |
| 3031 | 10.21233/n3gj4d | | 3861 | 10.21233/n3jb8g |
| 1796 | 10.21233/n3gn0x | | 776 | 10.21233/n3jd85 |
| 1137 | 10.21233/n3gt6x | | 997 | 10.21233/n3jd9j |
| 3563 | 10.21233/n3gz35 | | 1814 | 10.21233/n3jf27 |
| 2324 | 10.21233/n3h182 | | 2615 | 10.21233/n3jf40 |
| 3126 | 10.21233/n3h20c | | 2007 | 10.21233/n3jj28 |
| 256 | 10.21233/n3h402 | | 340 | 10.21233/n3jm53 |

| | |
|---|---|
| 1154 | 10.21233/n3jm87 |
| 1551 | 10.21233/n3jm9m |
| 1974 | 10.21233/n3jn0j |
| 307 | 10.21233/n3jq3c |
| 2951 | 10.21233/n3jr2b |
| 492 | 10.21233/n3jt55 |
| 1705 | 10.21233/n3jt89 |
| 2341 | 10.21233/n3jv10 |
| 272 | 10.21233/n3jx1p |
| 2884 | 10.21233/n3k18p |
| 3079 | 10.21233/n3k493 |
| 3473 | 10.21233/n3k501 |
| 2943 | 10.21233/n3kr31 |
| 3067 | 10.21233/n3m49d |
| 3561 | 10.21233/n3mn6f |
| 3771 | 10.21233/n3mn7t |
| 3125 | 10.21233/n3mv20 |
| 3494 | 10.21233/n3mz21 |
| 3054 | 10.21233/n3n48b |
| 2959 | 10.21233/n3nj5b |
| 2892 | 10.21233/n3nv29 |
| 3486 | 10.21233/n3nz2b |
| 2882 | 10.21233/n3pv17 |
| 3078 | 10.21233/n3pv2m |
| 3471 | 10.21233/n3pz18 |
| 2907 | 10.21233/n3qn37 |
| 3029 | 10.21233/n3r20j |
| 3493 | 10.21233/n3rr5b |
| 3454 | 10.21233/n3rv3m |
| 2925 | 10.21233/n3sf5j |
| 2890 | 10.21233/n3sj5k |
| 3485 | 10.21233/n3sn5m |
| 3859 | 10.21233/n3sv49 |
| 2949 | 10.21233/n3t77j |
| 3077 | 10.21233/n3tn4j |
| 3568 | 10.21233/n3v51w |
| 2906 | 10.21233/n3vb7w |
| 3559 | 10.21233/n3w516 |
| 3492 | 10.21233/n3wf8m |
| 3051 | 10.21233/n3wj5h |
| 3583 | 10.21233/n3wz3w |
| 2956 | 10.21233/n3x203 |
| 3484 | 10.21233/n3xf7j |

| | |
|---|---|
| 3043 | 10.21233/n3xj4f |
| 3858 | 10.21233/n3xj8z |
| 2948 | 10.21233/n3z183 |
| 2880 | 10.21233/n3zb6f |
| 3076 | 10.21233/n3zb7t |
| 3035 | 10.21233/n3zj54 |
| 40875 | 10.21233/n4bq-p922 |
| 40997 | 10.21233/n4sd-es14 |
| 23006 | 10.21233/n4x8-0k74 |
| 41366 | 10.21233/n4xw-kd72 |
| 21535 | 10.21233/n4y2-m886 |
| 20495 | 10.21233/n5d1-mq43 |
| 26446 | 10.21233/n5nj-5q34 |
| 4087 | 10.21233/n6bg-kc41 |
| 20167 | 10.21233/n7jz-0g74 |
| 24236 | 10.21233/n87r-7a39 |
| 14999 | 10.21233/n8dz-y640 |
| 24590 | 10.21233/n8w1-xz08 |
| 4347 | 10.21233/n8z7-nx69 |
| 4216 | 10.21233/n9a0-fk75 |
| 24467 | 10.21233/n9vy-3c22 |
| 15415 | 10.21233/na70-nc19 |
| 4377 | 10.21233/nbg7-rk14 |
| 26507 | 10.21233/nbnz-tz48 |
| 14637 | 10.21233/nc0w-mk91 |
| 4081 | 10.21233/nc7w-an65 |
| 2069 | 10.21233/nda9-aw65 |
| 4173 | 10.21233/ndhk-ws14 |
| 24325 | 10.21233/nep1-3z35 |
| 24900 | 10.21233/nf08-bn61 |
| 24534 | 10.21233/nfah-4t91 |
| 41146 | 10.21233/ngxf-6p79 |
| 4057 | 10.21233/nkzg-w156 |
| 15115 | 10.21233/nm9j-w372 |
| 24559 | 10.21233/nmwt-fc53 |
| 15366 | 10.21233/nn1h-xp19 |
| 16269 | 10.21233/nn2g-qt23 |
| 4169 | 10.21233/nnyx-kj31 |
| 14494 | 10.21233/npm2-8d68 |
| 17859 | 10.21233/npmk-6h16 |
| 4481 | 10.21233/nq7x-mx79 |
| 22373 | 10.21233/nqb2-na71 |
| 22693 | 10.21233/nrd3-gj67 |

| | |
|---|---|
| 15772 | 10.21233/nre4-q408 |
| 16086 | 10.21233/nrq0-e757 |
| 41201 | 10.21233/nrvm-5m89 |
| 21838 | 10.21233/nryw-9765 |
| 3890 | 10.21233/nsm3-eh40 |
| 25217 | 10.21233/nssh-az33 |
| 40999 | 10.21233/nt4c-9462 |
| 22982 | 10.21233/nt73-cr73 |
| 4502 | 10.21233/nvn7-f141 |
| 41604 | 10.21233/nwfm-st50 |
| 45629 | 10.21233/nwgq-s472 |
| 45281 | 10.21233/ny4b-e654 |
| 15635 | 10.21233/ny7s-f780 |
| 45383 | 10.21233/p0mn-7534 |
| 41230 | 10.21233/p1h4-4v17 |
| 22786 | 10.21233/p1tb-df93 |
| 4021 | 10.21233/p28f-qv78 |
| 15250 | 10.21233/p37m-jn82 |
| 41252 | 10.21233/p39d-rd93 |
| 25213 | 10.21233/p3md-9032 |
| 14529 | 10.21233/p4mc-9f34 |
| 20180 | 10.21233/p4nw-2d17 |
| 4549 | 10.21233/p4zh-1534 |
| 24281 | 10.21233/p5jz-zk41 |
| 4055 | 10.21233/p76r-3q63 |
| 16081 | 10.21233/p7fp-g179 |
| 22810 | 10.21233/p83k-v204 |
| 45200 | 10.21233/p8ac-sf20 |
| 24218 | 10.21233/p8jv-nx13 |
| 17340 | 10.21233/p96n-6v19 |
| 24846 | 10.21233/p9bs-0686 |
| 15649 | 10.21233/paxn-mh26 |
| 15746 | 10.21233/pbbn-5h88 |
| 24929 | 10.21233/pbc8-c366 |
| 25287 | 10.21233/pbmw-9s02 |
| 24626 | 10.21233/pbq8-yy15 |
| 16251 | 10.21233/pbwt-b156 |
| 15296 | 10.21233/pck3-ct03 |
| 16271 | 10.21233/pcmd-m560 |
| 25283 | 10.21233/pe6c-pb93 |
| 24277 | 10.21233/pegh-gp75 |
| 16063 | 10.21233/pfsb-bc85 |
| 24671 | 10.21233/pgj9-5v59 |

| | |
|---|---|
| 32258 | 10.21233/pgwj-9f54 |
| 24348 | 10.21233/ph7f-8b78 |
| 25078 | 10.21233/ph7n-8m49 |
| 16217 | 10.21233/phdj-h055 |
| 4103 | 10.21233/phdz-hs08 |
| 44941 | 10.21233/phqc-me73 |
| 25344 | 10.21233/pht8-mj95 |
| 4504 | 10.21233/pj9k-qg05 |
| 45097 | 10.21233/pkhx-pf62 |
| 16179 | 10.21233/pkt4-am12 |
| 15371 | 10.21233/pn5c-9x40 |
| 18123 | 10.21233/pnbx-1e72 |
| 14928 | 10.21233/pps9-sn76 |
| 45636 | 10.21233/pr2m-cc37 |
| 14556 | 10.21233/ps1x-es22 |
| 15811 | 10.21233/psnq-xv46 |
| 14446 | 10.21233/ptaw-4d96 |
| 14768 | 10.21233/ptjw-d867 |
| 40439 | 10.21233/ptq6-q055 |
| 14930 | 10.21233/ptx4-dz18 |
| 4286 | 10.21233/pvn2-a273 |
| 25386 | 10.21233/pvpd-tm53 |
| 45219 | 10.21233/pwgs-4g85 |
| 22851 | 10.21233/pxem-ce07 |
| 15342 | 10.21233/pyjt-xa38 |
| 41280 | 10.21233/pys7-k313 |
| 4047 | 10.21233/q057-d355 |
| 19993 | 10.21233/q20r-m406 |
| 22847 | 10.21233/q2tc-z758 |
| 4523 | 10.21233/q36c-4q49 |
| 21545 | 10.21233/q6p2-kw54 |
| 17292 | 10.21233/q744-pp22 |
| 42570 | 10.21233/q95g-2x92 |
| 41421 | 10.21233/q9sk-t894 |
| 45192 | 10.21233/q9wh-hq41 |
| 14104 | 10.21233/qaht-6745 |
| 14935 | 10.21233/qamn-c462 |
| 25400 | 10.21233/qbf7-hn56 |
| 21770 | 10.21233/qbkz-hz40 |
| 32260 | 10.21233/qd2q-5c39 |
| 24553 | 10.21233/qdx7-af11 |
| 4135 | 10.21233/qem1-6q63 |
| 13029 | 10.21233/qf6p-8j75 |

| | | | |
|---|---|---|---|
| 4209 | 10.21233/qf8f-me80 | 22842 | 10.21233/r8zp-ww92 |
| 22648 | 10.21233/qf8f-wb97 | 4102 | 10.21233/r9pn-tx36 |
| 4470 | 10.21233/qfrx-zf42 | 17338 | 10.21233/r9wb-gz24 |
| 42547 | 10.21233/qg8e-np07 | 22646 | 10.21233/rahy-8590 |
| 45173 | 10.21233/qgv1-sv24 | 4203 | 10.21233/raph-rb39 |
| 45164 | 10.21233/qh50-8208 | 41753 | 10.21233/rarx-fe28 |
| 15377 | 10.21233/qhtr-fk23 | 26425 | 10.21233/rb09-6144 |
| 41183 | 10.21233/qjda-f561 | 15325 | 10.21233/rb6f-1g25 |
| 4085 | 10.21233/qjjs-h634 | 15368 | 10.21233/rc8q-je06 |
| 4116 | 10.21233/qjxn-st60 | 4154 | 10.21233/rce2-en21 |
| 44723 | 10.21233/qk2b-8b38 | 41746 | 10.21233/rd01-0v56 |
| 22904 | 10.21233/qka4-j483 | 22757 | 10.21233/rd1y-0x62 |
| 46479 | 10.21233/qkmf-dk52 | 45716 | 10.21233/rfek-5a50 |
| 41417 | 10.21233/qmke-wg69 | 14650 | 10.21233/rg17-hr80 |
| 14558 | 10.21233/qmy5-xr61 | 42663 | 10.21233/rgr7-f616 |
| 4483 | 10.21233/qp8c-4854 | 15020 | 10.21233/rgvz-gj38 |
| 22344 | 10.21233/qphx-cn83 | 42677 | 10.21233/rhm6-pc30 |
| 4197 | 10.21233/qq1x-zk02 | 15009 | 10.21233/rhs1-wt07 |
| 41476 | 10.21233/qqk2-fy61 | 45324 | 10.21233/rj4f-fn81 |
| 15612 | 10.21233/qrjk-gj85 | 13060 | 10.21233/rja4-q512 |
| 24100 | 10.21233/qrwc-m362 | 4420 | 10.21233/rmkd-2162 |
| 22814 | 10.21233/qtf9-zf76 | 41411 | 10.21233/rmpn-9878 |
| 24300 | 10.21233/qtfz-0018 | 24931 | 10.21233/rn1j-tb02 |
| 41237 | 10.21233/qx7x-1245 | 25404 | 10.21233/rn65-2k02 |
| 24250 | 10.21233/qx94-hx64 | 4490 | 10.21233/rnkp-2723 |
| 4330 | 10.21233/qxv2-w585 | 15680 | 10.21233/rp2w-0753 |
| 40949 | 10.21233/qyrx-tz10 | 15785 | 10.21233/rp8s-2k53 |
| 42685 | 10.21233/qzek-wp34 | 41750 | 10.21233/rpzt-3p58 |
| 24956 | 10.21233/r08t-2829 | 15909 | 10.21233/rqzm-ma95 |
| 15294 | 10.21233/r0cq-9083 | 24574 | 10.21233/rrws-7771 |
| 20397 | 10.21233/r0kz-yt78 | 41521 | 10.21233/rrzj-sp79 |
| 20311 | 10.21233/r0mg-q111 | 24944 | 10.21233/rsdg-ep81 |
| 41339 | 10.21233/r11j-mj44 | 20042 | 10.21233/rt31-zw12 |
| 4058 | 10.21233/r27w-7c93 | 45207 | 10.21233/rt3s-g752 |
| 4532 | 10.21233/r2nw-6c72 | 45327 | 10.21233/rtcq-kn47 |
| 23019 | 10.21233/r37v-gx98 | 22761 | 10.21233/rw25-2v24 |
| 24193 | 10.21233/r3mj-5588 | 25372 | 10.21233/rwdw-1852 |
| 15704 | 10.21233/r4mc-tc72 | 21418 | 10.21233/rwpj-e158 |
| 25127 | 10.21233/r4xz-v815 | 24952 | 10.21233/ryar-ev70 |
| 16238 | 10.21233/r5wm-z686 | 22689 | 10.21233/rybb-mp20 |
| 26516 | 10.21233/r6kr-hg54 | 4134 | 10.21233/ryfa-y280 |
| 4351 | 10.21233/r749-6141 | 45725 | 10.21233/s0pq-6335 |
| 25318 | 10.21233/r7dr-q154 | 4545 | 10.21233/s1ae-ar89 |

| | |
|---|---|
| 40627 | 10.21233/s1va-qn28 |
| 3913 | 10.21233/s1x4-mq32 |
| 46485 | 10.21233/s2j6-5w36 |
| 17597 | 10.21233/s3er-nf45 |
| 15284 | 10.21233/s3ma-a925 |
| 18106 | 10.21233/s4jg-yh87 |
| 22119 | 10.21233/s4k3-4429 |
| 14370 | 10.21233/s4yh-p034 |
| 22738 | 10.21233/s54m-1k85 |
| 16073 | 10.21233/s6jx-f115 |
| 42673 | 10.21233/s6my-6052 |
| 45387 | 10.21233/s6wg-4d92 |
| 4288 | 10.21233/s7rb-ap65 |
| 41220 | 10.21233/s7t1-hx33 |
| 24520 | 10.21233/s85f-3f36 |
| 24837 | 10.21233/s8ht-3z18 |
| 4517 | 10.21233/s8kk-y022 |
| 17383 | 10.21233/s9wz-1g21 |
| 25013 | 10.21233/sa5v-5h89 |
| 24102 | 10.21233/sa88-mb64 |
| 14972 | 10.21233/sam2-cd21 |
| 45314 | 10.21233/sb36-c222 |
| 24284 | 10.21233/sbes-jp06 |
| 46462 | 10.21233/sc2f-5d04 |
| 24197 | 10.21233/sc9n-2r53 |
| 24578 | 10.21233/scew-1h82 |
| 45168 | 10.21233/scrs-nq43 |
| 4118 | 10.21233/sczz-2g33 |
| 22763 | 10.21233/sd3z-7v27 |
| 4554 | 10.21233/sdqe-ky81 |
| 3884 | 10.21233/sewp-vp39 |
| 25027 | 10.21233/sf2s-x915 |
| 4543 | 10.21233/sfsk-ej53 |
| 21422 | 10.21233/sg70-ew74 |
| 4150 | 10.21233/shhz-0f61 |
| 4005 | 10.21233/sj04-8c24 |
| 22793 | 10.21233/sjqw-qa18 |
| 15511 | 10.21233/sk02-jh34 |
| 17672 | 10.21233/skp4-td45 |
| 15713 | 10.21233/sn81-zc38 |
| 22753 | 10.21233/snt9-p808 |
| 20293 | 10.21233/snyf-b526 |
| 4030 | 10.21233/sq17-8k63 |

| | |
|---|---|
| 24110 | 10.21233/sq7c-9a90 |
| 25232 | 10.21233/sqtj-fr11 |
| 15381 | 10.21233/ss13-7n86 |
| 17880 | 10.21233/ssg9-gk36 |
| 40955 | 10.21233/stpw-ta17 |
| 24497 | 10.21233/sxfw-2b43 |
| 24181 | 10.21233/sxhe-sa20 |
| 24227 | 10.21233/sxhx-j135 |
| 22824 | 10.21233/sy1b-py46 |
| 41245 | 10.21233/syjt-3739 |
| 4002 | 10.21233/szbb-8k05 |
| 18159 | 10.21233/t00e-b616 |
| 22342 | 10.21233/t052-s807 |
| 22674 | 10.21233/t0ea-cd42 |
| 4383 | 10.21233/t0t5-0w85 |
| 24108 | 10.21233/t1ba-1q15 |
| 4032 | 10.21233/t1n7-cq82 |
| 45406 | 10.21233/t1p9-n569 |
| 41030 | 10.21233/t2fv-s268 |
| 41453 | 10.21233/t318-2y34 |
| 42712 | 10.21233/t318-n046 |
| 20223 | 10.21233/t3c4-7x23 |
| 45221 | 10.21233/t3tp-6180 |
| 16129 | 10.21233/t4b6-ax18 |
| 24669 | 10.21233/t4ct-mw46 |
| 24493 | 10.21233/t4n0-6s83 |
| 41210 | 10.21233/t55e-8g66 |
| 3973 | 10.21233/t57v-4b19 |
| 45144 | 10.21233/t6dq-4b03 |
| 24753 | 10.21233/t7ne-1h11 |
| 4117 | 10.21233/tab2-ts04 |
| 15462 | 10.21233/tadv-5c57 |
| 24145 | 10.21233/tbav-4v25 |
| 21415 | 10.21233/tbhk-kw62 |
| 3866 | 10.21233/tcav-fn36 |
| 20046 | 10.21233/td9p-m745 |
| 14676 | 10.21233/te1p-he56 |
| 24859 | 10.21233/te66-cy52 |
| 20512 | 10.21233/tegf-sg62 |
| 42745 | 10.21233/tekc-rz82 |
| 17398 | 10.21233/teyv-5z70 |
| 14510 | 10.21233/tfq7-s408 |
| 22955 | 10.21233/tgd4-nk81 |

| | | | | |
|---|---|---|---|---|
| 3872 | 10.21233/tk7w-th94 | | 4468 | 10.21233/vhxz-9969 |
| 26435 | 10.21233/tktg-sh56 | | 24032 | 10.21233/vj9n-bt57 |
| 4518 | 10.21233/tkzz-2t77 | | 22960 | 10.21233/vjdb-ec95 |
| 14641 | 10.21233/tmdc-mj73 | | 22845 | 10.21233/vkd7-vh44 |
| 1147 | 10.21233/tp4a-h605 | | 22652 | 10.21233/vmeb-gz58 |
| 4529 | 10.21233/tphv-z798 | | 15766 | 10.21233/vmwe-g130 |
| 45722 | 10.21233/tpvk-wa79 | | 15276 | 10.21233/vmxy-3004 |
| 24112 | 10.21233/tq8x-5z45 | | 22745 | 10.21233/vn5f-tz80 |
| 17328 | 10.21233/tr2m-fm40 | | 4200 | 10.21233/vpnw-pv89 |
| 25410 | 10.21233/ts6e-h593 | | 24410 | 10.21233/vq9d-x874 |
| 22938 | 10.21233/tt6f-9c78 | | 45410 | 10.21233/vqfq-2n70 |
| 15825 | 10.21233/ttaf-pc16 | | 16054 | 10.21233/vqnc-sx41 |
| 22771 | 10.21233/ttjd-w232 | | 4020 | 10.21233/vqqw-nc29 |
| 15665 | 10.21233/twkp-hz46 | | 15633 | 10.21233/vr5d-4130 |
| 45204 | 10.21233/tx0m-hk41 | | 1782 | 10.21233/vrv9-4t45 |
| 15323 | 10.21233/txns-sv58 | | 45115 | 10.21233/vs9g-8428 |
| 19931 | 10.21233/txpr-cb18 | | 41770 | 10.21233/vsfm-xz32 |
| 25131 | 10.21233/tz5g-8d59 | | 24902 | 10.21233/vtmh-9d28 |
| 4294 | 10.21233/v088-1d37 | | 15882 | 10.21233/vx1h-dj04 |
| 17359 | 10.21233/v0wk-b742 | | 4056 | 10.21233/vxzv-9k94 |
| 24048 | 10.21233/v1dw-9x16 | | 4452 | 10.21233/vy6k-x487 |
| 15363 | 10.21233/v2kw-6697 | | 15022 | 10.21233/w03f-sr85 |
| 24061 | 10.21233/v3yx-9c11 | | 14498 | 10.21233/w0ab-4w11 |
| 22640 | 10.21233/v4mx-q473 | | 42671 | 10.21233/w0c2-k512 |
| 22662 | 10.21233/v7ky-p784 | | 45719 | 10.21233/w0j3-9h31 |
| 17711 | 10.21233/v7nq-4w81 | | 17296 | 10.21233/w2gd-mz79 |
| 24343 | 10.21233/v8jm-v785 | | 43295 | 10.21233/w4n6-qv07 |
| 1582 | 10.21233/v98s-4073 | | 3999 | 10.21233/w4z8-3q21 |
| 4405 | 10.21233/v9em-3b09 | | 14564 | 10.21233/w4zt-yj57 |
| 25406 | 10.21233/vbwz-tg22 | | 25308 | 10.21233/w5qv-sv92 |
| 41401 | 10.21233/vc1n-v894 | | 15214 | 10.21233/w5rw-qd24 |
| 14955 | 10.21233/vcma-8k37 | | 25157 | 10.21233/w61s-9827 |
| 4533 | 10.21233/vdpe-dp57 | | 4069 | 10.21233/w63t-nf61 |
| 15403 | 10.21233/vds5-pf58 | | 3896 | 10.21233/w6j3-gn15 |
| 14803 | 10.21233/ve01-y167 | | 17324 | 10.21233/w72d-bx06 |
| 40990 | 10.21233/ve15-mn82 | | 41469 | 10.21233/w7bt-xr68 |
| 24279 | 10.21233/veqa-0n52 | | 45194 | 10.21233/w88p-xj69 |
| 14619 | 10.21233/vf6b-8k35 | | 41532 | 10.21233/w9z1-fn16 |
| 16016 | 10.21233/vffp-x175 | | 19933 | 10.21233/watr-af89 |
| 41460 | 10.21233/vg3e-tx36 | | 24637 | 10.21233/wb8q-wk43 |
| 41485 | 10.21233/vgxh-qv84 | | 4476 | 10.21233/wbfp-6424 |
| 4207 | 10.21233/vh8n-2090 | | 22638 | 10.21233/wc9m-v912 |
| 22664 | 10.21233/vhkm-se51 | | 41233 | 10.21233/wefk-sp33 |

| | | | |
|---|---|---|---|
| 4097 | 10.21233/weqb-ef25 | 24463 | 10.21233/x7a5-2c36 |
| 4380 | 10.21233/whjz-0k24 | 45816 | 10.21233/x7et-qe66 |
| 45710 | 10.21233/wjmk-7376 | 15719 | 10.21233/x7hq-ts05 |
| 4510 | 10.21233/wkz1-sd64 | 41758 | 10.21233/x8p5-xp81 |
| 41323 | 10.21233/wm29-be77 | 19820 | 10.21233/x8y7-re14 |
| 15103 | 10.21233/wn4z-7521 | 24927 | 10.21233/x90a-8v66 |
| 26510 | 10.21233/wnfe-f385 | 41171 | 10.21233/x91r-1a30 |
| 41587 | 10.21233/wp2z-kt38 | 4145 | 10.21233/x9b0-z153 |
| 17320 | 10.21233/wq2v-5053 | 24616 | 10.21233/x9jq-my02 |
| 3882 | 10.21233/wq2x-c027 | 4332 | 10.21233/xa3s-kj49 |
| 26520 | 10.21233/wqej-0r45 | 40914 | 10.21233/xa6z-jh17 |
| 14897 | 10.21233/wrb3-qd23 | 24639 | 10.21233/xadk-1x55 |
| 14959 | 10.21233/wrt5-qc27 | 24941 | 10.21233/xb5c-wb44 |
| 45389 | 10.21233/ws4c-an72 | 14512 | 10.21233/xc0y-6b19 |
| 22962 | 10.21233/wvsb-6130 | 4314 | 10.21233/xe1b-tf19 |
| 21796 | 10.21233/wwa4-0236 | 32262 | 10.21233/xe3h-wb42 |
| 13073 | 10.21233/wwc5-6x04 | 22953 | 10.21233/xeew-fr63 |
| 4466 | 10.21233/wwj8-s239 | 41203 | 10.21233/xeke-2h87 |
| 25382 | 10.21233/wxyd-q154 | 41506 | 10.21233/xett-n046 |
| 16181 | 10.21233/wy62-rm78 | 41911 | 10.21233/xfh1-6085 |
| 15385 | 10.21233/wy8b-3e83 | 41582 | 10.21233/xfrs-q084 |
| 3934 | 10.21233/wyaq-1b97 | 4391 | 10.21233/xj85-7c26 |
| 25003 | 10.21233/x0ba-j040 | 22801 | 10.21233/xm08-vp32 |
| 15594 | 10.21233/x0c3-ed21 | 26439 | 10.21233/xmbg-mz72 |
| 41228 | 10.21233/x0cd-0q06 | 41269 | 10.21233/xn7x-eb17 |
| 17404 | 10.21233/x0jy-tp30 | 3870 | 10.21233/xnjz-z143 |
| 24259 | 10.21233/x0z2-py02 | 21702 | 10.21233/xp8e-bd49 |
| 4304 | 10.21233/x11h-5275 | 40844 | 10.21233/xpy2-m222 |
| 4059 | 10.21233/x17y-bb18 | 4042 | 10.21233/xq55-z804 |
| 21706 | 10.21233/x19k-0e45 | 15904 | 10.21233/xqhj-c782 |
| 15878 | 10.21233/x1bk-ra92 | 22791 | 10.21233/xraa-re03 |
| 3894 | 10.21233/x1s2-s723 | 17417 | 10.21233/xrzb-zq82 |
| 15138 | 10.21233/x1ts-qh05 | 22691 | 10.21233/xsdn-cx54 |
| 18110 | 10.21233/x2md-aj39 | 41479 | 10.21233/xsk9-bx90 |
| 4024 | 10.21233/x2v2-4v32 | 24964 | 10.21233/xss2-e959 |
| 14110 | 10.21233/x34r-6x54 | 15406 | 10.21233/xstm-aq61 |
| 14633 | 10.21233/x35d-q248 | 24572 | 10.21233/xsxr-ky48 |
| 4041 | 10.21233/x43j-z193 | 4004 | 10.21233/xw05-mj88 |
| 4083 | 10.21233/x49t-wj37 | 25402 | 10.21233/xw8c-gq46 |
| 42655 | 10.21233/x4x4-tg29 | 3937 | 10.21233/xwhx-v671 |
| 3986 | 10.21233/x4xa-1k90 | 3874 | 10.21233/xwk9-q426 |
| 24593 | 10.21233/x6et-gw35 | 15244 | 10.21233/xyf7-9v15 |
| 26442 | 10.21233/x6sj-rv93 | 14491 | 10.21233/xyvh-px20 |

| | |
|---|---|
| 22736 | 10.21233/xyxq-gr38 |
| 15950 | 10.21233/xz5z-pr53 |
| 3928 | 10.21233/y0cr-gc25 |
| 20308 | 10.21233/y0pj-8d71 |
| 22352 | 10.21233/y1ej-xh18 |
| 24563 | 10.21233/y1f4-bz49 |
| 14125 | 10.21233/y28m-nq05 |
| 18051 | 10.21233/y3mq-m120 |
| 24106 | 10.21233/y42f-6218 |
| 3878 | 10.21233/y4e7-me97 |
| 16186 | 10.21233/y4mj-v434 |
| 44951 | 10.21233/y4ss-7811 |
| 41276 | 10.21233/y554-yd92 |
| 25835 | 10.21233/y5bx-3n98 |
| 22238 | 10.21233/y6je-9a17 |
| 4189 | 10.21233/y7b7-5a72 |
| 24759 | 10.21233/ya3m-xh05 |
| 24057 | 10.21233/ya53-0561 |
| 22346 | 10.21233/yb97-px46 |
| 24238 | 10.21233/ycfn-p887 |
| 14289 | 10.21233/yczc-8b58 |
| 4541 | 10.21233/yd8b-wb47 |
| 14925 | 10.21233/ydnc-0495 |
| 4249 | 10.21233/ye9x-z889 |
| 4539 | 10.21233/yeg0-6p54 |
| 24852 | 10.21233/yfcd-n026 |
| 25083 | 10.21233/yfwj-f753 |
| 40569 | 10.21233/ygbz-tf05 |
| 15939 | 10.21233/yhre-qg42 |
| 4187 | 10.21233/yj5d-ej18 |
| 24954 | 10.21233/yjbz-h684 |
| 15209 | 10.21233/yjhg-b425 |
| 41397 | 10.21233/yktz-5q37 |
| 14944 | 10.21233/ym56-bk43 |
| 21915 | 10.21233/ymye-gn96 |
| 41218 | 10.21233/yp3p-5z78 |
| 22931 | 10.21233/yp88-eq30 |
| 24166 | 10.21233/yr7f-7517 |
| 41562 | 10.21233/ysrz-ve64 |
| 15223 | 10.21233/yt10-kv93 |
| 24491 | 10.21233/ytdh-rq41 |
| 24358 | 10.21233/yvzf-7m50 |
| 15205 | 10.21233/yw2g-1t26 |

| | |
|---|---|
| 16175 | 10.21233/ywa8-4h38 |
| 15976 | 10.21233/ywkn-vk84 |
| 41580 | 10.21233/ywn4-f081 |
| 21772 | 10.21233/yxa5-7p09 |
| 14291 | 10.21233/yxsc-gf06 |
| 41121 | 10.21233/yxxp-as21 |
| 45707 | 10.21233/yxzd-xn66 |
| 22812 | 10.21233/yy4y-0p72 |
| 4340 | 10.21233/yys3-qz47 |
| 14792 | 10.21233/yzwy-0p89 |
| 17945 | 10.21233/yzzz-aa38 |
| 17377 | 10.21233/z08c-rb24 |
| 21766 | 10.21233/z256-fh92 |
| 15197 | 10.21233/z377-nz81 |
| 24611 | 10.21233/z3vf-sk02 |
| 17385 | 10.21233/z5j4-8v57 |
| 17366 | 10.21233/z627-yk93 |
| 41159 | 10.21233/z7gy-mw75 |
| 15247 | 10.21233/z7yn-c813 |
| 4323 | 10.21233/z80c-sf19 |
| 17347 | 10.21233/z81b-ch22 |
| 25289 | 10.21233/z8a8-vz94 |
| 4272 | 10.21233/z913-g676 |
| 3887 | 10.21233/z9wq-cc18 |
| 24030 | 10.21233/za1e-fb06 |
| 15730 | 10.21233/zakg-m595 |
| 20287 | 10.21233/zc0w-ge65 |
| 14539 | 10.21233/zcrn-2h47 |
| 4497 | 10.21233/zcst-x145 |
| 15660 | 10.21233/zd0z-ve57 |
| 24854 | 10.21233/zd47-af67 |
| 14941 | 10.21233/zdh7-r634 |
| 45398 | 10.21233/zdqt-kr34 |
| 41926 | 10.21233/zdyp-3t68 |
| 15796 | 10.21233/ze2y-2e90 |
| 15715 | 10.21233/zedd-cm35 |
| 2028 | 10.21233/zesr-6507 |
| 4309 | 10.21233/zf76-3490 |
| 16079 | 10.21233/zh76-mx93 |
| 14799 | 10.21233/zhp9-cm48 |
| 4359 | 10.21233/zhsh-zr85 |
| 14629 | 10.21233/zjqh-j697 |
| 22831 | 10.21233/zjr5-v667 |

| | |
|---|---|
| 4374 | 10.21233/zjtz-k726 |
| 20165 | 10.21233/zk15-wq18 |
| 40791 | 10.21233/zkeb-rj50 |
| 15933 | 10.21233/zkg7-4840 |
| 4505 | 10.21233/zkna-r256 |
| 15902 | 10.21233/zn7m-gc31 |
| 12 | 10.21233/znex-sp94 |
| 15682 | 10.21233/zpe5-s053 |
| 40168 | 10.21233/zq7z-7471 |
| 24078 | 10.21233/zqez-kr13 |
| 41335 | 10.21233/zqj0-ma56 |
| 24149 | 10.21233/zqk8-aa65 |
| 4372 | 10.21233/zqse-ep64 |
| 4507 | 10.21233/zr55-9j43 |
| 42714 | 10.21233/zspv-ta39 |
| 24243 | 10.21233/zvcd-yn53 |
| 13053 | 10.21233/zvn7-m788 |
| 24302 | 10.21233/zw4f-ce58 |
| 13071 | 10.21233/zw81-nc35 |
| 20372 | 10.21233/zwdj-4e50 |
| 4404 | 10.21233/zwpy-vk55 |
| 4498 | 10.21233/zx8x-vj28 |
| 4050 | 10.21233/zxz5-q261 |

---

## Author Comment (AC5)

**Letter to the Editor**

Laura Schild and Ulrike Herzschuh

Dear Kirsten Elger,

Thank you for opening our manuscript to community discussion. We appreciate the reviewers for their insightful feedback and constructive comments. We made use of a global compilation of pollen data, and our collected metadata, such as basin size and basin type to reconstruct past vegetation using REVEALS.Notably, none of the reviewers criticized our general implementation of the method to reconstruct past vegetation using lake and peat sediments, the details which are criticized can be easily addressed. In addition, the validation of our results shows clear improvements in forest cover reconstruction compared to pure, uncorrected pollen data highlighting the validity of our dataset. Any issues raised by the reviewers and community members are either easily resolved or have already been implemented in our revisions and responses.

Marie Gaillard pointed out a wording problem with the name of the 80% pollen source area. We agree with her feedback and will change the name to improve clarity. She also asked for clarification on the inclusion of small sites. Since our dataset is intended to be used as a gridded dataset, we have provided an additional script to facilitate the inclusion of small sites. We will also flag small basins and peatland reconstructions in the dataset to avoid using these unreliable site-wise reconstructions alone.

Michela Mariani raised the issue of continental RPP syntheses and the inclusion of hemispheric values being a strong generalization. While we agree that this is the case, continental syntheses are common in large-scale pollen-based vegetation reconstructions and have been shown to yield improved results (e.g. Dawson et al., 2024; Githumbi et al., 2022; Serge et al., 2023; Trondman et al., 2015). This is also true in our reconstruction, as demonstrated by our validations, which are free from circularity. We tried to highlight and discuss the uncertainty regarding reconstructions in the Southern Hemisphere in our manuscript, but agree with Michela Mariani that excluding the Southern Hemisphere would be more sensible and will implement this change.

Williams et al. emphasized the need for better open science practices. In response, we have added citations to Neotoma and included the DOIs of the datasets used from Neotoma. Similarly, Giesecke also raised the need for Neotoma citations, which we have addressed as described above. Additionally, he pointed out the uncertainty in the Southern Hemisphere, and as mentioned, we agree with this point and have decided to exclude it for consistency. Furthermore, we will only publish forest cover reconstructions from the optimized dataset and exclude taxonomic reconstructions to address the uncertainty regarding this approach.

We emphasize that all these changes are feasible and have either already been implemented or are in the process of being implemented. We believe these revisions significantly improve our manuscript and would like to submit our revised version for your consideration.

Best regards,

Laura Schild and Ulrike Herzschuh

Dawson, A., Williams, J.W., Gaillard, M.-J., Goring, S.J., Pirzamanbein, B., Lindstrom, J., Anderson, R.S., Brunelle, A., Foster, D., Gajewski, K., Gavin, D.G., Lacourse, T., Minckley, T.A., Oswald, W., Shuman, B., Whitlock, C., 2024. Holocene land cover change in North America: continental trends, regional drivers, and implications for vegetation-atmosphere feedbacks. Clim. Past Discuss. 1–52. https://doi.org/10.5194/cp-2024-6

Githumbi, E., Fyfe, R., Gaillard, M.-J., Trondman, A.-K., Mazier, F., Nielsen, A.-B., Poska, A., Sugita, S., Woodbridge, J., Azuara, J., Feurdean, A., Grindean, R., Lebreton, V., Marquer, L., Nebout-Combourieu, N., Stančikaitė, M., Tanţău, I., Tonkov, S., Shumilovskikh, L., LandClimII data contributors, 2022. European pollen-based REVEALS land-cover reconstructions for the Holocene: methodology, mapping and potentials. Earth Syst. Sci. Data 14, 1581–1619. https://doi.org/10.5194/essd-14-1581-2022

Serge, M.A., Mazier, F., Fyfe, R., Gaillard, M.-J., Klein, T., Lagnoux, A., Galop, D., Githumbi, E., Mindrescu, M., Nielsen, A.B., Trondman, A.-K., Poska, A., Sugita, S., Woodbridge, J., Abel-Schaad, D., Åkesson, C., Alenius, T., Ammann, B., Andersen, S.T., Anderson, R.S., Andrič, M., Balakauskas, L., Barnekow, L., Batalova, V., Bergman, J., Birks, H.J.B., Björkman, L., Bjune, A.E., Borisova, O., Broothaerts, N., Carrion, J., Caseldine, C., Christiansen, J., Cui, Q., Currás, A., Czerwiński, S., David, R., Davies, A.L., De Jong, R., Di Rita, F., Dietre, B., Dörfler, W., Doyen, E., Edwards, K.J., Ejarque, A., Endtmann, E., Etienne, D., Faure, E., Feeser, I., Feurdean, A., Fischer, E., Fletcher, W., Franco-Múgica, F., Fredh, E.D., Froyd, C., Garcés-Pastor, S., García-Moreiras, I., Gauthier, E., Gil-Romera, G., González-Sampériz, P., Grant, M.J., Grindean, R., Haas, J.N., Hannon, G., Heather, A.-J., Heikkilä, M., Hjelle, K., Jahns, S., Jasiunas, N., Jiménez-Moreno, G., Jouffroy-Bapicot, I., Kabailienė, M., Kamerling, I.M., Kangur, M., Karpińska-Kołaczek, M., Kasianova, A., Kołaczek, P., Lagerås, P., Latalowa, M., Lechterbeck, J., Leroyer, C., Leydet, M., Lindbladh, M., Lisitsyna, O., López-Sáez, J.-A., Lowe, J., Luelmo-Lautenschlaeger, R., Lukanina, E., Macijauskaitė, L., Magri, D., Marguerie, D., Marquer, L., Martinez-Cortizas, A., Mehl, I., Mesa-Fernández, J.M., Mighall, T., Miola, A., Miras, Y., Morales-Molino, C., Mrotzek, A., Sobrino, C.M., Odgaard, B., Ozola, I., Pérez-Díaz, S., Pérez-Obiol, R.P., Poggi, C., Rego, P.R., Ramos-Román, M.J., Rasmussen, P., Reille, M., Rösch, M., Ruffaldi, P., Goni, M.S., Savukynienė, N., Schröder, T., Schult, M., Segerström, U., Seppä, H., Vives, G.S., Shumilovskikh, L., Smettan, H.W., Stancikaite, M., Stevenson, A.C., Stivrins, N., Tantau, I., Theuerkauf, M., Tonkov, S., van der Knaap, W.O., van Leeuwen, J.F.N., Vecmane, E., Verstraeten, G., Veski, S., Voigt, R., Von Stedingk, H., Waller, M.P., Wiethold, J., Willis, K.J., Wolters, S., Zernitskaya, V.P., 2023. Testing the Effect of Relative Pollen Productivity on the REVEALS Model: A Validated Reconstruction of Europe-Wide Holocene Vegetation. Land 12, 986. https://doi.org/10.3390/land12050986

Trondman, A.-K., Gaillard, M.-J., Mazier, F., Sugita, S., Fyfe, R., Nielsen, A.B., Twiddle, C., Barratt, P., Birks, H.J.B., Bjune, A.E., Björkman, L., Broström, A., Caseldine, C., David, R., Dodson, J., Dörfler, W., Fischer, E., van Geel, B., Giesecke, T., Hultberg, T., Kalnina, L., Kangur, M., van der Knaap, P., Koff, T., Kuneš, P., Lagerås, P., Latałowa, M., Lechterbeck, J., Leroyer, C., Leydet, M., Lindbladh,

M., Marquer, L., Mitchell, F.J.G., Odgaard, B.V., Peglar, S.M., Persson, T., Poska, A., Rösch, M., Seppä, H., Veski, S., Wick, L., 2015. Pollen-based quantitative reconstructions of Holocene regional vegetation cover (plant-functional types and land-cover types) in Europe suitable for climate modelling. Glob. Change Biol. 21, 676–697. https://doi.org/10.1111/gcb.12737

---

## Author Response (AR1)

**Letter to the Editor**

Laura Schild

Dear Kirsten Elger,

Thank you for the opportunity to revise our manuscript. We have carefully considered the reviewers' comments and revised the manuscript accordingly. We are including detailed responses and changes to each comment in this document. Below we list the general changes we have made.

We are removing the optimization of RPP values and the reconstruction of the southern hemisphere vegetation from the manuscript because of the large uncertainties associated with these estimates. We also exclude all datasets that are not suitable for reconstruction with REVEALS, and we rename our calculated 80% pollen source area and expand on its applicability and calculation. We also add several more recent RPP studies to the synthesis of RPP values. To adequately acknowledge Neotoma and its constituent databases, we add citations and edit our Acknowledgments section. To clarify the use of the dataset, we expand the section on applications and limitations of the dataset and even provide a script with adjustable rasterization (https://zenodo.org/records/12800291). Records that are uncertain for site-specific analyses are highlighted in the dataset. We believe that these revisions have significantly improved our manuscript and have submitted the revised version.

We have updated our dataset accordingly and uploaded the new version to Zenodo (https://zenodo.org/records/12800159). Once the reviewers are satisfied with the changes made, we will update the dataset on PANGAEA and update the links provided in the Code and Data Availability section.

Best regards,

Laura Schild

**Reply to Gaillard**

Laura Schild & Ulrike Herzschuh

**General reply**

Dear Marie-José Gaillard,

We thank you for your extensive review of our manuscript and the constructive comments. While you welcome our work towards global pollen-based vegetation reconstructions and do not criticize the general method you stated helpful comments regarding some concerns. We are confident we have been able to implement all of them .

Large basins provide great value to large-scale reconstructions and site-wise reconstructions will only be kept if the original basin is of sufficient size (>= 50ha). We do agree with and recognize the limited suitability of small basins for site-wise reconstructions. Our previous intention was to create a dataset that could be gridded at different resolutions desired by the user, which we did not emphasize enough in our manuscript. We believe this flexibility would improve the usefulness of the reconstructions. To still acknowledge and highlight potential downfalls here, it is pertinent to both flag unreliable, small basins in the data set and provide a detailed description of potential uses and an adjustable script for rasterization. We implemented this in our revisions. Additionally, any non-lake or peat basins will be removed from the data set as they are unfitting for reconstruction by REVEALS.

The naming of our calculated source area is indeed unfortunate and will be changed. We will also expand on its calculation and usability in the manuscript text. Additionally, new RPP studies were added to our synthesis.

We have been able to implement all of these adjustments and believe it improved the clarity of the manuscript greatly. Below we respond in detail to each major issue raised and the detailed comments.

An updated version of the dataset can now be found here:
https://doi.org/10.5281/zenodo.12800159. The dataset on PANGAEA will be updated as soon as possible.

Best regards
Laura Schild and Ulrike Herzschuh

**Specific replies**

**Major Issues**

**Pollen records appropriate for the application of the REVEALS model to reconstruct REGIONAL plant cover**

Original comment

1. The REVEALS model was developed to reconstruct REGIONAL plant cover using pollen records from LARGE LAKES, alternatively multiple SMALL LAKES (Sugita 2007a, REVEALS model). Trondman et al. (2016) (VHA) tested the REVEALS model using MULTIPLE SMALL SITES (lakes and bogs) and concluded that pollen records from MULTIPLE SMALL BOGS could be used, ideally in mixture with pollen records from LARGE and/or SMALL lakes. Thus:
2. The REVEALS model is NOT appropriate to reconstruct regional plant cover using pollen records from SINGLE SMALL sites (lakes or bogs) and from LARGE BOGS (single or multiple). See Sugita (2007a, REVEALS model) for the definition of large lake, and Trondman et al. (2015) for the choice of 50 ha as a "practical" delimitation between small (< 50 ha) and large ( >50 ha) sites.
3. The REVEALS model IS appropriate to reconstruct plant cover using pollen records from SINGLE LARGE LAKES (however always better with records from SEVERAL LARGE LAKES in the same vegetation region); and it is also appropriate using pollen records from MULTIPLE SMALL LAKES (Sugita 2007a REVEALS model) and from a mixture of SMALL SITES (bogs and lakes) (Trondman et al., 2016).
4. In the LandCover6k protocol, LARGE BOGS are used, but the reconstructions are considered as not or less reliable (information provided in the publications) if they include: (1) only one large bog record, (2) several large bog records and no lake record or too few lake records relative to the number of large bog records.
5. The REVEALS model is NOT appropriate using pollen records from marine sediments or other types of sites receiving large amounts of pollen from rivers or surface run-off. The LandCover6k reconstructions have excluded marine and large deltas pollen records. Pollen records from lagunes that are sufficiently sheltered from the sea can be used.
6. **All the points made above, and the first point made below, imply that** (1) the dataset of single site REVEALS estimates of plant cover CANNOT BE USED AS SUCH as each REVEALS reconstruction from a single small site (bog or lake) and a single large bog is incorrect; (2) Only REVEALS estimates using pollen records from single LARGE LAKES or REVEALS **MEAN** ESTIMATES based on the REVEALS estimates from **MULTIPLE SITES** are correct and can therefore be used. This also implies that IF THIS DATASET IS MADE OPEN ACCESS FOR USE, **it MUST BE CLARIFIED for the user what can be done AND NOT DONE with these single sites REVEALS estimates**, i.e. (1) one CANNOT use the original single site REVEALS plant cover if the pollen record is from a SMALL SITE (lake or bog) or a LARGE BOG. (2) one **CAN** calculate **MEAN REVEALS estimates** within regions from ca. 50 km x 50 km (se Hellman et al., 2008b in VHA and Trondman et al., 2015 in GCB) up to whole regions or continents (the latter continental scale is provided in Schmid et al.'s dataset, but nothing else).

Reply

Our site-wise reconstructions using large lakes are still valid and their information can be used in gridded versions of this data set. We agree and recognize that reconstructions from small lakes and peatland sites should not be used alone as site-wise reconstructions. Our validation

was adjusted to also use these sites aggregated in grid cells and not individually. Revised Figures 9 and 11 below show this validation.

[Figure]

**Figure 9.** Remote sensing forest cover (LANDSAT) and modern reconstructed forest cover from Pollen and REVEALS (< 100 years BP) in 2x2° grid cells with mean absolute errors (MAE) and correlation coefficient ($R$) per group. Reconstructed forest cover from the original pollen data tends to overestimate observed (remote sensing) forest cover. Improvements with the REVEALS reconstruction are especially high in Europe. Validations with different grid cell sizes are available in the supplement (S3: Validation results for different spatial resolutions).

[Figure]

**Figure 11.** Map of the reconstruction error (in % forest cover) for forest cover reconstructed from Pollen and REVEALS data. Remaining errors with the overall better REVEALS reconstructions are especially high in North America (Northern West Coast, Labrador Peninsula).

Our aim is for the data set to be used flexibly, meaning that users can set their own temporal and spatial resolution for rasterization.This is why we did not prepare a set rasterization. To highlight this use case we **provided a script to rasterize the dataset dynamically and classify grid cell reliability by record availability (https://zenodo.org/records/12800291 )**. Additionally, we expanded on this in our data usability section and clarify how we intend the data set to be used reliably. Small sites and peatland will also receive an additional flag in the data set.

Tracked changes (data usability):

To ensure the correct utilization of the dataset and to obtain reliable analysis results, several key considerations should be followed. Firstly, rasterization mitigates individual errors by temporal and spatial averaging. This process is particularly

390 useful in reducing the variance that might arise from individual measurements, providing a more reliable representation of the underlying signal.The reliability of reconstructions varies among different taxa due to the quality of RPP values, and this is explicitly documented in a supplementary file that outlines the sources of RPP values (see Section Code and Data availability). Reconstructions of taxa with continental RPP values are the most reliable, followed by those based on hemispheric data, with standardized RPP values being the least reliable. This hierarchy should be taken into account when interpreting the results.

395 Higher certainty is associated with forest cover reconstruction, as it is based on aggregation among taxa. Reconstructions of temporal forest cover trends are reliable, as evidenced by high correlation coefficients, despite a tendency for absolute values to be overestimated, particularly in North America. For individual time series, the reliability of data varies with the size of the lakes from which samples were taken. Only data derived from large lakes ($\geq$ 50 ha) are reliable for site-wise analyses. This distinction is clearly indicated with validity flags in the dataset. Reconstructions from smaller basins should not be used alone.

400

**Implementation of the REVEALS model and pollen source areas**

**Original comment**

7. This first point is also relevant for the issue discussed above: The authors (Schild et al.) use Theuerkauf et al. (2016) "REVEALSinR" to implement the REVEALS model as an alternative to Sugita's REVEALS programme (last revised in 2022) or the R REVEALS program by Petr Kunes. The use of "REVEALSinR" implies that the REVEALS model assumptions (Sugita 2007a; further discussed and explained in e.g., Githumbi et al. (2022), Li et al. (2020; 2023) must be considered while implementing the model. For example, the selection of appropriate sites and the number of pollen records used for the reconstruction are essential. If using pollen records from SMALL sites, the larger the number of sites/pollen records the better. The use of a single small site or a single LARGE BOG will provide biased reconstructions that won't be useful for the analysis of past plant cover, neither at the regional scale nor at the local scale. For instance, Theuerkauf et al. 2016 write: "*Like the original REVEALS programme, 'REVEALSinR' includes a function to address deposition in lakes (for details see ESM). Both the original REVEALS programme and 'REVEALSinR' only consider atmospheric pollen deposition (and lake mixing); neither model is applicable to sites that receive significant amounts of pollen from rivers, streams or surface run-off*". Theuerkauf et al. (2016) do not say explicitly that REVEALS can be used with pollen records from SINGLE small sites. But, very unfortunately, they confuse the reader by introducing severe misunderstandings in their description of the REVEALS model and its application (selection of pollen records). For instance (under "Principles of 'REVEALSinR'": "*The REVEALS model (Sugita 2007a) is based on the assumption that pollen deposition of a plant taxon in a large lake or **peatland** is equal to the mean abundance of that taxon in the region, multiplied by its pollen productivity and its 'pollen dispersal-deposition coefficient' K….etc*".

Sugita 2007a calls his model REVEALS, i.e. Regional Estimates of Vegetation Abundance from Large **Sites" BUT** it is developed for pollen deposited in **LAKES** and tested theoretically with simulated pollen records from **LAKES**. Further, one of the assumptions of the REVEALS model is that **the deposition basin is NOT COVERED BY VEGETATION**. It follows, therefore, that REVEALS is not appropriate for pollen records from large bogs. Another unfortunate issue in Theuerkauf et al. (2016) is the use of small sites in one of the tests of the effect of different pollen dispersal models on the REVEALS reconstruction, **although the second experiment uses a pollen record from a LARGE LAKE, which is correct!**, i.e., (under Materials and Methods, in relation to the first experiment): "*We associate the record with lakes and peatlands of different size (100–10,000 m in diameter), using different cut-off distances for the tail of the GPM (50 km to infinity). This cut-off sets an arbitrary limit to the maximum distance pollen may travel (the region considered as pollen source area). The cut-off for the LSM is set to 100 km, which is the calculated average distance at which 95 % of the pollen has settled (cf. Fig. 1).*" The latter implies that the authors use REVEALS for single sites from 1 to 100 ha. Moreover, they use the fact that the "Radius of the 80 % source area of pollen" for sites of 1 ha or 100 ha are not significantly different to argue that what makes the largest difference between sites of different size is the pollen dispersal model. This is true for the "pollen source area" defined as the characteristic radius for 80% or 90% etc…. of the pollen reaching the site, but it is NOT true for the size of the area a quantitative pollen reconstruction of plant cover represents when pollen records are from **SMALL SITES. See the LOVE model (Sugita 2007b) and definition of RSAP (Sugita, 1994). This is a typical example of what a published paper having got weak reviews may lead to in studies by scientists that do not go back to the sources, in this case the description of the REVEALS model by its author!..... A good scientist should know that all what's published is not necessarily correct, especially today as the review system is close to collapse due to a too large article production in comparison to the number of reviewers that have the appropriate expertise to evaluate a new study. GO BACK TO THE ORIGINAL SOURCES!**

Reply

As outlined above, we agree that peatland sites as well as small lakes are unsuitable for a site-wise reconstruction, which is why we will **aggregate them in grid cells and highlight in the manuscript that this is the intended use case (https://zenodo.org/records/12800291 ).** This will be supplemented by a script to create flexibel rasterized data sets. See the tracked changes for the reconstruction methods below.

> Forest cover was reconstructed by summing up percentages of arboreal taxa (see S1S2: List of arboreal taxa) with Betulaceae, *Betula*, and *Alnus* being classified as arboreal at sites below 70° N. The mean reconstructed compositional coverages from the REVEALS results were used for the forest cover reconstructions. REVEALS results were then rasterized to aggregate and
> 155 include records from smaller basins as well. Reconstructed time series were averaged in 500 year bins and then rasterized in grids of differing spatial resolution. A grid cell was classified as having a valid reconstruction when it contained records from at least one large lake (>= 50 ha) or at least two small basins following Serge et al. (2023). Standard deviations of the REVEALS estimates were aggregated by applying the delta method by Stuart and Ord (1994), using the same equation as Wieczorek and Herzschuh (2020). We provide a script for rasterization with adjustable temporal and spatial resolution for users
> 160 of the dataset on Zenodo (https://zenodo.org/doi/10.5281/zenodo.12800290). For validation, the reconstructed forest cover of

Additionally, the 80% pollen source area is indeed less informative if calculated from small sites. We **removed calculated values from the small lake and peatland sites** in our data set. Of course we do not intend to recreate the RSAP as defined by Sugita in our data set creation, but the area where 80% of pollen originates from. The overlap in naming is indeed unfortunate,

which is why we see no problem **changing the name of our calculated parameter** to "80% pollen source area". See tracked changes for this terminology change below.

**2.2.2 Modifications in REVEALSinR**

We calculate the radius of  the 80% pollen source area by finding the radius in which the median influx of all taxa is 80% of the total influx (as defined by the total influx in the maximum extent of regional vegetation chosen). This is calculated

**Original Comment**

8. The authors of the discussed paper (Schmid et al.) claim that they are calculating the "RELEVANT SOURCE AREA" of each site, small and large (although it says RELATIVE pollen source area in the abstract". It is unclear how this is calculated. Under 2.2.2 it says, *"We calculate the radius of relevant pollen source area by FINDING THE RADIUS IN WHICH THE MEDIAN INFLUX OF ALL TAXA IS 80% OF THE TOTAL INFLUX (as defined by the total influx in the MAXIMUM extent of REGIONAL VEGETATION CHOSEN)"*. This seems to be the source area of pollen as defined by Theuerkauf et al (2016). **This in any case NOT the RELEVANT SOURCE AREA OF POLLEN (RSAP)**. RSAP was defined originally by Sugita (1994, Ecology) and can only be estimated for SMALL SITES using the LOVE model (backwards modelling approach; Sugita 2007b The LOVE model) or the ERV model and a forward modelling approach (Hellman et al., 2009, R. Pal. Pal). **The RSAP is the minimum size of the area for which the LOVE estimates of plant cover using pollen records from SMALL SITES is valid.** The maximum size of the area cannot be calculated. The definition of the pollen source area by Schmid et al. mentioned above seems to correspond to the "characteristic radius" approach first described by Prentice (1988). This method is generally used to estimate the parameter *Zmax* needed to apply the REVEALS model (see examples in Hellman et al. 2008b in VHA; and in Gaillard et al. 2022, see Figure below). Zmax is defined as the maximum extent of the regional vegetation and is not estimated in the REVEALS programme by Sugita. *Zmax* is not the same as RSAP and it is not either necessarily the size of the area for which a REVEALS plant cover reconstruction (using appropriate pollen records!) is valid (or most valid). See point 9 below.

From Supplementary Material for Gaillard, M.-J. Githumbi, E., Achoundong, G., Lézine, A.-M., Hély, C., Lebamba, J., Marquer, L., Mazier, F., Li, F., and Sugita, S. (2021). "The challenge of pollen-based quantitative reconstruction of Holocene plant cover in tropical regions: A pilot study in Cameroon." In: Runge, J., Gosling, W., Lézine, A.-M., and Scott, L. (eds) Quaternary Vegetation Dynamics. The African Pollen Database, pp. 183- 1518 205. CRC Press. eBook ISBN9781003162766, Taylor and Francis Group. https://doi.org/10.1201/9781003162766-12

*"$Z_{max}$ (distance within which most pollen comes from) is a parameter needed to apply the ERV model (see equation above). A way to estimate this distance is to calculate the "characteristic radius" (CR) sensu Prentice (1988) for each taxon involved in the ERV analysis and for the "basin size" (or radius) of the sample site (0.5 m for soil samples, lake size for sediment samples) using the taxa FSP (e.g. Hellman et al., 2008b). We calculated CR using Prentice's bog model (GPM) and the Sutton's parameters cz (vertical diffusion coefficient, 0.12); cy (horizontal diffusion coefficient, 0.21), n (empirical coefficient, 0.25), and u (wind speed, 3 m/s).The CR of the 12 taxa used in this study (Table 2, above) for a basin size of 0.5 m (soil sample) (Figure 1) implies that 90% of three pollen taxa are coming from > 200 km (e.g. Moraceae, ca. 250 km) and 90% of nine pollen taxa are coming from < 200 km (e.g. Syzygium, ca. 290 km (max CR); Macaranga, ca. 150 km; Podocarpus, ca. 100 km; Poaceae, ca. 20 km (min CR)). ≤ 85% of all 26 taxa used in the first ERV model run come from ≤ 200 km (all results not shown here). Therefore, $Z_{max}$ was set to 200 km."*

9. Theuerkauf et al. (2016) also discuss the size of the area represented by REVEALS estimated plant cover (under Discussion): *"REVEALS output is commonly interpreted as representing the regional vegetation composition— but how large is this region? Or, where does the pollen come from? There is no simple answer because pollen arrives from nearby as well as far away, with*

*nearby sources contributing (much) more (Janssen 1966). Prentice and Webb (1986) suggested approximating the source area as the area outside the basin from which e.g. 80 % of total pollen deposition arrives. For large lakes and peatlands with 1,000 m diameter (***MJG: e.g. large sites***!), the LSM predicts that the size of the 80 % source area is \*55 km for all taxa, whether with high or low fall speed. In contrast, the conventional GPM for neutral conditions predicts a large difference in the 80 % source area of taxa with low (\*120 km) and taxa with high fall speed (12 km; Table 1). Whereas the unrealistic GPM defies definition of a distinct source area, the realistic LSM offers a clear delineation." (……).* The latter result is perfectly logical but does not mean that it is not possible to define a pollen source area with the definition "*the area outside the basin from which e.g. 80 % of total pollen deposition arrives".* In the results presented by Theuerkauf et al. (2016) **the pollen source area is ca 55 km in diameter with the LSM and for the GPM it is maximum 120 km (i.e. distance for the low fall speed pollen, the distance for the high fall speed pollen being smaller, but for ALL the pollen types together, the max distance becomes 120 km.**

10. On the subject of "the size of the area represented by REVEALS estimated plant cover", Shinya Sugita writes in Li et al. (2022; pages 4-5): "*When REVEALS is applied using pollen records from multiple sites, one of the important assumptions is that there is no spatial gradients in vegetation composition within the multiple sites region (Sugita, 2007a). In addition, it is assumed (and computer simulations support it) that,* **when the basin size is >100 ha, the site-to-site variation of pollen assemblages becomes negligible even if the spatial structure of vegetation is highly patchy** *(Sugita, 2007a). Accordingly,* underlined *the averaged values* of the REVEALS estimates using pollen records from **multiple large sites** (**MJG: and multiple small sites, see Hellman et al., 2016**) *approximate the species composition of the regional vegetation reasonably well* as simulations and empirical studies have demonstrated (e.g. Hellman et al., 2008a, b). In theory and practice, however, the strict definition of the pollen source area is difficult for REVEALS application. **Sugita (2007a) defined it as the area within which most of the pollen comes from (Zmax).** Simulations and previous empirical studies (e.g. Sugita, 2007a, b; Hellman et al., 2008b; Sugita et al., 2010; Mazier et al., 2012) **have indicated that, when the radius of the source area defined varies from 50 km to 400 km, the REVEALS results of regional vegetation reconstruction do not change significantly.** The basin size is potentially important for REVEALS-based estimate of regional vegetation because differences in basin size among sites can lead to a significant site-to-site variation in the pollen assemblages. **However, as long as the multiple study sites are located within a region that satisfies the first assumption as described above (no gradients in the overall vegetation composition), the averaged REVEALS estimates effectively represent the regional vegetation composition as demonstrated in Hellman et al., 2008a.** The accuracy of the reconstructed vegetation against the observed vegetation composition was assessed for areas of 50 km × 50 km and 100 km × 100 km around each site in two regions of southern Sweden. The pollen records used are from 5 large lakes in each region, thus 10 lakes in total, that vary in size between 76 ha and 1965 ha. The results support the main conclusions and implications for the REVEALS application based on the theory and the simulations described in Sugita (2007a). Such evaluation is an essential step for credible application of the REVEALS model. Unfortunately, no other evaluation studies following the strategy of Hellman et al., 2008a have been published so far for other regions of the world.*"

11. Theuerkauf et al. also write: "*Therefore, in situations where **regional vegetation is expected to be patchy**, approaches that do not rely on **homogeneity** are preferable to REVEALS. For **a single site**, multiple scenario approaches allow the detection of vegetation mosaics (Fyfe 2006; Bunting et al. 2008)."* . "Patchy" is not the same as "non homogenous" (see e.g., Hellman et al., 2009a in Rev. Pal. Pal.) and above. The regional
vegetation can be patchy for a REVEALS application as long as the patchiness is
homogenous, see also point 10 above.

**12. All the points above imply that the authors of the discussed paper (Schmid et al.) MUST clarify how their calculate the "pollen source area" of each site (lake or bog, large or small), what the definition of that source area is, and what it can be used for when it is calculated for a small site, given that it is not the same as the RSAP and does not necessarily define the size of the area for which the REVEALS reconstruction of plant**

**cover is valid.**

As outlined above, we **adapted our terminology** and call the parameter calculated by us "80% pollen source area". We define it as the area from which the median relative influx of all taxa is 80%. This is calculated by employing the lake deposition model in Theuerkauf et al.'s REVEALSinR. Starting from zmax the deposited pollen is calculated per taxon. This is assumed to be the total pollen each taxon deposits. This is of course not the reality as pollen can originate from even much further and fluvial inputs into lakes are inevitable as well. But this is an assumption that REVEALS makes as well. In a step-wise process the radius around the basin is increased and the deposited pollen relative to the total influx at zmax calculated for each taxon. We define our 80% pollen source radius as the radius where the median of the relative influx of all taxa reaches 80%. The aim of this calculation is mainly to give an idea of the scale of source area to users not familiar with pollen data. It emphasizes the regional character of lacustrine pollen data and showcases how lake size influences this source area. Please see the tracked changes below with an expanded explanation.

**2.2.2 Modifications in REVEALSinR**

We calculate the radius of  the 80% pollen source area by finding the radius in which the median influx of all taxa is 80% of the total influx (as defined by the total influx in the maximum extent of regional vegetation chosen). This is calculated by employing the lake deposition model in REVEALSinR (Theuerkauf et al., 2016). Starting from $z_{max}$ the deposited pollen

140 is calculated per taxon. This is assumed to be the total pollen each taxon deposits. In a step-wise process the radius around the basin is increased and the deposited pollen relative to the total influx at $z_{max}$ is calculated for each taxon. We define our 80%

pollen source radius as the radius where the median of the relative influx of all taxa reaches 80%. The primary objective of this calculation is to provide a clear understanding of the scale of the source area for users unfamiliar with pollen data. It highlights the regional nature of lacustrine pollen data and demonstrates the influence of lake size on this source area.

145 We also reduced computational effort in REVEALSinR by implementing a maximum number of steps in the lake model used to model mixing in the basin. The number of steps was set to 500 unless $n$ falls below that maximum value for $n = basin\,radius/10$ for basins with a radius of at least 1000 m and $n = basin\,radius/2$ for basins with a radius smaller than 1000 m.

**Selection of RPP dataset and RPP values**

1. The RPP dataset used is the one published by Wieczorek and Herzschuh (2020). The RPP used in Schmid et al (this paper) are mean RPPs based on 1 to n original RPP values, **they are NOT original values**.
2. The Wieczorek and Herzschuh (2020) (WH) dataset does not include original RPP values from the southern hemisphere and doesn't use RPPs published since 2020 in China, Europe, and subtropical/tropical regions as well as Australia. As far as I can see the WH dataset uses only the original values in Commerford et al. (2013) for N America, but not the values from Calcote and others (?).
3. Both points MUST be clarified. As it stands now it looks like a) there are no original RPP values from the southern hemisphere/sub-tropical and tropical regions and b) northern hemisphere RPP values are used for the southern hemisphere and when the SH taxa do not exist in NH the RPP

is put to 1. Moreover, Tables A1 and A2 may give the impression to the reader that all these taxa have original RPP values in all continents. **It is VERY CONFUSING! CLARIFY. Do not call the values in Table A1 "Original RPP", these are means of original RPPs AS SELECTED AND CALCULATED BY WH! And provide the Tables A1 and A2 in a different format, with the list of taxa only ONCE in the first column and ascribe the following columns to the different continents/regions you have defined. Indicate for each continent whether the RPP value you use is a mean of original values (1-n) from a single continent or several (for instance with an asterisk for the mean value used and indicate with e.g. a cross the continents in which those values are used although there are no original values in those continents. Also indicate what RPP value used is based on a single original RPP value! It would also be useful if the taxa within a family for which a RPP value exists are named, for instance Thymelaceae (only one value and it is for *Stellara* (China), and Orobanchaceae, only one value for *Rhinanthus* type (Europe), etc. This will make the RPP dataset much more transparent, the reader will have the direct information of whether the RPP value is robust for the continent in question or not, without having to go back to WH (2020) and the ESM in there. You could also indicate for the taxa you have put RPP=1 in case there is an original value published since WH (2020) that can be used as an alternative to 1 or your "optimized values", for instance for Alchornea, Melastomataceae, Podocarpus etc (e.g., Gaillard et al., 2021). NOTE: correct the spelling errors of the plant taxa names!**

**Finally, the points mentioned on the first page of this comment related to other existing continental REVEALS reconstructions and their use could be included the introduction of the paper (or the Discussion). I.e. better describe the difference between this "Global" REVEALS reconstruction and the existing continental REVEALS reconstructions.**

Reply

We thank you for making us aware of this confusion. We used "original" here to describe the synthesis values and distinguish them from values that we tried to optimize. **As we omitted the Optimization from the manuscript, the reconstruction is now only titled REVEALS** and makes use of synthesized RPP values**.** The synthesis by Wieczorek and Herzschuh does indeed take the 1996 publication by Calcote into account.

Concerning syntheses from reconstructions following Wieczorek and Herzschuh (2020), the synthesis used in Githumbi et al. (2021) was finalized in 2019, before the publication of Wieczorek and Herzschuh, and the RPP studies used overlap with the ones used in Wieczorek and Herzschuh. A set of RPP values in southern France by Mazier was not used by WH, but is cited as "unpublished" in Githumbi et al. (2021) so we were unable to acquire these values or a description of the study. The preprint of Dawson et al.'s reconstruction unfortunately only became available after submission of this manuscript. The authors detail the use of the synthesis by Wieczorek and Herzschuh and the addition of RPP values for *Ambrosia* and *Tsuga* from a previous synthesis. **We examined the original publications for these values and and found that they did not meet the requirements for inclusion** as stated in Wieczorek and Herzschuh 2020 as they do not use ERV models. We did however include several RPP studies for Asia in our synthesis, which were published after 2020. Please see the tracked changes for the method description below and also the new Appendices with original RPP values and the RPP synthesis in the uploaded revised manuscript.

 **2.2.1  Parameters and Model Settings**

For each taxon, values for RPP (with uncertainties provided as standard deviation) and fall speeds are used.  We made use of the synthesis of Northern Hemisphere RPP and fall speed values by Wieczorek and Herzschuh (2020). Several RPP studies published since this synthesis were added to the compilation (Geng et al., 2022; Li et al., 2022b; Wang et al., 2021; Huang et al., 2021; Zhang et al., 2021a, b; Wan et al., 2020, 2023; Jian
125  . The methods by Wieczorek and Herzschuh (2020) were followed fore study selection and calculation of synthesis values. An overview of original values and synthesized values can be found in Appendix A and B respectively.
When available, we use continent-specific values in our reconstruction. For taxa with no continental values present, we use  Northern Hemispheric values. If no values exist for a taxon, RPP is set to a constant (RPP = 1, $\sigma$=0.25) and fall speeds are filled with mean continental fall speeds
130  Continental RPP values are available for the majority of pollen counts in all three continents (see Fig. 3). The fraction of pollen counts for which  standard RPP values were assumed is highest in North America but still < 10%. For each site, the REVEALS model also requires information on basin type, basin size and original pollen counts, all of which were collected in the LegacyPollen 2.0 dataset (Li et al., 2024b). Apart from taxon- and basin-specific parameters the REVEALS model requires several constant
135  parameters to be set, which can be found in Table 2.

As we perceive a high uncertainty for reconstructions in the southern hemisphere, due to a lack of RPP values, **we decide to exclude reconstructions from these records from the data set.**

**Detailed Comments**

**Abstract**

adjust after having considered all major comments above. Clarify how the   REVEALS dataset of plant-cover reconstructions for single sites of any type and size  should be used! Clarify the definition of your "relative pollen source area" that I  would rather call "characteristic pollen source area" or simply "pollen source area".  See above.

- We added a short explanation of the use of the dataset.

The improvements in forest cover reconstructions with the REVEALS reconstruction using continental RPP values range from 24% (North
15  America) to 72% (Europe) relative to the mean absolute error (MAE)  of the pollen-based reconstruction. The dataset can be used as a grid with binned and aggregated samples (adjustable script provided on Zenodo; https://zenodo.org/doi/10.5281/zenodo.12800290) or as individual timeseries if the record's basin size exceeds 50
20  ha.

Lines Comment

**Introduction**

26 akin ??

- We will change this wording to "comparable".

> and even trigger feedback effects with other earth system elements (IPCC, 2023; Armstrong McKay et al., 2022). Predicting these changes through modeling is challenging. A sufficient mechanistic understanding of vegetation dynamics and interactions with climate is needed, which requires validation and testing of model data with extensive vegetation data across climatic transitions  comparable to those anticipated in the future (Dearing et al., 2012). Given the relatively brief duration of available
>
> 30 instrumental climate and vegetation data, there is a clear need for long-term  vegetation records derived from paleoecological archives that cover broader climatic gradients than modern datasets (Dearing et al., 2010; Dallmeyer et al., 2023).

41 "real", perhaps "actual vegetation" is better4

- Implemented

> compositions do not accurately reflect vegetation (Davis, 1963; Prentice, 1985; Prentice and Webb III, 1986). This limita-
>
> 45 tion arises from variations in taxon-specific parameters  such as relative pollen productivity (RPP) and pollen dispersal characteristics, leading to discrepancies between the pollen record and  actual past vegetation. This hinders quantitative

47-49 "By accounting for ….. REVEALS models quantitative vegetation cover in relevant pollen source areas …." WRONG! Correct

- Implemented

> gional Estimates of Vegetation Abundance from Large Sites" (REVEALS) . By accounting for taxon-specific RPP and fall speed values, as well as basin-specific parameters such as basin size and type, REVEALS models quantitative vegetation cover in  the region surrounding a basin from pollen compositions. The model has been applied

63 "Yet, only Serge et al. …use the opportunity for extensive validation…" WRONG! See Pirzamanbein et al. (2014) for Europe: Pirzamanbein, B., Lindström, J., Poska, A., Sugita, S., Trondman, A. K., Fyfe, R., Mazier, F., Nielsen, A.B., Kaplan, J.O., Bjune, A.E., Birks, H.J.B., Giesecke, T., Kangur, M., Latałowa, M., Marquer, L., Smith, B., and Gaillard, M.J. (2014). "Creating spatially continuous maps of past land cover from point 1780 estimates: A new statistical approach applied to pollen data." Ecological Complexity 20: 127–141. https://doi.org/10.1016/j.ecocom.2014.09.005

- We added this reference here as well.

> constructions (Hjelle et al., 2015; Roberts et al., 2018). Yet, only Serge et al. (2023) and Pirzamanbein et al. (2014) use this
>
> 70 opportunity for extensive validation and even improvement of reconstructions from European pollen records. No

64 "No site-wise validations …. exist…." What do you mean? What about the the validations by Hellman et al., 2008a and b, and Sugita et al., 2010, and others?

- We changed this to grid-cell based.

> 70 dation of reconstructions (Hjelle et al., 2015; Roberts et al., 2018). Yet, only Serge et al. (2023) and Pirzamanbein et al. (2014)
>
> use this opportunity for extensive validation and even improvement of reconstructions from European pollen records. No

**Mention somewhere in the Introduction the available syntheses of RPP without forgetting the latest RPP synthesis for Europe published in Githumbi et al. (2022a) and the new REVEALS reconstruction for northern America: Dawson, A., Williams, J. W., Gaillard, M.-J., Goring, S. J., Pirzamanbein, B., Lindstrom, J., Anderson, R. S., Brunelle, A., Foster, D., Gajewski, K., Gavin, D. G., Lacourse, T., Minckley, T. A., Oswald, W., Shuman, B., and Whitlock, C. (2024). "Holocene land cover change in North America: continental 1410 trends, regional drivers, and implications for vegetation-atmosphere feedbacks." Climate of the Past 1411 Discussion [preprint].**

**https://doi.org/10.5194/cp-2024-6 , in review, 2024.**

Reply

Even though the reconstruction by Githumbi et al. was published in 2022, the RPP synthesis was completed in 2019 prior to Wieczorek and Herzschuh (2020). The same studies were considered in Githumbi et al. and Wieczorek and Herzschuh with the exception of one unpublished study concerning RPP values in southern France by Mazier. We were not able to acquire the values from this study. Githumbi et al. provide more RPP values because a combination of taxa was not done as in Wieczorek and Herzschuh.

The preprint by Dawson et al. (2024) was unfortunately only available after our submission. We found that the authors used RPP values as synthesized by Wieczorek and Herzschuh for Northern America. RPP values for two additional taxa were added but did not meet the requirements for synthesis as stated in Wieczorek and Herzschuh 2020. We added a mention of Githumbi et al. for their earlier synthesis.

though the model's performance heavily relies on accurate taxon-specific parameters. While Wieczorek and Herzschuh (2020) and Githumbi et al. (2022) provide a comprehensive compilation of RPP and fall speed values for taxa of the Northern Hemi-

60    sphere, the overall availability of RPP studies is still limited and regional variations in RPP values exist (Harris et al., 2020; Broström et al., 2008; Li et al., 2017; Mazier et al., 2012). This makes the application of REVEALS on larger scales particularly

**Methods**

Figure 1 Explain in the Figure caption what the different colours of dots mean,  I guess lakes versus bogs

- We have revised the figure to indicate different basin types.

[Figure]

**Figure 1.** Pollen record locations in the LegacyVegetation dataset. Colors indicate record type (large lake ≥ 50 ha). Record density is highest in Europe and Eastern North America, and lowest in Northern and Central Asia.

86-87 page 4 last lines until ca line 95 page 5
Explain the REVEALS model better and correctly OR simply  refer to Sugita (2007a).

- We feel that a concise description of the model is adequate and cite Sugita in line 85. We will add an additional reference to Githumbi et al. 2022 for further details the REVEALS model.

 For further details on the REVEALS model  see the original publication Sugita (2007) or Githumbi et al. (2022).

97 "using peatland records for reconstructions is, therefore, appropriate." NOT CORRECT. You MUST clarify that only pollen records from small bogs can be used if the mean REVEALS estimates from SEVERAL single small bogs is used, and even better, if the mean REVEALS estimates are from a mix of SEVERAL small bogs and small LAKES. Etc.... see major issues above

- Revised

instead of Sugita's deposition model (Sugita, 1993) in the dispersal and deposition function (see Eq. 1; Sugita, 2007). Previous 110 studies show that results from small bogs are still reliable when aggregated, while results from large bogs tend to deviate from
* * *
those of large lakes (Trondman et al., 2015; Mazier et al., 2012; Trondman et al., 2016 . Using peatland records for reconstructions is, therefore, appropriate when spatially averaging multiple sites. We use the implementation of REVEALS from the R package RE-VEALSinR (Theuerkauf et al., 2016).

8 (12)
Table 2 Several of these are not parameters but models, methods or function .....

- We changed the table title.

**Table 2.** Static model parameters and model settings for REVEALS runs using REVEALSinR (Theuerkauf et al., 2016).

Figure 3 Specify that the "available" RPP values are not necessarily original values obtained in these continents, but can be values obtained in other continents, right?

- We revised the figure to indicate RPP source.

[Figure]

**Figure 3.** Regional source of RPP values for percentage of pollen counts per continent. A majority of pollen counts is covered by continental RPP values with the highest fraction in Europe. Only a small percentage of pollen counts has only hemispheric RPP values available. No available RPP values lead to the use of a standardized RPP value of $1\pm0.25$.

110 Modifications in REVEALSinR:

a. What are these modifications? Are they modifications compared to REVEALSinR published in Theuerkauf et al (2016) or modifications compared to the REVEALS program by Sugita? Or anything else?

- These are modifications compared to Theuerkauf et al.'s REVEALSinR. We clarified this in the text.

145 We also reduced computational effort in REVEALSinR by implementing a maximum number of steps in the lake model used to model mixing in the basin. The number of steps was set to 500 unless $n$ falls below that maximum value for $n = basin\,radius/10$ for basins with a radius of at least 1000 m and $n = basin\,radius/2$ for basins with a radius smaller than 1000 m.

b. You did not calculate the "relevant source area of pollen" (RSAP) but something else that you define as "the radius in which etc…..total nflux (…..)." RSAP is something else, see my major comments above. Moreover the definition you provide is badly written, use instead the wording for the definition given in Theuerkauf et al. (2016)

- We adjusted the naming as described above and edited the description of the definition to contain more detail.

We calculate the radius of  the 80% pollen source area by finding the radius in which the median influx of all taxa is 80% of the total influx (as defined by the total influx in the maximum extent of regional vegetation chosen). This is calculated by employing the lake deposition model in REVEALSinR (Theuerkauf et al., 2016). Starting from $z_{max}$ the deposited pollen

140 is calculated per taxon. This is assumed to be the total pollen each taxon deposits. In a step-wise process the radius around the basin is increased and the deposited pollen relative to the total influx at $z_{max}$ is calculated for each taxon. We define our 80%

pollen source radius as the radius where the median of the relative influx of all taxa reaches 80%. The primary objective of this calculation is to provide a clear understanding of the scale of the source area for users unfamiliar with pollen data. It highlights the regional nature of lacustrine pollen data and demonstrates the influence of lake size on this source area.

145 We also reduced computational effort in REVEALSinR by implementing a maximum number of steps in the lake model used to model mixing in the basin. The number of steps was set to 500 unless $n$ falls below that maximum value for $n = basin\,radius/10$ for basins with a radius of at least 1000 m and $n = basin\,radius/2$ for basins with a radius smaller than 1000 m.

117-127 This validation is problematic as it uses the REVEALS estimates for individual sites which implies that the reconstructions using pollen records from small sites (lakes or bogs" will be biased compared to the REVEALS estimates obtained with pollen records from large sites. If you keep this "validation", you MUST clarify that the REVEALS estimates for the small sites can be strongly biased and therefore the correlation with the modern vegetation might be less good than if you would use the mean REVEALS estimates from several sites within a given area size (e.g., grid cells of 1 degree, or vegetation regions, biomes, or continents).

- We revised the validation to use gridded values.

[Figure]

**Figure 9.** Remote sensing forest cover (LANDSAT) and modern reconstructed forest cover from Pollen and REVEALS (< 100 years BP) in 2x2° grid cells with mean absolute errors (MAE) and correlation coefficient ($R$) per group. Reconstructed forest cover from the original pollen data tends to overestimate observed (remote sensing) forest cover. Improvements with the REVEALS reconstruction are especially high in Europe. Validations with different grid cell sizes are available in the supplement (S3: Validation results for different spatial resolutions).

134 I do not understand this equation, it doesn't make sense to me. This perhaps because it is not clear what you mean with "reconstructed tree cover", and "corrected tree cover". It is clear what "unvegetated" cover is, and it is clear that you have to adjust the modern vegetation cover by using the sum of open vegetated cover as 100% or (1) (i.e. total open cover – unvegetated cover = 1). Is your "reconstructed cover" the total open cover including the unvegetated cover? The use of "reconstructed" here is confusing

- Reconstructed tree cover pertains to the tree cover reconstructed from a pollen product. In our manuscript this includes the original pollen counts, REVEALS reconstructions and optimized REVEALS reconstructions. It is created by summing the percentage of arboreal taxa. Because of the closed compositional nature of pollen records, there is no way to reconstruct unvegetated areas with pollen counts. The value we get can be defined as the percentage of forested area in the vegetated source area around the basin. The forest cover available in remote sensing products is differently defined as the percentage of forested area in the total source area around the basin, which includes unvegetated areas. In order to correct for this within the validation we make use of land cover maps and extract the percentage of unvegetated area in the source area around the basin (Equation 2). This way we are able to convert the reconstructed forest cover to a value which also represents the percentage of forested area in the total source area around the basin and enables a comparison between remote sensing forest cover and corrected, reconstructed forest cover.

General comment for Methods: I do not understand from this description what is the time resolution of the reconstructions, from the Figure 8 it looks to be 1000 years. In this case, this should also be mentioned as a difference in comparison with the continental PAGES LandCover6k that use 500 years resolution up to recent times, and 3 shorter time windows, 350, 250 and ca 100 years between 0.7 k BP up to "present" (see e.g. Trondman et al., 2015).

- For the gridded dataset highlighted in the results a 500 yr temporal resolution was chosen. Validation makes use of the past 100 years.

Forest cover was reconstructed by summing up percentages of arboreal taxa (see S2: List of arboreal taxa) with Betulaceae, *Betula*, and *Alnus* being classified as arboreal at sites below 70° N. The mean reconstructed compositional coverages from the REVEALS results were used for the forest cover reconstructions. REVEALS results were then rasterized to aggregate and
155 include records from smaller basins as well. Reconstructed time series were averaged in 500 year bins and then rasterized in grids of differing spatial resolution. A grid cell was classified as having a valid reconstruction when it contained records from at least one large lake (>= 50 ha) or at least two small basins following Serge et al. (2023). Standard deviations of the REVEALS estimates were aggregated by applying the delta method by Stuart and Ord (1994), using the same equation as Wieczorek and Herzschuh (2020). We provide a script for rasterization with adjustable temporal and spatial resolution for users
160 of the dataset on Zenodo (https://zenodo.org/doi/10.5281/zenodo.12800290). For validation, the reconstructed forest cover of the past  100 years was rasterized and compared to modern remote sensing forest cover. Only valid grid cells as

**Data Summary**

3.1 Pollen source area Adjust according to my major comments above

- We changed the name to "80% pollen source area" as described above.

**3.1  80% Pollen Source Areas**

3.2 Comparison of original and optimized values Do not use the term "original" for the RPP values you are using but rather "WH mean RPP values" or something similar. The values you are using are not "original values from specific studies unless there is only ONE value that you are using. See my major comment re Tables A1 and A2

- We have omitted the optimization from the manuscript.

Figure 4 Map indicating the size of relevant pollen source areas: CORRECT! It is not RSAP!

- We changed the name to "80% pollen source area" as described above.

[Figure]

**Figure 5.** Scatterplot of basin diameter and 80% pollen source area of large lakes in the REVEALS data set. In general, larger basins have larger pollen source areas with the relationship between diameter and 80% pollen source radius being roughly logarithmic.

165-169 "The highest and lowest absolute change respectively occurred for Quercus (4.08) and Fabaceae (0.09) in Africa, etc...." What do you mean? Is it a +/- change or only + change, specify! I see that it is often a + change. I would write: "The highest respectively lowest absolute change (highest/lower) occurred for Quercus (+4.08)/Fabaceae (-0.09) in Africa. If this is not what you meant, CLARIFY!

- We omitted the Optimization and therefore also this paragraph from this manuscript.

175-197 The comparison presented in Figure 7 is fine as you have calculated average REVEALS-estimated cover for whole continents, which is OK even if you used pollen records from small sites. See my major comments above. My question is: did you calculate errors for the average REVEALS estimates using the errors produced by REVEALSinR from each individual pollen record?

- We did not calculate errors for this figure and only show mean values. Standard deviations were calculated using the delta method.

from at least one large lake (>= 50 ha) or at least two small basins following Serge et al. (2023). Standard deviations of the REVEALS estimates were aggregated by applying the delta method by Stuart and Ord (1994), using the same equation as Wieczorek and Herzschuh (2020). We provide a script for rasterization with adjustable temporal and spatial resolution for users

160 of the dataset on Zenodo (https://zenodo.org/doi/10.5281/zenodo.12800290). For validation, the reconstructed forest cover of

199-209 Similarly, Figures 8 and 9 present average forest cover using REVEALS estimates from pollen records available within 10 degrees grid cells, which means that most grid cells are represented by REVEALS estimates using several pollen records. As these data are also made accessible, it would be useful for the user if you added a file that provides the identity code of the grid cells for which the "average" REVEALS estimate is based on the reconstruction from a single pollen record from one-several large bog(2), or 1-2 small bog(2), or 1-2 small lakes, or 1 small bog + 1 small lake. See example in Githumbi et al. (2022a).

- We added this classification in this figure and included it in the script for rasterization to be used by the user. Temporal averages always include only valid grid cells.

[Figure]

**Figure 7.** Northern Hemisphere and continental average forest cover from 2x2° grid cell means for raw pollen data and the REVEALS reconstruction (Northern Hemisphere averages from different grid cell resolutions are available in S2: Reconstruction results for different spatial resolutions). Remotely sensed global average forest clover for the grid cells with valid pollen coverage is indicated with the diamond. Temporal trends are the same, but absolute forest cover is reduced in the REVEALS reconstructions compared to the original pollen data. Both reconstructions still overestimate forest cover.

[Figure]

**Figure 8.** Reconstructed forest cover in 2x2° grid cells from raw pollen data and the REVEALS reconstruction for 5 example time slices (reconstructions with different grid cell sizes are available in the in S2: Reconstruction results for different spatial resolutions). Valid cells are filled and include reconstructions from at least one large lake ($\geq$ 50 ha) or several smaller basins. Forest cover in Eastern North America is higher than in Europe and Asia. REVEALS reconstructed forest cover is generally lower than raw pollen reconstructions.

3.5 validation It is not correct to validate the REVEALS model with modern  vegetation data SITE BY

SITE, given that a REVEALS reconstruction using a pollen record from ONE large bog or ONE small site (bog or lake) will in most cases be biased. **A proper revision of this paper should/MUST use the 10 degree grid cell reconstructions to validate these new REVEALS reconstructions (using WH RPP dataset or optimized RPP), and use the cover of modern vegetation within those same grid cells for comparison. Even the SLOO validation should be redone using 10 degree grid cells as the basis for the validation.**

- We revised the validation to use gridded data.

[Figure]

**Figure 9.** Remote sensing forest cover (LANDSAT) and modern reconstructed forest cover from Pollen and REVEALS (< 100 years BP) in 2x2° grid cells with mean absolute errors (MAE) and correlation coefficient (R) per group. Reconstructed forest cover from the original pollen data tends to overestimate observed (remote sensing) forest cover. Improvements with the REVEALS reconstruction are especially high in Europe. Validations with different grid cell sizes are available in the supplement (S3: Validation results for different spatial resolutions).

[Figure]

**Figure 11.** Map of the reconstruction error (in % forest cover) for forest cover reconstructed from Pollen and REVEALS data. Remaining errors with the overall better REVEALS reconstructions are especially high in North America (Northern West Coast, Labrador Peninsula).

251-258 The major difference between N hemispheric vegetation and sub tropical-tropical vegetation is that: in northern and temperate (mediterranean) regions a majority of the tree species are wind pollinated and produce large quantities of pollen per unit area, while pollen of herbaceous plants use to be insect pollinated or both wind and insect pollinated and produce less quantities of pollen per unit area, which implies that trees often are overrepresented by pollen compared with herbs; in (sub) tropical regions it is the inverse, many trees are insect pollinated and often produce small quantities of pollen which implies that herbs may be overrepresented by pollen compared to trees. The latter is well illustrated by Figure 10 pollen % versus remote sensed plant cover.

**In this section you MUST clarify that you have not used the RPP values that have been obtained from modern pollen-vegetation datasets in (sub) tropical regions and are available today in published articles (China, Africa, southern America) and provide**
**example references** (you do not need to do a literature search given that you do not use them). It's however important that you inform the reader that such values exist. For instance, in Gaillard et al. (2021) the obtained RPP in Cameroon for 13 taxa are compared with values obtained for these taxa in Africa and China, which already provide a significant number of existing values. Another useful paper is that by Wan et al. (2020): Wan, Q., Zhang, Y., Huang, K., Sun, Q., Zhang, X., Gaillard, M.-J., Xu, Q., Li, F. and Zheng, Z., 2020, Evaluating quantitative pollen representation of vegetation in the tropics: A case study on the Hainan Island, tropical China. Ecological Indicators, 114, article: 106297, 10.1016/j.ecolind.2020.106297.

- We decided to exclude the Southern Hemisphere from this reconstruction as missing RPP and high fraction of insect pollination make our reconstruction too unreliable here.

**Dataset applications and limitations and Conclusions**

**Adjust these two sections according to the major comments explained in the first part of this review in addressing all the issues implied by your dataset, in particular the REVEALS estimates for single sites.**

285-286 "…with previous REVEALS applications and show an increase ….until roughly 4 ka BP (references). **This is not correct, the REVEALS reconstructions mentioned show an increase of forest cover/respectively a decrease in openland cover until around 6 ka BP.** The best reference for this is Strandberg et al. (2023) in Clim of the Past and Figure 1 therein that is based on the REVEALS reconstruction from Githumbi et al.

(2022a, in ESSD).

- We revised this.

previous reconstructions as well (Li et al., 2023, 2022b, 2024a). Results for European forest cover also roughly correspond
with previous REVEALS applications and show an increase of forest cover after the last glacial maximum until roughly
350 6 ka BP (Githumbi et al., 2022; Fyfe et al., 2015; Serge et al., 2023; Strandberg
. The gridded reconstruction by Serge et al. (2023) was even validated with modern remote sensing forest cover and showed a
good fit.

293-294 The deglacial forest conundrium (or Holocene temperature  conundrium (HTC)) is also
discussed in Strandberg et al. (2022, in  QSR).

- We added this reference.

reconstructions as shown by Binney et al. (2011). Additionally, this dataset can address unanswered questions about Holocene
360 vegetation dynamics, including the deglacial forest conundrum (Dallmeyer et al., 2022; Strandberg et al., 2022)
. It also serves as a valuable tool for validating models with coupled climate and vegetation,  which rely on extensive

**References**

387-389 replace this reference by/ or add : Dallmeyer et al. (2024) in Clim Past  Discussion: Dawson, A.,
Williams, J. W., Gaillard, M.-J., Goring, S. J.,  Pirzamanbein, B., Lindstrom, J., Anderson,
R. S., Brunelle, A., Foster,  D., Gajewski, K., Gavin, D. G., Lacourse, T., Minckley, T. A.,
Oswald,  W., Shuman, B., and Whitlock, C. (2024). "Holocene land cover  change in North
America: continental trends, regional drivers, and  implications for vegetation-atmosphere
feedbacks." Climate of the  Past Discussion [preprint]. https://doi.org/10.5194/cp-2024-6 ,
in  review, 2024

- We reference Dawson et al.'s (2024) manuscript now as it was not available at the time
of submission.

. It also serves as a valuable tool for validating models with coupled climate and vegetation,  which rely on extensive
time series and vegetation data for accurate predictions  (Dallmeyer et al., 2023; Dawson et al., 2024).

409-413 replace the Githumbi et al. 2021 in Clim Past Discussion by Githumbi  et al. (2022): Githumbi,
E., Fyfe, R., Gaillard, M.-J., Trondman, A.-K.,  Mazier, F., Nielsen, A.-B., Poska, A., Sugita,
S., Woodbridge, J.,  Azuara, J., Feurdean, A., Grindean, R., Lebreton, V., Marquer, L.,
Nebout - Combourieu, N., Stančikaitė, M., Tanţău, I., Tonkov, S.,  Shumilovskikh, L., and
LandClimII data contributors (2022a).  "European pollen-based REVEALS land-cover
reconstructions for the  Holocene: methodology, mapping and potentials." Earth System
Science Data 14: 1581–1619. https://doi.org/10.5194/essd-14-1581- 2022

- We corrected this citation.

Githumbi, E., Fyfe, R., Gaillard, M.-J., Trondman, A.-K., Mazier, F., Nielsen, A.-B., Poska, A., Sugita, S., Woodbridge, J., Azuara, J.,
Feurdean, A., Grindean, R., Lebreton, V., Marquer, L., Nebout-Combourieu, N., Stančikaitė, M., Tanţău, I., Tonkov, S., Shumilovskikh,
L., and data contributors, L.: European pollen-based REVEALS land-cover reconstructions for the Holocene: methodology, mapping and
520 potentials, Earth System Science Data, 14, 1581–1619, https://doi.org/10.5194/essd-14-1581-2022, publisher: Copernicus GmbH, 2022.

Do remember to also refer to Trondman et al. (2016) (see my major comment on the  application of
REVEALS using pollen records from small sites): Trondman, A.-K., Gaillard, M.-J., Sugita,
S., Björkman, L., Greisman,
A., Hultberg, T., Lagerås, P., and Lindbladh, M. (2016). "Are pollen  records from small
sites appropriate for REVEALS model-based quantitative reconstructions of past regional
vegetation? An empirical  test in southern Sweden." Vegetation History and

Archaeobotany 25: 131–151. https://doi.org/10.1007/s00334-015-0536-9

- We added this reference

instead of Sugita's deposition model (Sugita, 1993) in the dispersal and deposition function (see Eq. 1; Sugita, 2007). Previous

110    studies show that results from small bogs are still reliable when aggregated, while results from large bogs tend to deviate from

those of large lakes (Trondman et al., 2015; Mazier et al., 2012; Trondman et al., 2016)
. Using peatland records for reconstructions is, therefore, appropriate when spatially averaging multiple sites. We use the implementation of REVEALS from the R package RE-
VEALSinR (Theuerkauf et al., 2016).

**Reply to Mariani**

Laura Schild & Ulrike Herzschuh

**General reply**

Dear Michela Mariani,

We thank you for your careful review or our manuscript. We appreciate the valid points you have made and found them to be easily remediable. Regarding other matters you have mentioned, we see a good opportunity to provide clarification on how we follow common procedures and highlight the usability and validity of our dataset.

Firstly, the use of continental syntheses of RPP values is widely accepted in continental-scale reconstructions and has been applied in several previously published reconstructions in Europe, North America and Asia. The use of even hemispheric values, when continental ones are missing, maximizes the utility of available data while still producing improved reconstructions as our validations show. As we cannot achieve small-scale perfect reconstructions yet, we advocate for these broader and necessarily coarser insights into vegetation dynamics by generalizing reconstructions.

Furthermore, our REVEALS application using previously published synthesized RPP values is completely independent of remote sensing data. This means using remote sensing data allows for reliable validation. This shows us a clear improvement of forest cover reconstructions compared to raw pollen-data.

We do concur that uncertainties with Southern Hemispheric reconstructions are high and will exclude those from our evaluation along with samples prior to the 14 ka BP. Moreover, the few non-lake and non-peat records will be removed from the data set as they are unfit for reconstruction with REVEALS. Their removal has no effect on our spatial coverage and general reconstruction results. These adjustments were all completed.

We have added replies to your specific comments below.

An updated version of the dataset can now be found here:
https://doi.org/10.5281/zenodo.12800159 . The dataset on PANGAEA will be updated as soon as possible.

Best
Laura Schild and Ulrike Herzschuh

**Specific replies**

**Major issues:**

**Original Comment**

**Inadequate regional calibrations**: The generalization of RPPEs across broad geographical scales (hemispheres) ignores crucial ecological and bioclimatic regional variations. This approach most likely leads to significant inaccuracies in the vegetation reconstructions, in spite of what the presumed 'validation' approach suggests (see below).

**Reply**

While we agree that a reconstruction using synthesized values will not reflect reality exactly, we still argue for the usability and informativeness of the result of this generalized approach. Continental syntheses of RPP values are standard practice in large-scale reconstructions as they allow an approximation of past vegetation dynamics on a large scale. Notable examples of previously used continental-scale syntheses in Europe include Serge et al. (2023), Trondman et al. (2015) and Pirzamanbein et al. (2014). Githumbi et al. (2022) also synthesize values for Northern and Central Europe and treat only mediterranean records differently. Reconstructions in North America (Dawson et al. 2024) and Northern Asia (Cao et al. 2019) synthesize values on large or continental scales as well.

While we recognize the variability of relative pollen productivity (RPP), we advocate for the use of even hemispheric averages when continental values are lacking. The direction of taxon-specific correction (over- or underproduction of pollen) will generally be correct and provide a vast improvement of REVEALS estimates to using pollen percentages alone while being able to make the most of the data currently available. We highlight that compositional reconstructions using this method come with uncertainties, but are confident that aggregates, such as reconstructed forest cover, are much closer to reality than previous pollen-based estimates. By employing this methodology, our overarching goal is to generate reconstructions that facilitate comparisons across the Northern Hemisphere while shedding light on general vegetation dynamics. This approach mirrors the methodology utilized in large-scale climate models, where local nuances are necessarily sacrificed for broader insights. We highlight the uncertainty connected with hemispheric and standardized RPP values in the revised section "Data applications and limitations" (see tracked changes below).

Importantly, this is underlined by our validation which uses independent remote sensing data and demonstrates notable improvements in reconstruction accuracy compared to reconstructions based on raw pollen data. We will expand on this in our reply to an issue below.

Githumbi, Esther, Ralph Fyfe, Marie-Jose Gaillard, Anna-Kari Trondman, Florence Mazier, Anne-Birgitte Nielsen, Anneli Poska, u. a. „European Pollen-Based REVEALS Land-Cover Reconstructions for the Holocene: Methodology, Mapping and Potentials".

*Earth System Science Data* 14, Nr. 4 (8. April 2022): 1581–1619. https://doi.org/10.5194/essd-14-1581-2022.

Serge, M. A., F. Mazier, R. Fyfe, M.-J. Gaillard, T. Klein, A. Lagnoux, D. Galop, u. a. „Testing the Effect of Relative Pollen Productivity on the REVEALS Model: A Validated Reconstruction of Europe-Wide Holocene Vegetation". *Land* 12, Nr. 5 (Mai 2023): 986. https://doi.org/10.3390/land12050986.

Dawson, Andria, John W. Williams, Marie-José Gaillard, Simon J. Goring, Behnaz Pirzamanbein, Johan Lindstrom, R. Scott Anderson, u. a. „Holocene Land Cover Change in North America: Continental Trends, Regional Drivers, and Implications for Vegetation-Atmosphere Feedbacks". *Climate of the Past Discussions*, 20. Februar 2024, 1–52. https://doi.org/10.5194/cp-2024-6.

Trondman, A.-K., M.-J. Gaillard, F. Mazier, S. Sugita, R. Fyfe, A. B. Nielsen, C. Twiddle, u. a. „Pollen-Based Quantitative Reconstructions of Holocene Regional Vegetation Cover (Plant-Functional Types and Land-Cover Types) in Europe Suitable for Climate Modelling". *Global Change Biology* 21, Nr. 2 (2015): 676–97. https://doi.org/10.1111/gcb.12737.

Pirzamanbein, Behnaz, Johan Lindström, Anneli Poska, Shinya Sugita, Anna-Kari Trondman, Ralph Fyfe, Florence Mazier, u. a. „Creating Spatially Continuous Maps of Past Land Cover from Point Estimates: A New Statistical Approach Applied to Pollen Data". *Ecological Complexity* 20 (1. Dezember 2014): 127–41. https://doi.org/10.1016/j.ecocom.2014.09.005.

Cao, Xianyong, Fang Tian, Furong Li, Marie-José Gaillard, Natalia Rudaya, Qinghai Xu, und Ulrike Herzschuh. „Pollen-Based Quantitative Land-Cover Reconstruction for Northern Asia Covering the Last 40 Ka Cal BP". *Climate of the Past* 15, Nr. 4 (8. August 2019): 1503–36. https://doi.org/10.5194/cp-15-1503-2019.

signal.The reliability of reconstructions varies among different taxa due to the quality of RPP values, and this is explicitly documented in a supplementary file that outlines the sources of RPP values (see Section Code and Data availability). Reconstructions of taxa with continental RPP values are the most reliable, followed by those based on hemispheric data, with standardized RPP
385 values being the least reliable. This hierarchy should be taken into account when interpreting the results. Higher certainty is associated with forest cover reconstruction, as it is based on aggregation among taxa. Reconstructions of temporal forest cover

**Original comment**

**Questionable data assumptions and methodological gaps:** The use of northern hemisphere RPPE values for taxa not natively present in the southern hemisphere, such as *Alnus* in Australia, introduces substantial and confusing biases. Presumably, the authors have not consulted the relevant scholars who worked within this field and the geographical areas mentioned. Similarly, defaulting RPP to 1 for taxa without specific data oversimplifies pollen-vegetation relationships. The paper does not adequately address the absence of data for the Southern Hemisphere, leading to a misleading portrayal of global vegetation.

It is suggested >50% of RPPEs are missing for Australia and Oceanic pollen records. So, in this work a decision was made to run these records using the Northern Hemispheric RPPEs, despite very different bioclimatic and ecological contexts. This extrapolation of Northern Hemisphere RPPEs to southern locations missing PPEs without considering ecological or bioclimatic differences is particularly problematic. RPPEs empirically produced using ground truthing work (field surveys and surface pollen collection) were ignored, especially across the Southern Hemisphere (see some references below).

Duffin, K. I., & Bunting, M. J. (2008). Relative pollen productivity and fall speed estimates for southern African savanna taxa. Vegetation History and Archaeobotany, 17, 507-525.

Mariani, M., Connor, S. E., Theuerkauf, M., Kuneš, P., & Fletcher, M. S. (2016). Testing quantitative pollen dispersal models in animal-pollinated vegetation mosaics: An example from temperate Tasmania, Australia. Quaternary Science Reviews, 154, 214-225.

Mariani, M., Connor, S. E., Fletcher, M. S., Theuerkauf, M., Kuneš, P., Jacobsen, G., ... & Zawadzki, A. (2017). How old is the Tasmanian cultural landscape? A test of landscape openness using quantitative land‑cover reconstructions. Journal of Biogeography, 44(10), 2410-2420.

Mariani, M., Connor, S. E., Theuerkauf, M., Herbert, A., Kuneš, P., Bowman, D., ... & Briles, C. (2022). Disruption of cultural burning promotes shrub encroachment and unprecedented wildfires. Frontiers in Ecology and the Environment, 20(5), 292-300.

**Reply**

**We do concur that uncertainties with Southern Hemispheric reconstructions are high due to a lack of regional RPP values and will exclude those from our data set.**
We believe that including other observed taxa in the model and setting their RPP to 1, will still result in better estimates of aggregate values such as forest cover than the raw pollen data and therefore apply this standard value to include as much data as possible. Including missing RPP values by setting them to 1 or excluding taxa without RPP estimates leads to relatively similar coverage estimates as indicated by the figures below. Excluding any taxa tends to result in an overestimation of the remaining taxa as the total pollen count is reduced. By including all taxa we aim to account for this.

We have added several RPP studies published since the synthesis by Wieczorek and Herzschuh (2020) to our synthesis or RPP values and we highlight the percentage of pollen counts without RPP values (see tracked changes and revised Figure 3 below).

[Figure]

[Figure]

[Figure]

Fig 1: *(newly created)* Comparison of reconstructed cover values for taxa in a reconstruction excluding taxa, for which no RPP estimates are available, and in a reconstruction where all taxa are included and unknown RPP set to 1. **The results are highly correlated, showing that including taxa with unknown RPP does not impact the reconstruction of the taxa for which RPP were already known.**

For each taxon, values for RPP (with uncertainties provided as standard deviation) and fall speeds are used.  We made use of the synthesis of Northern Hemisphere RPP and fall speed values by Wieczorek and Herzschuh (2020). Several RPP studies published since this synthesis were added to the compilation (Geng et al., 2022; Li et al., 2022b; Wang et al., 2021; Huang et al., 2021; Zhang et al., 2021a, b; Wan et al., 2020, 2023; Jian

125 . The methods by Wieczorek and Herzschuh (2020) were followed fore study selection and calculation of synthesis values. An overview of original values and synthesized values can be found in Appendix A and B respectively. When available, we use continent-specific values in our reconstruction. For taxa with no continental values present, we use  Northern Hemispheric values. If no values exist for a taxon, RPP is set to a constant (RPP = 1, $\sigma$=0.25) and fall speeds are filled with mean continental fall speeds

130 Continental RPP values are available for the majority of pollen counts in all three continents (see Fig. 3). The fraction of pollen counts for which  standard RPP values were assumed is highest in North America but still < 10%. For each site, the REVEALS model also requires information on basin type, basin size and original pollen counts, all of which were collected in the LegacyPollen 2.0 dataset (Li et al., 2024b). Apart from taxon- and basin-specific parameters the REVEALS model requires several constant

135 parameters to be set, which can be found in Table 2.

[Figure]

**Figure 3.** Regional source of RPP values for percentage of pollen counts per continent. A majority of pollen counts is covered by continental RPP values with the highest fraction in Europe. Only a small percentage of pollen counts has only hemispheric RPP values available. No available RPP values lead to the use of a standardized RPP value of 1±0.25.

**Original Comment**

**Oversimplified and incorrect spatial and temporal settings:** The inclusion of incorrect basin types in the model without appropriate adjustments is very concerning. Why are marine records included for a model explicitly designed to work for large lakes of closed basins with wind dispersal as the only mechanism for pollen deposition?

The manuscripts states that 'all sites that were not classified as lakes were run with peatland settings' = can we consider the ocean a peatland? REVEALS cannot work with marine records and it definitely does not make sense to apply the 'peatland' settings for marine records with

some random arbitrary basin radius (100m?). Further, using a deep temporal scope (50ka) without any consideration for massive climatic shifts (likely larger than the effect of regional RPPEs values vs regional bioclimate variations) are concerning oversights, making any pre-Holocene glacial REVEALS reconstructions unrealistic with current interglacial PPEs.

**Reply**

We agree with the unsuitability of non-lake and non-peat records and apologize for their inclusion. **We will remove them from our data set** (see tracked changes below). We realize that many of the peat sites used do not have basin sizes assigned to them. However, peatlands tend to be relatively small and therefore similar in size, with the mean size of peatlands used in Trondman et al. (2016) being lower than 100m and the average peatland size in the data used by Githumbi et al. (2022) being 716 m (with a rather large standard deviation of 1901 m due to few unrealistically large peatlands). These differences of several hundred meters at most do not influence the reconstruction of REVEALS estimates considerably, which is why a standardization of peatland sizes is appropriate here. Please see Figure 2 below for an example peatland reconstruction using different basin diameters.

Trondman, Anna-Kari, Marie-José Gaillard, Shinya Sugita, Leif Björkman, Annica Greisman, Tove Hultberg, Per Lagerås, Matts Lindbladh, und Florence Mazier. „Are Pollen Records from Small Sites Appropriate for REVEALS Model-Based Quantitative Reconstructions of Past Regional Vegetation? An Empirical Test in Southern Sweden". *Vegetation History and Archaeobotany* 25, Nr. 2 (1. März 2016): 131–51. https://doi.org/10.1007/s00334-015-0536-9.

[Figure]

Fig 2: Example peatland site (Ageröds mosse, 13.42774W 55.93448N) with reconstructed vegetation at different set basin sizes (diameter in m). **The basin size has minimal impact on the reconstructed result when using peatlands and can therefore be standardized.**

Tracked changes (exclusion of non-lake and non-peat records):

90  Sediment and peat cores used for the creation of pollen data are of lacustrine, peat and marine origin. For the REVEALS reconstruction only lake and peat records in the Northern Hemisphere were used ($n = 2732$) Analogous to the preceding Lega-cyPollen 1.0 dataset (Herzschuh et al., 2022), the data synthesis involved revising age modeling and taxonomic harmonization for consistency of records. Spatial data coverage of records in the reconstruction is  dense in Europe (1275 records) and  North America (1016 records) and sparsest in Asia (

95   441) (see Fig. 1). The records' sample density decreases with age (see Fig. 2).

**Original Comment**

**Dubious optimization and validation:** Optimizing PPEs to match remote sensing data risks validating the model based on its own assumptions rather than providing an unbiased estimation of past vegetation, which REVEALS is designed to do. This circular reasoning undermines the scientific integrity of the model's outputs. While an interesting concept this

needs to be validated separately on a much smaller spatially and higher resolution scale before such a widespread application. This cannot really be called a 'validation'.

**Reply**

We apologize for any confusion that may have arisen here, but **there is no circularity associated with the validation of the REVEALS reconstruction** making use of published syntheses (titled "REVEALS (original RPP)" in our original manuscript). The remotely sensed forest cover is independent of the REVEALS reconstruction and has previously been used to validate large-scale reconstruction by Serge et al. (2023) and Pirzamanbein et al. (2014). We make use of the openness correction to account for predominantly urban structures influencing modern forest cover from remote sensing with large consolidated areas. However, we do this to both the original pollen data and the REVEALS reconstruction. **Our validations show a clear improvement in forest cover reconstruction compared to pure pollen data.**

> et al., 2013; Townshend, 2016). An openness correction was applied to sites containing urban areas and paved surfaces within the 80% pollen source areas (PSA) to correct for areas without any pollen sources and thus  ensure comparability to
> 165    modern remote sensing forest cover (see Equations 2-4). For this, the percentage of unvegetated land cover classes for the year 2015 in the ESA CCI land cover data set was used (ESA, 2017, see Table 3). Areas covered by water or ice are already considered as missing values in the remote sensing forest cover data set and do not need to be corrected for. Forest cover was validated  for each grid cell and mean absolute error (MAE) and correlation coefficients calculated for each continent. No openness correction was applied to the reconstruction values in the final dataset. Validation for a 2x2° grid is included in the
> 170    results section. Further validations using 1°,5°, and 10° resolution are included in the supplementary material (S3: Validation results for different spatial resolutions).

We have decided to **omit the optimization** from this manuscript.

**Original comment**

The reconstructed forest cover for the past 500 years was compared to modern remote sensed cover. Why not a smaller and more recent age bin was considered? In the past 500 years many areas of the world have been colonised by Europeans and have experienced major shifts in vegetation structure, as management transferred from Indigenous to colonial regimes (e.g. the Americas and Australia). This means that forest cover over the whole 500 years bin is not comparable to modern remote sensing data. This highlights a Eurocentric view of the global vegetation patterns.

An example of validation of RPPEs using modern vegetation data (with surveys) has been done in the following papers:

Mariani, M., Connor, S. E., Fletcher, M. S., Theuerkauf, M., Kuneš, P., Jacobsen, G., ... & Zawadzki, A. (2017). How old is the Tasmanian cultural landscape? A test of landscape openness using quantitative land‐cover reconstructions. Journal of Biogeography, 44(10), 2410-2420.

Mariani, M., Connor, S. E., Theuerkauf, M., Herbert, A., Kuneš, P., Bowman, D., ... & Briles, C. (2022). Disruption of cultural burning promotes shrub encroachment and unprecedented wildfires. Frontiers in Ecology and the Environment, 20(5), 292-300.

**Reply**

Our choice of 500 year age bins was founded in the aim to include as many records as possible, since not all have samples as young as 100 years BP. We have, however, **tested a smaller age bin for the REVEALS reconstruction** using published synthesis values and found similar validation results. In our revised manuscript we include a 100 year age bin with gridded data.

[Figure]

**Figure 9.** Remote sensing forest cover (LANDSAT) and modern reconstructed forest cover from Pollen and REVEALS (< 100 years BP) in 2x2° grid cells with mean absolute errors (MAE) and correlation coefficient ($R$) per group. Reconstructed forest cover from the original pollen data tends to overestimate observed (remote sensing) forest cover. Improvements with the REVEALS reconstruction are especially high in Europe. Validations with different grid cell sizes are available in the supplement (S3: Validation results for different spatial resolutions).

Tracked changes (smaller age bin):

160   of the dataset on Zenodo (https://zenodo.org/doi/10.5281/zenodo.12800290). For validation, the reconstructed forest cover of the past  100 years was rasterized and compared to modern remote sensing forest cover. Only valid grid cells as defined above were used for validation. Average tree canopy cover  for all grid cells was

**Reply to Williams et al.**

Laura Schild and Ulrike Herzschuh

Dear John Williams and colleagues,

We thank you for your comments on our manuscript and appreciate your special attention towards open science and open data issues.
We apologize for the omission of a reference to Neotoma connected to the LegacyPollen dataset and will of course rectify this oversight. We add citations in the description of LegacyPollen2.0 and expand our acknowledgements. A table listing all included Neotoma records and their DOIs is appended as well. We especially thank you for the helpful inclusion of the *doi()* function in the neotoma2 R package and the example code on github.
Please see our revised passages below.

An updated version of the dataset can now be found here: https://doi.org/10.5281/zenodo.12800159. The dataset on PANGAEA will be updated as soon as possible.

Best regards
Laura Schild and Ulrike Herzschuh

**Revised text at line 76:**

The pollen data synthesis LegacyPollen2.0 (Li et al., 2024b) includes 3728 temporally resolved records (time-series) distributed globally. Data were collected from individual publications and the Neotoma Paleoecology Database which includes data from the European Pollen Database, the QUAVIDA data base for Australasia, the Latin American Pollen Database, the African Pollen Database and the North American Pollen database (Flantua et al., 2015; Fyfe et al., 2009; Giesecke et al., 2014; Lézine et al., 2021; Rowe et al., 2007; Whitmore et al., 2005; Williams et al., 2018). An overview of Neotoma records included in LegacyPollen 2.0 can be found in Table S2.

Flantua, S.G.A., Hooghiemstra, H., Grimm, E.C., Behling, H., Bush, M.B., González-Arango, C., Gosling, W.D., Ledru, M.-P., Lozano-García, S., Maldonado, A., Prieto, A.R., Rull, V., Van Boxel, J.H., 2015. Updated site compilation of the Latin American Pollen Database. Rev. Palaeobot. Palynol. 223, 104–115. https://doi.org/10.1016/j.revpalbo.2015.09.008

Fyfe, R.M., de Beaulieu, J.-L., Binney, H., Bradshaw, R.H.W., Brewer, S., Le Flao, A., Finsinger, W., Gaillard, M.-J., Giesecke, T., Gil-Romera, G., Grimm, E.C., Huntley, B., Kunes, P., Kühl, N., Leydet, M., Lotter, A.F., Tarasov, P.E., Tonkov, S., 2009. The European Pollen Database: past

efforts and current activities. Veg. Hist. Archaeobotany 18, 417–424. https://doi.org/10.1007/s00334-009-0215-9

Giesecke, T., Davis, B., Brewer, S., Finsinger, W., Wolters, S., Blaauw, M., de Beaulieu, J.-L., Binney, H., Fyfe, R.M., Gaillard, M.-J., Gil-Romera, G., van der Knaap, W.O., Kuneš, P., Kühl, N., van Leeuwen, J.F.N., Leydet, M., Lotter, A.F., Ortu, E., Semmler, M., Bradshaw, R.H.W., 2014. Towards mapping the late Quaternary vegetation change of Europe. Veg. Hist. Archaeobotany 23, 75–86. https://doi.org/10.1007/s00334-012-0390-y

Lézine, A.-M., Ivory, S.J., Gosling, W.D., Scott, L., 2021. The African Pollen Database (APD) and tracing environmental change: State of the Art, in: Quaternary Vegetation Dynamics. CRC Press.

Rowe, C., Fraser, R., Harrison, S., Dodson, J., 2007. The QUAVIDA synergy: quaternary fire, vegetation and climate change in Australasia. Quat. Int. 167–168, 355–355. https://doi.org/10.1016/j.quaint.2007.04.001

Whitmore, J., Gajewski, K., Sawada, M., Williams, J.W., Shuman, B., Bartlein, P.J., Minckley, T., Viau, A.E., Webb, T., Shafer, S., Anderson, P., Brubaker, L., 2005. Modern pollen data from North America and Greenland for multi-scale paleoenvironmental applications. Quat. Sci. Rev. 24, 1828–1848. https://doi.org/10.1016/j.quascirev.2005.03.005

Williams, J.W., Grimm, E.C., Blois, J.L., Charles, D.F., Davis, E.B., Goring, S.J., Graham, R.W., Smith, A.J., Anderson, M., Arroyo-Cabrales, J., Ashworth, A.C., Betancourt, J.L., Bills, B.W., Booth, R.K., Buckland, P.I., Curry, B.B., Giesecke, T., Jackson, S.T., Latorre, C., Nichols, J., Purdum, T., Roth, R.E., Stryker, M., Takahara, H., 2018. The Neotoma Paleoecology Database, a multiproxy, international, community-curated data resource. Quat. Res. 89, 156–177. https://doi.org/10.1017/qua.2017.105

Tracked changes:

85   The pollen data synthesis LegacyPollen2.0 (Li et al., 2024b) includes 3728 3680 temporally resolved records (time-series) distributed globally. Data were collected from individual publications and the Neotoma Paleoecology Database which includes data from the European Pollen Database, the QUAVIDA data base for Australasia, the Latin American Pollen Database, the

African Pollen Database and the North American Pollen database (Flantua et al., 2015; Fyfe et al., 2009b; Giesecke et al., 2014; Lézine et a . An overview of Neotoma records included in LegacyPollen 2.0 and this reconstruction can be found in S1.

**Revised acknowledgements:**

We thank Thomas Böhmer for support with dataset curation and harmonization. The project was supported by the Bundesministerium für Bildung, Wissenschaft, Forschung und Technologie through the German Climate Modeling Initiative PALMOD (grant no. 01LP1510C to UH), the European Union (ERC, GlacialLegacy grant no. 772852 to UH), and the China Scholarship Council (grant no. 201908130165 to CL). Data were partly obtained from the Neotoma Paleoecology Database (http://www.neotomadb.org) and its constituent databases (European Pollen Database, QUAVIDA data base for Australasia, Latin American Pollen Database, African Pollen Database and the North American Pollen database). The work of data contributors, data stewards, and the Neotoma community is gratefully acknowledged.

**Tracked changes:**

*Acknowledgements.* We thank Thomas Böhmer for support with dataset curation and harmonization. The project was supported by the Bundesministerium für Bildung, Wissenschaft, Forschung und Technologie through the German Climate Modeling Initiative PALMOD

430  (grant no. 01LP1510C to UH), the European Union (ERC, GlacialLegacy grant no. 772852 to UH), and the China Scholarship Council (grant no. 201908130165 to CL). Data were partly obtained from the Neotoma Paleoecology Database (http://www.neotomadb.org) and its constituent databases (European Pollen Database, QUAVIDA data base for Australasia, Latin American Pollen Database, African Pollen Database and the North American Pollen database). The work of data contributors, data stewards, and the Neotoma community is gratefully acknowledged.

**Reply to Giesecke**

Laura Schild and Ulrike Herzschuh

**General reply**

Dear Thomas Giesecke,

Thank you for your careful review of our manuscript. While you welcome our effort to conduct a global pollen-based vegetation reconstruction you raise some concerns. We are confident that these were resolved and will ultimately underline the validity and usability of our reconstruction.

We addressed connectivity of published data to original Neotoma records by adding citations and DOIs and we are keen to add revised chronologies and taxonomic harmonizations to existing Neotoma records. We acknowledge that manuscripts such as ours would not be possible without the considerable effort toward the Neotoma Paleoecological Database and include this in our acknowledgments. Added clarification of how we envision the data set to be used is indeed needed and we will expand on this in the manuscript and provide an additional R script for dynamic rasterization. We agree with the high uncertainty connected to the reconstructions in the Southern Hemisphere and have decided to omit these reconstructions from our data set. Additionally, we changed the name of our calculated source area and omitted the optimization from our manuscript.

We have been able to make all of these adjustments and address your comments. An updated version of the dataset can now be found here: https://doi.org/10.5281/zenodo.12800159. The dataset on PANGAEA will be updated as soon as possible.

Best regards
Laura Schild and Ulrike Herzschuh

**Detailed replies**

**General comments**

**Original comment**

Before looking more closely at the manuscript I like to extend on the comment by Williams et al. of developing the underlying Legacy Pollen Dataset. Branching of a large dataset from Neotoma into the Legacy Pollen Dataset with vetting and adding metadata, results in the additional work not being linked back to Neotoma. In the spirit of open science it would be better practice to

contribute to Neotoma by uploading additional datasets and correcting or adding metadata. Look up tables for taxonomic harmonization or new chronologies could then be linked to the data in Neotoma. Republishing Neotoma derived data makes scientists using that data ignore the original data source. This also means that Neotoma loses recognition for the work of the data stewards and support for acquiring funding to maintain and develop the database.

**Reply**

We have added citations to Neotoma and constituent data bases and revised our acknowledgements. Additionally we provide a list of Neotoma records included in the LegacyPollen2.0 data set and their DOIs (see S1 in the revised Supplementary Material). We are also open to expanding Neotoma's records by adding revised chronologies and taxonomic harmonizations, but would require assistance from the Neotoma team to train one of our team members.

See tracked changes for revisions below:

85   The pollen data synthesis LegacyPollen2.0 (Li et al., 2024b) includes  3680 temporally resolved records (time-series) distributed globally. Data were collected from individual publications and the Neotoma Paleoecology Database which includes data from the European Pollen Database, the QUAVIDA data base for Australasia, the Latin American Pollen Database, the

African Pollen Database and the North American Pollen database (Flantua et al., 2015; Fyfe et al., 2009b; Giesecke et al., 2014; Lézine et . An overview of Neotoma records included in LegacyPollen 2.0 and this reconstruction can be found in S1.

*Acknowledgements.* We thank Thomas Böhmer for support with dataset curation and harmonization. The project was supported by the

430   Bundesministerium für Bildung, Wissenschaft, Forschung und Technologie through the German Climate Modeling Initiative PALMOD (grant no. 01LP1510C to UH), the European Union (ERC, GlacialLegacy grant no. 772852 to UH), and the China Scholarship Council (grant no. 201908130165 to CL). Data were partly obtained from the Neotoma Paleoecology Database (http://www.neotomadb.org) and its constituent databases (European Pollen Database, QUAVIDA data base for Australasia, Latin American Pollen Database, African Pollen Database and the North American Pollen database). The work of data contributors, data stewards, and the Neotoma community is gratefully

435   acknowledged.

**Original comment**

Regarding the here presented manuscript by Shild et al. I see several problems and directions of how to address them. I generally agree with the comments by Marie-Jose Gaillard and Michela Mariani regarding technical shortcomings, definition of the source area of pollen (NOT "relevant source area") and the recommendation to restrict a REVEALS application to the northern hemisphere. If attempting to include the southern hemisphere the authors should reduce the unrealistic assumptions as some more information could be gained. Fall speeds

could be estimated using the size of pollen grains and initial guesses of RPPEs could have been made by inviting experts working on the different continents and including recent publications on RPPEs.

**Reply**

We will adjust our terminology to refer to the parameter calculated by us as the "80% pollen source area." This term describes the area from which the median relative influx of all taxa reaches 80%. This calculation uses the lake deposition model described in Theuerkauf et al.'s REVEALSinR. Initially, pollen deposition is calculated per taxon from zmax, representing the maximum depth. While this assumes that each taxon deposits all its pollen, it simplifies the reality where pollen can originate from farther distances, and fluvial inputs into lakes are inevitable. Nonetheless, this assumption aligns with REVEALS. Through a stepwise process, the radius around the basin is incrementally expanded, and the relative influx of deposited pollen for each taxon is calculated relative to the total influx at zmax. We define our 80% pollen source radius as the radius at which the median relative influx of all taxa reaches 80%. This calculation primarily serves to provide a sense of the source area's scale to users unfamiliar with pollen data. It underscores the regional nature of lacustrine pollen data and illustrates how lake size influences this source area. We included this detailed explanation in the manuscript.

**Tracked changes:**

We calculate the radius of  the 80% pollen source area by finding the radius in which the median influx of all taxa is 80% of the total influx (as defined by the total influx in the maximum extent of regional vegetation chosen). This is calculated by employing the lake deposition model in REVEALSinR (Theuerkauf et al., 2016). Starting from $z_{max}$ the deposited pollen

140  is calculated per taxon. This is assumed to be the total pollen each taxon deposits. In a step-wise process the radius around the basin is increased and the deposited pollen relative to the total influx at $z_{max}$ is calculated for each taxon. We define our 80%

pollen source radius as the radius where the median of the relative influx of all taxa reaches 80%. The primary objective of this calculation is to provide a clear understanding of the scale of the source area for users unfamiliar with pollen data. It highlights the regional nature of lacustrine pollen data and demonstrates the influence of lake size on this source area.

145  We also reduced computational effort in REVEALSinR by implementing a maximum number of steps in the lake model used to model mixing in the basin. The number of steps was set to 500 unless $n$ falls below that maximum value for $n = basin\ radius/10$ for basins with a radius of at least 1000 m and $n = basin\ radius/2$ for basins with a radius smaller than 1000 m.

We agree that uncertainties in Southern Hemispheric reconstructions are considerable due to limited regional RPP values and have decided to exclude them from our dataset.

**example tracked changes:**

**LegacyVegetation 1.0:  Northern Hemisphere reconstruction of vegetation composition and forest cover from pollen archives of the last  14 ka**

We thank you for the suggestion to calculate missing fall speed values. While we agree that this would be an apt way to deal with missing fallspeeds, we decided against this due to the large number of taxa with missing fall speed values. As we were unable to find a way of programmatically accessing pollen morphological properties, this would have entailed considerable additional time spent. The proportion of pollen counts of taxa without RPP or fall speed information is very low and close to negligible especially when reconstructing forest cover (see revised Figure 3 below).

[Figure]

**Figure 3.** Regional source of RPP values for percentage of pollen counts per continent. A majority of pollen counts is covered by continental RPP values with the highest fraction in Europe. Only a small percentage of pollen counts has only hemispheric RPP values available. No available RPP values lead to the use of a standardized RPP value of 1±0.25.

**Original comment**

My concerns are particularly related to the aim of the authors to publish the data resulting from the analysis. REVEALS results not only provide information on past woodland cover, but also on bias reduced abundance of the major plant genera or families. Given the way the authors used RPPEs and fall speeds for the southern hemisphere such estimates are unlikely to improve the bias in pollen percentage data. However, they may invite researchers not understanding the limitations to misuse such data. This is particularly the case where the authors adjusted RPPEs to obtain an overall better fit with modern tree cover globally. Here the authors admit that the adjusted RPPEs are ecologically meaningless, and I therefore urge the authors not to publish the resulting vegetation reconstructions as they will also be meaningless. If the authors are convinced that the resulting tree cover is meaningful, they could restrict the data publication to that.

**Reply**

We consider the optimized RPP and the related reconstruction not as the most relevant outcome from this manuscript but rather an addition to the reconstruction using a synthesis of published RPP. However, due to the uncertainties associated with optimization results, we have decided to omit the reconstruction using optimized values entirely. In the reconstruction using synthesized RPP values, we highlight which taxa had continental or hemispheric RPP available and which used standardized RPP values in a separate file outlining RPP sources.

As stated above, we removed the Southern Hemisphere from the data set as we agree with the notable uncertainties.

See the updated dataset here: https://doi.org/10.5281/zenodo.12793806

**Original comment**

If a data publication is pursued, the authors should explain in which way they envision the data to be used. Is the attempt to estimate the source area of 80% of pollen to then relate the information on tree cover to that area, and if so how shall that be implemented? Also, how will overlapping areas be treated? Even on the northern hemisphere the gained information is not continues and I therefore wonder how the results will be used in climate modelling. Moreover, I did not find anything in the manuscript of how the obtained data informs on the position of northern or southern forest or tree limits, that are difficult to estimate from percentage pollen data.

**Reply**

We mainly include the calculation of the 80% source area to give an idea of the scale of source area to users not familiar with pollen data. This emphasizes the regional scale of pollen data from large basins, when reconstructions are used at site-level.
Site-wise reconstructions using large lakes are valid alone and their information can be used in gridded versions of this data set as well. We recognize that reconstructions from small lakes and peatland sites should not be used alone as site-wise reconstructions. Our aim is for the data set to be used flexibly, meaning that users can set their own temporal and spatial resolution for rasterization.This is why we did not prepare a set rasterization. To highlight this use case we provide a script to rasterize the dataset dynamically and classify grid cell reliability by record availability (https://zenodo.org/records/12800291).  Additionally, expand on this in our data usability section and clarify how we intend the data set to be used reliably. Small sites and peatland also received an additional flag in the data set as "unfit to be used on site-level" .
Tracked changes (how to use the dataset):

390     To ensure the correct utilization of the dataset and to obtain reliable analysis results, several key considerations should be followed. Firstly, rasterization mitigates individual errors by temporal and spatial averaging. This process is particularly useful in reducing the variance that might arise from individual measurements, providing a more reliable representation of the underlying signal. The reliability of reconstructions varies among different taxa due to the quality of RPP values, and this is explicitly documented in a supplementary file that outlines the sources of RPP values (see Section Code and Data availability). Reconstructions of taxa with continental RPP values are the most reliable, followed by those based on hemispheric data, with standardized RPP values being the least reliable. This hierarchy should be taken into account when interpreting the results.

395 Higher certainty is associated with forest cover reconstruction, as it is based on aggregation among taxa. Reconstructions of temporal forest cover trends are reliable, as evidenced by high correlation coefficients, despite a tendency for absolute values to be overestimated, particularly in North America. For individual time series, the reliability of data varies with the size of the lakes from which samples were taken. Only data derived from large lakes ($\geq$ 50 ha) are reliable for site-wise analyses. This distinction is clearly indicated with validity flags in the dataset. Reconstructions from smaller basins should not be used alone.

400

Even though using REVEALS improves the reconstruction of vegetation compared to pollen data, we highlight the difficulty of detecting tree lines with compositional data in our manuscript.

Tracked changes (usefulness of dataset):

    The REVEALS forest cover reconstructions presented here offer valuable insight into past vegetation changes. The global
355 dataset provides an opportunity to explore past vegetation dynamics, gaining a deeper understanding of responses, trajectories, and potential feedback mechanisms. Given the increasing discussions surrounding the possibility of tipping events in vegetation cover (Armstrong McKay et al., 2022; Lenton and Williams, 2013), this could be of considerable use. While a reconstruction of exact tree lines is not trivial with pollen data, the application of REVEALS and subsequent biomization improve treeline reconstructions as shown by Binney et al. (2011). Additionally, this dataset can address unanswered questions about Holocene
360 vegetation dynamics, including the deglacial forest conundrum (Dallmeyer et al., 2022)(Dallmeyer et al., 2022; Strandberg et al., 2022) . It also serves as a valuable tool for validating models with coupled climate and vegetation, relying which rely on extensive time series and vegetation data for accurate predictions (Dallmeyer et al., 2023). (Dallmeyer et al., 2023; Dawson et al., 2024). Comparing modeled vegetation to reconstructed vegetation could help uncover missing dynamics in coupled climate-vegetation models. New insights gained from these applications could enhance our ability to predict future changes.

365

**Original comment**

Retaining all the current aspects of the manuscript I would recommend to revise the manuscript and publish it in a disciplinary journal to discuss aspects of this analysis which yield new insights. If continuing with the modern comparison I would suggest to use available surface sample data or core tops marked as modern. Using top samples as old as 500 years in the comparison with modern woodland cover introduces a huge bias.

**Reply**

Thank you for your suggestion. We do believe that it is more useful to validate with the data set instead of using independent surface samples. We agree that 500 years constitute a large age bin, we decided to use this in order to include as many records as possible. We revised this to

include a validation using a smaller age bin in the manuscript. Below you find revised validations using gridded data from the past 100 years.

[Figure]

**Figure 9.** Remote sensing forest cover (LANDSAT) and modern reconstructed forest cover from Pollen and REVEALS (< 100 years BP) in 2x2° grid cells with mean absolute errors (MAE) and correlation coefficient ($R$) per group. Reconstructed forest cover from the original pollen data tends to overestimate observed (remote sensing) forest cover. Improvements with the REVEALS reconstruction are especially high in Europe. Validations with different grid cell sizes are available in the supplement (S3: Validation results for different spatial resolutions).

**Original comment**

In summary, I don't recommend the publication of the data from the current analysis and suggest restricting the analysis to the northern hemisphere with a publication of the findings in a topical journal.

**Reply**

The reviewers and commenters unanimously express interest in our dataset and acknowledge its potential usefulness. They also recognize the inherent challenges and uncertainties. A data publication allows for the thorough exploration and discussion of these limitations. This accurate documentation is especially necessary in the context of a PhD thesis. Publishing in a topical journal would be counterproductive as it might not allow the space or focus needed for comprehensive discussion of the data. The data have also significantly been altered compared to the original pollen data which justifies the new dataset.

**Detailed comments**

17: The uncertainties introduced here will not make the results invaluable for the investigation of past vegetation dynamics.

We rephrase this statement. But we do believe that usefulness is a given as we exclude the Southern Hemisphere and show a clear improvement of forest cover reconstructions compared to Pollen data in our validations.

20   This improved quantitative reconstruction of vegetation cover is  beneficial for the investigation of past vegetation dynamics and modern model validation. By collecting more RPP estimates  especially in

28: The study is on vegetation cover not climate.

We clarified that we are not providing past climate data.

sitions  comparable to those anticipated in the future (Dearing et al., 2012). Given the relatively brief duration of available
30   instrumental climate and vegetation data, there is a clear need for long-term  vegetation records derived from pa-leoecological archives that cover broader climatic gradients than modern datasets (Dearing et al., 2010; Dallmeyer et al., 2023).

34: Fyfe et al. 2009 is not in the reference list.

We added Fyfe et al. 2009 to the reference list and cite it at line 76 (revised).

85   The pollen data synthesis LegacyPollen2.0 (Li et al., 2024b) includes  3680 temporally resolved records (time-series) distributed globally. Data were collected from individual publications and the Neotoma Paleoecology Database which includes data from the European Pollen Database, the QUAVIDA data base for Australasia, the Latin American Pollen Database, the

African Pollen Database and the North American Pollen database (Flantua et al., 2015; Fyfe et al., 2009b; Giesecke et al., 2014; Lézine et . An overview of Neotoma records included in LegacyPollen 2.0 and this reconstruction can be found in S1.

36-38: This is not allowing for a broader but a more restricted application of pollen data as some aspects may not be possible to investigate with a reduced taxonomic depth. Please reword.

We reworded this section to highlight the improved comparability as a trade-off with taxonomic depth.

into the global database Neotoma (Williams et al., 2018). To allow for a broader application of pollen data, LegacyPollen

40 2.0 (Li et al., 2024b) offers a global, harmonized pollen dataset that underwent taxonomic standardization, metadata verification and consistent age modeling (Li et al., 2022a, 2021; Herzschuh et al., 2022). This taxonomic harmonization trades

off higher taxonomic resolution of some datasets for equivalence, resulting in overall comparability useful for analyses at

large spatial scales. Despite advances in harmonization, the use of pollen data remains limited due to the fact that pollen

49: "relevant" pollen source area is well defined, but this is not meant here.

Sugita (2007) tends to refer to the "region" or the area where "most of the pollen originates" for this model area. We have decided to use "region" here.

gional Estimates of Vegetation Abundance from Large Sites" (REVEALS) . By accounting for taxon-specific RPP and fall

speed values, as well as basin-specific parameters such as basin size and type, REVEALS models quantitative vegetation

cover in  the region surrounding a basin from pollen compositions. The model has been applied

50: Nielsen and Odgaard 2010 are a good example of applying the full Landscape Reconstruction Algorithm with a focus on LOVE not so much REVEALS.

We removed Nielsen and Odgaard from the citations here.

cover in  the region surrounding a basin from pollen compositions. The model has been applied
* * *
55 in several regional-scale studies

(Nielsen et al., 2012; Mazier et al., 2015; Hellman et al., 2008) and multiple validations have demonstrated its  ability

in approximating actual vegetation (Sugita et al., 2010; Hellman et al., 2008; Soepboer et al., 2010; Mazier et al., 2012), even

51: "ability" rather than "accuracy"

We changed "ability" to "accuracy" (see tracked changes for comment above).

105: No information is given where the RPPEs are coming from for the southern hemisphere if they are not set to 1 and not used from the northern Hemisphere. Please provide references.

We have decided to omit reconstructions for the Southern Hemisphere from the data set.

106: Fall speeds can be easily estimated based on the size of pollen grains.

Thank you for this suggestion. As outlined above we have decided against this.

111: Not "relevant source area"

We changed the name of this value to "80% pollen source area".

**2.2.2 Modifications in REVEALSinR**

We calculate the radius of  the 80% pollen source area by finding the radius in which the median influx of all taxa is 80% of the total influx (as defined by the total influx in the maximum extent of regional vegetation chosen). This is calculated

118: How was that latitudinal limit used or derived for the past?

It is constant in time.

120: In most situations the forest cover has changed dramatically over the last 500 years. Why did you not use surface sample datasets for this exercise or sites with the top sample marked as modern?

We reduced the size of the age bin for validation to the past 100 years. As outlined above, we prefer a validation with data from this dataset rather than a separate surface sample dataset.

160 of the dataset on Zenodo (https://zenodo.org/doi/10.5281/zenodo.12800290). For validation, the reconstructed forest cover of the past  100 years was rasterized and compared to modern remote sensing forest cover. Only valid grid cells as defined above were used for validation. Average tree canopy cover  for all grid cells was

168: Chenopodiaceae are an old classification and now included in Amaranthaceae. Why do they have such different RPPEs in Europe?

Chenopodiaceae are included in Amaranthaceae in our input data. They have the same RPP in Europe. The next RPP in this sentence refered to Australia and Oceania (now void as we omit the Southern Hemisphere).

189: Particularly in the southern hemisphere one would expect that the adjustments lead to improvements. It would be interesting to explore the reasons why that is not the case e.g. the low pollen productivity of many tropical trees. I cannot see that low data availability is a reason here.

We decided to exclude reconstructions from the Southern Hemisphere in our data set.

195: Starting with initial values different from 1 may also be a way to explore this further.

We have omitted the optimization of RPP values.

201: It would be interesting to look at these trends for different continents. If a first approximation of forest cover could be made by adjusting arboreal pollen percentages on a continental scale that could be used by modelers as a first order estimate. Again such comparisons should better be done using the modern analogue approach.

We now also supplied this trend on a continental scale. That this could provide a rough estimate of potential error and its direction is true and could indeed be useful for modelers. We agree that the modern analogue technique is also a valid reconstruction method for past vegetation. However, REVEALS has been used for several large-scale and continental scale reconstructions in the past, which is why we think it is valid as well (see for example Githumbi et al. 2021, Dawson et al. 2024, Serge et al. 2023).

Dawson, Andria, John W. Williams, Marie-José Gaillard, Simon J. Goring, Behnaz Pirzamanbein, Johan Lindstrom, R. Scott Anderson, u. a. „Holocene Land Cover Change in North America: Continental Trends, Regional Drivers, and Implications for Vegetation-Atmosphere Feedbacks". *Climate of the Past Discussions*, 20. Februar 2024, 1–52. https://doi.org/10.5194/cp-2024-6.

Githumbi, Esther, Ralph Fyfe, Marie-Jose Gaillard, Anna-Kari Trondman, Florence Mazier, Anne-Birgitte Nielsen, Anneli Poska, u. a. „European Pollen-Based REVEALS Land-Cover Reconstructions for the Holocene: Methodology, Mapping and Potentials". *Earth System Science Data* 14, Nr. 4 (8. April 2022): 1581–1619. https://doi.org/10.5194/essd-14-1581-2022.

Serge, M. A., F. Mazier, R. Fyfe, M.-J. Gaillard, T. Klein, A. Lagnoux, D. Galop, u. a. „Testing the Effect of Relative Pollen Productivity on the REVEALS Model: A Validated Reconstruction of Europe-Wide Holocene Vegetation". *Land* 12, Nr. 5 (Mai 2023): 986. https://doi.org/10.3390/land12050986.

Revised figure:

[Figure]

**Figure 7.** Northern Hemisphere and continental average forest cover from 2x2° grid cell means for raw pollen data and the REVEALS reconstruction (Northern Hemisphere and continental averages from different grid cell resolutions are available in S2: Reconstruction results for different spatial resolutions). Remotely sensed global average forest clover for the grid cells with valid pollen coverage is indicated with the diamond. Temporal trends are the same, but absolute forest cover is reduced in the REVEALS reconstructions compared to the original pollen data. Both reconstructions still overestimate forest cover.

289-296: I strongly disagree with these statements. Particularly regarding the northern and southern tree limits the manuscript is not demonstrating how their detection has improved.

We explain these statements. However, the validations show a clear improvement of forest cover reconstruction compared to pollen data which, at the very least, now constitutes a **better** reconstruction product. Due to the potentially high temporal and spatial variability of pollen data we cannot (and may not ever) be able to reconstruct true past vegetation (something that a modern analogue is also not able to do). Still we see that we reconstruct absolute modern vegetation better than before.

> The REVEALS forest cover reconstructions presented here offer valuable insight into past vegetation changes. The global
> 355   dataset provides an opportunity to explore past vegetation dynamics, gaining a deeper understanding of responses, trajectories,
> and potential feedback mechanisms. Given the increasing discussions surrounding the possibility of tipping events in vegetation
> cover (Armstrong McKay et al., 2022; Lenton and Williams, 2013), this could be of considerable use. While a reconstruction
> of exact tree lines is not trivial with pollen data, the application of REVEALS and subsequent biomization improve treeline
> reconstructions as shown by Binney et al. (2011). Additionally, this dataset can address unanswered questions about Holocene
> 360   vegetation dynamics, including the deglacial forest conundrum (Dallmeyer et al., 2022)(Dallmeyer et al., 2022; Strandberg et al., 2022)
> . It also serves as a valuable tool for validating models with coupled climate and vegetation, relying which rely on extensive
> time series and vegetation data for accurate predictions (Dallmeyer et al., 2023). (Dallmeyer et al., 2023; Dawson et al., 2024).
> Comparing modeled vegetation to reconstructed vegetation could help uncover missing dynamics in coupled climate-vegetation
> models. New insights gained from these applications could enhance our ability to predict future changes.
> 365

Data: I looked at the resulting data for a few sites in northern Patagonia where I am familiar with the vegetation. First of all I could not find where the RPPE for Nothofagus comes from. For a site in the steppe (Lago Mosquito) with Austrocedrus on its western shore, REVEALS estimated Holocene values around 60 % forest cover, which is too high. The adjusted run did indeed lower the forest cover to between 40 and 20 %. However, this reconstruction returned the lowest forest cover for the time that Austrocedrus woodlands were present on its western shore and tree cover was highest. This may be due to the fact that Cupressaceae was not part of the taxa used for optimization. Thus, while the amount of forest cover is more realistic in the adjusted run the reconstructed Holocene trend is the reverse of what was interpreted by the authors of the site based data. Moreover, the only tree growing abundantly near the site for the last 3000 years is not part of the reconstructed vegetation in the adjusted run. This is of cause just a single example, but it does illustrate my earlier points.

We have decided to omit reconstructions from the Southern Hemisphere from our data set.

---

## Referee Report (RR1)

**Report for the responsible Editor**

By Marie-José Gaillard

Dear Editor, dear authors

I focused mainly on the authors' responses to my comments on the first submitted version of the manuscript, and on the implementation of the related revisions. I did read the entire revised manuscript but only partly commented the manuscript for revisions that still need to be done. I provide here general comments on issues that I think require revisions. The authors will have to implement the revisions consistently throughout the text, figures, tables and figure/table captions, and not only in places where I have commented in the revised manuscript. I did not check the authors' responses to the other reviewers but have seen that the authors have considered those comments in the revision.

**General comments**

The authors have made substantial revisions that were necessary such as deleting the southern hemisphere from the reconstruction and producing REVEALS estimates based on pollen records from several sites within areas (grid cells) of various sizes and for time windows of various lengths. This leads to more acceptable results. I appreciate the hard work made to finalize this revision, but there are still misunderstandings that needs to be clarified in the paper.

**1.**One of my major concerns is the calculation of REVEALS mean estimates based on the REVEALS reconstructions for several sites within grid cells and several pollen counts within time windows, i.e. the step that the authors call "aggregation" in space and time.

For the **"aggregation" in space** the authors calculate the mean of the individual site REVEALS estimates without any weighting by the K coefficient that is dependent of basin size (the larger the basin, the heavier the weighting should be for each taxon, and vice versa). Such a weighting is implemented in Sugita's REVEALS computer program but not in REVEALSinR. In Sugita's method, the REVEALS estimates from individual sites within a grid cell are weighted with the taxon-specific "pollen dispersal-deposition coefficient K" of all pollen taxa involved, se e.g. Li et al. (2017). This should be clarified under METHODS.

For the **"aggregation" in time** the authors similarly calculate the mean of the individual counted level REVEALS estimates. The reliability of REVEALS estimates depends, among other things, on the size of the pollen count. In this context, the usual size of pollen counts (often around 1000, seldom more, quite often around 500 and sometimes less) is a low pollen count. This implies that all REVEALS estimates in the Schield et al. REVEALS dataset are of relatively low reliability and calculating the mean of these REVEALS estimates does not make them more reliable. All earlier continental Holocene REVEALS reconstructions have worked with time windows of such a length that it would maximize the size of the counts without using too long time windows (generally maximum 500 years). The compromise to make depends on the aim of the study. One has then to sum pollen counts within each time window and use this new pollen count for the REVEALS application to obtain the REVEALS estimates for the time window (see e.g. Githumbi et al., 2022). This procedure is very different from calculating mean REVEALS estimates and is statistically the correct

way to do. **I understand that it would be a huge work to redo the work in this way for this manuscript. But this should be listed as one of the many differences between this REVEALS dataset and earlier ones.** I do not know whether the error on REVEALS estimates as calculated by REVEALSinR (see my point below) is sensitive to the size of pollen counts. I guess not, but I can't find anything about this issue in the REVEALSinR original paper or elsewhere. In that case, this is also an aspect that makes REVEALS applications using REVEALSinR weaker if the size of pollen counts is not considered in the error estimate on REVEALS results.

**2.** Another major difference between implementation of the REVEALS model with the computer programs of Sugita and REVEALSinR of Theuerkauf et al. (2016) is **the calculation of the uncertainties (errors) on the REVEALS estimates**. The REVEALS standard error accounts for the standard errors (or deviations) of the relative pollen productivities for the individual pollen taxa and on the number of pollen counted; i.e. the size of the pollen count matters. The error calculated in REVEALSinR does not consider the RPP errors. I do not mean that the errors from the REVEALSinR program are wrong, but it is a pity not to use the errors on RPPs as this parameter is very influential on the final REVEALS estimate of plant cover. **This difference between the two applications should at least been mentioned.**

**3. 80% pollen source area**: this information should be presented as an alternative to estimate the size of the region that is represented by REVEALS estimates of plant cover. **Sugita (2007a) who developed the REVEALS model assumes that Zmax is the size of the region represented by REVEALS estimates (see also Li et al., 2017)**. Zmax can only be assumed (you assumed it to be 1000 km over the entire study region) and the region from which most of the pollen are coming (in your case 80%) can be estimated. See also Hellman et al., 2008b (in VHA) who assumed Zmax to be 400 km (distance from the pollen site) in S Sweden and the 90% source area (200 km) was considered to be the area from which most of the pollen came. **One should therefore state that the assumed value for Zmax influences the estimate of x% pollen source area**. Please, **also specify what dispersal model you use, the Gaussian Plume Model or the Lagrangian Stochastic Model, for estimating your 80% pollen source area, which makes also a difference** (see Theuerkauf et al., 2016).

Two additional comments, minor but still important:

**4. Avoid the term reconstruction for pollen percentages or raw pollen data**. These are simply data, pollen% are not a reconstruction of vegetation, they are proxy data of vegetation, while a traditional narrative interpreting the pollen percentages using various kind of information is a reconstruction, as REVEALS-based estimates of plant cover is a reconstruction of past plant cover. I advise you to revise this throughout the manuscript, text and Figures. I made comments in the manuscript about that, but not everywhere. Using "reconstruction" for pollen data is misleading, and makes the text difficult to understand in some places.

**5.** I would use the terms **"(total) tree pollen"** and **"(total) tree cover"** instead of "forest cover" when it refers to pollen % and REVEALS-based estimates of tree cover. It is important to be clear in terms of what you are comparing the satellite vegetation (forest cover) with. If you choose to follow my advice, revise the manuscript consequently. I made comments in the manuscript about that, but not everywhere.

**In conclusion:**

I miss a description of your new REVEALS dataset for the N Hemisphere in comparison to the earlier continental REVEALS dataset for Europe, China and N America. What is **different** and **what are the improvements**.

**1.**In terms of what is different in the methodology, **please see my major comments above, and specific comments in the revised manuscript**. Do not forget that you use different chronologies than those used in earlier reconstructions. They might not be so different, but we do not know. **The best solution is to describe all the differences in methodology already in the METHODS section, in the part describing REVEALSinR and in the part describing how you "aggregate" site-specific and level (time)-specific REVEALS estimates to mean REVEALS estimates (level-specific meaning using single analysed levels/samples to run REVEALS.**

**2. In my view, the improvements in your REVEALS dataset are:**

-You have included in your synthesis the pollen records from the northern hemisphere between Europe and China, those sites that were included in Cao et al (2019) REVEALS reconstruction, and applied REVEALS on them in accordance with the methodology you use for the rest of the Northern Hemisphere.

-Further, it would be informative to know how many pollen records you use overall and in specific continents (Europe, China, N America) for which earlier REVEALS reconstructions exist. For Europe, compare with Serge et al. (2023). In terms of RPP, you should also mention if you use more RPP values than in earlier studies and also clarify that your RPP synthesis is made in a different way (different rules) than those by Githumbi et al. (2022) for Europe and Li et al. (2018) for China. For China, the improvement is that you have added new recent RPP values from recent papers.

-Finally, your new REVEALS dataset should be presented as **an alternative dataset that is more flexible that the earlier continental ones as it allows users to amalgamate the REVEALS estimates in space choosing various sizes of grid cells, and in time choosing various length of time windows.** It should be stated, however, that mean REVEALS estimates over space do not weight the K coefficient according to lake/bog size, and that mean REVEALS estimates over time are not as reliable as REVEALS estimates based on the total pollen count in a time window (see my comment above). With flexibility you loose reliability. **This should be clarified for the users.**

**LegacyVegetation 1.0:  Northern Hemisphere reconstruction of vegetation composition and forest cover from pollen archives of the last  14 ka**

Laura Schild[1,2], Peter Ewald[1,2], Chenzhi Li[1,2], Raphaël Hébert[1], Thomas Laepple[1,3], and Ulrike Herzschuh[1,2,4]

[1]Helmholtz Centre for Polar and Marine Research, Research Unit Potsdam, Alfred Wegener Institute (AWI), Germany
[2]Institute of Environmental Sciences and Geography, University of Potsdam, Karl-Liebknecht-Straße 24-25, Potsdam, Germany
[3]MARUM-Center for Marine Environmental Sciences and Faculty of Geosciences, University of Bremen, Germany
[4]Institute of Biochemistry and Biology, University of Potsdam, Karl-Liebknecht-Straße 24-25, Potsdam, Germany

**Correspondence:** Ulrike Herzschuh (ulrike.herzschuh@awi.de)

**Abstract.** With rapid anthropogenic climate change future vegetation trajectories are uncertain. Climate-vegetation models can be useful for predictions but need extensive data on past vegetation for validation and improving systemic understanding. Even though pollen data provide a great source of this information, the data is compositionally biased due to differences in taxon-specific relative pollen productivity (RPP) and dispersal.

5   Here we present a Northern Hemisphere reconstruction of quantitative regional vegetation cover from a  sedimentary pollen data set for the last  14 ka using the REVEALS model to correct for taxon- and basin-specific biases.  For the reconstruction, we  expanded on a previously published synthesis of continental RPP values.

10   The data sets include taxonomic compositions as well as reconstructed forest cover for each original pollen sample.  80% pollen sources areas were  calculated for large lakes and are included in the data set . Additional metadata includes modeled ages, age model sources, basin locations, types and sizes.

The improvements in forest cover reconstructions with the REVEALS reconstruction using  continental RPP values range from 24% (North America) to 72% (Europe ) relative to the mean absolute error (MAE)  of the pollen  reconstruction.  The dataset can be used as a grid with binned and aggregated samples (adjustable script provided on Zenodo; https://zenodo.org/doi/10.5281/zenodo.12800290) or as individual timeseries if the record's basin size exceeds 50 20   ha.

This improved quantitative reconstruction of vegetation cover is  beneficial for the investigation of past vegetation dynamics and modern model validation. By collecting more RPP estimates  especially in

North America and adding more records to existing pollen data syntheses, reconstructions may be improved even further.  The REVEALS reconstruction is freely available on PANGAEA (see Data availability section).

**1 Introduction**

[revised manuscript text omitted]

Here we present  reconstructed quantitative vegetation cover for the Northern Hemisphere from the LegacyPollen2.0 dataset - an updated global taxonomically and temporally standardized fossil pollen dataset of  3680 palynological records - using REVEALS spanning the last 14k years. The data sets were created using existing estimates of taxon-specific parameters. The REVEALS reconstruction includes corrected vegetation compositions as well as reconstructed forest cover.

**2 Methods**

**2.1 Pollen Data Set**

The pollen data synthesis LegacyPollen2.0 (Li et al., 2024b) includes  3680 temporally resolved records (time-series) distributed globally. Data were collected from individual publications and the Neotoma Paleoecology Database which includes data from the European Pollen Database, the QUAVIDA data base for Australasia, the Latin American Pollen Database, the

African Pollen Database and the North American Pollen database (Flantua et al., 2015; Fyfe et al., 2009b; Giesecke et al., 2014; Lézine et a

90 . An overview of Neotoma records included in LegacyPollen 2.0 and this reconstruction can be found in S1.

Sediment and peat cores used for the creation of pollen data are of lacustrine, peat and marine origin. For the REVEALS reconstruction only lake and peat records in the Northern Hemisphere were used ($n = 2732$) Analogous to the preceding LegacyPollen 1.0 dataset (Herzschuh et al., 2022), the data synthesis involved revising age modeling and taxonomic harmonization for consistency of records. Spatial data coverage of records in the reconstruction is  dense in

95 Europe (1275 records) and  North America (1016 records) and sparsest in Asia ( 441) (see Fig. 1). The records' sample density decreases with age (see Fig. 2).

[Figure]

**Figure 1.** ==Pollen record locations== in the LegacyVegetation dataset. Colors indicate record type (large lake $\geq$ 50 ha). Record density is  highest in Europe and Eastern North America, and lowest in  Northern and  Central Asia.

**2.2 Implementing REVEALS**

The REVEALS model ("Regional Estimates of Vegetation Abundance from Large Sites") estimates quantitative vegetation

100 coverage from pollen assemblages using site and taxon-specific parameters (Sugita, 2007). Based on wind speed and taxon-specific fall speed, pollen dispersal is modeled in ring sources around the basin and deposition over the basin is integrated to give pollen influx. Together with RPP this dispersal factor is used to correct original pollen counts to better represent  actual vegetation (see Equation 1 and Table 1). By running the model with variations of relative pollen productivity (RPP) values, a statistical distribution of results is calculated.

[Figure]

**Figure 2.** Temporal coverage of records in the LegacyVegetation dataset per continent. Bins are  500 years wide. Sample count decreases with age  and Europe has the most samples overall.

$$\hat{V}_i = \frac{n_{i,k}/\hat{\alpha}_i \int_R^{Z_{max}} g_i(z)dz}{\sum_{j=1}^{m}(n_{j,k}/\hat{\alpha}_j \int_R^{Z_{max}} g_i(z)dz)} \tag{1}$$

The REVEALS model follows a set of assumptions. Firstly, neither directionality nor pollen transport through agents other than

**Table 1.** Algebraic terms in the REVEALS equation (see Equation 1)

| Function term |  definition |
|---:|:---|
| $\hat{V}_i$ | vegetation estimate of taxon i |
| $n_{i,k}$ | pollen counts of taxon i at site k |
| $\alpha_i$ | relative pollen productivity of taxon i |
| $R$ | basin radius |
| $Z_{max}$ | maximum extent of regional vegetation |
| $z$ | distance from a point in the center of a basin |
| $g_i$ | dispersal and deposition function for taxon i |

wind are considered in the model. Additionally, it is assumed that the basin is circular with no source of pollen within the basin radius. The peatland and bog sites used in our reconstructions inherently violate this assumption. Nevertheless, the quantitative reconstruction of vegetation cover from peatland cores is possible by using Prentice's deposition model (Prentice, 1985, 1988) instead of Sugita's deposition model (Sugita, 1993) in the dispersal and deposition function (see Eq. 1; Sugita, 2007). Previous studies show that results from small bogs are still reliable when aggregated, while results from large bogs tend to deviate from

those of large lakes (Trondman et al., 2015; Mazier et al., 2012; Trondman et al., 2016). Using peatland records for reconstructions is, therefore, appropriate when spatially averaging multiple sites. We use the implementation of REVEALS from the R package REVEALSinR (Theuerkauf et al., 2016).

**2.2.1**

 For further details on the REVEALS model see the original publication Sugita (2007) or Githumbi et al. (2022).

**2.2.1 Parameters and Model Settings**

For each taxon, values for RPP (with uncertainties provided as standard deviation) and fall speeds are used.  We made use of the synthesis of Northern Hemisphere RPP and fall speed values by Wieczorek and Herzschuh (2020). Several RPP studies published since this synthesis were added to the compilation (Geng et al., 2022; Li et al., 2022b; Wang et al., 2021; Huang et al., 2021; Zhang et al., 2021a, b; Wan et al., 2020, 2023; Jian . The methods by Wieczorek and Herzschuh (2020) were followed fore study selection and calculation of synthesis values. An overview of original values and synthesized values can be found in Appendix A and B respectively. When available, we use continent-specific values in our reconstruction. For taxa with no continental values present, we use  Northern Hemispheric values. If no values exist for a taxon, RPP is set to a constant (RPP = 1, $\sigma$=0.25) and fall speeds are filled with mean continental fall speeds. Continental RPP values are available for the majority of pollen counts in all three continents (see Fig. 3). The fraction of pollen counts for which standard RPP values were assumed is highest in North America but still < 10%. For each site, the REVEALS model also requires information on basin type, basin size and original pollen counts, all of which were collected in the LegacyPollen 2.0 dataset (Li et al., 2024b). Apart from taxon- and basin-specific parameters the REVEALS model requires several constant parameters to be set, which can be found in Table 2.

**2.2.2 Modifications in REVEALSinR**

We calculate the radius of  the 80% pollen source area by finding the radius in which the median influx of all taxa is 80% of the total influx (as defined by the total influx in the maximum extent of regional vegetation chosen). This is calculated by employing the lake deposition model in REVEALSinR (Theuerkauf et al., 2016). Starting from $z_{max}$ the deposited pollen is calculated per taxon. This is assumed to be the total pollen each taxon deposits. In a step-wise process the radius around the basin is increased and the deposited pollen relative to the total influx at $z_{max}$ is calculated for each taxon. We define our 80%

**Table 2.** Static model parameters and model settings for REVEALS runs using REVEALSinR (Theuerkauf et al., 2016).

| Parameter | Values and settings used in model run |
| --- | --- |
| atmospheric model | unstable atmosphere |
| dispersal model | gaussian plume |
| wind speed | $3m \times s^{-1}$ |
| maximum extent of regional vegetation (region cutoff) | 1000 km |
| number of RPP variations | 2000 |
| peatland basin radius | 100 m |
| function to randomize pollen counts | rmultinom_reveals |

[Figure]

**Figure 3.**  Regional source of RPP values for percentage of pollen counts per continent . A  majority of pollen counts is covered by continental RPP values with the highest fraction in Europe. Only a small percentage of pollen counts has only hemispheric RPP  values available. No available RPP values lead to the  use of  a standardized RPP value of 1±0.25.

pollen source radius as the radius where the median of the relative influx of all taxa reaches 80%. The primary objective of this calculation is to provide a clear understanding of the scale of the source area for users unfamiliar with pollen data. It highlights the regional nature of lacustrine pollen data and demonstrates the influence of lake size on this source area.

We also reduced computational effort in REVEALSinR by implementing a maximum number of steps in the lake model used to model mixing in the basin. The number of steps was set to 500 unless $n$ falls below that maximum value for $n = basin\,radius/10$ for basins with a radius of at least 1000 m and $n = basin\,radius/2$ for basins with a radius smaller than 1000 m.

150

**2.3 Reconstruction of forest cover and validation**

Forest cover was reconstructed by summing up percentages of arboreal taxa (see S2: List of arboreal taxa) with Betulaceae, *Betula*, and *Alnus* being classified as arboreal at sites below 70° N. The mean reconstructed compositional coverages from the REVEALS results were used for the forest cover reconstructions. REVEALS results were then rasterized to aggregate and
155 include records from smaller basins as well. Reconstructed time series were averaged in 500 year bins and then rasterized in grids of differing spatial resolution. A grid cell was classified as having a valid reconstruction when it contained records from at least one large lake (>= 50 ha) or at least two small basins following Serge et al. (2023). Standard deviations of the REVEALS estimates were aggregated by applying the delta method by Stuart and Ord (1994), using the same equation as Wieczorek and Herzschuh (2020). We provide a script for rasterization with adjustable temporal and spatial resolution for users
160 of the dataset on Zenodo (https://zenodo.org/doi/10.5281/zenodo.12800290). For validation, the reconstructed forest cover of the past  100 years was rasterized and compared to modern remote sensing forest cover. Only valid grid cells as defined above were used for validation. Average tree canopy cover  for all grid cells was extracted from the Landsat Global Forest Cover Change (GFCC) data set from the temporal average of the years 2000, 2005, 2010 and 2015 (Sexton et al., 2013; Townshend, 2016). An openness correction was applied to sites containing urban areas
165 and paved surfaces within the 80% pollen source areas (PSA) to correct for areas without any pollen sources and thus  ensure comparability to modern remote sensing forest cover (see Equations 2-4). For this, the percentage of unvegetated land cover classes for the year 2015 in the ESA CCI land cover data set was used (ESA, 2017, see Table 3). Areas covered by water or ice are already considered as missing values in the remote sensing forest cover data set and do not need to be corrected for. Forest cover was validated  for each grid cell and mean absolute error (MAE) and correlation coefficients calculated
170 for each continent. No openness correction was applied to the reconstruction values in the final dataset. Validation for a 2x2° grid is included in the results section. Further validations using 1°,5°, and 10° resolution are included in the supplementary material (S3: Validation results for different spatial resolutions).

**Table 3.** Unvegetated land cover classes in ESA CCI LC chosen for the openness correction.

| Name | Code |
| --- | --- |
| Urban areas | 190 |
| Bare areas | 200 |
| Consolidated bare areas | 201 |
| Unconsolidated bare areas | 202 |

$$unvegetated\ classes = \{190, 200, 201, 202\} \tag{2}$$

$$unvegetated\ (\%) = \frac{\sum cells\ in\ PSA \in open\ classes}{\sum cells\ in\ PSA} \frac{\sum cells\ in\ PSA \in unvegetated\ classes}{\sum cells\ in\ PSA} \tag{3}$$

$$corrected\ tree\ cover = reconstructed\ tree\ cover \times (1 - unvegetated) \tag{4}$$

**2.4**

180 ~~In addition to the REVEALS approach, which is motivated by a biophysical model but also based on a large number of model choices and parameters, we also apply a statistical approach. Here, RPP values for common taxa are estimated by minimizing the misfit of reconstructed and remote sensing forest cover. For the optimization we rely on the "L-BFGS-B" method (Byrd et al., 1995), which allows for box constraints, and minimize the residual sum of squares (RSS) of reconstructed forest cover with remote sensing forest cover. RPP values were bound by upper and lower limits based on original RPP values~~
185

$$original\ RPP \times 0.25 < new\ RPP < original\ RPP \times 4$$

190 ~~predefined radius (exclusion buffer) were excluded from the optimization to account for spatial autocorrelation. The optimized RPP values were then applied to the forest cover reconstruction of the site left-out and the absolute error with remote sensing forest cover recorded. This was repeated with 20 sites to estimate the spread of MAE. The exclusion buffer around the validation site was set to 200 km. Due to computational limitations (roughly 3 hours for one continental SLOO fold using 20 threads with 1.2 GHz CPU each), the number of sites used per continental optimization during the cross-validation was limited to 100,~~
195

**3 Data summary**

**3.1 80% Pollen Source Areas**

Using REVEALS, radii of  80% pollen source areas were calculated for  large lakes(see
200 Fig. 4). The  radii indicate in which area 80% of the deposited pollen originated from (see Section 2.2.2) and yield an understanding of which area the pollen record is representative of., which is especially useful when individual time series from large lakes are being used for analyses. The 80% pollen source areas are roughly a function of basin size (see Fig. 5) and range between  155 km and 762 km. The median 80% pollen source radius is  225 km including all  large lakes.

[Figure]

**Figure 4.** Map indicating the  relevant pollen source areas for large lakes. Many small basins in Europe lead to smaller 80% pollen source areas. Several large basins and correspondingly large 80% pollen source areas exist in Asia. In general the 80% pollen source areas highlight the regional nature of the pollen record signal.

[Figure]

**Figure 5.**  Scatterplot of basin diameter and 80% pollen source area of  large lakes in the REVEALS data set.  In general, larger basins have larger pollen source areas with the relationship between  diameter and 80% pollen source radius being roughly logarithmic.

**3.2**

205 ~~The calculated pollen source areas (see section 3.1) were used to extract modern remote sensing forest cover per site. Within the optimization, RPP values were adjusted for the ten most common taxa per continent to improve the fit between reconstructed and remotely sensed modern forest cover. The RPP values are one of the main correction factors applied in REVEALS. Here we compare original and optimized RPP values for the relevant continental taxa.~~

The magnitude of adjustment from original to optimized RPP values differs between continents (see Fig. **??**). The highest and lowest absolute change respectively occurred for *Quercus* (4.08) and Fabaceae (0.09) in Africa, for *Picea* (87.81) and *Ephedra* (0.43) in Asia, for *Pinus* (32.58) and Asteraceae (0.16) in Europe, for *Alnus* (1.79) and Amaranthaceae (in which we included Chenopodiaceae, 0.02) in Australia and Oceania, for Amaranthaceae (63.81) and *Tsuga* (0.43) in North America, and for Amaranthaceae (15.91) and Melastomataceae (0.74) in South America (see Appendix B). Relative change of RPP values is mostly positive with many taxa reaching an increase of three times the original RPP value. This is the maximum RPP value that can be reached, as the upper constraint for RPP optimization was set as 4 times the original RPP value (see Section 2.4). In most cases RPP values for arboreal taxa are increased. This increase represents reconstructed forest cover being regulated down as can be seen in the validations (see Fig. 9). Dumbbell graph illustrating original and optimized RPP values per continent and taxon. Arboreal taxa such as Pinus, Picea, Quercus have increases that are especially large.

**3.2    Reconstructed compositions**

[Figure]

**Figure 6.** Average continental taxonomic coverages per reconstruction for the 8 most common taxa per continent.  Differences are  especially evident for Pinus, Artemisia, and Betula, which all have decreased coverages after the  application of REVEALS, as well as Poaceae and Cyperaceae with increased coverages.

220  REVEALS was used to reconstruct quantitative vegetation cover.  Here we compared these reconstructed compositions  to the original pollen composition.

Differences in composition  between Pollen data and REVEALS are apparent for all continents of the Northern Hemisphere.
225 ~~optimized RPP values both increase *Larix* cover in Asia, Ericales cover in Europe, and decrease *Picea* cover in North America, although the version with optimized RPP values does so more strongly (see Fig. 6). The original and the optimized version also diverge in the adjustment of some taxa. *Artemisia* cover in Asia is reduced by the original version and increased by the optimized one. *Picea* cover stays roughly the same with original RPP values in North America and decreases with optimized ones and while Asteraceae cover in Europeis increased in the REVEALS version with original RPP values, it is considerably~~
230

~~In the Southern Hemisphere the differences between reconstructions are much less pronounced (see Fig. 6). The REVEALS reconstruction with original RPP values is almost indistinguishable from the original pollen spectra and adjustments in the optimized version are also much smaller than in the Northern Hemisphere. An increase in Cyperaceae cover in Australia and Oceania, decreases of Asteraceae and Cyperaceae in South America, and~~ Some clear examples include: increases of Cyperaceae
235 in all continents, decreases of *Quercus* in Africa

 Betula in Europe, decreases of Pinus in all continents, and increases of Acer in North America with the application of REVEALS and its intended correction of taxon-sepcific biases (see Fig.
240  6).

**3.3 Reconstructed forest cover**

Using the compositional data available from the original pollen data  and the RE-
245 VEALS run, we reconstructed forest cover for all sites and samples and rasterized the result with different spatial resolutions. The temporal trend in Northern Hemisphere forest cover is the same for  both reconstructions. Forest cover increases from  14 ka BP until roughly 6 ka BP and decreases again towards the present (see Fig. 7). REVEALS reconstructed forest cover is generally lower than forest cover from original pollen compositions. On average forest cover values from the REVEALS run  are roughly 14.54%
250 lower than values from original pollen compositions. The temporal trends in Asia and North America are positive, whereas forest cover in Europe has its maximum around 6 ka BP and has been decreasing since.

Forest cover is  generally highest in Eastern North America. This

[Figure]

**Figure 7.**  Northern Hemisphere and continental average forest cover from 2x2° grid cell means for raw pollen data  and the REVEALS reconstruction (Northern Hemisphere and continental averages from different grid cell resolutions are available in S2: Reconstruction results for different spatial resolutions). Remotely sensed global average forest  clover for the grid cells with valid pollen  coverage is indicated with the diamond. Temporal trends are the same, but absolute forest cover is reduced in the REVEALS reconstructions compared to the original pollen data.  Both reconstructions still overestimate forest cover.

is also where data coverage is best in North America (see Fig. 8).
255  Density of valid grid cells is very high in Europe, where forest cover increases until roughly 6 ka BP and then decreases. Data coverage in Asia is sparse, but valid grid cells indicate higher forest cover on the Southeastern coast and in the boreal biome. Rather open areas exist at the Tibetan Plateau and at very high latitudes. The forest cover derived from the  REVEALS reconstruc-
260 tion is generally lower. However, the difference between Pollen and REVEALS forest cover is smaller in North America than in Europe and Asia.

[Figure]

**Figure 8.** ==Reconstructed forest cover in== 2x2° grid cells from raw pollen data  and the REVEALS reconstruction for 5 example time slices (reconstructions with different grid cell sizes are available in the in S2: Reconstruction results for different spatial resolutions). Valid cells are filled and include reconstructions from at least one large lake ($\geq$ 50 ha) or several smaller basins. Forest cover in Eastern North America is  higher than in Europe and Asia.  REVEALS reconstructed forest cover  is generally lower than ==raw pollen reconstructions==.

**3.4  Validation with gridded data sets**

**3.4.1**

265  Remote sensing forest cover within relevant pollen source areas was used to validate the modern, reconstructed forest cover from the original pollen data and  the REVEALS run for each grid cell. Here we present validation of gridded data with a 2° spatial resolution. Validations with additional spatial resolutions differ only marginally and are included in the supplementary materials (S3: Validation results for different spatial resolutions).

270  Forest cover reconstructed from original pollen data is predominantly higher than remote sensing forest cover with a  mean absolute error (MAE) of  33.05% in the Northern Hemisphere (see Fig. 10a). As reconstructed forest cover is much lower for  the REVEALS reconstruction (see Fig.7),

reconstructions.  the MAE value is reduced significantly to 19.73% (see Fig.

275 9a).

[Figure]

**Figure 9.** Remote sensing forest cover (LANDSAT) and modern reconstructed forest cover from Pollen  and REVEALS (< 100 years BP) in 2x2° grid cells with  mean absolute errors (MAE) and  correlation coefficient (R) per group. Reconstructed forest cover from the original pollen data tends to overestimate observed (remote sensing) forest cover.  Improvements with the REVEALS  reconstruction are especially high in Europe. Validations with different grid cell sizes are available in the  supplement (S3: Validation results for different spatial resolutions).

Continental mean absolute errors (MAE) in forest cover from original pollen data range from 24.61% (Asia) to 37.49% forest cover (North America, see Fig. 9b). All continental MAE values are lower for the REVEALS reconstruction  and range from 9.44% (Europe) to 27.27% (North America). The

280 improvement is largest in Europe (72% relative to the initial MAE of the pollen-based reconstruction, see Fig. 9 and 10)

and smallest in North America (24%). ~~Forest cover from the REVEALS reconstruction with optimized RPP values reduces continental MAE values even further with values ranging between 9.1% (Africa) and 21.08% forest cover (South America). MAE are generally improved more with optimized RPP values with the exception of records in Australia and Oceania. The largest improvement (relative to the pollen-based forest cover MAE) was achieved in North America (65%) but reconstructions in Europe (61%) and Asia (48%) also reduced the original MAE by more than or roughly half. The REVEALS runwith optimized RPP valuesthebest with, with the exception of records from Australia and Oceania. Additionally, the reduction of forest cover MAE, and therefore the reconstruction improvement, was much larger in the continents of the Northern Hemisphere for both REVEALS runs~~. Nevertheless, forest cover still tends to be overestimated.

[Figure]

**Figure 10.**  Forest cover reconstruction  per continent  for a gridded 2x2° reconstruction.  Mean errors decreased with the  REVEALS reconstruction  for all continents but are still generally  > 0 (overestimation of forest cover). Lowest errors are present in Europe.

Spatial patterns are present for the errors of  both forest cover reconstructions (see Fig. 11). In  Europe the REVEALS  reconstruction manages to reduce errors extensively. In  Eastern and coastal Northwestern North America, the

REVEALS reconstruction still tends to overestimate forest cover, even with the application of REVEALS and after optimizing. This could be due to a lack of continental RPP values. The same is the case for several records in eastern AsiaIn North America, few RPP studies are available (see Appendix A) and more taxa are assigned hemispheric or standardized values than in the other continents.

[Figure]

**Figure 11.** Map of the reconstruction error (in % forest cover) for forest cover reconstructed from Pollen , REVEALS with original RPP values and REVEALS data. Remaining errors with optimized RPP valuesthe overall better REVEALS reconstructions are especially high in North America (Northern West Coast, Labrador Peninsula).

The large difference between forest cover reconstructed from original pollen compositions and remote sensing forest cover could be due to the difference in the signal that is recorded. Remote sensing forest cover records the canopy, whereas pollen data also records the vegetation present below the tallest canopy. Several layers of trees could, therefore, increase the percentage of arboreal taxa recorded. Even though this comparison between these data sources may not be straightforward, it is still necessary for this large-scale validation of reconstruction as few other vegetation data is available globally. Additionally, it is more likely that the overestimation of forest cover in the initial pollen data is due to the higher production of pollen by trees than by non-arboreal taxa. This leads to an overrepresentation of arboreal taxa in the pollen record. By using REVEALS, the pollen productivity of taxa is taken into account and corrected for. The proportion of arboreal taxa is therefore strongly reduced in the vegetation compositions reconstructed using REVEALS.

The reasons for the difference in reconstruction improvements between the hemispheres could lie both in the smaller number of records available and the lack of regional RPP estimates for continents of the Southern Hemisphere. The latter play an important role as the optimization is based on the original RPP estimates and can only determine better values if these are in

315

320 **3.4.1**

~~A spatial leave-one-out validation was conducted by excluding a subset of available records in the optimization (see Sect. 2.4). By separating testing and training sites, the true spread of forest cover error from the optimization of RPP values can be evaluated. This also indicates the potential error if the optimized parameters were to be applied to new records. The distribution of absolute error from the SLOO validation is comparable to that of the reconstruction utilizing the complete optimization for~~
325 ~~Africa, Asia, Europe and South America (see Fig. ??). In North America, the absolute error spread and media are larger in the SLOO validation than in both REVEALS reconstructions. As errors in North America were comparably large to begin with (see Fig. 10 and 12), this could be due to the small number of folds conducted in the SLOO validation (n = 20) as well as the small number of records used (n = 100). The same could be the case for Australia and Oceania. Additionally, the spatial buffer in the SLOO validation leads to even fewer records being available for optimization. This could further decrease improvements~~
330 ~~in Australia and Oceania optimization. Overall the SLOO validation results indicate that the optimization success is relatively stable in Africa, Asia, Europe and South America. In North America, the spatial variability leads to higher uncertainty and in Australia and Oceania the optimization is not able to decrease absolute errors considerably. Boxplot of absolute errors from continental SLOO validations (20 folds) and from validations with complete Pollen, REVEALS (original RPP) and REVEALS (optimized RPP) data sets. The SLOO validation shows how reliable the optimized parameters are when testing sites were not~~
335

**4 Dataset applications and limitations**

Our reconstructed quantitative vegetation cover datasets using REVEALS provide  reconstructions of taxonomic compositions as well as forest cover  in Europe, Asia, and North America and extend
340 to 14 ka BP. The reconstructions made use of taxon-specific parameters and were, thus, able to correct some of the compositional biases present in pollen compositions. Notably, the error in modern reconstructed forest cover was reduced compared to pollen-based reconstructions on all continents which shows that improvements in forest cover reconstructions from  REVEALS applications are considerable.

345 Reconstruction results are also similar to available large-scale pollen-based vegetation reconstructions. Increases in forest cover in northern and eastern Asia up until the Holocene thermal maximum as seen in our results are consistent with reconstructions by Cao et al. (2019) and Tian et al. (2016). The reconstructed spatial patterns of forest cover in China with low forest cover in the North China plain and the Tibetan Plateau and a higher forest cover along the east coast and the south agree with previous reconstructions as well (Li et al., 2023, 2022b, 2024a). Results for European forest cover also roughly correspond with previous REVEALS applications and show an increase of forest cover after the last glacial maximum until roughly  6 ka BP (Githumbi et al., 2022; Fyfe et al., 2015; Serge et al., 2023; Strandberg . The gridded reconstruction by Serge et al. (2023) was even validated with modern remote sensing forest cover and showed a good fit.

The REVEALS forest cover reconstructions presented here offer valuable insight into past vegetation changes. The global dataset provides an opportunity to explore past vegetation dynamics, gaining a deeper understanding of responses, trajectories, and potential feedback mechanisms. Given the increasing discussions surrounding the possibility of tipping events in vegetation cover (Armstrong McKay et al., 2022; Lenton and Williams, 2013), this could be of considerable use. While a reconstruction of exact tree lines is not trivial with pollen data, the application of REVEALS and subsequent biomization improve treeline reconstructions as shown by Binney et al. (2011). Additionally, this dataset can address unanswered questions about Holocene vegetation dynamics, including the deglacial forest conundrum (Dallmeyer et al., 2022; Strandberg et al., 2022) . It also serves as a valuable tool for validating models with coupled climate and vegetation,  which rely on extensive time series and vegetation data for accurate predictions  (Dallmeyer et al., 2023; Dawson et al., 2024). Comparing modeled vegetation to reconstructed vegetation could help uncover missing dynamics in coupled climate-vegetation models. New insights gained from these applications could enhance our ability to predict future changes.

However, the reconstructions are associated with some of the limitations of sedimentary pollen data. This includes age uncertainty, temporal mixing, and irregular spatial and temporal resolution of records. Age uncertainty is already treated as best as possible through consistent age modeling of the pollen dataset (Li et al., 2022a, 2021). Nevertheless, in general, replicating sediment and peat cores could provide more accurate estimates.

Moreover, there is uncertainty surrounding the success of the compositional reconstructions. As global compositional vegetation data is not readily available, using remote sensing forest cover poses as the best option for validation. Even with an accurate forest cover reconstruction, uncertainties persist regarding the abundance of individual taxa due to the aggregated nature of the forest cover measure. To address this, global syntheses of forest and other plant inventories or compositional remote sensing products could offer better validation. ~~The optimized RPP set can produce very unrealistic compositions, for example regarding Asteraceae in Europe. The optimization was conducted purely statistically and limited ecological information was provided as input. The use of original RPP values, originating from physical studies, is, therefore, the more conservative approach for compositional reconstructions and the optimized data set should be used with caution for compositional applications. Although, many missing RPP and fall speed values, especially for taxa in the Southern Hemisphere, result in uncertainties in the original REVEALS reconstruction as well. A higher number of RPP estimates could help increase not only the confidence~~

Another challenge lies in validating the results with past vegetation data. It is uncertain whether RPP values have remained stable over time, and historical compositional data are not only scarce but likely too recent to test this assumption (Baker et al., 2016). Vegetational compositions from sedimentary ancient DNA could provide a solution. Local aDNA vegetation signals could be averaged across multiple records within a pollen source area to generate a comparable reconstructed vegetation composition using a different proxy and to compare to pollen-based results (Niemeyer et al., 2017).

To ensure the correct utilization of the dataset and to obtain reliable analysis results, several key considerations should be followed. Firstly, rasterization mitigates individual errors by temporal and spatial averaging. This process is particularly useful in reducing the variance that might arise from individual measurements, providing a more reliable representation of the underlying signal. The reliability of reconstructions varies among different taxa due to the quality of RPP values, and this is explicitly documented in a supplementary file that outlines the sources of RPP values (see Section Code and Data availability). Reconstructions of taxa with continental RPP values are the most reliable, followed by those based on hemispheric data, with standardized RPP values being the least reliable. This hierarchy should be taken into account when interpreting the results. Higher certainty is associated with forest cover reconstruction, as it is based on aggregation among taxa. Reconstructions of temporal forest cover trends are reliable, as evidenced by high correlation coefficients, despite a tendency for absolute values to be overestimated, particularly in North America. For individual time series, the reliability of data varies with the size of the lakes from which samples were taken. Only data derived from large lakes ($\geq$ 50 ha) are reliable for site-wise analyses. This distinction is clearly indicated with validity flags in the dataset. Reconstructions from smaller basins should not be used alone.

**5 Conclusions**

We present data sets of reconstructed compositional vegetation and forest cover  in the Northern Hemisphere from a sedimentary pollen data set using the REVEALS model. We used  synthesized RPP values for  reconstruction and made use of hemispheric or standardized values, when continental ones were not available. This approach allowed us to address some of the inherent biases in pollen compositions. Considerable improvement in the reconstruction of forest cover is of the Northern Hemisphere. Even though improvements of reconstructions in the Southern Hemisphere were largely possible as well, the collection of more regional RPP values is indispensable for better reconstructions~~achieved in all continents. Improvements were smallest in North America, which suggest a need for further RPP studies.

[revised manuscript text omitted]

**Appendix A: Original RPP values**

[revised manuscript text omitted]

IPCC: Climate Change 2023: Synthesis Report. Contribution of Working Groups I, II and III to the Sixth Assessment Report of the Intergovernmental Panel on Climate Change [Core Writing Team, H. Lee and J. Romero (eds.)]. IPCC, Geneva, Switzerland., Tech. rep., Intergovernmental Panel on Climate Change (IPCC), https://www.ipcc.ch/report/ar6/syr/, 2023.

Jiang, F., Xu, Q., Zhang, S., Li, F., Zhang, K., Wang, M., Shen, W., Sun, Y., and Zhou, Z.: Relative pollen productivities of the major plant taxa of subtropical evergreen–deciduous mixed woodland in China, Journal of Quaternary Science, 35, 526–538, https://doi.org/10.1002/jqs.3197, _eprint: https://onlinelibrary.wiley.com/doi/pdf/10.1002/jqs.3197, 2020.

[revised manuscript text omitted]

Strandberg, G., Lindström, J., Poska, A., Zhang, Q., Fyfe, R., Githumbi, E., Kjellström, E., Mazier, F., Nielsen, A. B., Sugita, S., Trond-
man, A.-K., Woodbridge, J., and Gaillard, M.-J.: Mid-Holocene European climate revisited: New high-resolution regional climate model
simulations using pollen-based land-cover, Quaternary Science Reviews, 281, 107 431, https://doi.org/10.1016/j.quascirev.2022.107431,
2022.

Strandberg, G., Chen, J., Fyfe, R., Kjellström, E., Lindström, J., Poska, A., Zhang, Q., and Gaillard, M.-J.: Did the Bronze Age deforestation
of Europe affect its climate? A regional climate model study using pollen-based land cover reconstructions, Climate of the Past, 19,
1507–1530, https://doi.org/10.5194/cp-19-1507-2023, publisher: Copernicus GmbH, 2023.

Stuart, A. and Ord, J.: Kendall's Advanced Theory of Statistic, vol. Vol. 1 of *Distribution Theory*, Edward Arnold, London, 1994.

Sugita, S.: A Model of Pollen Source Area for an Entire Lake Surface, Quaternary Research, 39, 239–244,
https://doi.org/10.1006/qres.1993.1027, 1993.

Sugita, S.: Theory of quantitative reconstruction of vegetation I: pollen from large sites REVEALS regional vegetation composition, The
Holocene, 17, 229–241, https://doi.org/10.1177/0959683607075837, publisher: SAGE Publications Ltd, 2007.

Sugita, S., Parshall, T., Calcote, R., and Walker, K.: Testing the Landscape Reconstruction Algorithm for spatially explicit reconstruction
of vegetation in northern Michigan and Wisconsin, Quaternary Research, 74, 289–300, https://doi.org/10.1016/j.yqres.2010.07.008, pub-
lisher: Cambridge University Press, 2010.

Theuerkauf, M., Couwenberg, J., Kuparinen, A., and Liebscher, V.: A matter of dispersal: REVEALSinR introduces state-of-the-art dispersal
models to quantitative vegetation reconstruction, Vegetation History and Archaeobotany, 25, 541–553, https://doi.org/10.1007/s00334-
016-0572-0, 2016.

660  Tian, F., Cao, X., Dallmeyer, A., Ni, J., Zhao, Y., Wang, Y., and Herzschuh, U.: Quantitative woody cover reconstructions from eastern continental Asia of the last 22 kyr reveal strong regional peculiarities, Quaternary Science Reviews, 137, 33–44, https://doi.org/10.1016/j.quascirev.2016.02.001, 2016.

Townshend, J.: Global Forest Cover Change (GFCC) Tree Cover Multi-Year Global 30 m V003, https://doi.org/10.5067/MEASURES/GFCC/GFCC30TC.003, 2016.

665  Trondman, A.-K., Gaillard, M.-J., Mazier, F., Sugita, S., Fyfe, R., Nielsen, A. B., Twiddle, C., Barratt, P., Birks, H. J. B., Bjune, A. E., Björkman, L., Broström, A., Caseldine, C., David, R., Dodson, J., Dörfler, W., Fischer, E., van Geel, B., Giesecke, T., Hultberg, T., Kalnina, L., Kangur, M., van der Knaap, P., Koff, T., Kuneš, P., Lagerås, P., Latałowa, M., Lechterbeck, J., Leroyer, C., Leydet, M., Lindbladh, M., Marquer, L., Mitchell, F. J. G., Odgaard, B. V., Peglar, S. M., Persson, T., Poska, A., Rösch, M., Seppä, H., Veski, S., and Wick, L.: Pollen-based quantitative reconstructions of Holocene regional vegetation cover (plant-functional types and land-
670  cover types) in Europe suitable for climate modelling, Global Change Biology, 21, 676–697, https://doi.org/10.1111/gcb.12737, _eprint: https://onlinelibrary.wiley.com/doi/pdf/10.1111/gcb.12737, 2015.

Trondman, A.-K., Gaillard, M.-J., Sugita, S., Björkman, L., Greisman, A., Hultberg, T., Lagerås, P., Lindbladh, M., and Mazier, F.: Are pollen records from small sites appropriate for REVEALS model-based quantitative reconstructions of past regional vegetation? An empirical test in southern Sweden, Vegetation History and Archaeobotany, 25, 131–151, https://doi.org/10.1007/s00334-015-0536-9, 2016.

675  Viau, A. E., Ladd, M., and Gajewski, K.: The climate of North America during the past 2000 years reconstructed from pollen data, Global and Planetary Change, 84-85, 75–83, https://doi.org/10.1016/j.gloplacha.2011.09.010, 2012.

Vincens, A., Lézine, A.-M., Buchet, G., Lewden, D., and Le Thomas, A.: African pollen database inventory of tree and shrub pollen types, Review of Palaeobotany and Palynology, 145, 135–141, https://doi.org/10.1016/j.revpalbo.2006.09.004, 2007.

Wan, Q., Zhang, Y., Huang, K., Sun, Q., Zhang, X., Gaillard, M.-J., Xu, Q., Li, F., and Zheng, Z.: Evaluating quantitative pollen
680  representation of vegetation in the tropics: A case study on the Hainan Island, tropical China, Ecological Indicators, 114, 106 297, https://doi.org/10.1016/j.ecolind.2020.106297, 2020.

Wan, Q., Huang, K., Chen, C., Tang, Y., Zhang, X., Zhang, Z., and Zheng, Z.: Relative Pollen Productivity Estimates for Major Plant Taxa in Middle Subtropical China, Land, 12, 1337, https://doi.org/10.3390/land12071337, number: 7 Publisher: Multidisciplinary Digital Publishing Institute, 2023.

685  Wang, Y., Xu, Q., Zhang, S., Sun, Y., Li, Y., Hao, J., Huang, R., Shi, J., Wang, N., Wang, T., Li, Y., Zhang, R., Zhang, X., and Zhou, Z.: Relative pollen productivity estimates and landcover reconstruction of desert steppe in arid Western China: An example in Barkol Basin, Quaternary Sciences, 41, 1738–1748, https://doi.org/10.11928/j.issn.1001-7410.2021.06.19, publisher: , 2021.

Webb, T., Howe, S. E., Bradshaw, R. H. W., and Heide, K. M.: Estimating plant abundances from pollen percentages: The use of regression analysis, Review of Palaeobotany and Palynology, 34, 269–300, https://doi.org/10.1016/0034-6667(81)90046-4, 1981.

690  Whitmore, J., Gajewski, K., Sawada, M., Williams, J. W., Shuman, B., Bartlein, P. J., Minckley, T., Viau, A. E., Webb, T., Shafer, S., Anderson, P., and Brubaker, L.: Modern pollen data from North America and Greenland for multi-scale paleoenvironmental applications, Quaternary Science Reviews, 24, 1828–1848, https://doi.org/10.1016/j.quascirev.2005.03.005, 2005.

Wieczorek, M. and Herzschuh, U.: Compilation of relative pollen productivity (RPP) estimates and taxonomically harmonised RPP datasets for single continents and Northern Hemisphere extratropics, Earth System Science Data, 12, 3515–3528, https://doi.org/10.5194/essd-12-
695  3515-2020, publisher: Copernicus GmbH, 2020.

Williams, J. W., Grimm, E. C., Blois, J. L., Charles, D. F., Davis, E. B., Goring, S. J., Graham, R. W., Smith, A. J., Anderson, M., Arroyo-Cabrales, J., Ashworth, A. C., Betancourt, J. L., Bills, B. W., Booth, R. K., Buckland, P. I., Curry, B. B., Giesecke, T., Jackson, S. T.,

Latorre, C., Nichols, J., Purdum, T., Roth, R. E., Stryker, M., and Takahara, H.: The Neotoma Paleoecology Database, a multiproxy, international, community-curated data resource, Quaternary Research, 89, 156–177, https://doi.org/10.1017/qua.2017.105, publisher: Cambridge University Press, 2018.

Woodbridge, J., Fyfe, R. M., and Roberts, N.: A comparison of remotely sensed and pollen-based approaches to mapping Europe's land cover, Journal of Biogeography, 41, 2080–2092, https://doi.org/10.1111/jbi.12353, _eprint: https://onlinelibrary.wiley.com/doi/pdf/10.1111/jbi.12353, 2014.

Zhang, N., Ge, Y., Li, Y., Li, B., Zhang, R., Zhang, Z., Fan, B., Zhang, W., and Ding, G.: Modern pollen-vegetation relationships in the Taihang Mountains: Towards the quantitative reconstruction of land-cover changes in the North China Plain, Ecological Indicators, 129, 107 928, https://doi.org/10.1016/j.ecolind.2021.107928, 2021a.

Zhang, Y., Wei, Q., Zhang, Z., Xu, Q., Gao, W., and Li, Y.: Relative pollen productivity estimates of major plant taxa and relevant source area of pollen in the warm-temperate forest landscape of northern China, Vegetation History and Archaeobotany, 30, 231–241, https://doi.org/10.1007/s00334-020-00779-x, 2021b.

---

## Author Response (AR2)

**Letter to the editor**

Dear Kirsten Elgers,

We thank you for the opportunity to revise our manuscript once more. While all three reviewers had interesting suggestions, none criticized how we applied the REVEALS model to reconstruct vegetation from the input pollen data and we were able to accommodate a majority of recommendations. Following suggestions from all three reviewers, we added a comparison of our reconstructed tree cover with tree cover reconstructed by Serge et al. (2023) and found that our tree version has a significantly lower reconstruction error than the previous reconstruction. We also restructured and expanded our discussion to explore methodological shortcomings and differences to previous reconstructions, reasons for continued tree cover overestimation, and data set application. As suggested by both Thomas Giesecke and the anonymous reviewer, we decided to move the calculation of the 80% pollen source area to the supplementary materials (S5). An additional supplement (S6) also compares different rasterization methods and shows marginal differences..
We also corrected an error in the validation and reconstruction figures, where a 2x4° rasterization was mistakenly used instead of the intended 2x2° format. Importantly, this adjustment does not impact dataset validity. We believe that these and other minor changes improved the manuscript. We answer Reviewer comments in detail in the individual responses.
The most recent version of the dataset can be found on Zenodo (https://doi.org/10.5281/zenodo.13902921) as well as code for data set production (https://doi.org/10.5281/zenodo.10191859) and dynamic rasterization (https://doi.org/10.5281/zenodo.13902976). We will update the dataset currently deposited on PANGAEA once reviewers agree to the current version (https://doi.pangaea.de/10.1594/PANGAEA.961588). It is for this reason that we currently link to Zenodo in our manuscript instead of PANGAEA, but we will change this before publishing.

Best regards
Laura Schild and Ulrike Herzschuh

**Reply to Anonymous reviewer**

**General Reply**

Dear Reviewer,

Thank you very much for your valuable comments and suggestions. We are pleased to inform you that we have addressed all of your concerns and have implemented the majority of your suggestions.

To address your main concerns:

- **14 ka cutoff**: We now provide a clear justification for this cut-off, explaining its relevance to the climatic stability starting from this period.
- **80% PSA calculations**: As per your suggestion, we have moved these calculations to the supplementary materials.
- **Comparison with Serge et al.**: We have expanded the manuscript to include a direct comparison of our dataset with the previously published European REVEALS reconstruction by Serge et al.

Additionally, we have improved the structure of the discussion and expanded upon the validity of the dataset, ensuring a clearer presentation of our findings. We also corrected an error in the validation and reconstruction figures, where a 2x4° rasterization was mistakenly used instead of the intended 2x2° format. Importantly, this adjustment does not change the main messages of the paper. Once again, thank you for your thoughtful input. The dataset (https://doi.org/10.5281/zenodo.13902921), code (https://doi.org/10.5281/zenodo.10191859), and rasterization script (https://doi.org/10.5281/zenodo.13902976) can be found on Zenodo. We are confident that your suggestions have enhanced the quality and clarity of the manuscript and thank you for your review.

Best regards,
Laura Schild and Ulrike Herzschuh

**In-depth replies**

**Major Comments**

1. No justification is given for the 14 ka cutoff in the time series; please elaborate. Is this a result of historic climate values, the length of the sedimentary records used or both? 14 ka is ending up in the deglacial period, which is a bit of a push considering the abrupt climatic change at the onset of the Holocene. The prior LegacyPollen dataset paper (Herzschuh et al., 2022) states this as such, calling it a deglacial period/transition. Thus, at 14 ka, pollen productivity may not be

accounted for with Holocene interglacial PPEs. The authors should consider stopping at the Holocene boundary.

The climatic conditions during the past 14 ka BP were relatively stable which is why we consider it as a cutoff for the REVEALS reconstruction (Shakun et al., 2012). Additionally, the main peak of vegetation turnover had already happened during the late glacial, so that we can assume that forested landscapes, similar to today's landscapes, had already been established (Li et al., 2024; Mottl et al., 2021). We have added an explanation in the manuscript.

Li, C., Dallmeyer, A., Ni, J., Chevalier, M., Willeit, M., Andreev, A. A., Cao, X., Schild, L., Heim, B., and Herzschuh, U.: Global biome changes over the last 21,000 years inferred from model-data comparisons, EGUsphere, 1–26, https://doi.org/10.5194/egusphere-2024-1862, 2024.

Mottl, O., Flantua, S. G. A., Bhatta, K. P., Felde, V. A., Giesecke, T., Goring, S., Grimm, E. C., Haberle, S., Hooghiemstra, H., Ivory, S., Kuneš, P., Wolters, S., Seddon, A. W. R., and Williams, J. W.: Global acceleration in rates of vegetation change over the past 18,000 years, Science, 372, 860–864, https://doi.org/10.1126/science.abg1685, 2021.

Shakun, J. D., Clark, P. U., He, F., Marcott, S. A., Mix, A. C., Liu, Z., Otto-Bliesner, B., Schmittner, A., and Bard, E.: Global warming preceded by increasing carbon dioxide concentrations during the last deglaciation, Nature, 484, 49–54, https://doi.org/10.1038/nature10915, 2012.

90    taxonomic harmonization for consistency of records. Reconstruction chronologies may, therefore, differ slightly from previous reconstructions due to this revised age modeling. Spatial data coverage of records in the reconstruction is dense in Europe (1275 1287 records) and North America (1016 records 1040) and sparsest in Asia (441 446) (see Fig. 1). The records' sample density decreases with age (see Fig. 2). Only samples dated to 14 ka BP or younger were used to ensure that the climatic conditions of recorded vegetation were similar to the modern climate.

2.  Source Area: The selection of an 80% source area is justified by the following: "The primary objective of this calculation is to provide a clear understanding of the scale of the source area for users unfamiliar with pollen data. It highlights the regional nature of lacustrine pollen data and demonstrates the influence of lake size on this source area", which is a valid comment. However, no justification is given for using 80% pollen source areas; why not 70%, 90% or 95% etc? Is the 80% source area just a reflection of the value set for comparing the GPM to the LSM in Theuerkauf et al. (2016) based on Prentice and Webbs' (1986) somewhat arbitrary statement that "…significant amounts of pollen can be derived from far beyond the source area for, say, 80% of the pollen grains" found at a site.
    Also, it would be worthwhile to mention that when using the GPM source, areas have been found to vary greatly between taxa based on their individual fall speeds (Theuerkauf et al., 2016).
    The upper end of the 80% source area presented here is 762 km, which seems rather large.
    If the purpose of the 80% source area is to show that pollen source area increases with basin radius (Fig 5), I think the words from this section would be better spent elsewhere. The inclusion of an 80% source area here is confusing, and the manuscript may improve with this information being supplementary information instead.
    More importantly, no explanation of the dispersal model used to calculate the pollen source area is given, assumed to be the Gaussian plume with unstable conditions? The choice of dispersal model would substantially affect the results of this 'source area' calculation, hence this should be noted in the discussion. As paper is currently lacking in novelty, a good improvement would be to test dispersal models (e.g. changing settings in the GPM, but also trying the Lagrangian Stochastic model) in the validation for these regions. These is somewhat a low-hanging fruit, as the R code for REVEALSinR allows to change dispersal model, but the PPEs would need to be recalculated with the same settings to match the various attempts.
    -   Following your suggestion we decided to move the calculation and description of 80% pollen sources areas to the supplementary materials (S5, please find it with the new

manuscript upload). We still want to touch upon your points raised above here.

The threshold of 80% is indeed an arbitrary choice. However, we find this reasonable for this illustrative measure. We account for the fact that different area have different source areas by choosing the area where the median of pollen input for all taxa is 80%. This means that lakes with the same areas but different compositions could have different 80% source areas. However, we found basin area to have the bigger impact.

The dispersal model chosen plays an important role in the application of REVEALS and therefore the calculation of 80% pollen source area. This is why we state it in Table 2. Finally, we do not believe that the testing of different dispersal models and modes is within the scope of this data description paper, which ESSD states should "[...] describe original research data, databases, or combined datasets derived from them". [..] Although examples of data outcomes may prove necessary to demonstrate data quality, extensive interpretations of data – i.e. detailed analysis as an author might report in a research article – remain outside the scope of this data journal."
(https://www.earth-system-science-data.net/about/manuscript_types.html).

3. No comparison to prior European scale reconstructions is presented, e.g. (Serge et al., 2023), which would be very helpful to position where the methodology differs/falls in comparison to prior reconstructions.

   ○ We added a comparison with Serge et al. to our manuscript. Differences mainly exist in the choice of RPP values, arboreal taxa, and the aggregation procedure. Especially the latter is now described in detail in our methods section.
   We compare our reconstructed tree cover to Serge et al.'s and find that our tree cover tends to be lower. A comparison of validations shows a significant lower MAE with our dataset compared to Serge et al.'s dataset. Please see the relevant new manuscript sections pasted below.

Methodological rasterization differences:

This method of temporal and spatial averaging differs from several previous REVEALS applications. Pollen counts are often summed in temporal bins prior to running REVEALS to increase pollen counts and reduce uncertainty (Trondman et al., 2015; Githumbi et . However, temporally averaging after the REVEALs application, as implemented by us, increases the flexibility of the dataset

170  with the trade-off of potentially increased uncertainty. Rasterization has previously been perfomred by using a weighted average taking into account the basin size of the original record (Trondman et al., 2015; Githumbi et al., 2022; Serge et al., 2023). However, the most recent REVEALS-based North American vegetation reconstruction uses the same arithmetic mean as described above (Dawson et al., 2024b). When comparing our method of temporal and spatial aggregation to that used by previous European reconstructions (e.g. Serge et al., 2023), we also found no significant differences in the validation of reconstructed

175  tree cover (see S6).

Comparison with Serge et al. (methods):

 Additionally, we compare our REVEALS reconstruction to the most recently published REVEALS reconstruction in Europe by Serge et al. (2023, version: RPPs.st1). We average our reconstruction in the same grid and temporal bins as used by Serge et al. to compare the reconstructed tree cover between both reconstructions. To get the total tree cover,

200 we sum evergreen and summergreen tree cover values in Serge et al.'s dataset, while excluding broadleaved summergreen temperate warm shrubs (BSTWS) and broadleaved evergreen xeric shrubs (BEXS). We validate the previous reconstruction and our reconstruction in the most recent time slice available in Serge et al.'s reconstruction (-65 to 100 BP, https://doi.org/10.48579/PRO/J5( with the remote sensing forest cover and compare validations. Unfortunately, direct validation could only be performed with the most recent time slice available online, rather than the historical time slice used in the validation by Serge et al., which

205 limits the ability to reproduce their validation results exactly. We do not apply any openness correction here as we do not have comparable
* * *
  pollen source areas

210  available for the records used in Serge et al. (2023). The reconstruction by Serge et al. differs in the temporal as well as spatial aggregation routine, as described above. Definition of arboreal taxa varies, a different RPP-value set was used, and the amount of total records included is higher than in our reconstruction (Serge et al.: 1607, LegacyVegetation: 1287).

**Comparison with Serge et al. (results):**

The comparison between our reconstruction and tree cover reconstructed in Serge et al. (2023) shows that LegacyVegetation

280 (this publication) tends to have a lower tree cover independent of sample age. Serge et al. tend to overestimate forest cover even more than LegacyVegetation which leads to a much lower mean absolute error in LegacyVegetation compared to Serge et al. (Fig. 10). The MAE for LegacyVegetation is slightly higher than presented in Fig. 7 due to the difference in spatial resolution and the lack of openness correction.

[Figure]

**Figure 10.** (a) Comparison between LegacyVegetation (this publication) and the tree cover from Serge et al. (2023) and (b) validations with modern, remote-sensing forest cover for both data sets.

4. Section 3.2 is slightly arbitrary/worded strangely; trends in pollen and REVEALS are broadly the same, and the composition of the vegetation is what changes with PPEs adjusting the pollen values. Cyperaceae is a strange choice to include for the bogs and swamps as it would be locally deposited around the coring location. Whilst the Peatland setting in REVEALSinR would assume that the Cyperaceae originates from outside of the immediate area surrounding the core, many of such pollen grains would be local.

   - The section is supposed to be rather illustrative, showing the reader the compositional nature of the dataset produced and what differences between the original pollen record and the REVEALS estimate could look like. We illustrate continental mean compositions and highlight only the 8 most common taxa. So while peatlands are included, they only contribute to this mean together with small and large lakes. We highlight that the reconstruction is difficult to validate on a taxon-level and that we believe it is associated with higher uncertainty in the discussion. We also edited this Section 3.2 slightly to highlight its illustrative character (see tracked changes below).

 REVEALS was used to reconstruct quantitative vegetation cover.

225 Here we  illustrate a comparison between these reconstructed compositions to the original pollen composition. Differences in composition between  pollen data and REVEALS are apparent for all continents of the Northern Hemisphere. Some clear examples include: increases of Cyperaceae in all continents, decreases of  *Betula* in Europe, decreases of  *Pinus* in all continents, and increases of  *Acer* in North America with the application of REVEALS and its intended correction of  taxon-specific biases (see Fig. 4).

230

5. Figure 11 reveals that outside of Europe, continental vegetation reconstructions are not yet robust, likely due to the density of sedimentary records in North America and Asia or the implementation of continental averaged PPEs or another factor not explored here but nonetheless are better than unmodelled pollen reconstructions.

- This is correct. We highlight and discuss these limitations for North America and Asia in our edited discussion. We point out a lack of RPP studies in North America and hypothesize a higher regional variability of RPP values in Asia and North America than in Europe as potential reasons for higher reconstruction errors. Please see the tracked changes below.

 However, continental differences are evident in the quality of tree cover reconstruction, with Europe showing a significantly larger reduction in errors compared to other regions. North America and Asia exhibit larger reconstruction errors in the REVEALS estimates, though these are still lower than those derived from tree pollen percentages.

295 Notably, regions such as the Great Lakes, the Labrador Peninsula, and Pacific Northwest display particularly high errors in tree cover reconstruction. Asia, characterized by sparser coverage, presents fewer large errors. This highlights the need for improved vegetation reconstruction especially in North America and Asia. The reason for this reduced performance could lie in a lack of RPP studies, especially in North America, or in a significantly higher regional variability of RPP values compared to Europe. While differences in validation outcomes across varying spatial resolutions are marginal (see S4), some variability is observed

300 when different grids are employed, highlighting spatial heterogeneity in reconstruction success. Despite these caveats, overall trends appear consistent, with acceptable correlation coefficients, though absolute values in certain regions remain challenging to interpret with confidence as tree cover continues to be overestimated in all continents.

6. The discussion section would benefit from sub-sectioning, perhaps between (1) more methodological issues/limitations arising from the selected method and potential solutions or a forward outlook on how a northern hemisphere may become as robust as those seen prior for just continental scale European reconstructions. (2) more generalised and outlooking insights, e.g., those in relation to the validation and training of climate vegetation models, as mentioned early on in the manuscript (Abstract and Introduction). Section 2 may require more nuance and restraint when related back to some of the clearer limitations of the presented method, neatly summarised by the reconstruction error being much greater than Europe for North America and Asia in Figure 11.
    - We edited the discussion to follow a clearer structure and include a more extensive discussion of methodological limitations and data set validity.
    - Please see the new manuscript version for the edited discussion.

7. Moreover, there is a lack of discussion surrounding the role of human agency on forest cover through the Holocene, which is likely one of its biggest drivers.
    - Our aim is not really to discuss the reasons for forest cover trends, but rather the differences between Pollen and REVEALS estimates and the remote sensing forest cover used in our validation. In this context, we now highlight the anthropogenic impact on modern forest cover leading to potential mismatches between "modern" pollen-based tree cover (younger than 100 BP) and modern remote-sensing forest cover (2000 to 2015). Please see the tracked changes below for this edit.

is evident in the validations. Furthermore, pollen-based estimates are derived from records that span a much longer timescale than the modern forest cover data available, even though modern timeslices are used for validation. Increased anthropogenic impact could exacerbate discrepancies between pollen-based and

375  remote-sensing estimates. This could contribute to the overestimation of forest cover, which persists in all continents. Additionally, these modern and arguably unnatural vegetation conditions may not correspond to past vegetation and may therefore have reduced significance for the reconstruction of past, natural landscapes.

8. Case sensitivity in the axes and legends of figures should be consistent. Italics of species-genus names need to be implemented throughout.

○ We checked and implemented this.

**Other Comments by Line:**

1. L6: Why is the last 14 ka selected as the cutoff, 11.7 ka or 11 ka requiring less justification as clearer alternatives?
   - We have added an explanation for this cutoff in the manuscript (see tracked changes below).

90 taxonomic harmonization for consistency of records. Reconstruction chronologies may, therefore, differ slightly from previous reconstructions due to this revised age modeling. Spatial data coverage of records in the reconstruction is dense in Europe (1287 records) and North America (1040) and sparsest in Asia (446) (see Fig. 1). The records' sample density decreases with age (see Fig. 2). Only samples dated to 14 ka BP or younger were used to ensure that the climatic conditions of recorded vegetation were similar to the modern climate.

2. L8-9: does not read correctly, should be, e.g. "The pollen source area where 80% of the pollen originated within was calculated for large lakes (>50ha)".
   - Removed from main manuscript and added to supplementary materials (S5).
3. L40-45: called the Fagerlind Effect.
   - Added.

in the pollen record, while those with low pollen productivity and less effective dispersal are underrepresented. These factors, together with the compositional nature of pollen data, result in a non-linear relationship between pollen and vegetation, titled the Fagerlind effect (Prentice and Webb III, 1986; Fagerlind, 1952). Approaches such

4. L44: The R-value and ERV models are constituents of the LRA. The REVEALS model is not merely a "refinement" per se of the ERV. The ERV is a key part of the REVEALS model incorporated in the PPE calculation stage of the LRA.
   - We reworded this section. Please see tracked changes below.

tion, titled the Fagerlind effect (Prentice and Webb III, 1986; Fagerlind, 1952). Approaches such
50 as the R-value model (Davis, 1963; Webb et al., 1981) and the extended R-value model (Parsons and Prentice, 1981) were created to address this issue and were  later included into Sugita's (2007) model for "Regional Estimates of Vegetation Abundance from Large Sites" (REVEALS). By accounting for taxon-specific RPP and fall speed values, as well as basin-

5. L53: space needed after Githumbi et al. (2022)
   - Implemented.
6. L75: why are Australasian and Latin American pollen databases included here? The reconstruction is for the Northern Hemisphere. See the end of the next comment in relation to this.
   - These were related to the LegacyPollen dataset and were now removed to avoid confusion. (screenshot of tracked changes)

The pollen data synthesis LegacyPollen2.0 (Li et al., 2024b) includes 3680 temporally resolved records (time-series) distributed globally. Data were collected from individual publications and the Neotoma Paleoecology Database which includes data from the European Pollen Database,  and the North American Pollen database
85 (Fyfe et al., 2009b; Giesecke et al., 2014; Whitmore et al., 2005; Williams et al., 2018). An overview of Neotoma records included in LegacyPollen 2.0 and this reconstruction can be found in S1.

7. L79: why are some records "marine in origin"? Is this a typo? If it relates to the LegacyPollen 2.0 dataset creation, this should be covered in the LegacyPollen2.0 publication, not here it causes confusion.
   - These were related to the LegacyPollen dataset and were now removed to avoid confusion.
8. Fig1: consider separating this figure into four map panels, three regions and one hemispheric map as shown currently or reduce the point size slightly.
   - We reduced the point size.

[Figure]

9. Fig2: consider making the bars 3 stacked colours between the three categories of sites, large lakes, peatland and small lakes, or adding these as another facet element to the figures.
   - Implemented.

[Figure]

10. L85: REVEALS does not need to be defined again.
    - Removed definition.

The REVEALS model ("Regional Estimates of Vegetation Abundance from Large Sites") estimates quantitative vegetation coverage from pollen assemblages using site and taxon-specific parameters (Sugita, 2007). Based on wind speed and taxon-

11. L100: what version of the DISQOVER package is being used also, is REVEALSinR a function rather than a package itself?
    - We changed the package name and added the version used.

115  We use REVEALSinR from the DISQOVER package in R to implement REVEALS (Theuerkauf et al., 2016, Version 0.9.13, https://github . It mainly differs from the original program by Sugita (2007) in the process of error calculation. REVEALSinR includes repeated model runs with random error added to RPP values and pollen counts (see Table 2 for the number of variations). The resulting distribution of REVEALS results allows for an estimation of the standard deviation of vegetation cover per taxon. The program by Sugita (2007), however, derives error estimates with a hybrid method from a variance-covariance matrix of

120  PPE and Monte Carlo simulations. For further details on the REVEALS model see the original publication Sugita (2007) or

12. L104-130/Tab2: how many model runs (n) occurred for each site? It is not mentioned but is a key parameter of the REVEALSinR function defaulting to 1000; see the example given in the function information in the most recent DISQOVER package release version available here: https://github.com/MartinTheuerkauf/disqover.

      REVEALSinR(
      pollen,
      params,
      tBasin,
      dBasin,
      dwm = "lsm unstable",
      n = 1000,
      regionCutoff = 1e+05,
      ppefun = rnorm_reveals,
      pollenfun = rmultinom_reveals,
      writeresults = FALSE,
      verbose = TRUE
      n      number of model runs per time slice, by default 1000

- All static parameters used in the REVEALSinR function are listed in Table 2 (see below). The amount of pollen count and RPP variations (n) is set to 2000 in our REVEALS runs.

**Table 2.** Static model parameters and model settings for REVEALS runs using REVEALSinR (Theuerkauf et al., 2016).

| Parameter | Values and settings used in model run |
| --- | --- |
| atmospheric model | unstable atmosphere |
| dispersal model | gaussian plume |
| wind speed | $3m \times s^{-1}$ |
| maximum extent of regional vegetation ($Z_{max}$) | 1000 km |
| number of RPP and pollen count variations (n) | 2000 |
| peatland basin area (for missing sizes) | 31.41 ha |
| lake basin area (for missing sizes) | 49 ha |
| function to randomize pollen counts | rmultinom_reveals |

13. L125: To avoid confusion with the usage of (n), as mentioned previously, which is the number of model runs in the REVEALSinR implementation, consider the notation for the mixing value to something else.
- We changed the notation to m.

140  standard value which can be found in Table 2 , together with several constant parameters set in REVEALSinR. Lastly, we also reduced computational effort in REVEALSinR by implementing a maximum number of steps in the lake model used to model mixing in the basin. The number of steps was set to 500 unless $m$ falls below that maximum value for $m = basin\,radius/10$ for basins with a radius of at least 1000 m and $m = basin\,radius/2$ for basins with a radius smaller than 1000 m.

14. L160: why was 80% chosen to see the major comment.
- Please see our reply to your major comment on this.
15. Sec3.1: 762km is rather large as the 80% source area, especially if Zmax is set to 1000km (Tab 2 region cutoff value). Similarly, the lower 155km is still quite large, even for the GPM (Theuerkauf et al., 2016), which generally has larger pollen source areas for taxa.

- We only show the 80% pollen source areas for large lakes here. As they depend both on Zmax (rather large in our case) and the dispersal model this can be the case. Additionally, the pollen composition influences the 80% source area, by increasing it when far-dispersing taxa are dominant or decreasing it when short-dispersing taxa make up a majority of the sample.
- Following your suggestion above, we have decided to move the calculation and results of the 80% pollen source area to the supplementary materials (S5).

16. L168-9: Betula, Pinus, Acer. Italicisation and case sensitivity need checking throughout the manuscript, including figures and captions.
    - Implemented.
17. Fig 7: the key agebin (0 ka – modern) would benefit from being presented as point data rather than a time series. Four bar charts with the three series: pollen vs REVEALS vs remote sensing, might be a good idea. The time series is interesting but is not the impactful point.
    - We chose a time series here to illustrate continental and Northern Hemispheric trend, while adding the modern remote sensing forest cover to already indicate, the potential error in this reconstruction. As we go into a lot more detail with comparing modern pollen-based, REVEALS estimated and remote sensing tree and forest cover, we believe that this Figure does not have to focus on the modern age bin.
18. L184-185: implies that REVEALS is not working well in North America and Asia for continental averages. The difference should be larger and closer to the remote sensing for the modern modelled pollen values.
    - Yes, our REVEALS application is indeed not able to reduce the error more in Asia and North America, even though the error is reduced compared to tree pollen percentages. We chose to highlight and discuss this more in our revised manuscript (please see tracked changes below).

 However, continental differences are evident in the quality of tree cover reconstruction,
295   with Europe showing a significantly larger reduction in errors compared to other regions. North America and Asia exhibit larger reconstruction errors in the REVEALS estimates, though these are still lower than those derived from tree pollen percentages. Notably, regions such as the Great Lakes, the Labrador Peninsula, and the Pacific Northwest display particularly high errors in tree cover reconstruction. Asia, characterized by sparser coverage, presents fewer large errors increasing the overall continental reconstruction error. This highlights the need for improved vegetation reconstruction, especially in North
300   America and Asia. The reason for this reduced performance could lie in a lack of RPP studies, especially in North America, or in a significantly higher regional variability of RPP values compared to Europe. While differences in validation outcomes across varying spatial resolutions are marginal (see S4), some variability is observed when different grids are employed, highlighting spatial heterogeneity in reconstruction success. Despite these caveats, overall trends in tree cover appear consistent, with acceptable correlation coefficients, though absolute values in certain regions remain challenging to interpret with confidence
305   as tree cover continues to be overestimated in all continents.

19. Sec3.4: while MAE is useful, I would consider using additional measures like dissimilarity to help identify/quantify the difference between REVEALS vs remote sensing, pollen vs remote sensing, and split into the three continents (Jackson and Williams, 2004; Overpeck et al., 1985). It would be possible to estimate critical values.
    Jackson, S.T., Williams, J.W., 2004. MODERN ANALOGS IN QUATERNARY PALEOECOLOGY: Here Today, Gone Yesterday, Gone Tomorrow? Annu. Rev. Earth Planet. Sci. 32, 495–537. https://doi.org/10.1146/annurev.earth.32.101802.120435

Overpeck, J.T., Webb, T., Prentice, I.C., 1985. Quantitative Interpretation of Fossil Pollen Spectra: Dissimilarity Coefficients and the Method of Modern Analogs. Quat. res. 23, 87–108. https://doi.org/10.1016/0033-5894(85)90074-2

- While dissimilarity measures are useful to compare multivariate data, they are not applicable here as we are comparing only one variable (reconstructed tree cover, i.e. a univariate comparison). Multivariate comparisons would be great to validate REVEALS applications further. However large-scale standardized data is missing to do so. We also highlight the need for such data for better validation in the discussion (see manuscript excerpt below). In addition to the MAE, we also show the distribution of error in the boxplot in Figure 8 to give more information surrounding the error distribution.

Another challenge lies in validating the compositional reconstruction results. It remains uncertain whether RPP values have remained stable over time, and historical compositional data are not only scarce but also likely too recent to test this assumption effectively (Baker et al., 2016). Validating modern compositional reconstructions on large spatial scales is therefore difficult.  As global compositional vegeta-
375 tion data  are not readily available,  remote sensing of tree cover serves as the best option for validation.  But even with accurate tree cover reconstructions, uncertainties remain regarding the abundance of individual taxa due to the aggregated nature of the  tree cover measure. To address this issue, global syntheses of  tree and plant inventories or compositional remote sensing products could
380  provide more robust validation. Additionally, vegetational compositions derived from sedimentary ancient DNA  (sedaDNA) offer a promising avenue for comparing past vegetation data. Local quantitative sedaDNA vegetation signals could be averaged across multiple records
385  to compare with pollen-based results (Niemeyer et al., 2017; Capo et al., 2021).

20. L202: Europe shows the best results, but trees are still being overrepresented here in comparison to the remote sensed cover; what is driving the overrepresentation of trees here and beyond? This is not unpacked in great enough detail in the discussion.

- We expanded on this in the Discussion. (Please see tracked changes below.)

However, the reconstructions are associated with some of the limitations of sedimentary pollen data . This includes age uncertainty, temporal mixing, and irregular spatial and temporal resolution of records. Age uncertainty is already treated as best

355  as possible through consistent age modeling of the pollen dataset (Li et al., 2022a, 2021). Nevertheless, in general, replicating sediment and peat cores could provide more accurate estimates Validating pollen-based tree cover estimates with remote sensing-derived forest cover also presents a challenge. One key issue is the inherent errors associated with remote sensing forest cover data. While validation using other sensors is possible, only a limited subset of the available data is cross-validated with Lidar data, which itself is characterized by limited spatial coverage (Sexton et al., 2013). A critical limitation of surface

360  reflectance methods, as used in the Landsat-based forest cover, is their reliance on a 2D perspective, primarily capturing the forest canopy. This means that the understory is often not detected, resulting in an incomplete representation of the forest structure. In contrast, pollen-based estimates provide a more comprehensive, stratified view of the vegetation, as they
* * *
incorporate all contributing taxa, not just the tree canopy. Despite this broader scope, pollen data and REVEALS estimates tend to emphasize trees more than other vegetation types consistently as is evident in the validations. Furthermore, pollen-based

365  estimates are derived from records that span a much longer timescale than the modern forest cover data available, even though modern timeslices are used for validation. Increased anthropogenic impact could exacerbate discrepancies between pollen-based and remote-sensing estimates. This could contribute to the overestimation of forest cover, which persists in all continents. Additionally, these modern and arguably unnatural vegetation conditions may not correspond to past vegetation and may therefore have reduced significance for the reconstruction of past, natural landscapes.

21. Fig9: REVEALS reconstruction validations for Europe can be compared with the prior validations of (Serge et al., 2023).
    - We added this comparison in our manuscript. Please see relevant parts of the Methods, Results and Discussion sections in our reply to your general comment above.
22. Fig11: Strongly suggests that robust continental reconstruction in North America and Asia is not possible yet, even when validated just on the arboreal layer.
    - As stated in our reply to you "other comment" No. 18, we expand on this in the Discussion.
23. Sec4: see the main comments section.
    - We edited the Discussion to follow a clearer structure as suggested. Please see the tracked changes document.
24. L232: a comparison of the results of Serge et al. (2023) is warranted. Should the forest cover for Europe be better if not the same/similar?
    - We added this comparison as stated in our reply to your "other comment" no. 21. We find that our reconstruction is closer to remote sensing forest cover than the previous Serge et al. reconstruction and state some possible reasons. See more details in our reply to your general comment no 3.
25. L255-260: DNA comments do not make sense in terms of vegetation reconstructions. A reconstruction implies quantity. SedDNA/eDNA data today remains point data, which provides no information on the quantity of vegetation around a site, merely presence and absence.
    - Quantitative estimates of past vegetation from sedimentary ancient DNA are possible with meta barcoding and have become a more common method to reconstruct past

environments. We added an additional reference here (https://www.mdpi.com/2571-550X/4/1/6). (see tracked changes)

could provide more robust validation. Additionally, vegetational compositions derived from sedimentary ancient DNA  (sedaDNA) offer a promising avenue for comparing past vegetation data. Local quantitative aDNA vegetation signals could be averaged across multiple records  to compare with pollen-based results

390  (Niemeyer et al., 2017; Capo et al., 2021).

26. L269-271: These sentences about the reliability of data are more methodological considerations that belong in the methods or a separate discussion unpacking the limitations of the presented method.

- As stated in our reply to you major comment and "other comment" no. 23. We edited the Discussion to follow a clearer structure.

**Reply to Giesecke**

**General reply**

Dear Thomas Giesecke,

Thank you very much for your valuable comments and suggestions. We believe we were able to implement the majority of your recommendations.

- We validate our reconstructed modern vegetation using observed modern forest cover. Validation is a standard practice applied in many datasets, including those published in *Earth System Science Data* (ESSD).
- Regarding your question on why REVEALS still overestimates forest cover, we have expanded our discussion to highlight the differences between remote sensing-based forest cover and pollen-based forest cover. We also address potential deficiencies with RPP values in this context. We now compare our results directly with those of Serge et al., focusing on validation results, where we observe a significantly smaller error in our reconstruction.
- As suggested, we have moved the 80% source area calculation to the supplementary materials.
- Furthermore, we restructured the discussion to expand on dataset usage and challenges.
- Regarding the current dataset version, we believe that there may have been some confusion regarding the repository. We made the new version of the dataset available on Zenodo and will update the final version on PANGAEA, due to the longer processing times at PANGAEA. The dataset deposited on Zenodo does not include reconstructions from marine records or samples before 14 ka BP.

The dataset (https://doi.org/10.5281/zenodo.13902921), code (https://doi.org/10.5281/zenodo.10191859), and rasterization script (https://doi.org/10.5281/zenodo.13902976) can be found on Zenodo. Once again, thank you for your insightful feedback, which has contributed to the overall improvement of the manuscript. We are confident that the changes strengthened the clarity and depth of our work.

Best regards,
Laura Schild and Ulrike Herzschuh

**In-depth replies**

**General comments**

Reading the revised manuscript Schild et al. I am glad to see that all of the controversial analyses were dropped. However, the full initial data is still available online. The remaining manuscript has a strong focus on testing the REVEALS estimates of forest cover against remote sensing data. Like before, I feel this is a research question that should be individually addressed in a research and not in this data paper, unless the aim of the comparison is the justification of the chosen RPPE. If this topic is kept as part of this publication it needs more attention: the authors need to describe how they made sure that the samples used in the comparison represent the last 100 years; clearly state what was compared; differences of using averages of small lakes or a large lake should be evaluated; and the spatial differences nicely

displayed in Fig 11 should be discussed. Fig 11 also indicates that there are grids with an overestimate in tree cover (even more common at a higher resolution grid), which needs to be explained.

Firstly, we do believe that validations are an essential part of a data publication and would like to highlight that this is indeed a common practice in ESSD (e.g. https://essd.copernicus.org/articles/16/2917/2024/, https://essd.copernicus.org/articles/16/2465/2024/, https://essd.copernicus.org/articles/16/2449/2024/).

Secondly, we would like to address your concerns. We choose to include samples younger than 100 BP as determined by Age models used in the LegacyPollen dataset. We clearly state in our methods section which values are being compared. These are tree pollen percentages, tree REVEALS estimates (reconstructed tree cover), and remote sensing forest cover (a temporal average for the years 2000, 2005, 2010, and 2015). We do this for each grid cell that has a valid REVEALS tree cover estimate for the timeslice 100 BP - present. We define grid cells as valid, which include samples from at least one large lake or several smaller basins.
We chose to do this with the averaged gridded data since we recommend data to be used in this format. we provide additional validations with different grid cell sizes in the supplementary materials. In our revised discussion, we now mention the slight differences between validations results of differing spatial resolutions and highlight areas in which especially high errors occur (see tracked changes pasted below).

>  However, continental differences are evident in the quality of tree cover reconstruction,
> 295 with Europe showing a significantly larger reduction in errors compared to other regions. North America and Asia exhibit larger reconstruction errors in the REVEALS estimates, though these are still lower than those derived from tree pollen percentages. Notably, regions such as the Great Lakes, the Labrador Peninsula, and the Pacific Northwest display particularly high errors in tree cover reconstruction. Asia, characterized by sparser coverage, presents fewer large errors increasing the overall continental reconstruction error. This highlights the need for improved vegetation reconstruction, especially in North
> 300 America and Asia. The reason for this reduced performance could lie in a lack of RPP studies, especially in North America, or in a significantly higher regional variability of RPP values compared to Europe. While differences in validation outcomes across varying spatial resolutions are marginal (see S4), some variability is observed when different grids are employed, highlighting spatial heterogeneity in reconstruction success. Despite these caveats, overall trends in tree cover appear consistent, with acceptable correlation coefficients, though absolute values in certain regions remain challenging to interpret with confidence
> 305 as tree cover continues to be overestimated in all continents.

Also the calculation of the source area for 80% of the pollen is not well discussed in terms of its correctness, usefulness and implication. Personally, I don't think that the theoretical 80% area is useful with respect to openness comparisons where the signal may be determined by the theoretical source area of 30 to 50 % of pollen (compare Matthias & Giesecke (2014) QSR, 87, 12-23.). I recommend removing these additional analysis, presenting them elsewhere.

Following another reviewer's suggestion, we decided to move the calculation and description of the 80% pollen source area to the supplementary materials (see S5 of the uploaded new manuscript version).

Instead the authors could provide examples of past vegetation composition or compare their results in more detail to existing quantitative reconstructions. They could elaborate how this data differs from previous continental studies, stress what this specific dataset is better at, and where the limitations are (see below under Data).

In our edited manuscript, we expand on methodological differences to previous quantitative reconstructions, compare our reconstructed tree cover to the most recent European REVEALS reconstruction by Serge et al. (2023), and expand on the limitations of our dataset in the discussion. Our rasterization follows a different methodology, which features a trade-off between increased flexibility in temporal and spatial resolution with slightly increased uncertainty. The comparison with Serge et al. highlights much lower reconstruction errors in our dataset than in Serge et al.'s, while their data coverage is slightly better.

Differences in rasterization:

This method of temporal and spatial averaging differs from several previous REVEALS applications. Pollen counts are often summed in temporal bins prior to running REVEALS to increase pollen counts and reduce uncertainty (Trondman et al., 2015; Githumbi et . However, temporally averaging after the REVEALs application, as implemented by us, increases the flexibility of the dataset

170 with the trade-off of potentially increased uncertainty. Rasterization has previously been perfomred by using a weighted average taking into account the basin size of the original record (Trondman et al., 2015; Githumbi et al., 2022; Serge et al., 2023). However, the most recent REVEALS-based North American vegetation reconstruction uses the same arithmetic mean as described above (Dawson et al., 2024b). When comparing our method of temporal and spatial aggregation to that used by previous European reconstructions (e.g. Serge et al., 2023), we also found no significant differences in the validation of reconstructed

175 tree cover (see S6).

Comparing with Serge et al. (methods):

 Additionally, we compare our REVEALS reconstruction to the most recently published REVEALS reconstruction in Europe by Serge et al. (2023, version: RPPs.st1). We average our reconstruction in the same grid and temporal bins as used by Serge et al. to compare the reconstructed tree cover between both reconstructions. To get the total tree cover,

200 we sum evergreen and summergreen tree cover values in Serge et al.'s dataset, while excluding broadleaved summergreen temperate warm shrubs (BSTWS) and broadleaved evergreen xeric shrubs (BEXS). We validate the previous reconstruction and our reconstruction in the most recent time slice available in Serge et al.'s reconstruction (-65 to 100 BP, https://doi.org/10.48579/PRO/J5... with the remote sensing forest cover and compare validations. Unfortunately, direct validation could only be performed with the most recent time slice available online, rather than the historical time slice used in the validation by Serge et al., which

205 limits the ability to reproduce their validation results exactly. We do not apply any openness correction here as we do not have comparable 80

 % pollen source areas

210  available for the records used in Serge et al. (2023). The reconstruction by Serge et al. differs in the temporal as well as spatial aggregation routine, as described above. Definition of arboreal taxa varies, a different RPP-value set was used, and the amount of total records included is higher than in our reconstruction (Serge et al.: 1607, LegacyVegetation: 1287).

Comparison with Serge et al. (results):

A specific comparison with Serge et al. (2023) reveals that our reconstruction generally shows lower forest cover across
305 Europe. Unfortunately, direct validation could only be performed with the most recent time slice available online, rather than
the historical time slice used in the validation by Serge et al., which limits the ability to reproduce their validation results
exactly. Despite this limitation, our reconstruction demonstrates a lower MAE, indicating improved accuracy. This is notable
given that Serge et al. utilized a larger number of records in their study. One potential explanation for these differences could
lie in the variations in RPP values and the selection of arboreal taxa used in the reconstruction, as we employ an arboreal tree
310 treshold and include more taxa in our REVEALS reconstruction.

[Figure]

**Figure 10.** (a) Comparison between LegacyVegetation (this publication) and the tree cover from Serge et al. (2023) and (b) validations with
modern, remote-sensing forest cover for both data sets.

**Specific comments**

I am using the line numbering of the track changes document.
Title: I don't think the short title "LegacyVegetation" is informative and adding a version number already
here is confusing. What if there will not be an update?
- We removed the version number, but would like to maintain the dataset title.

**LegacyVegetation: Northern Hemisphere reconstruction of  past plant cover and  total tree cover from pollen archives of the last 14 ka**

11: Better "The source area of 80% of the deposited pollen ..."
- We moved calculations and descriptions of the 80% pollen source area to the supplementary
materials (S5).
22: Better "compiling" rather than "collecting". I thought that work was done by Wieczorek et al. 2020, so
perhaps better "updating".
- "collecting" here refers to conducting more RPP studies in North America to have a larger amount
of taxa and/or regional variability represented. It does not refer to the compilation of RPP values
in a synthesis.
70: This is not really related to the previous sentence, which outlines the need for reconstructing forest
cover. Validation is only possible for the modern situation which is often different from the natural that we

are aiming to reconstruct: e.g. trees are harvested when they are still young reducing the overall pollen production, alien taxa (Pseudotsuga) make up large proportions of the forest, N-fertilization may lead to higher pollen production ....

- We extended our discussion on remote sensing data characteristics that could influence validation.
  We also improved wording in this text passage in the introduction. (two screenshots)

With its importance for the assessment of biome stability, carbon storage, climatic feedbacks, and land-use-change,  cover is an often reconstructed variable (e.g. Fyfe et al., 2015; Githumbi et al., 2022; Serge et al., 2023). Due to the global availability of remote sensing data on contemporary

70  tree cover, reconstructions of tree cover in modern time slices may even be validated (Hjelle et al., 2015; Roberts et al., 2018). Yet, only Serge et al. (2023) and Pirzamanbein et al. (2014) use this opportunity for extensive validation and even improvement of reconstructions from European pollen records. No grid-cell based validations exist for the Northern Hemisphere.

Validating pollen-based tree cover estimates with remote sensing-derived forest cover also presents a challenge. One key issue is the inherent errors associated with remote sensing forest cover

365  data. While validation using other sensors is possible, only a limited subset of the available data is cross-validated with Lidar data, which itself is characterized by limited spatial coverage (Sexton et al., 2013). A critical limitation of surface reflectance methods, as used in the Landsat-based forest cover, is their reliance on a 2D perspective, primarily capturing the forest canopy. This means that the understory is often not detected, resulting in an incomplete representation of the forest structure. In contrast, pollen-based estimates provide a more

370 comprehensive, stratified view of the vegetation, as they incorporate all contributing taxa, not just the tree canopy. Despite this broader scope, pollen data and REVEALS estimates tend to emphasize trees more than other vegetation types consistently as is evident in the validations. Furthermore, pollen-based estimates are derived from records that span a much longer timescale than the modern forest cover data available, even though modern timeslices are used for validation. Increased anthropogenic impact could exacerbate discrepancies between pollen-based and

375 remote-sensing estimates. This could contribute to the overestimation of forest cover, which persists in all continents. Additionally, these modern and arguably unnatural vegetation conditions may not correspond to past vegetation and may therefore have reduced significance for the reconstruction of past, natural landscapes.

153: This cut off remains a weak point of the analysis, should be motivated and highlighted in what the data can be used for.

- We implemented this threshold (of 60°N, we apologize the typo in the manuscript) to reflect the present distribution of *Betula* plant functional types, as illustrated by GBIF occurrences in the two plots below. For this example we define both *Betula nana* and *Betula glandulosa* as shrub PFTs. We added an emphasis on the limitations of this static threshold in the manuscript's discussion (see tracked changes below). We also added the present Betula PFT distribution as illustrated below to the supplementary materials (S2).

tend to have higher certainty compared to taxon-specific reconstructions, as they are based on aggregation across taxa. However, the static latitudinal arboreal threshold for Betulaceae, *Betula*, and *Alnus* poses a limitation in our reconstruction. This could

350 be improved by incorporating a dynamic, climate-dependent threshold in future work.

- GBIF references:
    - GBIF.org (01 October 2024) GBIF Occurrence Download  https://doi.org/10.15468/dl.2pw3qw
    - GBIF.org (01 October 2024) GBIF Occurrence Download  https://doi.org/10.15468/dl.vgchrb
    - GBIF.org (7 October 2024) GBIF Occurrence Download https://doi.org/10.15468/dl.7fdwhs

- GBIF.org (7 October 2024) GBIF Occurrence Download https://doi.org/10.15468/dl.achmvv
- GBIF.org (1 October 2024) GBIF Occurrence Download https://doi.org/10.15468/dl.xyv4ge

161: How was the age of the samples determined?
- The ages of the samples were determined using standardized age modeling. The exact source of the age model is documented for each record in the LegacyPollen 2.0 dataset (please see Section 2.1, https://essd.copernicus.org/articles/14/3213/2022/essd-14-3213-2022-discussion.html, and https://doi.pangaea.de/10.1594/PANGAEA.965907)

reconstruction only lake and peat records in the Northern Hemisphere were used ($n = $ 2752) Analogous to the preceding
LegacyPollen 1.0 dataset (Herzschuh et al., 2022), the data synthesis involved revising and standardizing age modeling and
90     taxonomic harmonization for consistency of records. Reconstruction chronologies may, therefore, differ slightly from previous
reconstructions due to this revised age modeling. Spatial data coverage of records in the reconstruction is dense in Europe

Figure caption to Fig 4: Smaller than what? Also the results of this analysis should be discussed more using the original paper (Sugita 1993). Are the results making sense for the reconstruction of tree cover. Would it not be more useful to look at the distance that 80% of the herbs come from? If the results are

making sense, what are the consequences? Can result represent several grid cells? How do the result compare in cases were the area for one site is a subset of the area for another?

- Following another reviewer's suggestion we decided to move the calculation and description of the 80% pollen source are to the supplementary materials. The results depend both on the choice of the dispersal model and the maximum vegetation extent (Zmax). Both basin size and its composition will impact the calculated 80% pollen source radius. And have to be assumed. The results have, therefore, an illustrative character, aiming to highlight the regional nature of pollen-based vegetation data to potential data users. For rasterization we follow previous reconstructions in that we average compositions from records located within a grid cell not source areas located within a grid cell. Norms on Zmax, dispersal models and percentage of source area considered would need to be established first, before using pollen source areas of any kind for spatial averaging rather than record location.

Fig 6: The new title of the manuscript indicates reconstructions for the last 14 ka but here 20 ka are presented. X-axis label is missing.

- See corrected figure below.

[Figure]

-

260: "lower" than what?

- Than tree pollen percentages.

coast and in the boreal biome. Rather open areas exist at the Tibetan Plateau and at very high latitudes. The  tree cover derived from the REVEALS reconstruction is generally lower than tree pollen percentages. However, the difference between

265: Here you use "relevant". Is that the 80% absolute? If yes, I think this may be too large for some lakes. Or are you using the grid squares as suggested in the rest of the sentence. Please explain.

- Grid cell remote sensing forest cover values were used. Please see corrected sentence below.

Remote sensing forest cover within  grid cells was used to validate the modern, reconstructed  tree cover from the original pollen data and the REVEALS  estimates for each grid cell. Here we present validation of

307: This is a well-documented fact for areas with a dominance of wind pollinated trees - note not true in the subtropics and tropics.

- We remove this paragraph in favor of a more detailed discussion of remote sensing and pollen data differences in the discussion.

355: Global?
- We corrected this.

433: Did you use data from all these constituent databases?
- The LegacyPollen dataset uses data from all these constituent databases. We do not. We have removed the one we do not use. (screenshot)

*Acknowledgements.* We thank Thomas Böhmer for support with dataset curation and harmonization. The project was supported by the

445 Bundesministerium für Bildung, Wissenschaft, Forschung und Technologie through the German Climate Modeling Initiative PALMOD (grant no. 01LP1510C to UH), the European Union (ERC, GlacialLegacy grant no. 772852 to UH), and the China Scholarship Council (grant no. 201908130165 to CL). Data were partly obtained from the Neotoma Paleoecology Database (http://www.neotomadb.org) and its constituent databases (European Pollen Database,  and the North American Pollen database). The work of data contributors, data stewards, and the Neotoma community is gratefully

450 acknowledged.

Supplement Fig 4: I see large differences between the different grid resolutions. Particularly interesting are the overestimations in the finest grid in Central Europe.
- We would like to disagree concerning large differences between the different spatial resolutions. While we do see a bit of variability concerning the median and mean error values (see Figure below), we cannot discern a clear trend towards better or worse reconstructions with a lower or higher spatial resolution or any large differences. A possible trend could be an increase in variability in REVEALS errors in Europe with higher spatial resolution, but this does not seem to be the case in the other continents, nor the Northern Hemisphere as a whole. We do however add a sentence in our discussion, highlighting this small variability between different spatial resolutions (see tracked changes below).

[Figure]

**Data**

Reading the manuscript I got the impression that the data is presented in a gridded format with binned time steps, while the data on PANGAEA is the sample based REVEALS estimate.

At second reading I discovered that a script for rasterization and binning is available from Zenodo. The script works well but it is using the data from Zenodo where duplicate files are available. As criticized before the data still contains marine cores e.g. MD84-629 which is wrongly labeled as peatland. REVEALS estimates are still provided for sites beyond 14,000 years ago as now indicated by the title in response to earlier criticism.

- It seems you had a look at the deprecated and not the reviewed dataset. Due to the length of data editor processing time, the data on PANGAEA has not been updated yet. Instead, we uploaded the revised dataset to Zenodo to make it available to reviewers before being finalized on PANGAEA. We outlined this interim solution in our reply to your last review and updated the dataset links in the manuscript. The dataset deposited on Zenodo does not include any records other than lakes and peat and does not extend to samples older than 14 ka BP.

Looking at the data for Europe I see supposed links to the data in the EPD/Neotoma: e.g. Event = Handle (in Neotoma), Site_ID and Dataset_ID identical to Neotoma. It is nice to see them but I did not see any documentation, stating that indeed these columns are links to the data in Neotoma. Moreover, the Event

seems inconsistent e.g. "AGE_neotoma" versus AGE in Neotoma, while other EPD/Neotoma datasets don't have the suffix.

- The input dataset (LegacyPollen2.0) used for this REVEALS reconstruction includes additional columns indicating Neotoma_DOIs for records that were originally obtained from Neotoma. A list of Neotoma records used in the reconstruction is additionally given in S1.
  The event inconsistencies are due to PANGAEA restriction in event uniqueness. Each event name can only be awarded once. The Event "AGE" already existed on PANGAEA, but evidently for a different location. This is why PANGAEA requires suffixes to be added to unique Event-Names. Again the original Neotoma event names can be acquired from the input dataset (LegacyPollen2.0, https://doi.pangaea.de/10.1594/PANGAEA.965907).

The basin diameter for peat bogs seems to have been set to 100 (Table 2) without further explanation. However the 100 m are also given as basin size in the site metadata for all peatlands, which is not correct! Estimated lake diameters and derived data are given with 12 digits after the comma suggesting a precision that is not there.

- The peatland basin size was only set to 100 m for peatlands where any other basin size is missing (n = 488 for all continents). We correct this inaccuracy in table 2. In the course of the revisions we had added basin areas from previous reconstructions where possible. We have now also added this updated metadata to the zenodo upload. We agree that the amount decimals given is unsuitable and have now updated to two decimals in the Zenodo upload.

**Table 2.** Static model parameters and model settings for REVEALS runs using REVEALSinR (Theuerkauf et al., 2016).

| Parameter | Values and settings used in model run |
| --- | --- |
| atmospheric model | unstable atmosphere |
| dispersal model | gaussian plume |
| wind speed | $3m \times s^{-1}$ |
| maximum extent of regional vegetation ($Z_{max}$) | 1000 km |
| number of RPP  and pollen count variations (n) | 2000 |
| peatland basin  area (for missing sizes) | 31.41 ha |
| lake basin area (for missing sizes) | 49 ha |
| function to randomize pollen counts | rmultinom_reveals |

The data include for each taxon the 10_percentile, 90_percentile, mean, and median, as well as the sd of cover in %, while the manuscript does not mention these calculations and how they may be used. It needs to be better communicated what kind of data is presented and how the authors anticipate it to be used.

- We added a paragraph in the manuscript to describe the data structure and when and how it should be rasterized. Please see tracked changes below.

215   The published dataset includes vegetation reconstructions for individual records in Asia, Europe, and North America up until 14 ka BP. The reconstructed coverage values include mean, median, standard deviation, and 10% and 90% quantile values for each taxon. Mean values and standard deviations are given for tree cover. For each sample its validity as a site is given. Only reconstructions from large lakes are valid independently. To include all other records a spatial and temporal average is necessary (rasterization, https://doi.org/10.5281/zenodo.12800291).

I went on checking the REVEALS estimates for Grosser Treppelsee in Brandenburg, for which I counted the pollen. While the overall forest cover reconstruction through time looks reasonable the taxon specific

estimates are unlikely for several taxa. The region is dominated by Pinus forest while the estimate of Pinus cover in the surface sample is 8.5% mean cover, which is much too low. REVEALS estimates for Brassicaceae cover 100 to 200 years ago is with around 20% way too high. Thus the data may be useful for continental scale forest cover reconstructions while regional studies would benefit from regionally estimated PPEs. The caveats of using continental scale RPPEs and particularly of setting RPPEs to 1 for some taxa need to be discussed in the publication.

[Figure]

- We show the pollen percentages and the REVEALS reconstruction for 10 common taxa at Großer Treppelsee above. The cover of *Pinus* is especially corrected to be lower due to its high pollen productivity. It should also be remembered that the pollen record from a lake as large as Großer Treppelsee (~59 ha) will have a regional signal rather than a local one, including the mosaic of open and closed vegetation in Brandenburg and likely even Poland and not just the rather closed forest surrounding the actual lake. The modern forest cover (landsat) in a circle with a 100 km radius surrounding Großer Treppelsee is 28.194% and the reconstructed modern value (with the openness correction accounting for urban areas) is ~ 29%. We believe that this shows that tree cover is reconstructed well in this area. Nevertheless, the accuracy of the REVEALS reconstruction depends on RPP values, the availability and quality of which varies for each taxon. We already show this in Figure 3 and our discussion. We did however expand on the variation between RPP value sources and the validity of reconstructions on a larger spatial scale and the aggregated tree cover scale as opposed the compostion in the discussion. (see tracked changes)

The ~~REVEALS forest cover reconstructions presented here offer valuable insight into past vegetation changes. The global dataset provides an opportunity to explore past vegetation dynamics, gaining a deeper understanding of responses, trajectories, and potential feedback mechanisms. Given the increasing discussions surrounding the possibility of tipping events in vegetation cover (Armstrong McKay et al., 2022; Lenton and Williams, 2013), this could be of considerable use. While a reconstruction~~

335  reliability of reconstructions also varies among different taxa due to the quality of RPP values, which is documented in detail in a supplementary file outlining the sources of RPP values (see Section "Code and Data Availability"). Reconstructions based on taxa with continental RPP values are the most reliable, followed by those based on hemispheric data, with standardized RPP values being the least reliable. This hierarchy should be considered when interpreting results. The use of continental RPP values could also make our reconstruction more reliable

340 at larger spatial scales as opposed to local reconstructions. Additionally, uncertainties in RPP values themselves can affect reconstruction success and could be leading to the persistent overrepresentation of tree taxa despite the application of RE-VEALS

345  Tree cover reconstructions tend to have higher certainty compared to taxon-specific reconstructions, as they are based on aggregation across taxa. However, the static latitudinal arboreal threshold for Betulaceae, *Betula*, and *Alnus* poses a limitation in our reconstruction. This could

350 be improved by incorporating a dynamic, climate-dependent threshold in future work.

**Reply to Gaillard**

**General reply**

Dear Marie-Jose Gaillard,

Thank you very much for your thorough review of our manuscript. We are pleased to inform you that we were able to implement most of your suggestions and have addressed your concerns accordingly.

Regarding your larger comments:

- **Aggregation differences from previous continental-scale reconstructions:** We now describe these differences more clearly in the text and provide supplementary materials demonstrating that the method of aggregation does not significantly impact the absolute reconstructed tree cover values.
- **REVEALSinR error:** We have clarified the error calculation in REVEALSinR and now explain in more detail how coverage distributions are calculated across repeated model runs while accounting for total pollen counts and RPP variability.
- **Move 80% PSA to supplementary materials:** This suggestion has been implemented, and the relevant section has been moved to the supplementary materials.

In addition, we have addressed several smaller phrasing issues throughout the manuscript. We also corrected an error in the validation and reconstruction figures, where a 2x4° rasterization was mistakenly used instead of the intended 2x2° format. Importantly, this adjustment does not impact dataset validity.

Please see our detailed responses to each of your comments below. We also replied to your in-line comments in the previous manuscript version and append this document as well. The current dataset version (https://doi.org/10.5281/zenodo.13902921), code (https://doi.org/10.5281/zenodo.10191859), and rasterization script (https://doi.org/10.5281/zenodo.13902976) can be found on Zenodo.

Once again, we would like to express our thanks for your input, which we believe has improved the clarity of the manuscript and emphasized its usability compared to other datasets. We are confident that we have addressed all of your suggestions.

Best regards,
Laura Schild and Ulrike Herzschuh

**Specific replies**

**General comments**

The authors have made substantial revisions that were necessary such as deleting the southern hemisphere from the reconstruction and producing REVEALS estimates based on pollen records from several sites within areas (grid cells) of various sizes and for time windows of various lengths. This leads to more acceptable results. I appreciate the hard work made to finalize this revision, but there are still

misunderstandings that needs to be clarified in the paper.

1. One of my major concerns is the calculation of REVEALS mean estimates based on the REVEALS reconstructions for several sites within grid cells and several pollen counts within time windows, i.e. the step that the authors call "aggregation" in space and time. For the "aggregation" in space the authors calculate the mean of the individual site REVEALS estimates without any weighting by the K coefficient that is dependent of basin size (the larger the basin, the heavier the weighting should be for each taxon, and vice versa). Such a weighting is implemented in Sugita's REVEALS computer program but not in REVEALSinR. In Sugita's method, the REVEALS estimates from individual sites within a grid cell are weighted with the taxon-specific "pollen dispersal-deposition coefficient K" of all pollen taxa involved, se e.g. Li et al. (2017). This should be clarified under METHODS.

   For the "aggregation" in time the authors similarly calculate the mean of the individual counted level REVEALS estimates. The reliability of REVEALS estimates depends, among other things, on the size of the pollen count. In this context, the usual size of pollen counts (often around 1000, seldom more, quite often around 500 and sometimes less) is a low pollen count. This implies that all REVEALS estimates in the Schield et al. REVEALS dataset are of relatively low reliability and calculating the mean of these REVEALS estimates does not make them more reliable. All earlier continental Holocene REVEALS reconstructions have worked with time windows of such a length that it would maximize the size of the counts without using too long time windows (generally maximum 500 years). The compromise to make depends on the aim of the study. One has then to sum pollen counts within each time window and use this new pollen count for the REVEALS application to obtain the REVEALS estimates for the time window (see e.g. Githumbi et al., 2022). This procedure is very different from calculating mean REVEALS estimates and is statistically the correct  way to do. I understand that it would be a huge work to redo the work in this way for this manuscript. But this should be listed as one of the many differences between this REVEALS dataset and earlier ones. I do not know whether the error on REVEALS estimates as calculated by REVEALSinR (see my point below) is sensitive to the size of pollen counts. I guess not, but I can't find anything about this issue in the REVEALSinR original paper or elsewhere. In that case, this is also an aspect that makes REVEALS applications using REVEALSinR weaker if the size of pollen counts is not considered in the error estimate on REVEALS results.

Thank you for highlighting these differences. We added an explanation of these differences to previous reconstructions in our manuscript (see tracked changes below). We highlight the trade-off of reconstruction robustness with application flexibility with the changed temporal binning. While our spatial aggregation differs from previous European reconstructions, it is actually the same arithmetic mean used by Dawson et al. (2024, see lines 306-309,398 in https://github.com/andydawson/reveals-na/blob/master/r/reveals.r).

165 and Ord (1994), using the same equation as Wieczorek and Herzschuh (2020). We provide a script for rasterization with adjustable temporal and spatial resolution for users of the dataset on Zenodo (https://zenodo.org/doi/10.5281/zenodo.12800290). This method of temporal and spatial averaging differs from several previous REVEALS applications. Pollen counts are often summed in temporal bins prior to running REVEALS to increase pollen counts and reduce uncertainty (Trondman et al., 2015; Githumbi et . However, temporally averaging after the REVEALs application, as implemented by us, increases the flexibility of the dataset

170 with the trade-off of potentially increased uncertainty. Rasterization has previously been perfomred by using a weighted average taking into account the basin size of the original record (Trondman et al., 2015; Githumbi et al., 2022; Serge et al., 2023). However, the most recent REVEALS-based North American vegetation reconstruction uses the same arithmetic mean as described above (Dawson et al., 2024b). When comparing our method of temporal and spatial aggregation to that used by previous European reconstructions (e.g. Serge et al., 2023), we also found no significant differences in the validation of reconstructed

175 tree cover (see S6).

For validation, the reconstructed  tree cover of the past 100 years was rasterized and compared to modern remote

We did implement a temporal binning of pollen counts prior to REVEALS and did a weighted mean (based on basin size) to compare the tree cover reconstruction of the modern time slice between our method of rasterization and previous European reconstructions' method (titled "Gaillard suggestion" in the Fig. R1 below). We find that absolute values differ minimally and that the reconstruction error is virtually the same. We, therefore, feel confident in stating that our method of rasterization does not impact reconstruction success significantly negatively.
We explain the error calculation in REVEALSinR in detail in our reply to the next general comment.

[Figure]

Fig. R1: Overview of differences between modern tree cover timeslices of LegacyVegetation rasterization and methods used in previous European REVEALS reconstructions ("Gaillard suggestion"). (A) Gridcell (2x2°) differences in tree cover between the two versions. (B) Validations with landsat forest cover for both versions.

2. Another major difference between implementation of the REVEALS model with the computer programs of Sugita and REVEALSinR of Theuerkauf et al. (2016) is the calculation of the uncertainties (errors) on the REVEALS estimates. The REVEALS standard error accounts for the

standard errors (or deviations) of the relative pollen productivities for the individual pollen taxa and on the number of pollen counted; i.e. the size of the pollen count matters. The error calculated in REVEALSinR does not consider the RPP errors. I do not mean that the errors from the REVEALSinR program are wrong, but it is a pity not to use the errors on RPPs as this parameter is very influential on the final REVEALS estimate of plant cover. This difference between the two applications should at least been mentioned.

REVEALSinR runs REVEALS on each sample multiple times (n = 2000 in our study) while altering pollen counts as well as RPP each time. Errors are added (or subtracted) on the pollen counts. The size of the error depends on the total pollen counts with smaller total counts resulting in larger errors being added (hence the larger standard deviations of the randomized pollen counts with total pollen counts are low). This is visualized in Fig. R2 below. Not summing pollen counts in time slices before running REVEALS does therefore lead to higher standard deviations, but does not significantly effect absolute values as highlighted in our reply to your general comment 1.

During repeated REVEALS runs in REVEALSinR, RPP are also generated randomly from a normal distribution ($\mu$ = RPP value, $\sigma$ = RPP SD). The 2000 REVEALS results are then used to calculate statistics such as the mean and median REVEALS estimate, as well as quantiles and standard deviations. REVEALSinR therefore accounts both for total pollen count and RPP SDs in the calculation of REVEALS estimates. We include these statistics as REVEALS outputs in our description of the dataset in the manuscript (see tracked changes below).

Description of differences between Sugita's program and REVEALSinR:

115   We use REVEALSinR from the DISQOVER package in R to implement REVEALS (Theuerkauf et al., 2016, Version 0.9.13, https://github . It mainly differs from the original program by Sugita (2007) in the process of error calculation. REVEALSinR includes repeated model runs with random error added to RPP values and pollen counts (see Table 2 for the number of variations). The resulting distribution of REVEALS results allows for an estimation of the standard deviation of vegetation cover per taxon. The program by Sugita (2007), however, derives error estimates with a hybrid method from a variance-covariance matrix of

120   PPE and Monte Carlo simulations. For further details on the REVEALS model see the original publication Sugita (2007)  and for previous REVEALS applications on continental scales see e.g Li et al. (2017), Githumbi et al. (2022) , Serge et al. (2023), and Dawson et al. (2024a).

Description of values included in dataset:

**3.1 Dataset description**

The published dataset includes vegetation reconstructions for individual records in Asia, Europe, and North America up until
215   14 ka BP. The reconstructed coverage values include mean, median, standard deviation, and 10% and 90% quantile values for each taxon. Mean values and standard deviations are given for tree cover. For each sample its validity as a site is given. Only reconstructions from large lakes are valid independently. To include all other records a spatial and temporal average is necessary (rasterization, https://doi.org/10.5281/zenodo.12800291).

[Figure]

Fig. R2: Distribution of random pollen count standard deviations with changing total pollen counts in REVEALSinR.

3. 80% pollen source area: this information should be presented as an alternative to estimate the size of the region that is represented by REVEALS estimates of plant cover. Sugita (2007a) who developed the REVEALS model assumes that Zmax is the size of the region represented by REVEALS estimates (see also Li et al., 2017). Zmax can only be assumed (you assumed it to be 1000 km over the entire study region) and the region from which most of the pollen are coming (in your case 80%) can be estimated. See also Hellman et al., 2008b (in VHA) who assumed Zmax to be 400 km (distance from the pollen site) in S Sweden and the 90% source area (200 km) was considered to be the area from which most of the pollen came. One should therefore state that the assumed value for Zmax influences the estimate of x% pollen source area. Please, also specify what dispersal model you use, the Gaussian Plume Model or the Lagrangian Stochastic Model, for estimating your 80% pollen source area, which makes also a difference (see Theuerkauf et al., 2016).

Following the suggestion by two other reviewers, we decided to move the calculation and results of the 80% pollen source area into the supplementary materials (see S5). We add two additional sentences in the manuscript highlighting the assumption of a maximum spatial extent of regional vegetation (Zmax) and include a reference to Hellmann et al. 2008 (see tracked changes below). All parameters used in REVEALSinR are listed in Table 2, which is also referred to in the text (see below as well).
Zmax highlight:

The REVEALS model follows a set of assumptions. Firstly, neither directionality nor pollen transport through agents other than wind are considered in the model. The maximum spatial extent for this pollen transport ($Z_{max}$, see Table 2) has to be set

105 to define the region in which most of the pollen originates. This value will always be an assumption and has only been tested empirically by Hellman et al. (2008b). Additionally, it is assumed that the basin is circular with no source of pollen within the basin radius. The peatland and bog sites used in our reconstructions inherently violate this assumption. Nevertheless, the

Parameter table:

**Table 2.** Static model parameters and model settings for REVEALS runs using REVEALSinR (Theuerkauf et al., 2016).

| Parameter | Values and settings used in model run |
| --- | --- |
| atmospheric model | unstable atmosphere |
| dispersal model | gaussian plume |
| wind speed | $3m \times s^{-1}$ |
| maximum extent of regional vegetation ($Z_{max}$) | 1000 km |
| number of RPP  and pollen count variations (n) | 2000 |
| peatland basin  area (for missing sizes) | 31.41 ha |
| lake basin area (for missing sizes) | 49 ha |
| function to randomize pollen counts | rmultinom_reveals |

**Two additional comments, minor but still important:**

4. Avoid the term reconstruction for pollen percentages or raw pollen data. These are simply data, pollen% are not a reconstruction of vegetation, they are proxy data of vegetation, while a traditional narrative interpreting the pollen percentages using various kind of information is a reconstruction, as REVEALS-based estimates of plant cover is a reconstruction of past plant cover. I advise you to revise this throughout the manuscript, text and Figures. I made comments in the manuscript about that, but not everywhere. Using "reconstruction" for pollen data is misleading, and makes the text difficult to understand in some places.

We removed "reconstructions" in the context of pollen and changed it to your suggested wording of tree pollen percentages. Please see an example text passage below.

reconstructions compared to the original pollen data. Both  pollen percentages and REVEALS estimates still overestimate  tree cover.

5. I would use the terms "(total) tree pollen" and "(total) tree cover" instead of "forest cover" when it refers to pollen % and REVEALS-based estimates of tree cover. It is important to be clear in terms of what you are comparing the satellite vegetation (forest cover) with. If you choose to follow my advice, revise the manuscript consequently. I made comments in the manuscript about that, but not everywhere.

We changed it to "tree pollen" and "tree cover" at the applicable locations. Please see an example text passage below.

**LegacyVegetation: Northern Hemisphere reconstruction of  past plant cover and  total tree cover from pollen archives of the last 14 ka**

**In conclusion**

I miss a description of your new REVEALS dataset for the N Hemisphere in comparison to the earlier continental REVEALS dataset for Europe, China and N America. What is different and what are the improvements.

1. In terms of what is different in the methodology, please see my major comments above, and specific comments in the revised manuscript. Do not forget that you use different chronologies than those used in earlier reconstructions. They might not be so different, but we do not know. The best solution is to describe all the differences in methodology already in the METHODS section, in the part describing REVEALSinR and in the part describing how you "aggregate" site-specific and level (time)-specific REVEALS estimates to mean REVEALS estimates (level-specific meaning using single analysed levels/samples to run REVEALS.

We now note in the manuscript that chronoligies may differ to previous reconstructions (see tracked changes below).

> reconstruction only lake and peat records in the Northern Hemisphere were used ($n =$ 2752) Analogous to the preceding LegacyPollen 1.0 dataset (Herzschuh et al., 2022), the data synthesis involved revising and standardizing age modeling and
> 90  taxonomic harmonization for consistency of records. Reconstruction chronologies may, therefore, differ slightly from previous reconstructions due to this revised age modeling. Spatial data coverage of records in the reconstruction is dense in Europe

As described above we added a description of how temporal and spatial averaging/aggregation differ between our and previous reconstructions. Please see the tracked changes again below.

> 165  and Ord (1994), using the same equation as Wieczorek and Herzschuh (2020). We provide a script for rasterization with adjustable temporal and spatial resolution for users of the dataset on Zenodo (https://zenodo.org/doi/10.5281/zenodo.12800290). This method of temporal and spatial averaging differs from several previous REVEALS applications. Pollen counts are often summed in temporal bins prior to running REVEALS to increase pollen counts and reduce uncertainty (Trondman et al., 2015; Githumbi et . However, temporally averaging after the REVEALs application, as implemented by us, increases the flexibility of the dataset
> 170  with the trade-off of potentially increased uncertainty. Rasterization has previously been perfomred by using a weighted average taking into account the basin size of the original record (Trondman et al., 2015; Githumbi et al., 2022; Serge et al., 2023). However, the most recent REVEALS-based North American vegetation reconstruction uses the same arithmetic mean as described above (Dawson et al., 2024b). When comparing our method of temporal and spatial aggregation to that used by previous European reconstructions (e.g. Serge et al., 2023), we also found no significant differences in the validation of reconstructed
> 175  tree cover (see S6).
>    For validation, the reconstructed  tree cover of the past 100 years was rasterized and compared to modern remote

2. In my view, the improvements in your REVEALS dataset are:
- You have included in your synthesis the pollen records from the northern hemisphere between Europe and China, those sites that were included in Cao et al (2019) REVEALS reconstruction, and applied REVEALS on them in accordance with the methodology you use for the rest of the Northern Hemisphere.
- Further, it would be informative to know how many pollen records you use overall and in specific continents (Europe, China, N America) for which earlier REVEALS reconstructions exist.

The number of records used for each continent were already present in Section 2.1. We correct small errors here and added the amount of records used in Serge et al. in Section 2.3 (see tracked changes below).

> reconstructions due to this revised age modeling. Spatial data coverage of records in the reconstruction is dense in Europe (1287 records) and North America (1040) and sparsest in Asia (446) (see Fig. 1). The records' sample density decreases with age (see Fig. 2). Only samples dated to 14 ka BP or younger were used to ensure that the climatic conditions of recorded vegetation were similar to the modern climate.

For Europe, compare with Serge et al. (2023).

We added a comparison of modern forest cover between our and Serge et al.'s reconstruction. Please see the methods and the results of this comparison in the manuscript's tracked changes below. The tree cover in Serge et al. tends to be higher than our reconstructed tree cover, leading to higher reconstruction errors.

Serge comparison (method):

 Additionally, we compare our REVEALS reconstruction to the most recently published REVEALS reconstruction in Europe by Serge et al. (2023, version: RPPs.st1). We average our reconstruction in the same grid and temporal bins as used by Serge et al. to compare the reconstructed tree cover between both reconstructions. To get the total tree cover,

200 we sum evergreen and summergreen tree cover values in Serge et al.'s dataset, while excluding broadleaved summergreen temperate warm shrubs (BSTWS) and broadleaved evergreen xeric shrubs (BEXS). We validate the previous reconstruction and our reconstruction in the most recent time slice available in Serge et al.'s reconstruction (-65 to 100 BP, https://doi.org/10.48579/PRO/J5( with the remote sensing forest cover and compare validations. Unfortunately, direct validation could only be performed with the most recent time slice available online, rather than the historical time slice used in the validation by Serge et al., which

205 limits the ability to reproduce their validation results exactly. We do not apply any openness correction here as we do not have comparable 80

% pollen source areas

210  available for the records used in Serge et al. (2023). The reconstruction by Serge et al. differs in the temporal as well as spatial aggregation routine, as described above. Definition of arboreal taxa varies, a different RPP-value set was used, and the amount of total records included is higher than in our reconstruction (Serge et al.: 1607, LegacyVegetation: 1287).

Serge comparison (results):

A specific comparison with Serge et al. (2023) reveals that our reconstruction generally shows lower forest cover across

305 Europe. Unfortunately, direct validation could only be performed with the most recent time slice available online, rather than the historical time slice used in the validation by Serge et al., which limits the ability to reproduce their validation results exactly. Despite this limitation, our reconstruction demonstrates a lower MAE, indicating improved accuracy. This is notable given that Serge et al. utilized a larger number of records in their study. One potential explanation for these differences could lie in the variations in RPP values and the selection of arboreal taxa used in the reconstruction, as we employ an arboreal tree

310 treshold and include more taxa in our REVEALS reconstruction.

[Figure]

**Figure 10.** (a) Comparison between LegacyVegetation (this publication) and the tree cover from Serge et al. (2023) and (b) validations with modern, remote-sensing forest cover for both data sets.

In terms of RPP, you should also mention if you use more RPP values than in earlier studies and also clarify that your RPP synthesis is made in a different way (different rules) than those by Githumbi et al. (2022) for Europe and Li et al. (2018) for China. For China, the improvement is that you have added new recent RPP values from recent papers.

The same synthesis rules used by Githumbi et al. (2022) were used in our synthesis. We synthesized RPP values on different taxonomic levels to account for the harmonized pollen dataset used in this reconstruction. This is why more values are available. Please see the expanded explanations in the tracked changes below.

2021; Zhang et al., 2021a, b; Wan et al., 2020, 2023; Jiang et al., 2020). The methods  for study selection and calculation of synthesis values  follow Wieczorek and Herzschuh (2020) as well as Githumbi et al. (2022). We expanded the synthesis calculation of RPP to different taxonomic levels (genus, family, and order) to account for the taxonomic harmonization in the pollen dataset. An overview of original values and synthesized values can be found in Appendix A and B respectively. The amount of RPP values in Asia (59) and Europe (69) is higher than in previous RPP synthesis due to the inclusion of multiple taxonomic levels (Li et al., 2018; Githumbi et al., 2022).

130

Finally, your new REVEALS dataset should be presented as an alternative dataset that is more flexible that the earlier continental ones as it allows users to amalgamate the REVEALS estimates in space choosing various sizes of grid cells, and in time choosing various length of time windows. It should be stated, however, that mean REVEALS estimates over space do not weight the K coefficient according to lake/bog size, and that mean REVEALS estimates over time are not as reliable as REVEALS estimates based on the total pollen count in a time window (see my comment above). With flexibility you loose reliability. This should be clarified for the users.

In addition to the description of differences in the methods section (see our reply to your general comment 1), we point out the trade-off once more in the discussion. Please see the tracked changes below.

Although our reconstruction method is more flexible than previous efforts, the temporal and spatial aggregation used may reduce its reliability, due to smaller total pollen counts used in REVEALS runs and the use of an arithmetic as opposed to a weighted spatial mean.

330

**LegacyVegetation 1.0:  Northern Hemisphere reconstruction of vegetation composition and forest cover from pollen archives of the last  14 ka**

Laura Schild[1,2], Peter Ewald[1,2], Chenzhi Li[1,2], Raphaël Hébert[1], Thomas Laepple[1,3], and Ulrike Herzschuh[1,2,4]

[1]Helmholtz Centre for Polar and Marine Research, Research Unit Potsdam, Alfred Wegener Institute (AWI), Germany
[2]Institute of Environmental Sciences and Geography, University of Potsdam, Karl-Liebknecht-Straße 24-25, Potsdam, Germany
[3]MARUM-Center for Marine Environmental Sciences and Faculty of Geosciences, University of Bremen, Germany
[4]Institute of Biochemistry and Biology, University of Potsdam, Karl-Liebknecht-Straße 24-25, Potsdam, Germany

**Correspondence:** Ulrike Herzschuh (ulrike.herzschuh@awi.de)

**Abstract.** With rapid anthropogenic climate change future vegetation trajectories are uncertain. Climate-vegetation models can be useful for predictions but need extensive data on past vegetation for validation and improving systemic understanding. Even though pollen data provide a great source of this information, the data is compositionally biased due to differences in taxon-specific relative pollen productivity (RPP) and dispersal.

5  Here we present a Northern Hemisphere reconstruction of quantitative regional vegetation cover from a  sedimentary pollen data set for the last  14 ka using the REVEALS model to correct for taxon- and basin-specific biases.  For the reconstruction, we  expanded on a previously published synthesis of continental RPP values.

10  The data sets include taxonomic compositions as well as reconstructed forest cover for each original pollen sample.  sources areas were  calculated for large lakes and are included in the data set . Additional metadata includes modeled ages, age model sources, basin locations, types and sizes.

The improvements in forest cover reconstructions with the REVEALS reconstruction using  continental RPP values range from 24% (North America) to 72% (Europe) relative to the mean absolute error (MAE)  of the pollen-based reconstruction.  The dataset can be used as a grid with binned and aggregated samples (adjustable script provided on Zenodo; https://zenodo.org/doi/10.5281/zenodo.12800290) or as individual timeseries if the record's basin size exceeds 50 ha.

This improved quantitative reconstruction of vegetation cover is  beneficial for the investigation of past vegetation dynamics and modern model validation. By collecting more RPP estimates  especially in

North America and adding more records to existing pollen data syntheses, reconstructions may be improved even further.  The REVEALS reconstruction is freely available on PANGAEA (see Data availability section).

**1 Introduction**

[revised manuscript text omitted]

Here we present  reconstructed quantitative vegetation cover for the Northern Hemisphere from the LegacyPollen2.0 dataset - an updated global taxonomically and temporally standardized fossil pollen dataset of  3680 palynological records - using REVEALS spanning  the last 14k years. The data sets were created using existing estimates of taxon-specific parameters  . The REVEALS reconstruction includes corrected vegetation compositions as well as reconstructed forest cover.

**2 Methods**

**2.1 Pollen Data Set**

The pollen data synthesis LegacyPollen2.0 (Li et al., 2024b) includes  3680 temporally resolved records (time-series) distributed globally. Data were collected from individual publications and the Neotoma Paleoecology Database which includes data from the European Pollen Database, the QUAVIDA data base for Australasia, the Latin American Pollen Database, the

African Pollen Database and the North American Pollen database (Flantua et al., 2015; Fyfe et al., 2009b; Giesecke et al., 2014; Lézine et a

90 . An overview of Neotoma records included in LegacyPollen 2.0 and this reconstruction can be found in S1.

Sediment and peat cores used for the creation of pollen data are of lacustrine, peat and marine origin. For the REVEALS reconstruction only lake and peat records in the Northern Hemisphere were used ($n = 2732$) Analogous to the preceding Lega-cyPollen 1.0 dataset (Herzschuh et al., 2022), the data synthesis involved revising age modeling and taxonomic harmonization for consistency of records. Spatial data coverage of records in the reconstruction is  dense in

95 Europe (1275 records) and  North America (1016 records) and sparsest in Asia ( 441) (see Fig. 1). The records ' sample density decreases with age (see Fig. 2).

[Figure]

**Figure 1.** Pollen record locations in the LegacyVegetation dataset. Colors indicate record type (large lake $\geq$ 50 ha). Record density is  highest in Europe and Eastern North America, and lowest in  Northern and  Central Asia.

**2.2 Implementing REVEALS**

The REVEALS model ("Regional Estimates of Vegetation Abundance from Large Sites") estimates quantitative vegetation

100 coverage from pollen assemblages using site and taxon-specific parameters (Sugita, 2007). Based on wind speed and taxon-specific fall speed, pollen dispersal is modeled in ring sources around the basin and deposition over the basin is integrated to give pollen influx. Together with RPP this dispersal factor is used to correct original pollen counts to better represent  actual vegetation (see Equation 1 and Table 1). By running the model with variations of relative pollen productivity (RPP) values, a statistical distribution of results is calculated.

[Figure]

**Figure 2.** Temporal coverage of records in the LegacyVegetation dataset per continent. Bins are  500 years wide. Sample count decreases with age  and Europe has the most samples overall.

$$\hat{V}_i = \frac{n_{i,k}/\hat{\alpha}_i \int_R^{Z_{max}} g_i(z)dz}{\sum_{j=1}^m (n_{j,k}/\hat{\alpha}_j \int_R^{Z_{max}} g_i(z)dz)} \tag{1}$$

The REVEALS model follows a set of assumptions. Firstly, neither directionality nor pollen transport through agents other than

**Table 1.** Algebraic terms in the REVEALS equation (see Equation 1)

| Function term |  definition |
|---|---|
| $\hat{V}_i$ | vegetation estimate of taxon i |
| $n_{i,k}$ | pollen counts of taxon i at site k |
| $\alpha_i$ | relative pollen productivity of taxon i |
| $R$ | basin radius |
| $Z_{max}$ | maximum extent of regional vegetation |
| $z$ | distance from a point in the center of a basin |
| $g_i$ | dispersal and deposition function for taxon i |

wind are considered in the model. Additionally, it is assumed that the basin is circular with no source of pollen within the basin radius. The peatland and bog sites used in our reconstructions inherently violate this assumption. Nevertheless, the quantitative reconstruction of vegetation cover from peatland cores is possible by using Prentice's deposition model (Prentice, 1985, 1988) instead of Sugita's deposition model (Sugita, 1993) in the dispersal and deposition function (see Eq. 1; Sugita, 2007). Previous studies show that results from small bogs are still reliable when aggregated, while results from large bogs tend to deviate from

those of large lakes (Trondman et al., 2015; Mazier et al., 2012; Trondman et al., 2016). Using peatland records for reconstructions is, therefore, appropriate when spatially averaging multiple sites. We use the implementation of REVEALS from the R package REVEALSinR (Theuerkauf et al., 2016).

**2.2.1**

 For further details on the REVEALS model see the original publication Sugita (2007) or Githumbi et al. (2022).

120

**2.2.1 Parameters and Model Settings**

For each taxon, values for RPP (with uncertainties provided as standard deviation) and fall speeds are used.  We made use of the synthesis of Northern Hemisphere RPP and fall speed values by Wieczorek and Herzschuh (2020). Several RPP studies published since this synthesis were added to the
125 compilation (Geng et al., 2022; Li et al., 2022b; Wang et al., 2021; Huang et al., 2021; Zhang et al., 2021a, b; Wan et al., 2020, 2023; Jian . The methods by Wieczorek and Herzschuh (2020) were followed fore study selection and calculation of synthesis values. An overview of original values and synthesized values can be found in Appendix A and B respectively.
When available, we use continent-specific values in our reconstruction. For taxa with no continental values present, we use  Northern Hemispheric values. If no values exist for a taxon, RPP is set to a constant (RPP = 1, $\sigma$=0.25)
130 and fall speeds are filled with mean continental fall speeds. Continental RPP values are available for the majority of pollen counts in all three continents (see Fig. 3). The fraction of pollen counts for which  standard RPP values were assumed is highest in North America but still < 10%. For each site, the REVEALS model also requires information on basin type, basin size and original pollen counts, all of which were collected in the LegacyPollen
135 2.0 dataset (Li et al., 2024b). Apart from taxon- and basin-specific parameters the REVEALS model requires several constant parameters to be set, which can be found in Table 2.

**2.2.2 Modifications in REVEALSinR**

We calculate the radius of  the 80% pollen source area by finding the radius in which the median influx of all taxa is 80% of the total influx (as defined by the total influx in the maximum extent of regional vegetation chosen). This is calculated
140 by employing the lake deposition model in REVEALSinR (Theuerkauf et al., 2016). Starting from $z_{max}$ the deposited pollen is calculated per taxon. This is assumed to be the total pollen each taxon deposits. In a step-wise process the radius around the basin is increased and the deposited pollen relative to the total influx at $z_{max}$ is calculated for each taxon. We define our 80%

**Table 2.** Static model parameters and model settings for REVEALS runs using REVEALSinR (Theuerkauf et al., 2016).

| Parameter | Values and settings used in model run |
|---|---|
| atmospheric model | unstable atmosphere |
| dispersal model | gaussian plume |
| wind speed | $3m \times s^{-1}$ |
| maximum extent of regional vegetation (region cutoff) | 1000 km |
| number of RPP variations | 2000 |
| peatland basin radius | 100 m |
| function to randomize pollen counts | rmultinom_reveals |

[Figure]

**Figure 3.** Percentage Regional source of RPP values for percentage of pollen counts per continent for which RPP estimates are available. A higher majority of pollen counts is covered by continental RPP values with the highest fraction in Europe. Only a small percentage of pollen counts has only hemispheric RPP information in the Northern Hemisphere compared values available. No available RPP values lead to the continents use of the Southern Hemisphere a standardized RPP value of 1±0.25.

pollen source radius as the radius where the median of the relative influx of all taxa reaches 80%. The primary objective of this calculation is to provide a clear understanding of the scale of the source area for users unfamiliar with pollen data. It highlights the regional nature of lacustrine pollen data and demonstrates the influence of lake size on this source area.

We also reduced computational effort in REVEALSinR by implementing a maximum number of steps in the lake model used to model mixing in the basin. The number of steps was set to 500 unless $n$ falls below that maximum value for $n = basin\,radius/10$ for basins with a radius of at least 1000 m and $n = basin\,radius/2$ for basins with a radius smaller than 1000 m.

**2.3 Reconstruction of forest cover and validation**

Forest cover was reconstructed by summing up percentages of arboreal taxa (see S1 S2: List of arboreal taxa) with Betulaceae, *Betula*, and *Alnus* being classified as arboreal at sites below 70° N. The mean reconstructed compositional coverages from the REVEALS results were used for the forest cover reconstructions. REVEALS results were then rasterized to aggregate and

155 include records from smaller basins as well. Reconstructed time series were averaged in 500 year bins and then rasterized in grids of differing spatial resolution. A grid cell was classified as having a valid reconstruction when it contained records from at least one large lake (>= 50 ha) or at least two small basins following Serge et al. (2023). Standard deviations of the REVEALS estimates were aggregated by applying the delta method by Stuart and Ord (1994), using the same equation as Wieczorek and Herzschuh (2020). We provide a script for rasterization with adjustable temporal and spatial resolution for users

160 of the dataset on Zenodo (https://zenodo.org/doi/10.5281/zenodo.12800290). For validation, the reconstructed forest cover of the past 500 years was 100 years was rasterized and compared to modern remote sensing forest cover. Only valid grid cells as defined above were used for validation. Average tree canopy cover within pollen source areas of all sites for all grid cells was extracted from the Landsat Global Forest Cover Change (GFCC) data set from the temporal average of the years 2000, 2005, 2010 and 2015 (Sexton et al., 2013; Townshend, 2016). An openness correction was applied to sites containing urban areas

165 and paved surfaces within the 80% pollen source areas (PSA) to correct for areas without any pollen sources and thus improve ensure comparability to modern remote sensing forest cover (see Equations 2-4). For this, the percentage of unvegetated land cover classes for the year 2015 in the ESA CCI land cover data set was used (ESA, 2017, see Table 3). Areas covered by water or ice are already considered as missing values in the remote sensing forest cover data set and do not need to be corrected for. Forest cover was validated site-wise for each grid cell and mean absolute error (MAE) and correlation coefficients calculated

170 for each continent. No openness correction was applied to the reconstruction values in the final dataset. Validation for a 2x2° grid is included in the results section. Further validations using 1°, 5°, and 10° resolution are included in the supplementary material (S3: Validation results for different spatial resolutions).

**Table 3.** Unvegetated land cover classes in ESA CCI LC chosen for the openness correction.

| Name | Code |
| --- | --- |
| Urban areas | 190 |
| Bare areas | 200 |
| Consolidated bare areas | 201 |
| Unconsolidated bare areas | 202 |

$$unvegetated\ classes = \{190, 200, 201, 202\} \tag{2}$$

$$unvegetated\ (\%) = \frac{\sum cells\ in\ PSA \in open\ classes}{\sum cells\ in\ PSA} \frac{\sum cells\ in\ PSA \in unvegetated\ classes}{\sum cells\ in\ PSA} \tag{3}$$

$$corrected\ tree\ cover = reconstructed\ tree\ cover \times (1 - unvegetated) \tag{4}$$

**2.4**

~~In addition to the REVEALS approach, which is motivated by a biophysical model but also based on a large number of model choices and parameters, we also apply a statistical approach. Here, RPP values for common taxa are estimated by minimizing the misfit of reconstructed and remote sensing forest cover. For the optimization we rely on the "L-BFGS-B" method (Byrd et al., 1995), which allows for box constraints, and minimize the residual sum of squares (RSS) of reconstructed forest cover with remote sensing forest cover. RPP values were bound by upper and lower limits based on original RPP values (see Equation ??). Fall speeds and standard deviations of RPP were kept constant to the REVEALS approach.~~

$original\ RPP \times 0.25 < new\ RPP < original\ RPP \times 4$

~~The RPP values were optimized for the ten most common taxa in the REVEALS reconstruction for all sites on a continent, forest cover reconstructed, and the residual sum of squares (RSS) with remote sensing forest cover calculated. The results were validated using a spatial leave-one-out (SLOO) cross-validation. In this cross-validation one site and all sites within a predefined radius (exclusion buffer) were excluded from the optimization to account for spatial autocorrelation. The optimized RPP values were then applied to the forest cover reconstruction of the site left-out and the absolute error with remote sensing forest cover recorded. This was repeated with 20 sites to estimate the spread of MAE. The exclusion buffer around the validation site was set to 200 km. Due to computational limitations (roughly 3 hours for one continental SLOO fold using 20 threads with 1.2 GHz CPU each), the number of sites used per continental optimization during the cross-validation was limited to 100, leading to a rather conservative estimate of the true error.~~

**3 Data summary**

**3.1 80% Pollen Source Areas**

Using REVEALS, radii of  80% pollen source areas were calculated for  large lakes (see Fig. 4). The  radii indicate in which area 80% of the deposited pollen originated from (see Section 2.2.2) and yield an understanding of which area the pollen record is representative of., which is especially useful when individual time series from large lakes are being used for analyses. The 80% pollen source areas are roughly a function of basin size (see Fig. 5) and range between  155 km and 762 km. The median 80% pollen source radius is  225 km including all  large lakes.

[Figure]

**Figure 4.** Map indicating the  relevant pollen source areas for large lakes. Many small basins in Europe lead to smaller 80% pollen source areas. Several large basins and correspondingly large 80% pollen source areas exist in Asia. In general the 80% pollen source areas highlight the regional nature of the pollen record signal.

[Figure]

**Figure 5.**  Scatterplot of basin diameter and 80% pollen source area of  large lakes in the REVEALS data set.  In general, larger basins have larger pollen source areas with the relationship between  diameter and 80% pollen source radius being roughly logarithmic.

**3.2**

205 ~~The calculated pollen source areas (see section 3.1) were used to extract modern remote sensing forest cover per site. Within the optimization, RPP values were adjusted for the ten most common taxa per continent to improve the fit between reconstructed and remotely sensed modern forest cover. The RPP values are one of the main correction factors applied in REVEALS. Here we compare original and optimized RPP values for the relevant continental taxa.~~

The magnitude of adjustment from original to optimized RPP values differs between continents (see Fig. **??**). The highest
210  and lowest absolute change respectively occurred for *Quercus* (4.08) and Fabaceae (0.09) in Africa, for *Picea* (87.81) and
*Ephedra* (0.43) in Asia, for *Pinus* (32.58) and Asteraceae (0.16) in Europe, for *Alnus* (1.79) and Amaranthaceae (in which we
included Chenopodiaceae, 0.02) in Australia and Oceania, for Amaranthaceae (63.81) and *Tsuga* (0.43) in North America, and
for Amaranthaceae (15.91) and Melastomataceae (0.74) in South America (see Appendix B). Relative change of RPP values
is mostly positive with many taxa reaching an increase of three times the original RPP value. This is the maximum RPP value
215  that can be reached, as the upper constraint for RPP optimization was set as 4 times the original RPP value (see Section 2.4). In
most cases RPP values for arboreal taxa are increased. This increase represents reconstructed forest cover being regulated down
as can be seen in the validations (see Fig. 9). Dumbbell graph illustrating original and optimized RPP values per continent and
taxon. Arboreal taxa such as Pinus, Picea, Quercus have increases that are especially large.

**3.2 Reconstructed compositions**

[Figure]

**Figure 6.** Average continental taxonomic coverages per reconstruction for the 8 most common taxa per continent.
Differences are  especially evident for Pinus, Artemisia, and Betula, which all have decreased coverages after the
 application of REVEALS, as well as Poaceae and Cyperaceae with increased coverages.

220  REVEALS was used to reconstruct quantitative vegetation cover.  Here we compared these reconstructed compositions  to the original pollen composition.

Differences in composition  between Pollen data and REVEALS are apparent for all continents of the Northern Hemisphere.

225 ~~optimized RPP values both increase *Larix* cover in Asia, Ericales cover in Europe, and decrease *Picea* cover in North America, although the version with optimized RPP values does so more strongly (see Fig. 6). The original and the optimized version also diverge in the adjustment of some taxa. *Artemisia* cover in Asia is reduced by the original version and increased by the optimized one. *Picea* cover stays roughly the same with original RPP values in North America and decreases with optimized ones and while Asteraceae cover in Europeis increased in the REVEALS version with original RPP values, it is considerably~~

230

~~In the Southern Hemisphere the differences between reconstructions are much less pronounced (see Fig. 6). The REVEALS reconstruction with original RPP values is almost indistinguishable from the original pollen spectra and adjustments in the optimized version are also much smaller than in the Northern Hemisphere. An increase in Cyperaceae cover in Australia and Oceania, decreases of Asteraceae and Cyperaceae in South America, and~~ Some clear examples include: increases of Cyperaceae

235 in all continents, decreases of

 Betula in Europe, decreases of Pinus in all continents, and increases of Acer in North America with the application of REVEALS and its intended correction of taxon-sepcific biases (see Fig.

240  6).

**3.3 Reconstructed forest cover**

Using the compositional data available from the original pollen data  and the RE-

245 VEALS run, we reconstructed forest cover for all sites and samples and rasterized the result with different spatial resolutions. The temporal trend in Northern Hemisphere forest cover is the same for  both reconstructions. Forest cover increases from  14 ka BP until roughly 6 ka BP and decreases again towards the present (see Fig. 7). REVEALS reconstructed forest cover is generally lower than forest cover from original pollen compositions. On average forest cover values from the REVEALS run are roughly 14.54%

250 lower than values from original pollen compositions. The temporal trends in Asia and North America are positive, whereas forest cover in Europe has its maximum around 6 ka BP and has been decreasing since.

Forest cover is  generally highest in Eastern North America. This

[Figure]

**Figure 7.**  Northern Hemisphere and continental average forest cover from 2x2° grid cell means for raw pollen data  and the REVEALS reconstruction (Northern Hemisphere and continental averages from different grid cell resolutions are available in S2: Reconstruction results for different spatial resolutions). Remotely sensed global average forest  clover for the grid cells with valid pollen  coverage is indicated with the diamond. Temporal trends are the same, but absolute forest cover is reduced in the REVEALS reconstructions compared to the original pollen data.  Both reconstructions still overestimate forest cover.

is also where data coverage is best in North America (see Fig. 8).

255  Density of valid grid cells is very high in Europe, where forest cover increases until roughly 6 ka BP and then decreases. Data coverage in Asia is sparse, but valid grid cells indicate higher forest cover on the Southeastern coast and in the boreal biome. Rather open areas exist at the Tibetan Plateau and at very high latitudes. The forest cover derived from the  REVEALS reconstruc-

260 tion is generally lower. However, the difference between Pollen and REVEALS forest cover is smaller in North America than in Europe and Asia.

[Figure]

**Figure 8.** Reconstructed forest cover in 2x2° grid cells from raw pollen data  and the REVEALS reconstruction for 5 example time slices (reconstructions with different grid cell sizes are available in the in S2: Reconstruction results for different spatial resolutions). Valid cells are filled and include reconstructions from at least one large lake ($\geq$ 50 ha) or several smaller basins. Forest cover in Eastern North America is  higher than in Europe and Asia.  REVEALS reconstructed forest cover  is generally lower than raw pollen reconstructions.

**3.4 Validation with gridded data sets**

**3.4.1**

265 Remote sensing forest cover within relevant pollen source areas was used to validate the modern, reconstructed forest cover from the original pollen data and  the REVEALS run for each grid cell. Here we present validation of gridded data with a 2° spatial resolution. Validations with additional spatial resolutions differ only marginally and are included in the supplementary materials (S3: Validation results for different spatial resolutions).

270 Forest cover reconstructed from original pollen data is predominantly higher than remote sensing forest cover with a  mean absolute error (MAE) of  33.05% in the Northern Hemisphere (see Fig. 10a). As reconstructed forest cover is much lower for  the REVEALS reconstruction (see Fig.7),

reconstructions.  the MAE value is reduced significantly to 19.73% (see Fig.
275   9a).

[Figure]

**Figure 9.** Remote sensing forest cover (LANDSAT) and modern reconstructed forest cover from Pollen  and REVEALS (< 100 years BP) in 2x2° grid cells with  mean absolute errors (MAE) and  correlation coefficient (R) per group. Reconstructed forest cover from the original pollen data tends to overestimate observed (remote sensing) forest cover.  Improvements with the REVEALS  reconstruction are especially high in Europe. Validations with different grid cell sizes are available in the supplement (S3: Validation results for different spatial resolutions).

Continental mean absolute errors (MAE) in forest cover from original pollen data range from 24.61% (Asia) to 37.49% forest cover (North America, see Fig. 9b). All continental MAE values are lower for the REVEALS reconstruction  and range from 9.44% (Europe) to 27.27% (North America). The
280   improvement is largest in Europe (72% relative to the initial MAE of the pollen-based reconstruction, see Fig. 9 and 10)

and smallest in North America (24%). ~~Forest cover from the REVEALS reconstruction with optimized RPP values reduces continental MAE values even further with values ranging between 9.1% (Africa) and 21.08% forest cover (South America). MAE are generally improved more with optimized RPP values with the exception of records in Australia and Oceania. The largest improvement (relative to the pollen-based forest cover MAE) was achieved in North America (65%) but reconstructions in Europe (61%) and Asia (48%) also reduced the original MAE by more than or roughly half. The REVEALS runwith optimized RPP valuesthebest with, with the exception of records from Australia and Oceania. Additionally, the reduction of forest cover MAE, and therefore the reconstruction improvement, was much larger in the continents of the Northern Hemisphere for both REVEALS runs~~. Nevertheless, forest cover still tends to be overestimated.

[Figure]

**Figure 10.**  Forest cover reconstruction  per continent  for a gridded 2x2° reconstruction.  Mean errors decreased with the  REVEALS reconstruction  for all continents buthigherthe Northern Hemisphere~~Europe.

Spatial patterns are present for the errors of  both forest cover reconstructions (see Fig. 11). In  Europe the REVEALS  reconstruction manages to reduce errors extensively. In  Eastern and coastal Northwestern North America, the

REVEALS reconstruction still tends to overestimate forest cover, even with the application of REVEALS and after optimizing . This could be due to a lack of continental RPP values. The same is the case for several records in eastern AsiaIn North America, few RPP studies are available (see Appendix A) and more taxa are assigned hemispheric or standardized values than in the other continents.

[Figure]

**Figure 11.** Map of the reconstruction error (in % forest cover) for forest cover reconstructed from Pollen , REVEALS with original RPP values and REVEALS data. Remaining errors with optimized RPP valuesthe overall better REVEALS reconstructions are especially high in North America (Northern West Coast, Labrador Peninsula).

[revised manuscript text omitted]

To ensure the correct utilization of the dataset and to obtain reliable analysis results, several key considerations should be followed. Firstly, rasterization mitigates individual errors by temporal and spatial averaging. This process is particularly useful in reducing the variance that might arise from individual measurements, providing a more reliable representation of the underlying signal. The reliability of reconstructions varies among different taxa due to the quality of RPP values, and this is explicitly documented in a supplementary file that outlines the sources of RPP values (see Section Code and Data availability). Reconstructions of taxa with continental RPP values are the most reliable, followed by those based on hemispheric data, with standardized RPP values being the least reliable. This hierarchy should be taken into account when interpreting the results. Higher certainty is associated with forest cover reconstruction, as it is based on aggregation among taxa. Reconstructions of temporal forest cover trends are reliable, as evidenced by high correlation coefficients, despite a tendency for absolute values to be overestimated, particularly in North America. For individual time series, the reliability of data varies with the size of the lakes from which samples were taken. Only data derived from large lakes ($\geq$ 50 ha) are reliable for site-wise analyses. This distinction is clearly indicated with validity flags in the dataset. Reconstructions from smaller basins should not be used alone.

**5  Conclusions**

We present data sets of reconstructed compositional vegetation and forest cover  in the Northern Hemisphere from a sedimentary pollen data set using the REVEALS model. We used  synthesized RPP values for reconstruction and made use of hemispheric or standardized values, when continental ones were not available. This approach allowed us to address some of the inherent biases in pollen compositions. Considerable improvement in the reconstruction of forest cover is achieved in all continents. Improvements were smallest in North America, which suggest a need for further RPP studies.

Accurate data on past vegetation is invaluable for the validation of coupled climate-vegetation models and the testing of hypotheses on feedback effects and vegetation dynamics. This knowledge is essential for modeling and predicting vegetation
415 trajectories under anthropogenic climate change.

**6 Code and data availability**

The produced datasets are freely available from Zenodo (https://doi.
org/10.5281/zenodo.12800159).

Input data from LegacyPollen 2.0 is available on PANGAEA  (https://doi.pangaea.de/10.1594/PANGAEA.965907, Li
420 et al. 2024b).

The code used to produce the datasets  and adjustable rasterization code are freely available from Zenodo (https://doi.org/10.
5281/zenodo.10191859, https://doi.org/10.5281/zenodo.12800291, Schild and Ewald 2023).

---

## Author Response (AR3)

Dear Kirsten Elgers,

Thank you for the opportunity to revise our manuscript.

We have carefully reviewed and addressed the reviewer's comments, making several minor adjustments accordingly. The reviewer's primary concerns regarding tree cover overestimates and certainty were already discussed in the manuscript, but we have expanded upon these explanations and discussions where necessary.

We believe these revisions have further refined our manuscript. Additionally, the dataset has been updated on Zenodo, and we have requested an update at PANGAEA, which will be processed next weel. We intend for the final data description paper to reference the PANGAEA dataset, which is why we have included this reference in the updated manuscript. Once the dataset update has been confirmed by the PANGAEA data steward, we will inform you promptly.

Thank you for your attention and support.

Best regards,
Laura Schild

**Reply to Thomas Giesecke**

Dear Thomas Giesecke,

Thank you for taking the time to review our manuscript and for providing your thoughtful feedback. We have addressed your inline comments directly within the PDF and made changes where appropriate.

Below, we respond to your two remaining major comments:

1. **Forest Cover Overestimates**: We appreciate your observation regarding the overestimation of forest cover. This topic was already discussed in the manuscript, where we also addressed potential discrepancies between remote sensing data and pollen-based reconstructions. For your reference, we have included the relevant paragraphs below.
2. **Taxonomic Reconstruction Uncertainty**: We have provided further commentary on the uncertainties associated with taxonomic reconstructions, as distinct from tree cover reconstructions, and have highlighted the error estimates included in the dataset.

We hope these responses address your concerns and provide further clarity on our approach.

Best regards,
Laura Schild

**Original comment**

Reading the revised manuscript I noted some detailed problems and marked them on the PDF. The manuscript is not ready for publication, requiring at least one last careful round of revision to clarify what is meant what was done, and why.

The authors did not address two of my earlier comments/concerns that I find important.
1. "the spatial differences nicely displayed in Fig 11 should be discussed. Fig 11 also indicates that there are grids with an overestimate in tree cover (even more common at a higher resolution grid), which needs to be explained."
    A new figure (9) has the value on the negative error removed, but there is no explanation given in which cases REVEALS overestimates forest cover. These may be interesting cases providing insights into potential errors of the comparison (e.g. recent forest felling) or locally inadequate PPEs.

**Reply**

We agree with the assessment of the continuous overestimate of forest cover even when using the REVEALS reconstruction. We already highlight regional differences in reconstruction success and discuss potential reasons for this persistent overestimate. Very few grid cells underestimate forest cover and we added a connection to the difference in remote sensing forest cover and pollen-based forest cover in the manuscript.
We include relevant paragraphs below.

Regional differences in reconstruction success are discussed here. (The underestimate in few grid cells is included.)

230    tree cover that corresponds better remote sensing forest cover. Nevertheless, tree cover still tends to be overestimated. Spatial patterns are present for the errors of both tree cover reconstructions (see Fig. 9). In Europe the REVEALS reconstruction manages to reduce errors extensively. In Eastern and coastal Northwestern North America, the REVEALS reconstruction still tends to overestimate tree cover.

However, continental differences are evident in the quality of tree cover reconstruction, with Europe showing a significantly larger reduction in errors compared to other regions. North America and Asia exhibit larger reconstruction errors in the RE-
250    VEALS estimates, though these are still lower than those derived from tree pollen percentages. Notably, regions such as the Great Lakes, the Labrador Peninsula, and the Pacific Northwest display particularly high errors in tree cover reconstruction. Asia, characterized by sparser coverage, presents fewer large errors increasing the overall continental reconstruction error. This highlights the need for improved vegetation reconstruction, especially in North America and Asia. The reason for this reduced performance could lie in a lack of RPP studies, especially in North America, or in a significantly higher regional variability

Reasons for differences between continents:

highlights the need for improved vegetation reconstruction, especially in North America and Asia. The reason for this reduced performance could lie in a lack of RPP studies, especially in North America, or in a significantly higher regional variability

255    of RPP values compared to Europe. While differences in validation outcomes across varying spatial resolutions are marginal (see S4), some variability is observed when different grids are employed, highlighting spatial heterogeneity in reconstruction

Potential reason for overall overestimates:

success. Despite these caveats, overall trends in tree cover appear consistent, with acceptable correlation coefficients, though absolute values in certain regions remain challenging to interpret with confidence as tree cover continues to be overestimated in all continents. As a result, arboreal taxa might still be overrepresented in the reconstruction because of inaccurate or missing
260    RPP values. Very few grid cells, predominantly in central Europe, slightly underestimate forest cover.

Differences between remote sensing data and reasons for discrepancies are discussed here (including the potentially increased anthropogenic impact)

Validating pollen-based tree cover estimates with remote sensing-derived forest cover also presents a challenge. One key issue  are the inherent errors associated with remote sensing forest cover data. While validation using other sensors is possible, only a limited subset of the available data is cross-validated with Lidar data, which itself is characterized by limited spatial coverage (Sexton et al., 2013). A critical limitation of surface reflectance methods, as used in the Landsat-based forest cover, is their reliance on a 2D perspective, primarily capturing the forest canopy. This means that the understory is often not detected, resulting in an incomplete representation of the forest structure. In contrast, pollen-based estimates provide a more comprehensive, stratified view of the vegetation, as they incorporate all contributing taxa, not just the tree canopy. Despite this broader scope, pollen data and REVEALS estimates tend to emphasize trees more than other vegetation types consistently as is evident in the validations. Few records in central Europe pose an exception to this where forest cover seems to be slightly underestimated (see Fig. 9). Furthermore, pollen-based estimates are derived from records that span a much longer timescale than the modern forest cover data available, even though modern timeslices are used for validation. Increased anthropogenic impact could exacerbate discrepancies between pollen-based and remote-sensing estimates. This could contribute to the overestimation of forest cover, which persists in all continents. Additionally, these modern and arguably unnatural vegetation conditions may not correspond to past vegetation and may therefore have reduced significance for the reconstruction of past, natural landscapes.

**Original comment**

2. "Thus the data may be useful for continental-scale forest cover reconstructions while regional studies would benefit from regionally estimated PPEs. The caveats of using continental scale RPPEs and particularly of setting RPPEs to 1 for some taxa need to be discussed in the publication."

   I highlighted detailed problems using one of the sites that I am most familiar with. Shield at al. respond to my comments "It should also be remembered that the pollen record from a lake as large as Großer Treppelsee (~59 ha) will have a regional signal rather than a local one, including the mosaic of open and closed vegetation in Brandenburg and likely even Poland and not just the rather closed forest surrounding the actual lake. The modern forest cover (landsat) in a circle with a 100 km radius surrounding Großer Treppelsee is 28.194% and the reconstructed modern value (with the openness correction accounting for urban areas) is ~ 29%." However, they don't respond to the point I raised, that while the overall forest values seem to be in the right order the values for individual taxa are off. (Although I would argue that the lake is divided into three basins and therefore not sensing a region as large as a 100 km radius.) Assuming the authors are correct with 28% forest cover, assuming a general proportion of Pinus around 70% (average proportion in Brandenburg https://www.sdw-brandenburg.de/ueber-den-wald/wald-in-brandenburg/ and there is rather more pine in this region including the Polish side) would result in a proportion for Pinus of around 20% while the authors reconstruct 10%. The reconstructed 8% – 17% Brassicaceae cover are not

even commented on. Yes, they could represent rape fields in the modern situation but 30% for 200 years ago are difficult to explain.

This dataset needs a warning for users: OK FOR CONTINENTAL QUESTIONS BUT POTENTIALLY BIASED FOR INDIVIDUAL SITES!

**Reply**

We agree that reconstructions of specific taxa are less reliable than the reconstruction of tree cover, where several taxa are aggregated. We highlight this in several paragraphs in our manuscript. We believe that our focus on tree cover reconstructions in the manuscript's abstract makes this clear as well. Namely here, where we point out the higher certainty of tree cover:

> The reliability of reconstructions also varies among different taxa due to the quality of RPP values, which is documented in detail in a supplementary file outlining the sources of RPP values (see Section "Code and Data Availability"). Reconstruc-
> 290 tions based on taxa with continental RPP values are the most reliable, followed by those based on hemispheric data, with standardized RPP values being the least reliable. This hierarchy should be considered when interpreting results. The use of continental RPP values could also make our reconstruction more reliable at larger spatial scales as opposed to local reconstructions. Additionally, uncertainties in RPP values themselves can affect reconstruction success and could be leading to the persistent overrepresentation of tree taxa despite the application of REVEALS. Tree cover reconstructions tend to have higher
> 295 certainty compared to taxon-specific reconstructions, as they are based on aggregation across taxa. However, the static latitudinal arboreal threshold for Betulaceae, *Betula*, and *Alnus* poses a limitation in our reconstruction. This could be improved by incorporating a dynamic, climate-dependent threshold in future work.

And here, where we discuss the challenges in validating reconstructions of specific taxa at a large spatial scale.

> 315      Another challenge lies in validating the compositional reconstruction results. It remains uncertain whether RPP values have remained stable over time, and historical compositional data are not only scarce but also likely too recent to test this assumption effectively (Baker et al., 2016). Validating modern compositional reconstructions on large spatial scales is therefore difficult. As global compositional vegetation data are not readily available, remote sensing of tree cover serves as the best option for validation. But even with accurate tree cover reconstructions, uncertainties remain regarding the abundance of individual taxa
> 320 due to the aggregated nature of the tree cover measure. To address this issue, global syntheses of tree and plant inventories or compositional remote sensing products could provide more robust validation. Additionally, vegetational compositions derived from sedimentary ancient DNA (sedaDNA) offer a promising avenue for comparing past vegetation data. Local quantitative sedaDNA vegetation signals could be averaged across multiple records to compare with pollen-based results (Niemeyer et al., 2017; Capo et al., 2021).

**Original comment**

> While the data for the southern hemisphere has been removed southern hemisphere taxa are still included in the dataset resulting in empty columns. While the manuscript now mentions the different values included (mean, median, standard deviation, and 10% and 90% quantile values), it still is not stating why these values are provided or how they may be used. For the above case of *Pinus* the uncertainty range is not including the minimum suggested value of 20%. Thus based on this single example the uncertainty range provided suggests an accuracy that is not there.

**Reply**

Thank you for noticing empty columns in the dataset. We removed any taxa that were not present in the continental datasets (Asia: 11, Europe: 3, North America: 17).

While we do describe how these values are calculated, we failed to explicitly highlight that these are included in the dataset. We now do so here and include an explanation of why the model's certainty/uncertainty can still be wrong regarding reality:

105    sion 0.9.13, https://github.com/MartinTheuerkauf/disqover/blob/main/disqover). It mainly differs from the original program by Sugita (2007) in the process of error calculation. REVEALSinR includes repeated model runs with random error added to RPP values and pollen counts (see Table 2 for the number of variations). The resulting distribution of REVEALS results allows for an estimation of the standard deviation of vegetation cover per taxon.  These error estimates are included in our dataset as the standard deviations as well as the 10th and 90th percentile estimates for each taxon. However, the program by Sugita

110    (2007)  derives error estimates with a hybrid method from a variance-covariance matrix of PPE and Monte Carlo simulations. For further details on the REVEALS model see the original publication Sugita (2007) and for previous REVEALS applications on continental scales see e.g Li et al. (2017), Githumbi et al. (2022), Serge et al. (2023), and Dawson et al. (2024).

While the error estimates give insights into how certain the model is, estimates can still be wrong depending on the quality of RPP values provided.

**LegacyVegetation: Northern Hemisphere reconstruction of past plant cover and total tree cover from pollen archives of the last 14 ka**

Laura Schild[1,2], Peter Ewald[1,2], Chenzhi Li[1,2], Raphaël Hébert[1], Thomas Laepple[1,3], and Ulrike Herzschuh[1,2,4]

[1]Helmholtz Centre for Polar and Marine Research, Research Unit Potsdam, Alfred Wegener Institute (AWI), Germany
[2]Institute of Environmental Sciences and Geography, University of Potsdam, Karl-Liebknecht-Straße 24-25, Potsdam, Germany
[3]MARUM-Center for Marine Environmental Sciences and Faculty of Geosciences, University of Bremen, Germany
[4]Institute of Biochemistry and Biology, University of Potsdam, Karl-Liebknecht-Straße 24-25, Potsdam, Germany

**Correspondence:** Ulrike Herzschuh (ulrike.herzschuh@awi.de)

**Abstract.** With rapid anthropogenic climate change future vegetation trajectories are uncertain. Climate-vegetation models can be useful for predictions but need extensive data on past vegetation for validation and improving systemic understanding. Even though pollen data provide a great source of this information, the data is compositionally biased due to differences in taxon-specific relative pollen productivity (RPP) and dispersal.

Here we present a Northern Hemisphere reconstruction of quantitative regional vegetation cover from a sedimentary pollen data set for the last 14 ka using the REVEALS model to correct for taxon- and basin-specific biases. For the reconstruction, we expanded on a previously published synthesis of continental RPP values.

The data sets include taxonomic compositions as well as reconstructed tree cover for each original pollen sample. Additional metadata includes modeled ages, age model sources, basin locations, types, and sizes.

The improvements in tree cover reconstructions with the REVEALS reconstruction using continental RPP values range from 22% (Asia) to 67% (Europe) relative to the mean absolute error (MAE) of the pollen-based tree cover. The dataset can be used as a grid with binned and aggregated samples (adjustable script provided on Zenodo; https://doi.org/10.5281/zenodo.13902976) or as individual time series if the record's basin size exceeds 50 ha.

This alternative quantitative reconstruction of vegetation cover is beneficial for the investigation of past vegetation dynamics and modern model validation when varying spatial and temporal resolutions may be required. By collecting more RPP estimates, especially in North America, and adding more records to existing pollen data syntheses, reconstructions may be improved even further. The new REVEALS dataset is freely available on PANGAEA (see Data availability section).

**1 Introduction**

Anthropogenic climate change is driving vegetation shifts that could lead to disruptions in ecosystem functions and services, and even trigger feedback effects with other earth system elements (IPCC, 2023; Armstrong McKay et al., 2022). Predicting these changes through modeling is challenging. A thorough mechanistic understanding of vegetation dynamics and their interactions with climate is essential. This requires validating and testing data from coupled climate-vegetation models, which in turn depends on the availability of extensive vegetation data from periods spanning climatic transitions. (Dearing et al., 2012). Given the relatively brief duration of available instrumental climate and vegetation data, there is a clear need for long-term vegetation records derived from paleoecological archives that cover broader climatic gradients than modern datasets (Dearing et al., 2010; Dallmeyer et al., 2023).

Pollen data as a direct proxy for paleo-vegetation is especially useful for comparisons with modeled data as it can be used to reconstruct land-use (Fyfe et al., 2015; Davis et al., 2015), biomes (Woodbridge et al., 2014; Prentice et al., 1996), and climate (Herzschuh et al., 2023a, b; Bartlein et al., 2011; Viau et al., 2012). The compilation of pollen data syntheses is essential to aid this purpose (Anderson et al., 2006; Gaillard et al., 2010; Strandberg et al., 2014). Several subcontinental and continental collections of pollen data already exist, spanning regions such as Europe, North America, Africa, Siberia, and China (Fyfe et al., 2009a; Whitmore et al., 2005; Vincens et al., 2007; Cao et al., 2014, 2020) and have been integrated into the global database Neotoma (Williams et al., 2018). To allow for a broader application of pollen data, LegacyPollen 2.0 (Li et al., 2024b) offers a global, harmonized pollen dataset that underwent taxonomic standardization, metadata verification and consistent age modeling (Li et al., 2022a, 2021; Herzschuh et al., 2022). This taxonomic harmonization trades off the higher taxonomic resolution of some datasets for equivalence, resulting in overall comparability useful for analyses at large spatial scales. Despite advances in harmonization, the use of pollen data remains limited due to the fact that pollen compositions do not accurately reflect vegetation (Davis, 1963; Prentice, 1985; Prentice and Webb III, 1986). This limitation arises from variations in taxon-specific parameters such as relative pollen productivity (RPP) and pollen dispersal characteristics, leading to discrepancies between the pollen record and actual past vegetation. This hinders quantitative vegetation assessment as taxa with high pollen productivity and efficient pollen dispersal tend to be overrepresented in the pollen record, while those with low pollen productivity and less effective dispersal are underrepresented. These factors, together with the compositional nature of pollen data, result in a non-linear relationship between pollen and vegetation, titled the Fagerlind effect (Prentice and Webb III, 1986; Fagerlind, 1952). Approaches such as the R-value model (Davis, 1963; Webb et al., 1981) and the extended R-value model (Parsons and Prentice, 1981) were created to address this issue and were later included into Sugita's (2007) model for "Regional Estimates of Vegetation Abundance from Large Sites" (REVEALS). By accounting for taxon-specific RPP and fall speed values, as well as basin-specific parameters such as basin size and type, REVEALS estimates regional vegetation cover from pollen counts. The model has been applied in several regional-scale studies (Nielsen et al., 2012; Mazier et al., 2015; Hellman et al., 2008a) and multiple validations have demonstrated its ability to approximate actual vegetation (Sugita et al., 2010; Hellman et al., 2008a; Soepboer et al., 2010; Mazier et al., 2012), even though the model's performance heavily relies on accurate taxon-specific parameters. While Li et al. (2017), Wieczorek and Herzschuh (2020), and Githumbi et al. (2022) provide comprehensive compilations of RPP and fall speed values for taxa of China, the Northern Hemisphere, and Europe the Northern Hemisphere respectively, the overall availability of RPP studies is still limited and regional variations in RPP values exist (Harris et al., 2020; Broström et al., 2008; Li et al., 2017; Mazier et al., 2012). This makes the application of REVEALS on larger scales particularly challenging. Only some (sub-) continental REVEALS reconstructions are available for

Europe (Trondman et al., 2015; Roberts et al., 2018; Githumbi et al., 2022; Serge et al., 2023), Asia (Cao et al., 2019; Li et al., 2022b, 2023, 2024a), and North America (Dawson et al., 2024a). Currently, no global or Northern Hemispheric quantitative vegetation cover reconstructions using REVEALS exist.

60

With its importance for the assessment of biome stability, carbon storage, climatic feedbacks, and land-use-change, tree cover is an often reconstructed variable (e.g. Fyfe et al., 2015; Githumbi et al., 2022; Serge et al., 2023). Due to the global availability of remote sensing data on contemporary tree cover, reconstructions of tree cover in modern time slices may even be validated (Hjelle et al., 2015; Roberts et al., 2018). Yet, only Serge et al. (2023) and Pirzamanbein et al. (2014) use this op-

65 portunity for extensive validation and even improvement of reconstructions from European pollen records. No grid-cell based validations exist for the Northern Hemisphere.

Here we present reconstructed quantitative vegetation cover for the Northern Hemisphere from the LegacyPollen2.0 dataset - an updated global taxonomically and temporally standardized fossil pollen dataset of 3680 palynological records - using

70 REVEALS spanning the last 14k years. The data sets were created using existing estimates of taxon-specific parameters. The REVEALS reconstruction includes corrected vegetation compositions as well as reconstructed tree cover.

**2 Methods**

**2.1 Pollen Data Set**

The pollen data synthesis LegacyPollen2.0 (Li et al., 2024b) includes 3680 temporally resolved records (time-series) distributed

75 globally. Data were collected from individual publications and the Neotoma Paleoecology Database which includes data from the European Pollen Database, and the North American Pollen database (Fyfe et al., 2009b; Giesecke et al., 2014; Whitmore et al., 2005; Williams et al., 2018). An overview of Neotoma records included in LegacyPollen 2.0 and this reconstruction can be found in

For the REVEALS reconstruction only lake and peat records in the Northern Hemisphere were used ($n$ = 2752) Analogous

80 to the preceding LegacyPollen 1.0 dataset (Herzschuh et al., 2022), the data synthesis involved revising and standardizing age modeling and taxonomic harmonization for consistency of records.  chronologies may, therefore, differ slightly from previous reconstructions due to this revised age modeling. Spatial data coverage of records in the reconstruction is dense in Europe (1287 records) and North America (1040) and sparsest in Asia (446) (see Fig. 1). The records' sample density decreases with age (see Fig. 2). Only samples dated to 14 ka BP or younger were used to ensure that the climatic conditions of

85 recorded vegetation were similar to the modern climate ?Mottl et al. (2021); ?.

[Figure]

**Figure 1.** Pollen record locations in the LegacyVegetation dataset. Colors indicate record type (large lake $\geq$ 50 ha). Record density is highest in Europe and Eastern North America, and lowest in Northern and Central Asia.

[Figure]

**Figure 2.** Temporal coverage of records in the LegacyVegetation dataset per continent. Bins are 500 years wide. Sample count decreases with age and Europe has the most samples overall.

**2.2 Implementing REVEALS**

The REVEALS model estimates quantitative vegetation coverage from pollen assemblages using site and taxon-specific parameters (Sugita, 2007). Based on wind speed and taxon-specific fall speed, pollen dispersal is modeled in ring sources around the basin and deposition over the basin is integrated to give pollen influx. Together with RPP this dispersal factor is used to correct original pollen counts to better represent actual vegetation (see Equation 1 and Table 1).

90

$$\hat{V}_i = \frac{n_{i,k}/\hat{\alpha}_i \int_R^{Z_{max}} g_i(z)dz}{\sum_{j=1}^m (n_{j,k}/\hat{\alpha}_j \int_R^{Z_{max}} g_i(z)dz)} \tag{1}$$

The REVEALS model follows a set of assumptions. Firstly, neither directionality nor pollen transport through agents other

**Table 1.** Algebraic terms in the REVEALS equation (see Equation 1)

| Function term | definition |
|---|---|
| $\hat{V}_i$ | vegetation estimate of taxon i |
| $n_{i,k}$ | pollen counts of taxon i at site k |
| $\alpha_i$ | relative pollen productivity of taxon i |
| $R$ | basin radius |
| $Z_{max}$ | maximum extent of regional vegetation |
| $z$ | distance from a point in the center of a basin |
| $g_i$ | dispersal and deposition function for taxon i |

than wind are considered in the model. The maximum spatial extent for this pollen transport ($Z_{max}$, see Table 2) has to be set to define the region in which most of the pollen originates. This value will always be an assumption and has only been tested em-
95 pirically by Hellman et al. (2008b). Additionally, it is assumed that the basin is circular with no source of pollen within the basin radius. The peatland and bog sites used in our reconstructions inherently violate this assumption. Nevertheless, the quantitative reconstruction of vegetation cover from peatland cores is possible by using Prentice's deposition model (Prentice, 1985, 1988) instead of Sugita's deposition model (Sugita, 1993) in the dispersal and deposition function (see Eq. 1; Sugita, 2007). Previous studies show that results from small bogs are still reliable when aggregated, while results from large bogs alone tend to deviate
100 from those of large lakes due to the violation of the aforementioned assumption (Trondman et al., 2016). Using small peatland records for reconstructions is, therefore, appropriate when spatially averaging multiple sites. Following Trondman et al. (2015), we do so by using both large and small peatlands. We use REVEALSinR from the DISQOVER package in R to implement REVEALS (Theuerkauf et al., 2016, Version 0.9.13, https://github.com/MartinTheuerkauf/disqover/blob/main/disqover). It mainly differs from the original program by Sugita (2007) in the process of error calculation. REVEALSinR includes re-
105 peated model runs with random error added to RPP values and pollen counts (see Table 2 for the number of variations). The resulting distribution of REVEALS results allows for an estimation of the standard deviation of vegetation cover per taxon. The program by Sugita (2007), however, derives error estimates with a hybrid method from a variance-covariance matrix of PPE and Monte Carlo simulations. For further details on the REVEALS model see the original publication Sugita (2007) and for previous REVEALS applications on continental scales see e.g Li et al. (2017), Githumbi et al. (2022), Serge et al. (2023),
110 and Dawson et al. (2024a).

**2.2.1 Parameters and Model Settings**

For each taxon, values for RPP (with uncertainties provided as standard deviation) and fall speeds are used. We made use of the synthesis of Northern Hemisphere RPP and fall speed values by Wieczorek and Herzschuh (2020). Several RPP studies published since this synthesis were added to the compilation (Geng et al., 2022; Li et al., 2022b; Wang et al., 2021; Huang et al., 2021; Zhang et al., 2021a, b; Wan et al., 2020, 2023; Jiang et al., 2020). The methods for study selection and calculation of synthesis values follow Wieczorek and Herzschuh (2020) as well as Githumbi et al. (2022). We expanded the synthesis calculation of RPP to different taxonomic levels (genus, family, and order) to account for the taxonomic harmonization in the pollen dataset. An overview of original values and synthesized values can be found in Appendix A and B respectively. The amount of RPP values in Asia (59) and Europe (69) is higher than in previous RPP synthesis due to the inclusion of multiple taxonomic levels (Li et al., 2018; Githumbi et al., 2022).

When available, we use continent-specific values in our reconstruction. For taxa with no continental values present, we use Northern Hemispheric values. If no values exist for a taxon, RPP is set to a constant (RPP = 1, $\sigma$=0.25) and fall speeds are filled with mean continental fall speeds. Continental RPP values are available for the majority of pollen counts in all three continents (see Fig. 3). The fraction of pollen counts for which standard RPP values were assumed is highest in North America but still < 10%. For each site, the REVEALS model also requires information on basin type, basin size and original pollen counts, all of which were collected in the LegacyPollen 2.0 dataset (Li et al., 2024b). Missing basin areas for lakes and peatlands are set to a standard value which can be found in Table 2 together with several constant parameters set in REVEALSinR. Lastly, we also reduced computational effort in REVEALSinR by implementing a maximum number of steps in the lake model used to model mixing in the basin. The number of steps was set to 500 unless $m$ falls below that maximum value for $m = basin\,radius/10$ for basins with a radius of at least 1000 m and $m = basin\,radius/2$ for basins with a radius smaller than 1000 m.

**Table 2.** Static model parameters and model settings for REVEALS runs using REVEALSinR (Theuerkauf et al., 2016).

| Parameter | Values and settings used in model run |
| --- | --- |
| atmospheric model | unstable atmosphere |
| dispersal model | gaussian plume |
| wind speed | $3m \times s^{-1}$ |
| maximum extent of regional vegetation ($Z_{max}$) | 1000 km |
| number of RPP and pollen count variations (n) | 2000 |
| peatland basin area (for missing sizes) | 31.41 ha |
| lake basin area (for missing sizes) | 49 ha |
| function to randomize pollen counts | rmultinom_reveals |

[Figure]

**Figure 3.** Percentage of the total pollen counts for which either continental, hemispheric, or "standard" RPP values were used. The standard value (1+-0,5) is used when no RPP value is available for a specific taxon.

**2.3 Reconstruction of tree cover and validation**

Tree cover was reconstructed by summing up percentages of arboreal taxa (see S2: List of arboreal taxa) with Betulaceae, *Betula*, and *Alnus* being classified as arboreal at sites below 60° N. The mean reconstructed compositional coverages from the REVEALS results were used for the tree cover reconstructions. REVEALS results were then rasterized to also include records from smaller basins in a temporal and spatial aggregation. Reconstructed time series were averaged in 500 year bins and then rasterized and averaged in grids of differing spatial resolution. A grid cell was classified as having a valid reconstruction when it contained records from at least one large lake (>= 50 ha) or at least two small basins following Serge et al. (2023). Standard deviations of the REVEALS estimates were aggregated by applying the delta method by Stuart and Ord (1994), using the same equation as Wieczorek and Herzschuh (2020). We provide a script for rasterization with adjustable temporal and spatial resolution for users of the dataset on Zenodo (https://zenodo.org/doi/10.5281/zenodo.12800290).

This method of temporal and spatial averaging differs from several previous REVEALS applications. Pollen counts are often summed in temporal bins prior to running REVEALS to increase pollen counts and reduce uncertainty (Trondman et al., 2015; Githumbi et al., 2022; Serge et al., 2023; Dawson et al., 2024a). However, temporally averaging after the REVEALs application, as implemented by us, increases the flexibility of the dataset with the trade-off of potentially increased uncertainty. Rasterization has previously been perfomred by using a weighted average taking into account the basin size of the original record (Trondman et al., 2015; Githumbi et al., 2022; Serge et al., 2023). However, the most recent REVEALS-based North American vegetation reconstruction uses the same arithmetic mean as described above (Dawson et al., 2024b). When comparing our method of temporal and spatial aggregation to that used by previous European reconstructions (e.g. Serge et al., 2023), we also found no significant differences in the validation of reconstructed tree cover (see S6).

For validation, the reconstructed tree cover of the past 100 years was rasterized and compared to modern remote sensing forest cover. Only valid grid cells as defined above were used for validation. Average forest canopy cover for all grid cells was extracted from the Landsat Global Forest Cover Change (GFCC) data set from the temporal average of the years 2000, 2005, 2010, and 2015 (Sexton et al., 2013; Townshend, 2016). An openness correction was applied to sites containing urban areas and paved surfaces within the 80% pollen source areas (Supplementary Materials S5) to correct for areas without any pollen sources and thus ensure comparability to modern remote sensing forest cover (see Equations 2-4). For this, the percentage of unvegetated land cover classes for the year 2015 in the ESA CCI land cover data set was used (ESA, 2017, see Table 3). Areas covered by water or ice are already considered as missing values in the remote sensing forest cover data set and do not need to be corrected for. Reconstructed tree cover was validated for each grid cell and mean absolute error (MAE $= \frac{1}{n} \sum_{i=1}^{n} |y_i - \hat{y}_i|$) and correlation coefficients were calculated for each continent. No openness correction was applied to the reconstruction values in the final dataset. Validation for a 2x2° grid is included in the results section. Further validations using 1°, 5°, and 10° resolution are included in the supplementary material (S4).

**Table 3.** Unvegetated land cover classes in ESA CCI LC chosen for the openness correction.

| Name | Code |
| --- | --- |
| Urban areas | 190 |
| Bare areas | 200 |
| Consolidated bare areas | 201 |
| Unconsolidated bare areas | 202 |

$$unvegetated\ classes = \{190, 200, 201, 202\} \tag{2}$$

$$unvegetated\ (\%) = \frac{\sum cells\ in\ PSA \in unvegetated\ classes}{\sum cells\ in\ PSA} \tag{3}$$

$$corrected\ tree\ cover = reconstructed\ tree\ cover \times (1 - unvegetated) \tag{4}$$

Additionally, we compare our REVEALS reconstruction to the most recently published REVEALS reconstruction in Europe by Serge et al. (2023, version: RPPs.st1). We average our reconstruction in the same grid and temporal bins as used by Serge et al. to compare the reconstructed tree cover between both reconstructions. To get the total tree cover, we sum evergreen and summergreen tree cover values in Serge et al.'s dataset, while excluding broadleaved summergreen temperate warm shrubs

(BSTWS) and broadleaved evergreen xeric shrubs (BEXS). We validate the previous reconstruction and our reconstruction in the most recent time slice available in Serge et al.'s reconstruction (-65 to 100 BP, https://doi.org/10.48579/PRO/J5GZUO)

175 with the remote sensing forest cover and compare validations. Unfortunately, direct validation could only be performed with the most recent time slice available online, rather than the historical time slice used in the validation by Serge et al., which limits the ability to reproduce their validation results exactly. We do not apply any openness correction here as we do not have comparable 80% pollen source areas available for the records used in Serge et al. (2023). The reconstruction by Serge et al. differs in the temporal as well as spatial aggregation routine, as described above. Definition of arboreal taxa varies, a different

180 RPP-value set was used, and the amount of total records included is higher than in our reconstruction (Serge et al.: 1607, LegacyVegetation: 1287).

**3 Data summary**

**3.1 Dataset description**

The published dataset includes vegetation reconstructions for individual records in Asia, Europe, and North America up until

185 14 ka BP. The reconstructed coverage values include mean, median, standard deviation, and 10% and 90% quantile values for each taxon. Mean values and standard deviations are given for tree cover. For each sample its validity as a site is given. Only reconstructions from large lakes are valid independently. To include all other records a spatial and temporal average is necessary (rasterization, https://doi.org/10.5281/zenodo.12800291).

REVEALS was used to reconstruct quantitative vegetation cover. Here we illustrate a comparison between these recon-

190 structed compositions to the original pollen composition. Differences in composition between pollen data and REVEALS are apparent for all continents of the Northern Hemisphere. Some clear examples include: increases of Cyperaceae in all continents, decreases of *Betula* in Europe, decreases of *Pinus* in all continents, and increases of *Acer* in North America with the application of REVEALS and its intended correction of taxon-specific biases (see Fig. 4).

195 Using the compositional data available from the original pollen data and the REVEALS run, we reconstructed tree cover for all sites and samples and rasterized the result with different spatial resolutions. The temporal trend in Northern Hemisphere tree cover is the same for both pollen and REVEALS data. Tree cover increases from 14 ka BP until roughly 6 ka BP and decreases again towards the present (see Fig. 5). REVEALS reconstructed tree cover is generally lower than tree cover from original pollen compositions. On average tree cover values from the REVEALS run are roughly 14.54% lower than values

200 from original pollen compositions. The temporal trends in Asia and North America are positive, whereas tree cover in Europe reached its maximum around 6 ka BP and has been decreasing since.

Tree cover is generally highest in Eastern North America. This is also where data coverage is best in North America (see Fig. 6). The density of valid grid cells is very high in Europe. Data coverage in Asia is sparse, but valid grid cells indicate higher tree cover on the Southeastern coast and in the boreal biome. Rather open areas exist at the Tibetan Plateau and at very high

[Figure]

**Figure 4.** Average continental taxonomic coverages per reconstruction for the 8 most common taxa per continent. Differences are especially evident for *Pinus*, *Artemisia*, and *Betula*, which all have decreased coverages after the application of REVEALS, as well as Poaceae and Cyperaceae with increased coverages.

205    latitudes. The tree cover derived from the REVEALS reconstruction is generally lower than tree pollen percentages. However, the difference between pollen and REVEALS tree cover is smaller in North America than in Europe and Asia.

**3.2   Validation with gridded data sets**

Remote sensing forest cover within grid cells was used to validate the modern, reconstructed tree cover from the original pollen data and the REVEALS estimates for each grid cell. Here we present validation of gridded data with a 2° spatial resolution.

210    Validations with additional spatial resolutions differ only marginally and are included in the supplementary materials (S4). Tree cover from original pollen percentages is predominantly higher than remote sensing forest cover with a mean absolute error (MAE) of 31.67% in the Northern Hemisphere (see Fig. 7). As reconstructed tree cover is much lower for the REVEALS reconstruction (see Fig.5), the MAE value is reduced significantly to 20.03% (see Fig. 7).

[Figure]

**Figure 5.** Northern Hemisphere and continental mean tree pollen percentage and mean REVEALS tree cover for 2°x2° grid cells through the Holocene. (Northern Hemisphere and continental averages from different grid cell resolutions are available in S3: Reconstruction results for different spatial resolutions). Remotely sensed average forest cover for the grid cells with valid pollen coverage (at least one large lake or multiple other basins present in the time slice) is indicated with the diamond. Temporal trends are the same, but absolute tree cover is reduced in the REVEALS reconstructions compared to the original pollen data. Both pollen percentages and REVEALS estimates still overestimate tree cover.

215   Continental mean absolute errors (MAE) in tree cover from original pollen data range from 24.7% (Asia) to 35.87% tree cover (North America, see Fig. 7b). All continental MAE values are lower for the REVEALS reconstruction and range from 9.67% (Europe) to 26.43% (North America). The improvement is largest in Europe (67% relative to the initial MAE of the pollen-based reconstruction, see Fig. 7 and 8) and smallest in Asia (22%). REVEALS reconstructed tree cover also increases correlation coefficients with the exception of Asia. The REVEALS run, therefore, produced reconstructed tree cover that cor-

220   responds better remote sensing forest cover. Nevertheless, tree cover still tends to be overestimated. Spatial patterns are present for the errors of both tree cover reconstructions (see Fig. 9). In Europe the REVEALS reconstruction manages to reduce errors extensively. In Eastern and coastal Northwestern North America, the REVEALS reconstruction still tends to overestimate tree cover.

225   The comparison between our reconstruction and tree cover reconstructed in Serge et al. (2023) shows that LegacyVegetation (this publication) tends to have a lower tree cover independent of sample age. Serge et al. tend to overestimate forest cover even more than LegacyVegetation which leads to a much lower mean absolute error in LegacyVegetation compared to Serge et al.

[Figure]

**Figure 6.** Total tree pollen percentages and REVEALS reconstructed tree cover in 2x2° grid cells for 5 example time slices (reconstructions with different grid cell sizes are available in the in S3: Reconstruction results for different spatial resolutions). Valid cells are filled and include reconstructions from at least one large lake (≥ 50 ha) or several smaller basins. Tree cover in Eastern North America is higher than in Europe and Asia. REVEALS reconstructed tree cover is generally lower than tree pollen percentages.

(Fig. 10). The MAE for LegacyVegetation is slightly higher than presented in Fig. 7 due to the difference in spatial resolution and the lack of openness correction.

230 **4 Discussion**

**4.1 Continental patterns in reconstruction validity**

Our reconstructed quantitative vegetation cover datasets using REVEALS provide reconstructions of taxonomic compositions as well as tree cover in Europe, Asia, and North America and extend to 14 ka BP. The reconstructions made use of taxon-specific parameters and were, thus, able to correct some of the compositional biases present in pollen compositions. Notably,

235 the error in modern reconstructed tree cover was reduced compared to pollen-based reconstructions on all continents which shows that improvements in tree cover reconstructions from REVEALS applications are considerable.

[Figure]

**Figure 7.** Remote sensing tree cover (LANDSAT) and modern tree cover from tree pollen and REVEALS estimates (< 100 years BP) in 2x2° grid cells with mean absolute errors (MAE, see Methods section) and correlation coefficient ($R$) per group. Reconstructed tree cover from the original pollen data tends to overestimate observed (remote sensing) forest cover. Tree pollen percentages tend to overestimate observed tree cover from remote sensing data more than REVEALS estimated tree cover. The correlation between REVEALS estimates of tree cover and observed data is generally better, especially for Europe. Validations with different grid cell sizes are available in the supplement (S4).

However, continental differences are evident in the quality of tree cover reconstruction, with Europe showing a significantly larger reduction in errors compared to other regions. North America and Asia exhibit larger reconstruction errors in the RE-

240    VEALS estimates, though these are still lower than those derived from tree pollen percentages. Notably, regions such as the Great Lakes, the Labrador Peninsula, and the Pacific Northwest display particularly high errors in tree cover reconstruction. Asia, characterized by sparser coverage, presents fewer large errors increasing the overall continental reconstruction error. This highlights the need for improved vegetation reconstruction, especially in North America and Asia. The reason for this reduced performance could lie in a lack of RPP studies, especially in North America, or in a significantly higher regional variability

245    of RPP values compared to Europe. While differences in validation outcomes across varying spatial resolutions are marginal (see S4), some variability is observed when different grids are employed, highlighting spatial heterogeneity in reconstruction

[Figure]

**Figure 8.** Tree cover reconstruction error per continent for a gridded 2x2° reconstruction. Mean errors decreased with the REVEALS reconstruction for all continents but are still generally > 0 (overestimation of tree cover). Lowest errors are present in Europe.

[Figure]

**Figure 9.** Map of the reconstruction error (in % tree cover) for tree cover from pollen counts and REVEALS estimates. Remaining errors with the overall better REVEALS reconstructions are especially high in North America (Northern West Coast, Labrador Peninsula).

success. Despite these caveats, overall trends in tree cover appear consistent, with acceptable correlation coefficients, though absolute values in certain regions remain challenging to interpret with confidence as tree cover continues to be overestimated

[Figure]

**Figure 10.** (a) Comparison between LegacyVegetation (this publication) and the tree cover from Serge et al. (2023) and (b) validations with modern, remote-sensing forest cover for both data sets.

in all continents.

250

    A specific comparison with the previous European REVEALS reconstruction by Serge et al. (2023) reveals that our reconstruction generally shows lower forest cover across Europe and demonstrates a much lower MAE, indicating improved accuracy. This is notable given that Serge et al. utilized a larger number of records in their study. One potential explanation for these differences could lie in the variations in RPP values and the selection of arboreal taxa used in the reconstruction, as we

255    employ an arboreal tree threshold and include more taxa in our REVEALS reconstruction.

    In general, the tree cover trends in our reconstruction results are similar to available large-scale pollen-based vegetation reconstructions. Increases in tree cover in northern and eastern Asia up until the Holocene thermal maximum as seen in our results are consistent with reconstructions by Cao et al. (2019) and Tian et al. (2016). The reconstructed spatial patterns of

260    tree cover in China with low tree cover in the North China plain and the Tibetan Plateau and a higher tree cover along the east coast and the south agree with previous reconstructions as well (Li et al., 2023, 2022b, 2024a). Results for European tree cover trends also roughly correspond with previous REVEALS applications and show an increase of tree cover after the last glacial maximum until roughly 6 ka BP (Githumbi et al., 2022; Fyfe et al., 2015; Serge et al., 2023; Strandberg et al., 2023).

**4.2 Data use and methodological limitations**

To ensure proper dataset utilization and obtain reliable analytical results, several key considerations must be followed. The reliability of individual time series data varies based on the size of the lakes from which samples were taken. Only data from large lakes ($\geq$ 50 ha) are considered reliable for site-specific analyses, and these are clearly marked with validity flags in the dataset. When incorporating records from smaller lakes or other sources, rasterization is necessary (https://zenodo.org/records/12800291). Although our rasterization method is more flexible than previous efforts, the temporal and spatial aggregation used may reduce its reliability, due to smaller total pollen counts used in REVEALS runs and the use of an arithmetic as opposed to a weighted spatial mean. We do however find that reconstructions differences between these methods are marginal (S6).

The reliability of reconstructions also varies among different taxa due to the quality of RPP values, which is documented in detail in a supplementary file outlining the sources of RPP values (see Section "Code and Data Availability"). Reconstructions based on taxa with continental RPP values are the most reliable, followed by those based on hemispheric data, with standardized RPP values being the least reliable. This hierarchy should be considered when interpreting results. The use of continental RPP values could also make our reconstruction more reliable at larger spatial scales as opposed to local reconstructions. Additionally, uncertainties in RPP values themselves can affect reconstruction success and could be leading to the persistent overrepresentation of tree taxa despite the application of REVEALS. Tree cover reconstructions tend to have higher certainty compared to taxon-specific reconstructions, as they are based on aggregation across taxa. However, the static latitudinal arboreal threshold for Betulaceae, *Betula*, and *Alnus* poses a limitation in our reconstruction. This could be improved by incorporating a dynamic, climate-dependent threshold in future work.

Validating pollen-based tree cover estimates with remote sensing-derived forest cover also presents a challenge. One key issue is the inherent errors associated with remote sensing forest cover data. While validation using other sensors is possible, only a limited subset of the available data is cross-validated with Lidar data, which itself is characterized by limited spatial coverage (Sexton et al., 2013). A critical limitation of surface reflectance methods, as used in the Landsat-based forest cover, is their reliance on a 2D perspective, primarily capturing the forest canopy. This means that the understory is often not detected, resulting in an incomplete representation of the forest structure. In contrast, pollen-based estimates provide a more comprehensive, stratified view of the vegetation, as they incorporate all contributing taxa, not just the tree canopy. Despite this broader scope, pollen data and REVEALS estimates tend to emphasize trees more than other vegetation types consistently as is evident in the validations. Furthermore, pollen-based estimates are derived from records that span a much longer timescale than the modern forest cover data available, even though modern timeslices are used for validation. Increased anthropogenic impact could exacerbate discrepancies between pollen-based and remote-sensing estimates. This could contribute to the overestimation of forest cover, which persists in all continents. Additionally, these modern and arguably unnatural vegetation conditions may not correspond to past vegetation and may therefore have reduced significance for the reconstruction of past, natural landscapes.

Another challenge lies in validating the compositional reconstruction results. It remains uncertain whether RPP values have remained stable over time, and historical compositional data are not only scarce but also likely too recent to test this assumption effectively (Baker et al., 2016). Validating modern compositional reconstructions on large spatial scales is therefore difficult. As global compositional vegetation data are not readily available, remote sensing of tree cover serves as the best option for validation. But even with accurate tree cover reconstructions, uncertainties remain regarding the abundance of individual taxa due to the aggregated nature of the tree cover measure. To address this issue, global syntheses of tree and plant inventories or compositional remote sensing products could provide more robust validation. Additionally, vegetational compositions derived from sedimentary ancient DNA (sedaDNA) offer a promising avenue for comparing past vegetation data. Local quantitative sedaDNA vegetation signals could be averaged across multiple records to compare with pollen-based results (Niemeyer et al., 2017; Capo et al., 2021).

Lastly, the reconstructions are subject to certain limitations inherent in sedimentary pollen data, such as age uncertainty, temporal mixing, and irregular spatial and temporal resolution of records. Age uncertainty has been addressed as effectively as possible through consistent age modeling of the pollen dataset (Li et al., 2022a, 2021). However, replicating sediment and peat cores could generally provide more accurate estimates of record variability. Moreover, sampling more large lakes and ensuring precise dating would improve spatial coverage. Further, additional RPP studies are necessary to provide more accurate RPP estimates, including the development of regional RPP datasets to enhance reconstruction accuracy. This is especially the case in North America.

**4.3 Outlook**

The REVEALS tree cover reconstructions presented here offers insight into past vegetation changes and is a valuable alternative to already existing regional reconstructions, which follow different temporal and spatial aggregation methods. The Northern Hemisphere dataset provides an opportunity to explore past vegetation dynamics, gaining a deeper understanding of responses, trajectories, and potential feedback mechanisms. This is especially the case in Europe, whereas trend-based analyses should be the focus in North America and Asia. Given the increasing discussions surrounding the possibility of tipping events in vegetation cover (Armstrong McKay et al., 2022; Lenton and Williams, 2013), this could be of considerable use. While a reconstruction of exact tree lines is not trivial with pollen data, the application of REVEALS and subsequent biomization improve treeline reconstructions as shown by Binney et al. (2011). Additionally, this dataset can help address unanswered questions about Holocene vegetation dynamics, including the deglacial forest conundrum (Dallmeyer et al., 2022; Strandberg et al., 2022). It could also serve as a valuable tool for validating Earth System Models that require extensive time series and vegetation data for accurate predictions (Dallmeyer et al., 2023). Comparing modeled vegetation to reconstructed vegetation could help uncover missing dynamics in coupled climate-vegetation models and new insights gained from these applications could enhance our ability to predict future changes.

**5 Conclusions**

We present data sets of reconstructed past plant cover and tree cover in the Northern Hemisphere from a sedimentary pollen
data set using the REVEALS model. We used synthesized RPP values for reconstruction and made use of hemispheric or
standardized values, when continental ones were not available. This approach allowed us to address some of the inherent
biases in pollen compositions. Considerable improvement in the reconstruction of tree cover compared to pollen percentages is
achieved in all continents and reconstruction errors in Europe are lower compared to previous reconstructions. However, strong
overestimation of tree cover persisted in North America and Asia highlighting the need for improved regional RPP syntheses.
Extensive data on past vegetation is invaluable for the validation of coupled climate-vegetation models and the testing of
hypotheses on feedback effects and vegetation dynamics. This knowledge is essential for modeling and predicting vegetation
trajectories under anthropogenic climate change.

**6 Code and data availability**

The produced datasets are freely available from Zenodo (https://doi.org/10.5281/zenodo.13902921).

Input data from LegacyPollen 2.0 is available on PANGAEA (https://doi.pangaea.de/10.1594/PANGAEA.965907, Li et al.
2024b).

The code used to produce the datasets and adjustable rasterization code are freely available from Zenodo (https://doi.org/10.
5281/zenodo.10191859, https://doi.org/10.5281/zenodo.13902976, Schild and Ewald 2023).

**Appendix A: Original RPP values**

[revised manuscript text omitted]

ESA: Land Cover CCI Product User Guide Version 2., maps.elie.ucl.ac.be/CCI/viewer/download/ESACCI-LC-Ph2-PUGv2_2.0.pdf, 2017.

Fagerlind, F.: The real signification of pollen diagrams., Botaniska Notiser, p. 40, 1952.

Fyfe, R. M., de Beaulieu, J.-L., Binney, H., Bradshaw, R. H. W., Brewer, S., Le Flao, A., Finsinger, W., Gaillard, M.-J., Giesecke, T., Gil-Romera, G., Grimm, E. C., Huntley, B., Kunes, P., Kühl, N., Leydet, M., Lotter, A. F., Tarasov, P. E., and Tonkov, S.: The European Pollen Database: past efforts and current activities, Vegetation History and Archaeobotany, 18, 417–424, https://doi.org/10.1007/s00334-009-0215-9, 2009a.

Fyfe, R. M., de Beaulieu, J.-L., Binney, H., Bradshaw, R. H. W., Brewer, S., Le Flao, A., Finsinger, W., Gaillard, M.-J., Giesecke, T., Gil-Romera, G., Grimm, E. C., Huntley, B., Kunes, P., Kühl, N., Leydet, M., Lotter, A. F., Tarasov, P. E., and Tonkov, S.: The European Pollen Database: past efforts and current activities, Vegetation History and Archaeobotany, 18, 417–424, https://doi.org/10.1007/s00334-009-0215-9, 2009b.

Fyfe, R. M., Woodbridge, J., and Roberts, N.: From forest to farmland: pollen-inferred land cover change across Europe using the pseudobiomization approach, Global Change Biology, 21, 1197–1212, https://doi.org/10.1111/gcb.12776, _eprint: https://onlinelibrary.wiley.com/doi/pdf/10.1111/gcb.12776, 2015.

Gaillard, M.-J., Sugita, S., Mazier, F., Trondman, A.-K., Broström, A., Hickler, T., Kaplan, J. O., Kjellström, E., Kokfelt, U., Kuneš, P., Lemmen, C., Miller, P., Olofsson, J., Poska, A., Rundgren, M., Smith, B., Strandberg, G., Fyfe, R., Nielsen, A. B., Alenius, T., Balakauskas, L., Barnekow, L., Birks, H. J. B., Bjune, A., Björkman, L., Giesecke, T., Hjelle, K., Kalnina, L., Kangur, M., van der Knaap, W. O., Koff, T., Lagerås, P., Latałowa, M., Leydet, M., Lechterbeck, J., Lindbladh, M., Odgaard, B., Peglar, S., Segerström, U., von Stedingk,

H., and Seppä, H.: Holocene land-cover reconstructions for studies on land cover-climate feedbacks, Climate of the Past, 6, 483–499, https://doi.org/10.5194/cp-6-483-2010, publisher: Copernicus GmbH, 2010.

440 Geng, R., Andreev, A., Kruse, S., Heim, B., van Geffen, F., Pestryakova, L., Zakharov, E., Troeva, E., Shevtsova, I., Li, F., Zhao, Y., and Herzschuh, U.: Modern Pollen Assemblages From Lake Sediments and Soil in East Siberia and Relative Pollen Productivity Estimates for Major Taxa, Frontiers in Ecology and Evolution, 10, 837 857, https://doi.org/10.3389/fevo.2022.837857, publisher: Frontiers, 2022.

Giesecke, T., Ammann, B., and Brande, A.: Palynological richness and evenness: insights from the taxa accumulation curve, Vegetation History and Archaeobotany, 23, 217–228, https://doi.org/10.1007/s00334-014-0435-5, 2014.

445 Githumbi, E., Fyfe, R., Gaillard, M.-J., Trondman, A.-K., Mazier, F., Nielsen, A.-B., Poska, A., Sugita, S., Woodbridge, J., Azuara, J., Feurdean, A., Grindean, R., Lebreton, V., Marquer, L., Nebout-Combourieu, N., Stančikaitė, M., Tanţău, I., Tonkov, S., Shumilovskikh, L., and data contributors, L.: European pollen-based REVEALS land-cover reconstructions for the Holocene: methodology, mapping and potentials, Earth System Science Data, 14, 1581–1619, https://doi.org/10.5194/essd-14-1581-2022, publisher: Copernicus GmbH, 2022.

Harris, I., Osborn, T. J., Jones, P., and Lister, D.: Version 4 of the CRU TS monthly high-resolution gridded multivariate climate dataset,
450 Scientific Data, 7, 109, https://doi.org/10.1038/s41597-020-0453-3, number: 1 Publisher: Nature Publishing Group, 2020.

Hellman, S., Gaillard, M.-J., Broström, A., and Sugita, S.: The REVEALS model, a new tool to estimate past regional plant abundance from pollen data in large lakes: validation in southern Sweden, Journal of Quaternary Science, 23, 21–42, https://doi.org/10.1002/jqs.1126, _eprint: https://onlinelibrary.wiley.com/doi/pdf/10.1002/jqs.1126, 2008a.

Hellman, S. E. V., Gaillard, M.-j., Broström, A., and Sugita, S.: Effects of the sampling design and selection of parameter values on pollen-
455 based quantitative reconstructions of regional vegetation: a case study in southern Sweden using the REVEALS model, Vegetation History and Archaeobotany, 17, 445–459, https://doi.org/10.1007/s00334-008-0149-7, 2008b.

Herzschuh, U., Li, C., Böhmer, T., Postl, A. K., Heim, B., Andreev, A. A., Cao, X., Wieczorek, M., and Ni, J.: LegacyPollen 1.0: A taxonomically harmonized global Late Quaternary pollen dataset of 2831 records with standardized chronologies, Earth System Science Data Discussions, pp. 1–25, https://doi.org/10.5194/essd-2022-37, publisher: Copernicus GmbH, 2022.

460 Herzschuh, U., Böhmer, T., Li, C., and Cao, X.: Northern Hemisphere temperature and precipitation reconstruction from taxonomically harmonized pollen data set with revised chronologies using WA-PLS and MAT (LegacyClimate 1.0), https://doi.org/10.1594/PANGAEA.930512, artwork Size: 12 datasets Medium: application/zip Publisher: PANGAEA, 2023a.

Herzschuh, U., Böhmer, T., Li, C., Chevalier, M., Hébert, R., Dallmeyer, A., Cao, X., Bigelow, N. H., Nazarova, L., Novenko, E. Y., Park, J., Peyron, O., Rudaya, N. A., Schlütz, F., Shumilovskikh, L. S., Tarasov, P. E., Wang, Y., Wen, R., Xu, Q., and Zheng, Z.: LegacyClimate
465 1.0: a dataset of pollen-based climate reconstructions from 2594 Northern Hemisphere sites covering the last 30 kyr and beyond, Earth System Science Data, 15, 2235–2258, https://doi.org/10.5194/essd-15-2235-2023, publisher: Copernicus GmbH, 2023b.

Hjelle, K. L., Mehl, I. K., Sugita, S., and Andersen, G. L.: From pollen percentage to vegetation cover: evaluation of the Landscape Reconstruction Algorithm in western Norway, Journal of Quaternary Science, 30, 312–324, https://doi.org/10.1002/jqs.2769, _eprint: https://onlinelibrary.wiley.com/doi/pdf/10.1002/jqs.2769, 2015.

470 Huang, R., Xu, Q., Tian, F., Li, J., Wang, Y., and Hao, J.: Re-estimated relative pollen productivity of typical steppe and meadow steppe in Inner Mongolia, Quaternary Sciences, 41, 1727–1737, https://doi.org/10.11928/j.issn.1001-7410.2021.06.18, publisher: , 2021.

IPCC: Climate Change 2023: Synthesis Report. Contribution of Working Groups I, II and III to the Sixth Assessment Report of the Intergovernmental Panel on Climate Change [Core Writing Team, H. Lee and J. Romero (eds.)]. IPCC, Geneva, Switzerland., Tech. rep., Intergovernmental Panel on Climate Change (IPCC), https://www.ipcc.ch/report/ar6/syr/, 2023.

475    Jiang, F., Xu, Q., Zhang, S., Li, F., Zhang, K., Wang, M., Shen, W., Sun, Y., and Zhou, Z.: Relative pollen productivities of the major plant taxa of subtropical evergreen–deciduous mixed woodland in China, Journal of Quaternary Science, 35, 526–538, https://doi.org/10.1002/jqs.3197, _eprint: https://onlinelibrary.wiley.com/doi/pdf/10.1002/jqs.3197, 2020.

[revised manuscript text omitted]

Strandberg, G., Lindström, J., Poska, A., Zhang, Q., Fyfe, R., Githumbi, E., Kjellström, E., Mazier, F., Nielsen, A. B., Sugita, S., Trondman, A.-K., Woodbridge, J., and Gaillard, M.-J.: Mid-Holocene European climate revisited: New high-resolution regional climate model simulations using pollen-based land-cover, Quaternary Science Reviews, 281, 107 431, https://doi.org/10.1016/j.quascirev.2022.107431, 2022.

570   Strandberg, G., Chen, J., Fyfe, R., Kjellström, E., Lindström, J., Poska, A., Zhang, Q., and Gaillard, M.-J.: Did the Bronze Age deforestation of Europe affect its climate? A regional climate model study using pollen-based land cover reconstructions, Climate of the Past, 19, 1507–1530, https://doi.org/10.5194/cp-19-1507-2023, publisher: Copernicus GmbH, 2023.

Stuart, A. and Ord, J.: Kendall's Advanced Theory of Statistic, vol. Vol. 1 of *Distribution Theory*, Edward Arnold, London, 1994.

Sugita, S.: A Model of Pollen Source Area for an Entire Lake Surface, Quaternary Research, 39, 239–244,
575   https://doi.org/10.1006/qres.1993.1027, 1993.

Sugita, S.: Theory of quantitative reconstruction of vegetation I: pollen from large sites REVEALS regional vegetation composition, The Holocene, 17, 229–241, https://doi.org/10.1177/0959683607075837, publisher: SAGE Publications Ltd, 2007.

Sugita, S., Parshall, T., Calcote, R., and Walker, K.: Testing the Landscape Reconstruction Algorithm for spatially explicit reconstruction of vegetation in northern Michigan and Wisconsin, Quaternary Research, 74, 289–300, https://doi.org/10.1016/j.yqres.2010.07.008, pub-
580   lisher: Cambridge University Press, 2010.

Theuerkauf, M., Couwenberg, J., Kuparinen, A., and Liebscher, V.: A matter of dispersal: REVEALSinR introduces state-of-the-art dispersal models to quantitative vegetation reconstruction, Vegetation History and Archaeobotany, 25, 541–553, https://doi.org/10.1007/s00334-016-0572-0, 2016.

Tian, F., Cao, X., Dallmeyer, A., Ni, J., Zhao, Y., Wang, Y., and Herzschuh, U.: Quantitative woody cover reconstructions
585   from eastern continental Asia of the last 22 kyr reveal strong regional peculiarities, Quaternary Science Reviews, 137, 33–44, https://doi.org/10.1016/j.quascirev.2016.02.001, 2016.

Townshend, J.: Global Forest Cover Change (GFCC) Tree Cover Multi-Year Global 30 m V003, https://doi.org/10.5067/MEASURES/GFCC/GFCC30TC.003, 2016.

Trondman, A.-K., Gaillard, M.-J., Mazier, F., Sugita, S., Fyfe, R., Nielsen, A. B., Twiddle, C., Barratt, P., Birks, H. J. B., Bjune, A. E., Björkman, L., Broström, A., Caseldine, C., David, R., Dodson, J., Dörfler, W., Fischer, E., van Geel, B., Giesecke, T., Hultberg, T., Kalnina, L., Kangur, M., van der Knaap, P., Koff, T., Kuneš, P., Lagerås, P., Latałowa, M., Lechterbeck, J., Leroyer, C., Leydet, M., Lindbladh, M., Marquer, L., Mitchell, F. J. G., Odgaard, B. V., Peglar, S. M., Persson, T., Poska, A., Rösch, M., Seppä, H., Veski, S., and Wick, L.: Pollen-based quantitative reconstructions of Holocene regional vegetation cover (plant-functional types and land-cover types) in Europe suitable for climate modelling, Global Change Biology, 21, 676–697, https://doi.org/10.1111/gcb.12737, _eprint: https://onlinelibrary.wiley.com/doi/pdf/10.1111/gcb.12737, 2015.

Trondman, A.-K., Gaillard, M.-J., Sugita, S., Björkman, L., Greisman, A., Hultberg, T., Lagerås, P., Lindbladh, M., and Mazier, F.: Are pollen records from small sites appropriate for REVEALS model-based quantitative reconstructions of past regional vegetation? An empirical test in southern Sweden, Vegetation History and Archaeobotany, 25, 131–151, https://doi.org/10.1007/s00334-015-0536-9, 2016.

Viau, A. E., Ladd, M., and Gajewski, K.: The climate of North America during the past 2000 years reconstructed from pollen data, Global and Planetary Change, 84-85, 75–83, https://doi.org/10.1016/j.gloplacha.2011.09.010, 2012.

Vincens, A., Lézine, A.-M., Buchet, G., Lewden, D., and Le Thomas, A.: African pollen database inventory of tree and shrub pollen types, Review of Palaeobotany and Palynology, 145, 135–141, https://doi.org/10.1016/j.revpalbo.2006.09.004, 2007.

Wan, Q., Zhang, Y., Huang, K., Sun, Q., Zhang, X., Gaillard, M.-J., Xu, Q., Li, F., and Zheng, Z.: Evaluating quantitative pollen representation of vegetation in the tropics: A case study on the Hainan Island, tropical China, Ecological Indicators, 114, 106 297, https://doi.org/10.1016/j.ecolind.2020.106297, 2020.

Wan, Q., Huang, K., Chen, C., Tang, Y., Zhang, X., Zhang, Z., and Zheng, Z.: Relative Pollen Productivity Estimates for Major Plant Taxa in Middle Subtropical China, Land, 12, 1337, https://doi.org/10.3390/land12071337, number: 7 Publisher: Multidisciplinary Digital Publishing Institute, 2023.

Wang, Y., Xu, Q., Zhang, S., Sun, Y., Li, Y., Hao, J., Huang, R., Shi, J., Wang, N., Wang, T., Li, Y., Zhang, R., Zhang, X., and Zhou, Z.: Relative pollen productivity estimates and landcover reconstruction of desert steppe in arid Western China: An example in Barkol Basin, Quaternary Sciences, 41, 1738–1748, https://doi.org/10.11928/j.issn.1001-7410.2021.06.19, publisher: , 2021.

Webb, T., Howe, S. E., Bradshaw, R. H. W., and Heide, K. M.: Estimating plant abundances from pollen percentages: The use of regression analysis, Review of Palaeobotany and Palynology, 34, 269–300, https://doi.org/10.1016/0034-6667(81)90046-4, 1981.

Whitmore, J., Gajewski, K., Sawada, M., Williams, J. W., Shuman, B., Bartlein, P. J., Minckley, T., Viau, A. E., Webb, T., Shafer, S., Anderson, P., and Brubaker, L.: Modern pollen data from North America and Greenland for multi-scale paleoenvironmental applications, Quaternary Science Reviews, 24, 1828–1848, https://doi.org/10.1016/j.quascirev.2005.03.005, 2005.

Wieczorek, M. and Herzschuh, U.: Compilation of relative pollen productivity (RPP) estimates and taxonomically harmonised RPP datasets for single continents and Northern Hemisphere extratropics, Earth System Science Data, 12, 3515–3528, https://doi.org/10.5194/essd-12-3515-2020, publisher: Copernicus GmbH, 2020.

Williams, J. W., Grimm, E. C., Blois, J. L., Charles, D. F., Davis, E. B., Goring, S. J., Graham, R. W., Smith, A. J., Anderson, M., Arroyo-Cabrales, J., Ashworth, A. C., Betancourt, J. L., Bills, B. W., Booth, R. K., Buckland, P. I., Curry, B. B., Giesecke, T., Jackson, S. T., Latorre, C., Nichols, J., Purdum, T., Roth, R. E., Stryker, M., and Takahara, H.: The Neotoma Paleoecology Database, a multiproxy, international, community-curated data resource, Quaternary Research, 89, 156–177, https://doi.org/10.1017/qua.2017.105, publisher: Cambridge University Press, 2018.

625 Woodbridge, J., Fyfe, R. M., and Roberts, N.: A comparison of remotely sensed and pollen-based approaches to mapping Europe's land cover, Journal of Biogeography, 41, 2080–2092, https://doi.org/10.1111/jbi.12353, _eprint: https://onlinelibrary.wiley.com/doi/pdf/10.1111/jbi.12353, 2014.

Zhang, N., Ge, Y., Li, Y., Li, B., Zhang, R., Zhang, Z., Fan, B., Zhang, W., and Ding, G.: Modern pollen-vegetation relationships in the Taihang Mountains: Towards the quantitative reconstruction of land-cover changes in the North China Plain, Ecological Indicators, 129, 630 107 928, https://doi.org/10.1016/j.ecolind.2021.107928, 2021a.

Zhang, Y., Wei, Q., Zhang, Z., Xu, Q., Gao, W., and Li, Y.: Relative pollen productivity estimates of major plant taxa and relevant source area of pollen in the warm-temperate forest landscape of northern China, Vegetation History and Archaeobotany, 30, 231–241, https://doi.org/10.1007/s00334-020-00779-x, 2021b.